# Federated Multi-Task Learning under a Mixture of Distributions

Othmane Marfoq[1,3], Giovanni Neglia[1], Aurélien Bellet[2], Laetitia Kameni[3], and Richard Vidal[3]

[1]Inria, Université Côte d'Azur, France, {othmane.marfoq, giovanni.neglia}@inria.fr
[2]Inria, Université de Lille, France, aurelien.bellet@inria.fr
[3]Accenture Labs, France, {richard.vidal, laetitia.kameni}@accenture.com

## Abstract

The increasing size of data generated by smartphones and IoT devices motivated the development of *Federated Learning* (FL), a framework for on-device collaborative training of machine learning models. First efforts in FL focused on learning a single global model with good average performance across clients, but the global model may be arbitrarily bad for a given client, due to the inherent heterogeneity of local data distributions. Federated *multi-task learning* (MTL) approaches can learn *personalized models* by formulating an opportune penalized optimization problem. The penalization term can capture complex relations among personalized models, but eschews clear statistical assumptions about local data distributions.

In this work, we propose to study federated MTL under the flexible assumption that each local data distribution is a *mixture of unknown underlying distributions*. This assumption encompasses most of the existing personalized FL approaches and leads to federated EM-like algorithms for both client-server and fully decentralized settings. Moreover, it provides a principled way to serve personalized models to clients not seen at training time. The algorithms' convergence is analyzed through a novel federated surrogate optimization framework, which can be of general interest. Experimental results on FL benchmarks show that our approach provides models with higher accuracy and fairness than state-of-the-art methods.

## 1 Introduction

Federated Learning (FL) [28] allows a set of clients to collaboratively train models without sharing their local data. Standard FL approaches train a unique model for all clients [47, 32, 38, 29, 48]. However, as discussed in [56], the existence of such a global model suited for all clients is at odds with the statistical heterogeneity observed across different clients [37, 28]. Indeed, clients can have non-iid data and *varying preferences*. Consider for example a language modeling task: given the sequence of tokens "*I love eating*," the next word can be arbitrarily different from one client to another. Thus, having personalized models for each client is a necessity in many FL applications.

**Previous work on personalized FL.** A naive approach for FL personalization consists in learning first a global model and then fine-tuning its parameters at each client via a few iterations of stochastic gradient descent [58]. In this case, the global model plays the role of a meta-model to be used as initialization for few-shot adaptation at each client. In particular, the connection between FL and Model Agnostic Meta Learning (MAML) [27] has been studied in [19, 30, 1] in order to build a more suitable meta-model for local personalization. Unfortunately, these methods can fail to build a model with low generalization error (as exemplified by LEAF synthetic dataset [7, App. 1]). An alternative

approach is to jointly train a global model and one local model per client and then let each client build a personalized model by interpolating them [14, 9, 44]. However, if local distributions are far from the average distribution, a relevant global model does not exist and this approach boils down to every client learning only on its own local data. This issue is formally captured by the generalization bound in [14, Theorem 1].

Clustered FL [56, 20, 44] addresses the potential lack of a global model by assuming that clients can be partitioned into several clusters. Clients belonging to the same cluster share the same optimal model, but those models can be arbitrarily different across clusters (see [56, Assumption 2] for a rigorous formulation). During training, clients learn the cluster to which they belong as well as the cluster model. The Clustered FL assumption is also quite limiting, as no knowledge transfer is possible across clusters. In the extreme case where each client has its own optimal local model (recall the example on language modeling), the number of clusters coincides with the number of clients and no federated learning is possible.

Multi-Task Learning (MTL) has recently emerged as an alternative approach to learn personalized models in the federated setting and allows for more nuanced relations among clients' models [59, 63, 67, 24, 16]. The authors of [59, 63] were the first to frame FL personalization as a MTL problem. In particular, they defined federated MTL as a penalized optimization problem, where the penalization term models relationships among tasks (clients). The work [59] proposed the MOCHA algorithm for the client-server scenario, while [63, 67] presented decentralized algorithms for the same problem. Unfortunately, these algorithms can only learn simple models (linear models or linear combination of pre-trained models), because of the complex penalization term. Other MTL-based approaches [24, 23, 16, 26, 36] are able to train more general models at the cost of considering simpler penalization terms (e.g., the distance to the average model), thereby losing the capability to capture complex relations among tasks. Moreover, a general limitation of this line of work is that the penalization term is justified qualitatively and not on the basis of clear statistical assumptions on local data distributions.

More recently, [57] proposed pFedHN. pFedHN feeds local clients' representations to a global (across clients) hypernetwork, which can output personalized heterogeneous models. Unfortunately, the hypernetwork has a large memory footprint already for small clients' models (e.g., the hypernetwork in the experiments in [57] has 100 more parameters than the output model). Hence, it is not clear if pFedHN can scale to more complex models. Moreover, pFedHN requires each client to communicate multiple times for the server to learn meaningful representations. Therefore, its performance is likely to deteriorate when clients participate only once (or few times) to training, as it is the case for large-scale cross-device FL training. Furthermore, even once the hypernetwork parameters have been learned, training personalized models for new clients still requires multiple client-server communication rounds. More similar to our approach, FedFOMO [68] lets each client interpolate other clients' local models with opportune weights learned during training. However, this method lacks both theoretical justifications for such linear combinations and convergence guarantees. Moreover, FedFOMO requires the presence of a powerful server able to 1) store all individual local models and 2) learn for each client—through repeated interactions—which other clients' local models may be useful. Therefore, FedFOMO is not suited for cross-device FL where the number of clients may be very large (e.g., $10^5$–$10^7$ participating clients [28, Table 2]) and a given client may only participate in a single training round.

Overall, although current personalization approaches can lead to superior empirical performance in comparison to a shared global model or individually trained local models, it is still not well understood whether and under which conditions clients are guaranteed to benefit from collaboration.

**Our contributions.** In this work, we first show that federated learning is impossible without assumptions on local data distributions. Motivated by this negative result, we formulate a general and flexible assumption: *the data distribution of each client is a mixture of $M$ underlying distributions*. The proposed formulation has the advantage that each client can benefit from knowledge distilled from all other clients' datasets (even if any two clients can be arbitrarily different from each other). We also show that this assumption encompasses most of the personalized FL approaches previously proposed in the literature.

In our framework, a personalized model is a linear combination of $M$ shared component models. All clients jointly learn the $M$ components, while each client learns its personalized mixture weights. We show that federated EM-like algorithms can be used for training. In particular, we propose FedEM and D-FedEM for the client-server and the fully decentralized settings, respectively, and we

prove convergence guarantees. Our approach also provides a principled and efficient way to infer personalized models for clients unseen at training time. Our algorithms can easily be adapted to solve more general problems in a novel framework, which can be seen as a federated extension of the centralized surrogate optimization approach in [43]. To the best of our knowledge, our paper is the first work to propose federated surrogate optimization algorithms with convergence guarantees.

Through extensive experiments on FL benchmark datasets, we show that our approach generally yields models that 1) are on average more accurate, 2) are fairer across clients, and 3) generalize better to unseen clients than state-of-the-art personalized and non-personalized FL approaches.

**Paper outline.** The rest of the paper is organized as follows. In Section 2 we provide our impossibility result, introduce our main assumptions, and show that several popular personalization approaches can be obtained as special cases of our framework. Section 3 describes our algorithms, states their convergence results, and presents our general federated surrogate optimization framework. Finally, we provide experimental results in Section 4 before concluding in Section 5.

## 2 Problem Formulation

We consider a (countable) set $\mathcal{T}$ of classification (or regression) tasks which represent the set of possible clients. We will use the terms task and client interchangeably. Data at client $t \in \mathcal{T}$ is generated according to a local distribution $\mathcal{D}_t$ over $\mathcal{X} \times \mathcal{Y}$. Local data distributions $\{\mathcal{D}_t\}_{t \in \mathcal{T}}$ are in general different, thus it is natural to fit a separate model (hypothesis) $h_t \in \mathcal{H}$ to each data distribution $\mathcal{D}_t$. The goal is then to solve (in parallel) the following optimization problems

$$\forall t \in \mathcal{T}, \quad \minimize_{h_t \in \mathcal{H}} \mathcal{L}_{\mathcal{D}_t}(h_t), \tag{1}$$

where $h_t : \mathcal{X} \mapsto \Delta^{|\mathcal{Y}|}$ ($\Delta^D$ denoting the unitary simplex of dimension $D$), $l : \Delta^{|\mathcal{Y}|} \times \mathcal{Y} \mapsto \mathbb{R}^+$ is a loss function,[1] and $\mathcal{L}_{\mathcal{D}_t}(h_t) = \mathbb{E}_{(\mathbf{x},y) \sim \mathcal{D}_t} [l(h_t(\mathbf{x}), y)]$ is the true risk of a model $h_t$ under data distribution $\mathcal{D}_t$. For $(\mathbf{x}, y) \in \mathcal{X} \times \mathcal{Y}$, we will denote the joint distribution density associated to $\mathcal{D}_t$ by $p_t(\mathbf{x}, y)$, and the marginal densities by $p_t(\mathbf{x})$ and $p_t(y)$.

A set of $T$ clients $[T] \triangleq \{1, 2, \ldots T\} \subseteq \mathcal{T}$ participate to the initial training phase; other clients may join the system in a later stage. We denote by $\mathcal{S}_t = \{s_t^{(i)} = (\mathbf{x}_t^{(i)}, y_t^{(i)})\}_{i=1}^{n_t}$ the dataset at client $t \in [T]$ drawn i.i.d. from $\mathcal{D}_t$, and by $n = \sum_{t=1}^{T} n_t$ the total dataset size.

The idea of federated learning is to enable each client to benefit from data samples available at other clients in order to get a better estimation of $\mathcal{L}_{\mathcal{D}_t}$, and therefore get a model with a better generalization ability to unseen examples.

### 2.1 An Impossibility Result

We start by showing that some assumptions on the local distributions $p_t(\mathbf{x}, y)$, $t \in \mathcal{T}$ are needed for federated learning to be possible, i.e., for each client to be able to take advantage of the data at other clients. This holds even if all clients participate to the initial training phase (i.e., $\mathcal{T} = [T]$).

We consider the classic PAC learning framework where we fix a class of models $\mathcal{H}$ and seek a learning algorithm which is guaranteed, for all possible data distributions over $\mathcal{X} \times \mathcal{Y}$, to return with high probability a model with expected error $\epsilon$-close to the best possible error in the class $\mathcal{H}$. The worst-case sample complexity then refers to the minimum amount of labeled data required by any algorithm to reach a given $\epsilon$-approximation.

Our impossibility result for FL is based on a reduction to an impossibility result for Semi-Supervised Learning (SSL), which is the problem of learning from a training set with only a small amount of labeled data. The authors of [4] conjectured that, when the quantity of unlabeled data goes to infinity, the worst-case sample complexity of SSL improves over supervised learning at most by a constant factor that only depends on the hypothesis class [4, Conjecture 4]. This conjecture was later proved for the realizable case and hypothesis classes of finite VC dimension [13, Theorem 1], even when the marginal distribution over the domain set $\mathcal{X}$ is known [21, Theorem 2]. [2]

---

[1]In the case of (multi-output) regression, we have $h_t : \mathcal{X} \mapsto \mathbb{R}^d$ for some $d \geq 1$ and $l : \mathbb{R}^d \times \mathbb{R}^d \mapsto \mathbb{R}^+$.

[2]We note that whether the conjecture in [4] holds in the agnostic case is still an open problem.

In the context of FL, if the marginal distributions $p_t(\mathbf{x})$ are identical, but the conditional distributions $p_t(y|\mathbf{x})$ can be arbitrarily different, then each client $t$ can learn using: 1) its own local labeled dataset, and 2) the other clients' datasets, but only as unlabeled ones (because their labels have no relevance for $t$). The FL problem, with $T$ clients, then reduces to $T$ parallel SSL problems, or more precisely, it is at least as difficult as $T$ parallel SSL problems (because client $t$ has no direct access to the other local datasets but can only learn through the communication exchanges allowed by the FL algorithm). The SSL impossibility result implies that, without any additional assumption on the local distributions $p_t(\mathbf{x}, y)$, $t \in [T]$, any FL algorithm can reduce the sample complexity of client-$t$'s problem in (1) only by a constant in comparison to local learning, independently of how many other clients participate to training and how large their datasets' sizes are.

## 2.2 Learning under a Mixture Model

Motivated by the above impossibility result, in this work we propose to consider that each local data distribution $\mathcal{D}_t$ is a mixture of $M$ underlying distributions $\tilde{\mathcal{D}}_m$, $1 \le m \le M$, as formalized below.

**Assumption 1.** *There exist $M$ underlying (independent) distributions $\tilde{\mathcal{D}}_m$, $1 \le m \le M$, such that for $t \in \mathcal{T}$, $\mathcal{D}_t$ is mixture of the distributions $\{\tilde{\mathcal{D}}_m\}_{m=1}^{M}$ with weights $\pi_t^* = [\pi_{t1}^*, \ldots, \pi_{tM}^*] \in \Delta^M$, i.e.*

$$z_t \sim \mathcal{M}(\pi_t^*), \quad ((\mathbf{x}_t, y_t)\,|z_t = m) \sim \tilde{\mathcal{D}}_m, \quad \forall t \in \mathcal{T}, \tag{2}$$

*where $\mathcal{M}(\pi)$ is a multinomial (categorical) distribution with parameters $\pi$.*

Similarly to what was done above, we use $p_m(\mathbf{x}, y)$, $p_m(\mathbf{x})$, and $p_m(y)$ to denote the probability distribution densities associated to $\tilde{\mathcal{D}}_m$. We further assume that marginals over $\mathcal{X}$ are identical.

**Assumption 2.** *For all $m \in [M]$, we have $p_m(\mathbf{x}) = p(\mathbf{x})$.*

Assumption 2 is not strictly required for our analysis to hold, but, in the most general case, solving Problem (1) requires to learn generative models. Instead, under Assumption 2 we can restrict our attention to discriminative models (e.g., neural networks). [3] More specifically, we consider a parameterized set of models $\tilde{\mathcal{H}}$ with the following properties.

**Assumption 3.** *$\tilde{\mathcal{H}} = \{h_\theta\}_{\theta \in \mathbb{R}^d}$ is a set of hypotheses parameterized by $\theta \in \mathbb{R}^d$, whose convex hull is in $\mathcal{H}$. For each distribution $\tilde{\mathcal{D}}_m$ with $m \in [M]$, there exists a hypothesis $h_{\theta_m^*}$, such that*

$$l\left(h_{\theta_m^*}(\mathbf{x}), y\right) = -\log p_m(y|\mathbf{x}) + c, \tag{3}$$

*where $c \in \mathbb{R}$ is a normalization constant. The function $l(\cdot, \cdot)$ is then the log-loss associated to $p_m(y|\mathbf{x})$.*

We refer to the hypotheses in $\tilde{\mathcal{H}}$ as *component models* or simply *components*. We denote by $\Theta^* \in \mathbb{R}^{M \times d}$ the matrix whose $m$-th row is $\theta_m^*$, and by $\Pi^* \in \Delta^{T \times M}$ the matrix whose $t$-th row is $\pi_t^* \in \Delta^M$. Similarly, we will use $\Theta$ and $\Pi$ to denote arbitrary parameters.

**Remark 1.** *Assumptions 2–3 are mainly technical and are not required for our approach to work in practice. Experiments in Section 4 show that our algorithms perform well on standard FL benchmark datasets, for which these assumptions do not hold in general.*

Note that, under the above assumptions, $p_t(\mathbf{x}, y)$ depends on $\Theta^*$ and $\pi_t^*$. Moreover, we can prove (see App. A) that the optimal local model $h_t^* \in \mathcal{H}$ for client $t$ is a weighted average of models in $\tilde{\mathcal{H}}$.

**Proposition 2.1.** *Let $l(\cdot, \cdot)$ be the mean squared error loss, the logistic loss or the cross-entropy loss, and $\breve{\Theta}$ and $\breve{\Pi}$ be a solution of the following optimization problem:*

$$\underset{\Theta, \Pi}{\text{minimize}} \ \underset{t \sim D_{\mathcal{T}}}{\mathbb{E}} \ \underset{(\mathbf{x}, y) \sim \mathcal{D}_t}{\mathbb{E}} \left[-\log p_t(\mathbf{x}, y|\Theta, \pi_t)\right], \tag{4}$$

*where $D_{\mathcal{T}}$ is any distribution with support $\mathcal{T}$. Under Assumptions 1, 2, and 3, the predictors*

$$h_t^* = \sum_{m=1}^{M} \breve{\pi}_{tm} h_{\breve{\theta}_m}(\mathbf{x}), \quad \forall t \in \mathcal{T} \tag{5}$$

*minimize $\mathbb{E}_{(\mathbf{x}, y) \sim \mathcal{D}_t}[l(h_t(\mathbf{x}), y)]$ and thus solve Problem (1).*

---

[3]A possible way to ensure that Assumption 2 holds is to use the batch normalization technique from [40] to account for feature shift.

Proposition 2.1 suggests the following approach to solve Problem (1). First, we estimate the parameters $\breve{\Theta}$ and $\breve{\pi}_t$, $1 \le t \le T$, by minimizing the empirical version of Problem (4) on the training data, i.e., minimizing:

$$f(\Theta, \Pi) \triangleq -\frac{\log p(\mathcal{S}_{1:T}|\Theta, \Pi)}{n} \triangleq -\frac{1}{n}\sum_{t=1}^{T}\sum_{i=1}^{n_t} \log p(s_t^{(i)}|\Theta, \pi_t), \qquad (6)$$

which is the (negative) likelihood of the probabilistic model (2). [4] Second, we use (5) to get the client predictor for the $T$ clients present at training time. Finally, to deal with a client $t_{\text{new}} \notin [T]$ not seen during training, we keep the mixture component models fixed and simply choose the weights $\pi_{t_{\text{new}}}$ that maximize the likelihood of the client data and get the client predictor via (5).

## 2.3 Generalizing Existing Frameworks

Before presenting our federated learning algorithms in Section 3, we show that the generative model in Assumption 1 extends some popular multi-task/personalized FL formulations in the literature.

**Clustered Federated Learning [56, 20]** assumes that each client belongs to one among $C$ clusters and proposes that all clients in the same cluster learn the same model. Our framework recovers this scenario considering $M = C$ and $\pi_{tc}^* = 1$ if task (client) $t$ is in cluster $c$ and $\pi_{tc}^* = 0$ otherwise.

**Personalization via model interpolation [44, 14]** relies on learning a global model $h_{\text{glob}}$ and $T$ local models $h_{\text{loc},t}$, and then using at each client the linear interpolation $h_t = \alpha_t h_{\text{loc},t} + (1 - \alpha_t)h_{\text{glob}}$. Each client model can thus be seen as a linear combination of $M = T + 1$ models $h_m = h_{\text{loc},m}$ for $m \in [T]$ and $h_0 = h_{\text{glob}}$ with specific weights $\pi_{tt}^* = \alpha_t$, $\pi_{t0}^* = 1 - \alpha_t$, and $\pi_{tt'}^* = 0$ for $t' \in [T] \setminus \{t\}$.

**Federated MTL via task relationships.** The authors of [59] proposed to learn personalized models by solving the following optimization problem inspired from classic MTL formulations:

$$\min_{W,\Omega} \sum_{t=1}^{T}\sum_{i=1}^{n_t} l(h_{w_t}(\mathbf{x}_t^{(i)}), y_t^{(i)}) + \lambda \operatorname{tr}(W\Omega W^{\intercal}), \qquad (7)$$

where $h_{w_t}$ are linear predictors parameterized by the rows of matrix $W$ and the matrix $\Omega$ captures task relationships (similarity). This formulation is motivated by the alternating structure optimization method (ASO) [2, 70]. In App. B, we show that, when predictors $h_{\theta_m^*}$ are linear and have bounded norm, our framework leads to the same ASO formulation that motivated Problem (7). Problem (7) can also be justified by probabilistic priors [69] or graphical models [35] (see [59, App. B.1]). Similar considerations hold for our framework (see again App. B). Reference [67] extends the approach in [59] by letting each client learn a personalized model as a weighted combination of $M$ *known* hypotheses. Our approach is more general and flexible as clients learn both the weights and the hypotheses. Finally, other personalized FL algorithms, like pFedMe [16], FedU [17], and those studied in [24] and in [23], can be framed as special cases of formulation (7). Their assumptions can thus also be seen as a particular case of our framework.

# 3 Federated Expectation-Maximization

## 3.1 Centralized Expectation-Maximization

Our goal is to estimate the optimal components' parameters $\Theta^* = (\theta_m^*)_{1 \le m \le M}$ and mixture weights $\Pi^* = (\pi_t^*)_{1 \le t \le T}$ by minimizing the negative log-likelihood $f(\Theta, \Pi)$ in (6). A natural approach to solve such non-convex problems is the Expectation-Maximization algorithm (EM), which alternates between two steps. Expectation steps update the distribution (denoted by $q_t$) over the latent variables $z_t^{(i)}$ for every data point $s_t^{(i)} = (\mathbf{x}_t^{(i)}, y_t^{(i)})$ given the current estimates of the parameters $\{\Theta, \Pi\}$. Maximization steps update the parameters $\{\Theta, \Pi\}$ by maximizing the expected log-likelihood, where the expectation is computed according to the current latent variables' distributions.

The following proposition provides the EM updates for our problem (proof in App. C).

---

[4]As the distribution $\mathcal{D}_{\mathcal{T}}$ over tasks in Proposition 2.1 is arbitrary, any positively weighted sum of clients' empirical losses could be considered.

**Proposition 3.1.** *Under Assumptions 1 and 2, at the $k$-th iteration the EM algorithm updates parameter estimates through the following steps:*

**E-step:** $\quad q_t^{k+1}(z_t^{(i)} = m) \propto \pi_{tm}^k \cdot \exp\left(-l(h_{\theta_m^k}(\mathbf{x}_t^{(i)}), y_t^{(i)})\right), \quad t \in [T], \; m \in [M], \; i \in [n_t] \quad (8)$

**M-step:** $\quad \pi_{tm}^{k+1} = \dfrac{\sum_{i=1}^{n_t} q_t^{k+1}(z_t^{(i)} = m)}{n_t}, \qquad\qquad\qquad\qquad t \in [T], \; m \in [M] \quad (9)$

$$\theta_m^{k+1} \in \underset{\theta \in \mathbb{R}^d}{\arg\min} \sum_{t=1}^{T} \sum_{i=1}^{n_t} q_t^{k+1}(z_t^{(i)} = m) l\big(h_\theta(\mathbf{x}_t^{(i)}), y_t^{(i)}\big), \qquad m \in [M] \quad (10)$$

The EM updates in Proposition 3.1 have a natural interpretation. In the E-step, given current component models $\Theta^k$ and mixture weights $\Pi^k$, (8) updates the a-posteriori probability $q_t^{k+1}(z_t^{(i)} = m)$ that point $s_t^{(i)}$ of client $t$ was drawn from the $m$-th distribution based on the current mixture weight $\pi_{tm}^k$ and on how well the corresponding component $\theta_m^k$ classifies $s_t^{(i)}$. The M-step consists of two updates under fixed probabilities $q_t^{k+1}$. First, (9) updates the mixture weights $\pi_t^{k+1}$ to reflect the prominence of each distribution $\tilde{\mathcal{D}}_m$ in $\mathcal{S}_t$ as given by $q_t^{k+1}$. Finally, (10) updates the components' parameters $\Theta^{k+1}$ by solving $M$ independent, weighted empirical risk minimization problems with weights given by $q_t^{k+1}$. These weights aim to construct an unbiased estimate of the true risk over each underlying distribution $\tilde{\mathcal{D}}_m$ using only points sampled from the client mixtures, similarly to importance sampling strategies used to learn from data with sample selection bias [61, 11, 10, 64].

### 3.2 Client-Server Algorithm

Federated learning aims to train machine learning models directly on the clients, without exchanging raw data, and thus we should run EM while assuming that only client $t$ has access to dataset $\mathcal{S}_t$. The E-step (8) and the $\Pi$ update (9) in the M-step operate separately on each local dataset $\mathcal{S}_t$ and can thus be performed locally at each client $t$. On the contrary, the $\Theta$ update (10) requires interaction with other clients, since the computation spans all data samples $\mathcal{S}_{1:T}$.

In this section, we consider a client-server setting, in which each client $t$ can communicate only with a centralized server (the orchestrator) and wants to learn components' parameters $\Theta^* = (\theta_m^*)_{1 \le m \le M}$ and its own mixture weights $\pi_t^*$.

We propose the algorithm `FedEM` for *Federated Expectation-Maximization* (Alg. 1). `FedEM` proceeds through communication rounds similarly to most FL algorithms including `FedAvg` [47], `FedProx` [38], SCAFFOLD [29], and `pFedMe` [16]. At each round, 1) the central server broadcasts the (shared) component models to the clients, 2) each client locally updates components and its personalized mixture weights, and 3) sends the updated components back to the server, 4) the server aggregates the updates. The local update performed at client $t$ consists in performing the steps in (8) and (9) and updating the local estimates of $\theta_m$ through a solver which approximates the exact minimization in (10) using only the local dataset $\mathcal{S}_t$ (see line 7). `FedEM` can operate with different local solvers—even different across clients—as far as they satisfy some local improvement guarantees (see the discussion in App. H). In what follows, we restrict our focus on the practically important case where the local solver performs multiple stochastic gradient descent updates (local SGD [60]). Under the following standard assumptions (see e.g., [66]), `FedEM` converges to a stationary point of $f$. Below, we use the more compact notation $l(\theta; s_t^{(i)}) \triangleq l(h_\theta(\mathbf{x}_t^{(i)}), y_t^{(i)})$.

**Assumption 4.** *The negative log-likelihood $f$ is bounded below by $f^* \in \mathbb{R}$.*

**Assumption 5.** *(Smoothness) For all $t \in [T]$ and $i \in [n_t]$, the function $\theta \mapsto l(\theta; s_t^{(i)})$ is $L$-smooth and twice continuously differentiable.*

**Assumption 6.** *(Unbiased gradients and bounded variance) Each client $t \in [T]$ can sample a random batch $\xi$ from $\mathcal{S}_t$ and compute an unbiased estimator $g_t(\theta, \xi)$ of the local gradient with bounded variance, i.e., $\mathbb{E}_\xi[g_t(\theta, \xi)] = \frac{1}{n_t} \sum_{i=1}^{n_t} \nabla_\theta l(\theta; s_t^{(i)})$ and $\mathbb{E}_\xi \| g_t(\theta, \xi) - \frac{1}{n_t} \sum_{i=1}^{n_t} \nabla_\theta l(\theta; s_t^{(i)}) \|^2 \le \sigma^2$.*

**Assumption 7.** *(Bounded dissimilarity) There exist $\beta$ and $G$ such that for any set of weights $\alpha \in \Delta^M$:*

$$\sum_{t=1}^{T} \frac{n_t}{n} \left\| \frac{1}{n_t} \sum_{i=1}^{n_t} \sum_{m=1}^{M} \alpha_m \cdot l(\theta; s_t^{(i)}) \right\|^2 \le G^2 + \beta^2 \left\| \frac{1}{n} \sum_{t=1}^{T} \sum_{i=1}^{n_t} \sum_{m=1}^{M} \alpha_m \cdot l(\theta; s_t^{(i)}) \right\|^2.$$

**Algorithm 1:** FedEM (see also the more detailed Alg. 2 in App. D.1)

**Input** : Data $\mathcal{S}_{1:T}$; number of mixture distributions $M$; number of communication rounds $K$
**Output** : $\theta_m^K, \; m \in [M]$

**1 for** iterations $k = 1, \ldots, K$ **do**
**2**    server broadcasts $\theta_m^{k-1}, \; 1 \le m \le M$, **to the** $T$ **clients**;
**3**    **for** tasks $t = 1, \ldots, T$ **in parallel over** $T$ **clients do**
**4**      **for** component $m = 1, \ldots, M$ **do**
**5**        update $q_t^k(z_t^{(i)} = m)$ as in (8), $\; \forall i \in \{1, \ldots, n_t\}$;
**6**        update $\pi_{tm}^k$ as in (9);
**7**        $\theta_{m,t}^k \leftarrow \texttt{LocalSolver}(m, \theta_m^{k-1}, q_t^k, \mathcal{S}_t)$;
**8**      **client** $t$ **sends** $\theta_{m,t}^k, \; 1 \le m \le M$, **to the server**;
**9**    **for** component $m = 1, \ldots, M$ **do**
**10**      $\theta_m^k \leftarrow \sum_{t=1}^T \frac{n_t}{n} \times \theta_{m,t}^k$;

Assumption 7 limits the level of dissimilarity of the different tasks, similarly to what is done in [66].

**Theorem 3.2.** *Under Assumptions 1–7, when clients use SGD as local solver with learning rate $\eta = \frac{a_0}{\sqrt{K}}$, after a large enough number of communication rounds $K$, FedEM's iterates satisfy:*

$$\frac{1}{K} \sum_{k=1}^K \mathbb{E} \left\| \nabla_\Theta f \left( \Theta^k, \Pi^k \right) \right\|_F^2 \le \mathcal{O} \left( \frac{1}{\sqrt{K}} \right), \qquad \frac{1}{K} \sum_{k=1}^K \Delta_\Pi f(\Theta^k, \Pi^k) \le \mathcal{O} \left( \frac{1}{K^{3/4}} \right), \quad (11)$$

*where the expectation is over the random batches samples, and $\Delta_\Pi f(\Theta^k, \Pi^k) \triangleq f \left( \Theta^k, \Pi^k \right) - f \left( \Theta^k, \Pi^{k+1} \right) \ge 0$.*

Theorem 3.2 (proof in App. G.1) expresses the convergence of both sets of parameters ($\Theta$ and $\Pi$) to a stationary point of $f$. Indeed, the gradient of $f$ with respect to $\Theta$ becomes arbitrarily small (left inequality in (11)) and the update in Eq. (9) leads to arbitrarily small improvements of $f$ (right inequality in (11)).

We conclude this section observing that FedEM allows an *unseen client*, i.e., a client $t_{\text{new}} \notin [T]$ arriving after the distributed training procedure, to learn its personalized model. The client simply retrieves the learned components' parameters $\Theta^K$ and computes its personalized weights $\pi_{t_{\text{new}}}$ (starting for example from a uniform initialization) through one E-step (8) and the first update in the M-step (9).

## 3.3 Fully Decentralized Algorithm

In some cases, clients may want to communicate directly in a peer-to-peer fashion instead of relying on the central server mediation [see 28, Section 2.1]. In fact, fully decentralized schemes may provide stronger privacy guarantees [12] and speed-up training as they better use communication resources [41, 46] and reduce the effect of stragglers [50]. For these reasons, they have attracted significant interest recently in the machine learning community [41, 63, 42, 62, 3, 51, 46, 31]. We refer to [49] for a comprehensive survey of fully decentralized optimization (also known as consensus-based optimization), and to [31] for a unified theoretical analysis of decentralized SGD.

We propose D-FedEM (Alg. 4 in App. D.2), a *fully decentralized version* of our federated expectation maximization algorithm. As in FedEM, the M-step for $\Theta$ update is replaced by an approximate maximization step consisting of local updates. The global aggregation step in FedEM (Alg. 1, line 10) is replaced by a partial aggregation step, where each client computes a weighted average of its current components and those of a subset of clients (its *neighborhood*), which may vary over time. The convergence of decentralized optimization schemes requires certain assumptions to guarantee that each client can influence the estimates of other clients over time. In our paper, we consider the general assumption in [31, Assumption 4] (restated as Assumption 8 in App. E for completeness). For instance, this assumption is satisfied if the graph of clients' communications is strongly connected every $\tau$ rounds.

D-FedEM converges to a stationary point of $f$ (formal statement in App. E and proof in App. G.2).

**Theorem 3.3** (Informal). *In the same setting of Theorem 3.2 and under the additional Assumption 8,* `D-FedEM`*'s individual estimates* $(\Theta_t^k)_{1 \le t \le T}$ *converge to a common value* $\bar{\Theta}^k$*. Moreover,* $\bar{\Theta}^k$ *and* $\Pi^k$ *converge to a stationary point of* $f$.

### 3.4 Federated Surrogate Optimization

`FedEM` and `D-FedEM` can be seen as particular instances of a more general framework—of potential interest for other applications—that we call *federated surrogate optimization*.

The standard majorization-minimization principle [34] iteratively minimizes, at each iteration $k$, a surrogate function $g^k$ majorizing the objective function $f$. The work [43] studied this approach when each $g^k$ is a first-order surrogate of $f$ (the formal definition from [43] is given in App. F.1).

Our novel federated surrogate optimization framework considers that the objective function $f$ is a weighted sum $f = \sum_{t=1}^{T} \omega_t f_t$ of $T$ functions and iteratively minimizes $f$ in a distributed fashion using *partial* first-order surrogates $g_t^k$ for each function $f_t$. "Partial" refers to the fact that $g_t^k$ is not required to be a first order surrogate wrt the whole set of parameters, as defined formally below.

**Definition 1** (Partial first-order surrogate). *A function* $g(\mathbf{u}, \mathbf{v}) : \mathbb{R}^{d_u} \times \mathcal{V} \to \mathbb{R}$ *is a partial-first-order surrogate of* $f(\mathbf{u}, \mathbf{v})$ *wrt* $\mathbf{u}$ *near* $(\mathbf{u}_0, \mathbf{v}_0) \in R^{d_u} \times \mathcal{V}$ *when the following conditions are satisfied:*

1. $g(\mathbf{u}, \mathbf{v}) \ge f(\mathbf{u}, \mathbf{v})$ *for all* $\mathbf{u} \in \mathbb{R}^{d_u}$ *and* $\mathbf{v} \in \mathcal{V}$;
2. $r(\mathbf{u}, \mathbf{v}) \triangleq g(\mathbf{u}, \mathbf{v}) - f(\mathbf{u}, \mathbf{v})$ *is differentiable and L-smooth with respect to* $\mathbf{u}$*. Moreover, we have* $r(\mathbf{u}_0, \mathbf{v}_0) = 0$ *and* $\nabla_{\mathbf{u}} r(\mathbf{u}_0, \mathbf{v}_0) = 0$.
3. $g(\mathbf{u}, \mathbf{v}_0) - g(\mathbf{u}, \mathbf{v}) = d_{\mathcal{V}}(\mathbf{v}_0, \mathbf{v})$ *for all* $\mathbf{u} \in \mathbb{R}^{d_u}$ *and* $\mathbf{v} \in \arg\min_{\mathbf{v}' \in \mathcal{V}} g(\mathbf{u}, \mathbf{v}')$*, where* $d_{\mathcal{V}}$ *is non-negative and* $d_{\mathcal{V}}(\mathbf{v}, \mathbf{v}') = 0 \iff \mathbf{v} = \mathbf{v}'$.

Under the assumption that each client $t$ can compute a partial first-order surrogate of $f_t$, we propose algorithms for federated surrogate optimization in both the client-server setting (Alg. 3) and the fully decentralized one (Alg. 5) and prove their convergence under mild conditions (App. G.1 and G.2). `FedEM` and `D-FedEM` can be seen as particular instances of these algorithms and Theorem. 3.2 and Theorem. 3.3 follow from the more general convergence results for federated surrogate optimization. We can also use our framework to analyze the convergence of other FL algorithms such as `pFedMe` [16], as we illustrate in App. F.3.

## 4 Experiments

**Datasets and models.** We evaluated our method on five federated benchmark datasets spanning a wide range of machine learning tasks: image classification (CIFAR10 and CIFAR100 [33]), handwritten character recognition (EMNIST [8] and FEMNIST [7]),[5] and language modeling (Shakespeare [7, 47]). Shakespeare dataset (resp. FEMNIST) was naturally partitioned by assigning all lines from the same characters (resp. all images from the same writer) to the same client. We created federated versions of CIFAR10 and EMNIST by distributing samples with the same label across the clients according to a symmetric Dirichlet distribution with parameter $0.4$, as in [65]. For CIFAR100, we exploited the availability of "coarse" and "fine" labels, using a two-stage Pachinko allocation method [39] to assign $600$ sample to each of the $100$ clients, as in [54]. We also evaluated our method on a synthetic dataset verifying Assumptions 1–3. For all tasks, we randomly split each local dataset into training ($60\%$), validation ($20\%$) and test ($20\%$) sets. Table 1 summarizes datasets, models, and number of clients (more details can be found in App. I.1). Code is available at `https://github.com/omarfoq/FedEM`.

**Other FL approaches.** We compared our algorithms with global models trained with `FedAvg` [47] and `FedProx` [38] as well as different personalization approaches: a personalized model trained only on the local dataset, `FedAvg` with local tuning (`FedAvg+`) [27], Clustered FL [56] and `pFedMe` [16]. For each method and each task, the learning rate and the other hyperparameters were tuned via grid search (details in App. I.2). `FedAvg+` updated the local model through a single pass on the local dataset. Unless otherwise stated, the number of components considered by `FedEM` was $M = 3$, training occurred over 80 communication rounds for Shakespeare and 200 rounds for all other datasets.

---

[5] For training, we sub-sampled $10\%$ and $15\%$ from EMNIST and FEMNIST datasets respectively.

Table 1: Datasets and models (details in App. I.1).

| Dataset | Task | Clients | Total samples | Model |
|---|---|---|---|---|
| FEMNIST [7] | Handwritten character recognition | 539 | 120, 772 | 2-layer CNN + 2-layer FFN |
| EMNIST [8] | Handwritten character recognition | 100 | 81, 425 | 2-layer CNN + 2-layer FFN |
| CIFAR10 [33] | Image classification | 80 | 60, 000 | MobileNet-v2 [55] |
| CIFAR100 [33] | Image classification | 100 | 60, 000 | MobileNet-v2 [55] |
| Shakespeare [7, 47] | Next-Character Prediction | 778 | 4, 226, 158 | Stacked-LSTM [25] |
| Synthetic | Binary Classification | 300 | 1, 570, 507 | Linear model |

Table 2: Test accuracy: average across clients / bottom decile.

| Dataset | Local | FedAvg [47] | FedProx [38] | FedAvg+ [27] | Clustered FL [56] | pFedMe [16] | FedEM (Ours) |
|---|---|---|---|---|---|---|---|
| FEMNIST | 71.0 / 57.5 | 78.6 / 63.9 | 78.9 / 64.0 | 75.3 / 53.0 | 73.5 / 55.1 | 74.9 / 57.6 | **79.9** / **64.8** |
| EMNIST | 71.9 / 64.3 | 82.6 / 75.0 | 83.0 / 75.4 | 83.1 / 75.8 | 82.7 / 75.0 | 83.3 / 76.4 | **83.5** / **76.6** |
| CIFAR10 | 70.2 / 48.7 | 78.2 / 72.4 | 78.0 / 70.8 | 82.3 / 70.6 | 78.6 / 71.2 | 81.7 / 73.6 | **84.3** / **78.1** |
| CIFAR100 | 31.5 / 19.9 | 40.9 / 33.2 | 41.0 / 33.2 | 39.0 / 28.3 | 41.5 / 34.1 | 41.8 / 32.5 | **44.1** / **35.0** |
| Shakespeare | 32.0 / 16.6 | **46.7** / 42.8 | 45.7 / 41.9 | 40.0 / 25.5 | 46.6 / 42.7 | 41.2 / 36.8 | **46.7** / **43.0** |
| Synthetic | 65.7 / 58.4 | 68.2 / 58.9 | 68.2 / 59.0 | 68.9 / 60.2 | 69.1 / 59.0 | 69.2 / 61.2 | **74.7** / **66.7** |

At each round, clients train for one epoch. Results for `D-FedEM` are in App. J.1. A comparison with `MOCHA` [59], which can only train linear models, is presented in App. J.2.

**Average performance of personalized models.** The performance of each personalized model (which is the same for all clients in the case of `FedAvg` and `FedProx`) is evaluated on the local test dataset (unseen at training). Table 2 shows the average weighted accuracy with weights proportional to local dataset sizes. We observe that `FedEM` obtains the best performance across all datasets.

**Fairness across clients.** `FedEM`'s improvement in terms of average accuracy could be the result of learning particularly good models for some clients at the expense of bad models for other clients. Table 2 shows the bottom decile of the accuracy of local models, i.e., the $(T/10)$-th worst accuracy (the minimum accuracy is particularly noisy, notably because some local test datasets are very small). Even clients with the worst personalized models are still better off when `FedEM` is used for training.

**Clients sampling.** In cross-device federated learning, only a subset of clients may be available at each round. We ran CIFAR10 experiments with different levels of participation: at each round a given fraction of all clients were sampled uniformly without replacement. We restrict the comparison to `FedEM` and `FedAvg+`, as 1) `FedAvg+` performed better than `FedProx` and `FedAvg` in the previous CIFAR10 experiments, 2) it is not clear how to extend `pFedMe` and Clustered FL to handle client sampling. Results in Fig. 1 (left) show that `FedEM` is more robust to low clients' participation levels. We provide additional results on client sampling, including a comparison with `APFL` [14], in App. J.6.

**Generalization to unseen clients.** As discussed in Section 3.2, `FedEM` allows new clients arriving after the distributed training to easily learn their personalized models. With the exception of `FedAvg+`, it is not clear how the other personalized FL algorithms should be extended to tackle the same goal (see discussion in App. J.3). In order to evaluate the quality of new clients' personalized models, we performed an experiment where only 80% of the clients ("old" clients) participate to the training. The remaining 20% join the system in a second phase and use their local training datasets to learn their personalized weights. Table 3 shows that `FedEM` allows new clients to learn a personalized model at least as good as `FedAvg`'s global one and always better than `FedAvg+`'s one. Unexpectedly, new clients achieve sometimes a significantly higher test accuracy than old clients (e.g., 47.5% against 44.1% on CIFAR100). Our investigation in App. J.3 suggests that, by selecting their mixture weights on local datasets that were not used to train the components, new clients can compensate for potential overfitting in the initial training phase. We also investigate in App. J.3 the effect of the local dataset size on the accuracy achieved by unseen clients, showing that personalization is effective even when unseen clients have small datasets.

**Effect of $M$.** A limitation of `FedEM` is that each client needs to update and transmit $M$ components at each round, requiring roughly $M$ times more computation and $M$ times larger messages. Nevertheless, the number of components to consider in practice is quite limited. We used $M = 3$ in our previous experiments, and Fig. 1 (right) shows that larger values do not yield much improvement and $M = 2$ already provides a significant level of personalization. In all experiments above, the number of communication rounds allowed all approaches to converge. As a consequence, even if other methods

Table 3: Average test accuracy across clients unseen at training (train accuracy in parenthesis).

| Dataset | FedAvg [47] | FedAvg+ [27] | FedEM (Ours) |
|---|---|---|---|
| FEMNIST | 78.3 (80.9) | 74.2 (84.2) | **79.1** (81.5) |
| EMNIST | 83.4 (82.7) | 83.7 (92.9) | **84.0** (83.3) |
| CIFAR10 | 77.3 (77.5) | 80.4 (80.5) | **85.9** (90.7) |
| CIFAR100 | 41.1 (42.1) | 36.5 (55.3) | **47.5** (46.6) |
| Shakespeare | **46.7** (47.1) | 40.2 (93.0) | **46.7** (46.6) |
| Synthetic | 68.6 (70.0) | 69.1 (72.1) | **73.0** (74.1) |

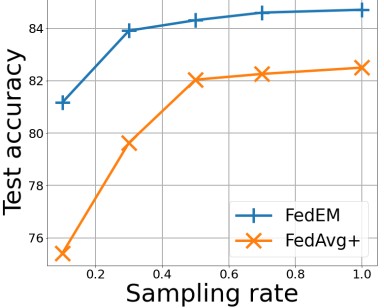 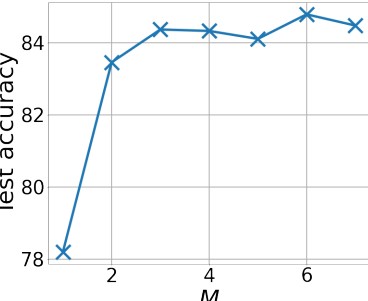

Figure 1: Effect of client sampling rate (left) and `FedEM` number of mixture components $M$ (right) on the test accuracy for CIFAR10 [33].

trained over $M = 3$ times more rounds—in order to have as much computation and communication as `FedEM`—the conclusions would not change. As a final experiment, we considered a time-constrained setting, where `FedEM` is limited to run one third ($= 1/M$) of the rounds (Table 7 in App. J.5). Even if `FedEM` does not reach its maximum accuracy, it still outperforms the other methods on 3 datasets.

## 5 Conclusion

In this paper, we proposed a novel federated MTL approach based on the flexible assumption that local data distributions are mixtures of underlying distributions. Our EM-like algorithms allow clients to jointly learn shared component models and personalized mixture weights in client-server and fully decentralized settings. We proved convergence guarantees for our algorithms through a general federated surrogate optimization framework which can be used to analyze other FL formulations. Extensive empirical evaluation shows that our approach learns models with higher accuracy and fairness than state-of-the-art FL algorithms, even for clients not present at training time.

In future work, we aim to reduce the local computation and communication of our algorithms. Aside from standard compression schemes [22], a promising direction is to limit the number of component models that a client updates/transmits at each step. This could be done in an adaptive manner based on the client's current mixture weights. A simultaneously published work [15] proposes a federated EM algorithm (also called `FedEM`), which does not address personalization but reduces communication requirements by compressing appropriately defined complete data sufficient statistics.

A second interesting research direction is to study personalized FL approaches under privacy constraints (quite unexplored until now with the notable exception of [3]). Some features of our algorithms may be beneficial for privacy (e.g., the fact that personalized weights are kept locally and that all users contribute to all shared models). We hope to design differentially private versions of our algorithms and characterize their privacy-utility trade-offs.

## 6 Acknowledgements

This work has been supported by the French government, through the 3IA Côte d'Azur Investments in the Future project managed by the National Research Agency (ANR) with the reference number ANR-19-P3IA-0002, and through grants ANR-16-CE23-0016 (Project PAMELA) and ANR-20-CE23-0015 (Project PRIDE). The authors are grateful to the OPAL infrastructure from Université Côte d'Azur for providing computational resources and technical support.

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
