# Appendix

## Table of Contents

# A Proof of Proposition 2.1

For $h \in \mathcal{H}$ and $(\mathbf{x}, y) \in \mathcal{X} \times \mathcal{Y}$, let $p_h(y|\mathbf{x})$ denote the conditional probability distribution of $y$ given $\mathbf{x}$ under model $h$, i.e.,

$$p_h(y|\mathbf{x}) \triangleq e^{c_h(\mathbf{x})} \times \exp\left\{ -l(h(\mathbf{x}), y) \right\}, \tag{12}$$

where

$$c_h(\mathbf{x}) \triangleq -\log\left[ \int_{y \in \mathcal{Y}} \exp\left\{ -l(h(\mathbf{x}), y) \right\} \mathrm{d}\, y \right]. \tag{13}$$

We also remind that the entropy of a probability distribution $q$ over $\mathcal{Y}$ is given by

$$H(q) \triangleq -\int_{y \in \mathcal{Y}} q(y) \cdot \log q(y) \,\mathrm{d}\, y, \tag{14}$$

and that the Kullback-Leibler divergence between two probability distributions $q_1$ and $q_2$ over $\mathcal{Y}$ is given by

$$\mathcal{KL}(q_1 \| q_2) \triangleq \int_{y \in \mathcal{Y}} q_1(y) \cdot \log \frac{q_1(y)}{q_2(y)} \,\mathrm{d}\, y. \tag{15}$$

**Proposition 2.1.** *Let $l(\cdot, \cdot)$ be the mean squared error loss, the logistic loss or the cross-entropy loss, and $\breve{\Theta}$ and $\breve{\Pi}$ be a solution of the following optimization problem:*

$$\underset{\Theta, \Pi}{\text{minimize}} \, \underset{t \sim D_\mathcal{T}}{\mathbb{E}} \, \underset{(\mathbf{x}, y) \sim \mathcal{D}_t}{\mathbb{E}} \left[ -\log p_t(\mathbf{x}, y | \Theta, \pi_t) \right], \tag{4}$$

*where $D_\mathcal{T}$ is any distribution with support $\mathcal{T}$. Under Assumptions 1, 2, and 3, the predictors*

$$h_t^* = \sum_{m=1}^{M} \breve{\pi}_{tm} h_{\breve{\theta}_m}, \quad \forall t \in \mathcal{T} \tag{5}$$

*minimize $\mathbb{E}_{(\mathbf{x}, y) \sim \mathcal{D}_t}\left[ l(h_t(\mathbf{x}), y) \right]$ and thus solve Problem (1).*

*Proof.* We prove the result for each of the three possible cases of the loss function. We verify that $c_h$ does not depend on $h$ in each of the three cases, then we use Lemma A.3 to conclude.

**Mean Squared Error Loss** This is the case of a regression problem where $\mathcal{Y} = \mathbb{R}^d$ for some $d > 0$. For $\mathbf{x}, y \in \mathcal{X} \times \mathcal{Y}$ and $h \in \mathcal{H}$, we have

$$p_h(y|\mathbf{x}) = \frac{1}{\sqrt{(2\pi)^d}} \cdot \exp\left\{ -\frac{\|h(\mathbf{x}) - y\|^2}{2} \right\}, \tag{16}$$

and

$$c_h(\mathbf{x}) = -\log\left( \sqrt{(2\pi)^d} \right) \tag{17}$$

**Logistic Loss** This is the case of a binary classification problem where $\mathcal{Y} = \{0, 1\}$. For $\mathbf{x}, y \in \mathcal{X} \times \mathcal{Y}$ and $h \in \mathcal{H}$, we have

$$p_h(y|\mathbf{x}) = (h(\mathbf{x}))^y \cdot (1 - h(\mathbf{x}))^{1-y}, \tag{18}$$

and

$$c_h(\mathbf{x}) = 0 \tag{19}$$

**Cross-entropy loss** This is the case of a classification problem where $\mathcal{Y} = [L]$ for some $L > 1$. For $\mathbf{x}, y \in \mathcal{X} \times \mathcal{Y}$ and $h \in \mathcal{H}$, we have

$$p_h(y|\mathbf{x}) = \prod_{l=1}^{L} (h(\mathbf{x}))^{\mathbb{1}_{\{y=l\}}}, \tag{20}$$

and

$$c_h(\mathbf{x}) = 0 \tag{21}$$

**Conclusion** For $t \in \mathcal{T}$, consider a predictor $h_t^*$ minimizing $\mathbb{E}_{(\mathbf{x},y)\sim\mathcal{D}_t}[l(h_t(\mathbf{x}),y)]$. Using Lemma A.3, for $(\mathbf{x}, y) \in \mathcal{X} \times \mathcal{Y}$, we have

$$p_{h_t^*}(y|\mathbf{x}) = \sum_{m=1}^{M} \breve{\pi}_{tm} \cdot p_m\left(y|\mathbf{x}, \breve{\theta}_m\right). \tag{22}$$

We multiply both sides of this equality by $y$ and we integrate over $y \in \mathcal{Y}$. Note that in all three cases we have

$$\forall \mathbf{x} \in \mathcal{X}, \quad \int_{y\in\mathcal{Y}} y \cdot p_h(\cdot|\mathbf{x})\, \mathrm{d}\, y = h(\mathbf{x}). \tag{23}$$

It follows that

$$h_t^* = \sum_{m=1}^{M} \breve{\pi}_{tm} h_{\breve{\theta}_m}, \quad \forall t \in \mathcal{T}. \tag{24}$$

$\square$

## Supporting Lemmas

**Lemma A.1.** *Suppose that Assumptions 1 and 3 hold, and consider $\breve{\Theta}$ and $\breve{\Pi}$ to be a solution of Problem (4). Then*

$$p_t(\mathbf{x}, y|\breve{\Theta}, \breve{\pi}_t) = p_t(\mathbf{x}, y|\Theta^*, \pi_t^*), \ \forall t \in \mathcal{T}. \tag{25}$$

*Proof.* For $t \in \mathcal{T}$,

$$\mathbb{E}_{(\mathbf{x},y)\sim\mathcal{D}_t}\left[-\log p_t(\mathbf{x}, y|\breve{\Theta}, \breve{\pi}_t)\right] \tag{26}$$

$$= -\int_{(\mathbf{x},y)\in\mathcal{X}\times\mathcal{Y}} p_t(\mathbf{x}, y|\Theta^*, \pi_t^*) \cdot \log p_t(\mathbf{x}, y|\breve{\Theta}, \breve{\pi}_t)\, \mathrm{d}\,\mathbf{x}\,\mathrm{d}\, y \tag{27}$$

$$= -\int_{(\mathbf{x},y)\in\mathcal{X}\times\mathcal{Y}} p_t(\mathbf{x}, y|\Theta^*, \pi_t^*) \cdot \log \frac{p_t(\mathbf{x}, y|\breve{\Theta}, \breve{\pi}_t)}{p_t(\mathbf{x}, y|\Theta^*, \pi_t^*)}\, \mathrm{d}\,\mathbf{x}\,\mathrm{d}\, y$$

$$\quad - \int_{(\mathbf{x},y)\in\mathcal{X}\times\mathcal{Y}} p_t(\mathbf{x}, y|\Theta^*, \pi_t^*) \cdot \log p_t(\mathbf{x}, y|\Theta^*, \pi_t^*)\, \mathrm{d}\,\mathbf{x}\,\mathrm{d}\, y \tag{28}$$

$$= \mathcal{KL}\left(p_t(\cdot|\Theta^*, \pi_t^*) \,\|\, p_t(\cdot|\breve{\Theta}, \breve{\pi}_t)\right) + H\left[p_t(\cdot|\Theta^*, \pi_t^*)\right], \tag{29}$$

Since the $\mathcal{KL}$ divergence is non-negative, we have

$$\mathbb{E}_{(\mathbf{x},y)\sim\mathcal{D}_t}\left[-\log p_t(\mathbf{x}, y|\breve{\Theta}, \breve{\pi}_t)\right] \geq H\left[p_t(\cdot|\Theta^*, \pi_t^*)\right] = \mathbb{E}_{(\mathbf{x},y)\sim\mathcal{D}_t}\left[-\log p_t(\mathbf{x}, y|\Theta^*, \pi_t^*)\right]. \tag{30}$$

Taking the expectation over $t \sim \mathcal{D}_\mathcal{T}$, we write

$$\mathbb{E}_{t\sim\mathcal{D}_\mathcal{T}}\mathbb{E}_{(\mathbf{x},y)\sim\mathcal{D}_t}\left[-\log p_t(\mathbf{x}, y|\breve{\Theta}, \breve{\pi}_t)\right] \geq \mathbb{E}_{t\sim\mathcal{D}_\mathcal{T}}\mathbb{E}_{(\mathbf{x},y)\sim\mathcal{D}_t}\left[-\log p_t(\mathbf{x}, y|\Theta^*, \pi_t^*)\right]. \tag{31}$$

Since $\breve{\Theta}$ and $\breve{\Pi}$ is a solution of Problem (4), we also have

$$\mathbb{E}_{t\sim\mathcal{D}_\mathcal{T}}\mathbb{E}_{(\mathbf{x},y)\sim\mathcal{D}_t}\left[-\log p_t(\mathbf{x}, y|\breve{\Theta}, \breve{\pi}_t)\right] \leq \mathbb{E}_{t\sim\mathcal{D}_\mathcal{T}}\mathbb{E}_{(\mathbf{x},y)\sim\mathcal{D}_t}\left[-\log p_t(\mathbf{x}, y|\Theta^*, \pi_t^*)\right]. \tag{32}$$

Combining (31), (32), and (29), we have

$$\mathbb{E}_{t\sim\mathcal{D}_\mathcal{T}} \mathcal{KL}\left(p_t(\cdot|\Theta^*, \pi_t^*) \,\|\, p_t(\cdot|\breve{\Theta}, \breve{\pi}_t)\right) = 0. \tag{33}$$

Since $\mathcal{KL}$ divergence is non-negative, and the support of $\mathcal{D}_\mathcal{T}$ is the countable set $\mathcal{T}$, it follows that

$$\forall t \in \mathcal{T}, \quad KL\left(p_t(\cdot|\Theta^*, \pi_t^*) \,\|\, p_t(\cdot|\breve{\Theta}, \breve{\pi}_t)\right) = 0. \tag{34}$$

Thus,

$$p_t(\mathbf{x}, y|\breve{\Theta}, \breve{\pi}_t) = p_t(\mathbf{x}, y|\Theta^*, \pi_t^*), \quad \forall t \in \mathcal{T}. \tag{35}$$

$\square$

**Lemma A.2.** *Consider $M$ probability distributions on $\mathcal{Y}$, that we denote $q_m$, $m \in [M]$, and $\alpha = (\alpha_1, \ldots, \alpha_m) \in \Delta^M$. For any probability distribution $q$ over $\mathcal{Y}$, we have*

$$\sum_{m=1}^{M} \alpha_m \cdot \mathcal{KL}\left(q_m \| \sum_{m'=1}^{M} \alpha_{m'} \cdot q_{m'}\right) \leq \sum_{m=1}^{M} \alpha_m \cdot \mathcal{KL}\left(q_m \| q\right), \tag{36}$$

*with equality if and only if,*

$$q = \sum_{m=1}^{M} \alpha_m \cdot q_m. \tag{37}$$

*Proof.*

$$\sum_{m=1}^{M} \alpha_m \cdot \mathcal{KL}\left(q_m \| q\right) - \sum_{m=1}^{M} \alpha_m \cdot \mathcal{KL}\left(q_m \| \sum_{m'=1}^{M} \alpha_{m'} \cdot q_{m'}\right)$$

$$= \sum_{m=1}^{M} \alpha_m \cdot \left[\mathcal{KL}\left(q_m \| q\right) - \mathcal{KL}\left(q_m \| \sum_{m'=1}^{M} \alpha_{m'} \cdot q_{m'}\right)\right] \tag{38}$$

$$= -\sum_{m=1}^{M} \alpha_m \int_{y \in \mathcal{Y}} q_m(y) \cdot \log\left(\frac{q(y)}{\sum_{m'=1}^{M} \alpha_{m'} \cdot q_{m'}(y)}\right) \tag{39}$$

$$= -\int_{y \in \mathcal{Y}} \left\{\sum_{m=1}^{M} \alpha_m \cdot q_m(y)\right\} \cdot \log\left(\frac{q(y)}{\sum_{m'=1}^{M} \alpha_{m'} \cdot q_{m'}(y)}\right) \mathrm{d}y \tag{40}$$

$$= \mathcal{KL}\left(\sum_{m=1}^{M} \alpha_m \cdot q_m \| q\right) \geq 0. \tag{41}$$

The equality holds, if and only if,

$$q = \sum_{m=1}^{M} \alpha_m \cdot q_m. \tag{42}$$

$\square$

**Lemma A.3.** *Consider $\breve{\Theta}$ and $\breve{\Pi}$ to be a solution of Problem (4). Under Assumptions 1, 2, and 3, if $c_h$ does not depend on $h \in \mathcal{H}$, then the predictors $h_t^*$, $t \in \mathcal{T}$, minimizing $\mathbb{E}_{(\mathbf{x},y) \sim \mathcal{D}_t}[l(h_t(\mathbf{x}), y)]$, verify for $(\mathbf{x}, y) \in \mathcal{X} \times \mathcal{Y}$*

$$p_{h_t^*}(y|\mathbf{x}) = \sum_{m=1}^{M} \breve{\pi}_{tm} \cdot p_m\left(y|\mathbf{x}, \breve{\theta}_m\right). \tag{43}$$

*Proof.* For $t \in \mathcal{T}$ and $h_t \in \mathcal{H}$, under Assumptions 1, 2, and 3, we have

$$\mathbb{E}_{(\mathbf{x},y) \sim \mathcal{D}_t}[l(h_t(\mathbf{x}), y)] = \int_{\mathbf{x}, y \in \mathcal{X} \times \mathcal{Y}} l(h_t(\mathbf{x}), y) \cdot p_t\left(\mathbf{x}, y | \Theta^*, \pi_t^*\right) \mathrm{d}\mathbf{x} \, \mathrm{d}y. \tag{44}$$

Using Lemma A.1, it follows that

$$\mathbb{E}_{(\mathbf{x},y) \sim \mathcal{D}_t}[l(h_t(\mathbf{x}), y)] = \int_{\mathbf{x}, y \in \mathcal{X} \times \mathcal{Y}} l(h_t(\mathbf{x}), y) \cdot p_t\left(\mathbf{x}, y | \breve{\Theta}, \breve{\pi}_t\right) \mathrm{d}\mathbf{x} \, \mathrm{d}y. \tag{45}$$

Thus, using Assumptions 1 and 2 we have,

$$\mathbb{E}_{(\mathbf{x},y) \sim \mathcal{D}_t}[l(h_t(\mathbf{x}), y)] \tag{46}$$

$$= \int_{\mathbf{x}, y \in \mathcal{X} \times \mathcal{Y}} l(h_t(\mathbf{x}), y) \cdot p_t\left(\mathbf{x}, y | \breve{\Theta}, \breve{\pi}_t\right) \mathrm{d}\mathbf{x} \, \mathrm{d}y \tag{47}$$

$$= \int_{\mathbf{x}, y \in \mathcal{X} \times \mathcal{Y}} l(h_t(\mathbf{x}), y) \cdot \left(\sum_{m=1}^{M} \breve{\pi}_{tm} \cdot p_m\left(y|\mathbf{x}, \breve{\theta}_m\right)\right) p(\mathbf{x}) \, \mathrm{d}\mathbf{x} \, \mathrm{d}y \tag{48}$$

$$= \int_{\mathbf{x} \in \mathcal{X}} \left[ \sum_{m=1}^{M} \breve{\pi}_{tm} \int_{y \in \mathcal{Y}} l(h_t(\mathbf{x}), y) \cdot p_m\left(y | \mathbf{x}, \breve{\theta}_m\right) \mathrm{d}\, y \right] p(\mathbf{x}) \,\mathrm{d}\,\mathbf{x} \tag{49}$$

$$= \int_{\mathbf{x} \in \mathcal{X}} \left[ \sum_{m=1}^{M} \breve{\pi}_{tm} \left\{ c_{h_t}(\mathbf{x}) - \int_{y \in \mathcal{Y}} p_m\left(y | \mathbf{x}, \breve{\theta}_m\right) \log p_{h_t}(y | \mathbf{x}) \,\mathrm{d}\, y \right\} \right] p(\mathbf{x}) \,\mathrm{d}\,\mathbf{x} \tag{50}$$

$$= \int_{\mathbf{x} \in \mathcal{X}} \left[ c_{h_t}(\mathbf{x}) - \sum_{m=1}^{M} \breve{\pi}_{tm} \int_{y \in \mathcal{Y}} p_m\left(y | \mathbf{x}, \breve{\theta}_m\right) \log p_{h_t}(y | \mathbf{x}) \,\mathrm{d}\, y \right] p(\mathbf{x}) \,\mathrm{d}\,\mathbf{x} \tag{51}$$

$$= \int_{\mathbf{x} \in \mathcal{X}} \left[ c_{h_t}(\mathbf{x}) + \sum_{m=1}^{M} \breve{\pi}_{tm} \cdot H\left( p_m\left( \cdot | \mathbf{x}, \breve{\theta}_m\right) \right) \right] p(\mathbf{x}) \,\mathrm{d}\,\mathbf{x}$$

$$+ \int_{\mathbf{x} \in \mathcal{X}} \left[ \sum_{m=1}^{M} \breve{\pi}_{tm} \cdot \mathcal{KL}\left( p_m\left( \cdot | \mathbf{x}, \breve{\theta}_m\right) \| p_{h_t}(\cdot | \mathbf{x}) \right) \right] p(\mathbf{x}) \,\mathrm{d}\,\mathbf{x}. \tag{52}$$

Let $h_t^{\circ}$ be a predictor satisfying the following equality:

$$p_{h_t^{\circ}}(y | \mathbf{x}) = \sum_{m=1}^{M} \breve{\pi}_{tm} \cdot p_m\left(y | \mathbf{x}, \breve{\theta}_m\right).$$

Using Lemma A.2, we have

$$\sum_{m=1}^{M} \breve{\pi}_{tm} \cdot \mathcal{KL}\left( p_m\left( \cdot | \mathbf{x}, \breve{\theta}_m\right) \| p_{h_t}(\cdot | \mathbf{x}) \right) \geq \sum_{m=1}^{M} \breve{\pi}_{tm} \cdot \mathcal{KL}\left( p_m\left( \cdot | \mathbf{x}, \breve{\theta}_m\right) \| p_{h_t^{\circ}}(\cdot | \mathbf{x}) \right) \tag{53}$$

with equality if and only if

$$p_{h_t}(\cdot | \mathbf{x}) = p_{h_t^{\circ}}(\cdot | \mathbf{x}). \tag{54}$$

Since $c_h$ does not depend on $h$, replacing (53) in (52), it follows that

$$\mathbb{E}_{(\mathbf{x},y) \sim \mathcal{D}_t} \left[ l(h_t(\mathbf{x}), y) \right] \geq \mathbb{E}_{(\mathbf{x},y) \sim \mathcal{D}_t} \left[ l(h_t^{\circ}(\mathbf{x}), y) \right]. \tag{55}$$

This inequality holds for any predictor $h_t$ and in particular for $h_t^* \in \arg\min \mathbb{E}_{(\mathbf{x},y) \sim \mathcal{D}_t} \left[ l(h_t(\mathbf{x}), y) \right]$, for which it also holds the opposite inequality, then:

$$\mathbb{E}_{(\mathbf{x},y) \sim \mathcal{D}_t} \left[ l(h_t^*(\mathbf{x}), y) \right] = \mathbb{E}_{(\mathbf{x},y) \sim \mathcal{D}_t} \left[ l(h_t^{\circ}(\mathbf{x}), y) \right], \tag{56}$$

and the equality implies that

$$p_{h_t^*}(\cdot | \mathbf{x}) = p_{h_t^{\circ}}(\cdot | \mathbf{x}) = \sum_{m=1}^{M} \breve{\pi}_{tm} \cdot p_m\left( \cdot | \mathbf{x}, \breve{\theta}_m\right). \tag{57}$$

$\square$

# B   Relation with Other Multi-Task Learning Frameworks

In this appendix, we give more details about the relation of our formulation with existing frameworks for (federated) MTL sketched in Section 2.3. We suppose that Assumptions 1–3 hold and that each client learns a predictor of the form (5). Note that this is more general than [67], where each client learns a personal hypothesis as a weighted combination of a set of $M$ base *known* hypothesis, since the base hypothesis and *not only the weights* are learned in our case.

**Alternating Structure Optimization [70].**   Alternating structure optimization (ASO) is a popular MTL approach that learns a shared low-dimensional predictive structure on hypothesis spaces from multiple related tasks, i.e., all tasks are assumed to share a common feature space $P \in \mathbb{R}^{d' \times d}$, where $d' \leq \min(T, d)$ is the dimensionality of the shared feature space and $P$ has orthonormal columns ($PP^{\mathsf{T}} = I_{d'}$), i.e., $P$ is *semi-orthogonal matrix*. ASO leads to the following formulation:

$$\underset{W, P: PP^{\mathsf{T}} = I_{d'}}{\text{minimize}} \quad \sum_{t=1}^{T} \sum_{i=1}^{n_t} l\left(h_{w_t}\left(\mathbf{x}_t^{(i)}\right), y_t^{(i)}\right) + \alpha\left(\operatorname{tr}\left(WW^{\mathsf{T}}\right) - \operatorname{tr}\left(WP^{\mathsf{T}}PW^{\mathsf{T}}\right)\right) + \beta \operatorname{tr}\left(WW^{\mathsf{T}}\right),$$
(58)

where $\alpha \geq 0$ is the regularization parameter for task relatedness and $\beta \geq 0$ is an additional L2 regularization parameter.

When the hypothesis $(h_\theta)_\theta$ are assumed to be linear, Eq. (5) can be written as $W = \Pi\Theta$. Writing the LQ decomposition[6] of matrix $\Theta$, i.e., $\Theta = LQ$, where $L \in \mathbb{R}^{M \times M}$ is a lower triangular matrix and $Q \in \mathbb{R}^{M \times d}$ is a semi-orthogonal matrix ($QQ^{\mathsf{T}} = I_M$), (5) becomes $W = \Pi LQ \in \mathbb{R}^{T \times d}$, thus, $W = WQ^{\mathsf{T}}Q$, leading to the constraint $\|W - WQ^{\mathsf{T}}Q\|_F^2 = \operatorname{tr}\left(WW^{\mathsf{T}}\right) - \operatorname{tr}\left(WQ^{\mathsf{T}}QW^{\mathsf{T}}\right) = 0$. If we assume $\|\theta_m\|_2^2$ to be bounded by a constant $B > 0$ for all $m \in [M]$, we get the constraint $\operatorname{tr}\left(WW^{\mathsf{T}}\right) \leq TB$. It means that minimizing $\sum_{t=1}^{T} \sum_{i=1}^{n_t} l\left(h_{w_t}\left(\mathbf{x}_t^{(i)}\right), y_t^{(i)}\right)$ under our Assumption 1 can be formulated as the following constrained optimization problem

$$
\begin{aligned}
\underset{W, Q: QQ^{\mathsf{T}} = I_M}{\text{minimize}} \quad & \sum_{t=1}^{T} \sum_{i=1}^{n_t} l\left(h_{w_t}\left(\mathbf{x}_t^{(i)}\right), y_t^{(i)}\right), \\
\text{subject to} \quad & \operatorname{tr}\left\{WW^{\mathsf{T}}\right\} - \operatorname{tr}\left\{WQ^{\mathsf{T}}QW^{\mathsf{T}}\right\} = 0, \\
& \operatorname{tr}\left(WW^{\mathsf{T}}\right) \leq TB.
\end{aligned}
$$
(59)

Thus, there exists Lagrange multipliers $\alpha \in \mathbb{R}$ and $\beta > 0$, for which Problem (59) is equivalent to the following regularized optimization problem

$$\underset{W, Q: QQ^{\mathsf{T}} = I_M}{\text{minimize}} \sum_{t=1}^{T} \sum_{i=1}^{n_t} l\left(h_{w_t}\left(\mathbf{x}_t^{(i)}\right), y_t^{(i)}\right) + \alpha\left(\operatorname{tr}\left\{WW^{\mathsf{T}}\right\} - \operatorname{tr}\left\{WQ^{\mathsf{T}}QW^{\mathsf{T}}\right\}\right) + \beta \operatorname{tr}\left\{WW^{\mathsf{T}}\right\},$$
(60)

which is exactly Problem (58).

**Federated MTL via task relationships.**   The ASO formulation above motivated the authors of [59] to learn personalized models by solving the following problem

$$\min_{W, \Omega} \sum_{t=1}^{T} \sum_{i=1}^{n_t} l\left(h_{w_t}\left(\mathbf{x}_t^{(i)}\right), y_t^{(i)}\right) + \lambda \operatorname{tr}\left(W\Omega W^{\mathsf{T}}\right),$$
(61)

Two alternative MTL formulations are presented in [59] to justify Problem (61): MTL with probabilistic priors [69] and MTL with graphical models [35]. Both of them can be covered using our Assumption 1 as follows:

- Considering $T = M$ and $\Pi = I_M$ in Assumption 1 and introducing a prior on $\Theta$ of the form

$$\Theta \sim \left(\prod \mathcal{N}\left(0, \sigma^2 I_d\right)\right) \mathcal{MN}\left(I_d \otimes \Omega\right)$$
(62)

  lead to a formulation similar to MTL with probabilistic priors [69].

---

[6]Note that when $\Theta$ is a full rank matrix, this decomposition is unique.

- Two tasks $t$ and $t'$ are independent if $\langle \pi_t, \pi_{t'} \rangle = 0$, thus using $\Omega_{t,t'} = \langle \pi_t, \pi_{t'} \rangle$ leads to the same graphical model as in [35].

Several personalized FL formulations, e.g., `pFedMe`[16], `FedU` [17] and the formulation studied in [24] and in [23], are special cases of formulation (62).

## C  Centralized Expectation Maximization

**Proposition 3.1.** *Under Assumptions 1 and 2, at the $k$-th iteration the EM algorithm updates parameter estimates through the following steps:*

**E-step:**  $$q_t^{k+1}(z_t^{(i)} = m) \propto \pi_{tm}^k \cdot \exp\left(-l(h_{\theta_m^k}(\mathbf{x}_t^{(i)}), y_t^{(i)})\right), \quad t \in [T], \; m \in [M], \; i \in [n_t] \quad (8)$$

**M-step:**  $$\pi_{tm}^{k+1} = \frac{\sum_{i=1}^{n_t} q_t^{k+1}(z_t^{(i)} = m)}{n_t}, \qquad\qquad t \in [T], \; m \in [M] \quad (9)$$

$$\theta_m^{k+1} \in \arg\min_{\theta \in \mathbb{R}^d} \sum_{t=1}^{T} \sum_{i=1}^{n_t} q_t^{k+1}(z_t^{(i)} = m) l(h_\theta(\mathbf{x}_t^{(i)}), y_t^{(i)}), \qquad m \in [M] \quad (10)$$

*Proof.* The objective is to learn parameters $\{\breve{\Theta}, \breve{\Pi}\}$ from the data $\mathcal{S}_{1:T}$ by maximizing the likelihood $p(\mathcal{S}_{1:T}|\Theta, \Pi)$. We introduce functions $q_t(z)$, $t \in [T]$ such that $q_t \geq 0$ and $\sum_{z=1}^{M} q_t(z) = 1$ in the expression of the likelihood. For $\Theta \in \mathbb{R}^{M \times d}$ and $\Pi \in \Delta^{T \times M}$, we have

$$\log p(\mathcal{S}_{1:T}|\Theta, \Pi) = \sum_{t=1}^{T} \sum_{i=1}^{n_t} \log p_t\left(s_t^{(i)}|\Theta, \pi_t\right) \tag{63}$$

$$= \sum_{t=1}^{T} \sum_{i=1}^{n_t} \log\left[\sum_{m=1}^{M} \left(\frac{p_t\left(s_t^{(i)}, z_t^{(i)} = m|\Theta, \pi_t\right)}{q_t\left(z_t^{(i)} = m\right)}\right) q_t\left(z_t^{(i)} = m\right)\right] \tag{64}$$

$$\geq \sum_{t=1}^{T} \sum_{i=1}^{n_t} \sum_{m=1}^{M} q_t\left(z_t^{(i)} = m\right) \log \frac{p_t\left(s_t^{(i)}, z_t^{(i)} = m|\Theta, \pi_t\right)}{q_t\left(z_t^{(i)} = m\right)} \tag{65}$$

$$= \sum_{t=1}^{T} \sum_{i=1}^{n_t} \sum_{m=1}^{M} q_t\left(z_t^{(i)} = m\right) \log p_t\left(s_t^{(i)}, z_t^{(i)} = m|\Theta, \pi_t\right)$$

$$- \sum_{t=1}^{T} \sum_{i=1}^{n_t} \sum_{m=1}^{M} q_t\left(z_t^{(i)} = m\right) \log q_t\left(z_t^{(i)} = m\right) \tag{66}$$

$$\triangleq \mathfrak{L}(\Theta, \Pi, Q_{1:T}), \tag{67}$$

where we used Jensen's inequality because log is concave. $\mathfrak{L}(\Theta, \Pi, Q_{1:T})$ is an *evidence lower bound*. The centralized EM-algorithm corresponds to iteratively maximizing this bound with respect to $Q_{1:T}$ (E-step) and with respect to $\{\Theta, \Pi\}$ (M-step).

**E-step.**  The difference between the log-likelihood and the evidence lower bound $\mathfrak{L}(\Theta, \Pi, Q_{1:T})$ can be expressed in terms of a sum of $\mathcal{KL}$ divergences:

$$\log p(\mathcal{S}_{1:T}|\Theta, \Pi) - \mathfrak{L}(\Theta, \Pi, Q_{1:T}) =$$

$$= \sum_{t=1}^{T} \sum_{i=1}^{n_t} \left\{ \log p_t\left(s_t^{(i)}|\Theta, \pi_t\right) - \sum_{m=1}^{M} q_t\left(z_t^{(i)} = m\right) \log \frac{p_t\left(s_t^{(i)}, z_t^{(i)} = m|\Theta, \pi_t\right)}{q_t\left(z_t^{(i)} = m\right)} \right\} \tag{68}$$

$$= \sum_{t=1}^{T} \sum_{t=1}^{n_t} \sum_{m=1}^{M} q_t\left(z_t^{(i)} = m\right) \left( \log p_t\left(s_t^{(i)}|\Theta, \pi_t\right) - \log \frac{p_t\left(s_t^{(i)}, z_t^{(i)} = m|\Theta, \pi_t\right)}{q_t\left(z_t^{(i)} = m\right)} \right) \tag{69}$$

$$= \sum_{t=1}^{T} \sum_{t=1}^{n_t} \sum_{m=1}^{M} q_t\left(z_t^{(i)} = m\right) \log \frac{p_t\left(s_t^{(i)}|\Theta, \pi_t\right) \cdot q_t\left(z_t^{(i)} = m\right)}{p_t\left(s_t^{(i)}, z_t^{(i)} = m|\Theta, \pi_t\right)} \tag{70}$$

$$= \sum_{t=1}^{T} \sum_{t=1}^{n_t} \sum_{m=1}^{M} q_t\left(z_t^{(i)} = m\right) \log \frac{q_t\left(z_t^{(i)} = m\right)}{p_t\left(z_t^{(i)} = m|s_t^{(i)}, \Theta, \pi_t\right)} \tag{71}$$

$$= \sum_{t=1}^{T} \sum_{i=1}^{n_t} \mathcal{KL} \left( q_t \left( z_t^{(i)} \right) \| p_t \left( z_t^{(i)} | s_t^{(i)}, \Theta, \pi_t \right) \right) \geq 0. \tag{72}$$

For fixed parameters $\{\Theta, \Pi\}$, the maximum of $\mathfrak{L}(\Theta, \Pi, Q_{1:T})$ is reached when

$$\sum_{t=1}^{T} \sum_{i=1}^{n_t} \mathcal{KL} \left( q_t \left( z_t^{(i)} \right) \| p_t \left( z_t^{(i)} | s_t^{(i)}, \Theta, \pi_t \right) \right) = 0.$$

Thus for $t \in [T]$ and $i \in [n_t]$, we have:

$$q_t(z_t^{(i)} = m) = p_t(z_t^{(i)} = m | s_t^{(i)}, \Theta, \pi_t) \tag{73}$$

$$= \frac{p_t(s_t^{(i)} | z_t^{(i)} = m, \Theta, \pi_t) \times p_t(z_t^{(i)} = m | \Theta, \pi_t)}{p_t \left( s_t^{(i)} | \Theta, \pi_t \right)} \tag{74}$$

$$= \frac{p_m(s_t^{(i)} | \theta_m) \times \pi_{tm}}{\sum_{m'=1}^{M} p_{m'}(s_t^{(i)}) \times \pi_{tm'}} \tag{75}$$

$$= \frac{p_m \left( y_t^{(i)} | \mathbf{x}_t^{(i)}, \theta_m \right) \times p_m \left( \mathbf{x}_t^{(i)} \right) \times \pi_{tm}}{\sum_{m'=1}^{M} p_{m'} \left( y_t^{(i)} | \mathbf{x}_t^{(i)}, \theta_{m'} \right) \times p_{m'} \left( \mathbf{x}_t^{(i)} \right) \times \pi_{tm'}} \tag{76}$$

$$= \frac{p_m \left( y_t^{(i)} | \mathbf{x}_t^{(i)}, \theta_m \right) \times p \left( \mathbf{x}_t^{(i)} \right) \times \pi_{tm}}{\sum_{m'=1}^{M} p_{m'} \left( y_t^{(i)} | \mathbf{x}_t^{(i)}, , \theta_{m'} \right) \times p \left( \mathbf{x}_t^{(i)} \right) \times \pi_{tm'}}, \tag{77}$$

where (77) relies on Assumption 2. It follows that

$$q_t(z_t^{(i)} = m) = p_t(z_t^{(i)} = m | s_t^{(i)}, \Theta, \pi_t) = \frac{p_m \left( y_t^{(i)} | \mathbf{x}_t^{(i)}, \theta_m \right) \times \pi_{tm}}{\sum_{m'=1}^{M} p_{m'} \left( y_t^{(i)} | \mathbf{x}_t^{(i)}, \theta_{m'} \right) \times \pi_{tm'}}. \tag{78}$$

**M-step.** We maximize now $\mathfrak{L}(\Theta, \Pi, Q_{1:T})$ with respect to $\{\Theta, \Pi\}$. By dropping the terms not depending on $\{\Theta, \Pi\}$ in the expression of $\mathfrak{L}(\Theta, \Pi, Q_{1:T})$ we write:

$$\mathfrak{L}(\Theta, \Pi, Q_{1:T})$$

$$= \sum_{t=1}^{T} \sum_{i=1}^{n_t} \sum_{m=1}^{M} q_t \left( z_t^{(i)} = m \right) \log p_t \left( s_t^{(i)}, z_t^{(i)} = m | \Theta, \pi_t \right) + c \tag{79}$$

$$= \sum_{t=1}^{T} \sum_{i=1}^{n_t} \sum_{m=1}^{M} q_t \left( z_t^{(i)} = m \right) \left[ \log p_t \left( s_t^{(i)} | z_t^{(i)} = m, \Theta, \pi_t \right) + \log p_t \left( z_t^{(i)} = m | \Theta, \pi_t \right) \right] + c \tag{80}$$

$$= \sum_{t=1}^{T} \sum_{i=1}^{n_t} \sum_{m=1}^{M} q_t \left( z_t^{(i)} = m \right) \left[ \log p_{\theta_m} \left( s_t^{(i)} \right) + \log \pi_{tm} \right] + c \tag{81}$$

$$= \sum_{t=1}^{T} \sum_{i=1}^{n_t} \sum_{m=1}^{M} q_t \left( z_t^{(i)} = m \right) \left[ \log p_{\theta_m} \left( y_t^{(i)} | \mathbf{x}_t^{(i)} \right) + \log p_m \left( \mathbf{x}_t^{(i)} \right) + \log \pi_{tm} \right] + c \tag{82}$$

$$= \sum_{t=1}^{T} \sum_{i=1}^{n_t} \sum_{m=1}^{M} q_t \left( z_t^{(i)} = m \right) \left[ \log p_{\theta_m} \left( y_t^{(i)} | \mathbf{x}_t^{(i)} \right) + \log \pi_{tm} \right] + c', \tag{83}$$

$$\tag{84}$$

where $c$ and $c'$ are constant not depending on $\{\Theta, \Pi\}$.

Thus, for $t \in [T]$ and $m \in [M]$, by solving a simple optimization problem we update $\pi_{tm}$ as follows:

$$\pi_{tm} = \frac{\sum_{i=1}^{n_t} q_t(z_t^{(i)} = m)}{n_t}. \tag{85}$$

On the other hand, for $m \in [M]$, we update $\theta_m$ by solving:

$$\theta_m \in \arg\min_{\theta \in \mathbb{R}^d} \sum_{t=1}^{T} \sum_{i=1}^{n_t} q_t(z_t^{(i)} = m) \times l\left(h_\theta(\mathbf{x}_t^{(i)}), y_t^{(i)}\right). \tag{86}$$

$\square$

# D Detailed Algorithms

## D.1 Client-Server Algorithm

Alg. 2 is a detailed version of Alg. 1 (FedEM), with local SGD used as local solver.

Alg. 3 gives our general algorithm for federated surrogate optimization, from which Alg. 2 is derived.

---

**Algorithm 2:** FedEM: Federated Expectation-Maximization

**Input** : Data $\mathcal{S}_{1:T}$; number of mixture components $M$; number of communication rounds $K$; number of local steps $J$

**Output** : $\theta_m^K$ for $1 \in [M]$; $\pi_t^K$ for $t \in [T]$

// Initialization

1 **server** randomly initialize $\theta_m^0 \in \mathbb{R}^d$ for $1 \le m \le M$;

2 **for** tasks $t = 1, \ldots, T$ **in parallel over** $T$ **clients do**

3 $\quad$ Randomly initialize $\pi_t^0 \in \Delta^M$;

// Main loop

4 **for** iterations $k = 1, \ldots, K$ **do**

5 $\quad$ server broadcasts $\theta_m^{k-1}$, $1 \le m \le M$ **to the** $T$ **clients**;

6 $\quad$ **for** tasks $t = 1, \ldots, T$ **in parallel over** $T$ **clients do**

7 $\quad\quad$ **for** component $m = 1, \ldots, M$ **do**

$\quad\quad\quad$ // E-step

8 $\quad\quad\quad$ **for** sample $i = 1, \ldots, n_t$ **do**

9 $\quad\quad\quad\quad q_t^k\left(z_t^{(i)} = m\right) \leftarrow \dfrac{\pi_{tm}^k \cdot \exp\left(-l(h_{\theta_m^k}(\mathbf{x}_t^{(i)}), y_t^{(i)})\right)}{\sum_{m'=1}^M \pi_{tm'}^k \cdot \exp\left(-l(h_{\theta_{m'}^k}(\mathbf{x}_t^{(i)}), y_t^{(i)})\right)}$ ;

$\quad\quad\quad$ // M-step

10 $\quad\quad\quad \pi_{tm}^k \leftarrow \dfrac{\sum_{i=1}^{n_t} q_t^k(z_t^{(i)}=m)}{n_t}$ ;

11 $\quad\quad\quad \theta_{m,t}^k \leftarrow \texttt{LocalSolver}(J, m, \theta_m^{k-1}, q_t^k, \mathcal{S}_t)$ ;

12 $\quad\quad$ **client** $t$ **sends** $\theta_{m,t}^k$, $1 \le m \le M$ **to the server**;

13 $\quad$ **for** component $m = 1, \ldots, M$ **do**

14 $\quad\quad \theta_m^k \leftarrow \sum_{t=1}^T \frac{n_t}{n} \cdot \theta_{m,t}^k$;

15 **Function** $\texttt{LocalSolver}(J, m, \theta, q, \mathcal{S})$**:**

16 $\quad$ **for** $j = 0, \ldots, J-1$ **do**

17 $\quad\quad$ Sample indexes $\mathcal{I}$ uniformly from $1, \ldots, |\mathcal{S}|$;

18 $\quad\quad \theta \leftarrow \theta - \eta_{k-1,j} \sum_{i \in \mathcal{I}} q(z^{(i)} = m) \cdot \nabla_\theta l\left(h_\theta\left(\mathbf{x}^{(i)}\right), y^{(i)}\right)$;

19 $\quad$ **return** $\theta$;

---

---

**Algorithm 3:** Federated Surrogate Optimization

---

**Input** : $\mathbf{u}^0 \in \mathbb{R}^{d_u}$; $\mathbf{V}^0 = \left(\mathbf{v}_t^0\right)_{1 \leq t \leq T} \in \mathcal{V}^T$; number of iterations $K$; number of local steps $J$

**Output :** $\mathbf{u}^K$; $\mathbf{v}_t^K$

**1 for iterations** $k = 1, \ldots, K$ **do**

2      **server broadcasts** $\mathbf{u}^{k-1}$ **to the** $T$ **clients**;

3      **for tasks** $t = 1, \ldots, T$ **in parallel over** $T$ **clients do**

4          Compute partial first-order surrogate function $g_t^k$ of $f_t$ near $\left\{\mathbf{u}^{k-1}, \mathbf{v}_t^{k-1}\right\}$;

5          $\mathbf{v}_t^k \leftarrow \underset{\mathbf{v} \in \mathcal{V}}{\arg\min}\, g_t^k\left(\mathbf{u}^{k-1}, \mathbf{v}\right)$;

6          $u_t^k \leftarrow \texttt{LocalSolver}(J, \mathbf{u}_t^{k-1}, \mathbf{v}_t^{k-1}, g_t^k, \mathcal{S}_t)$;

7          **client** $t$ **sends** $\mathbf{u}_t^k$ **to the server**;

8      $\mathbf{u}^k \leftarrow \sum_{t=1}^{T} \omega_t \cdot \mathbf{u}_t^k$;

9 **Function** $\texttt{LocalSolver}(J, \mathbf{u}, \mathbf{v}, g, \mathcal{S})$**:**

10      **for** $j = 0, \ldots, J-1$ **do**

11          sample $\xi^{k-1,j}$ from $\mathcal{S}$;

12          $\mathbf{u} \leftarrow \mathbf{u} - \eta_{k-1,j} \cdot \nabla_{\mathbf{u}} g(\mathbf{u}, \mathbf{v}; \xi^{k-1,j})$;

13      **return** $\Theta$;

---

### D.2 Fully Decentralized Algorithm

Alg. 4 shows D-FedEM, the fully decentralization version of our federated expectation maximization algorithm.

Alg. 5 gives our general fully decentralized algorithm for federated surrogate optimization, from which Alg. 4 is derived.

---

**Algorithm 4:** D-FedEM: Fully Decentralized Federated Expectation-Maximization

**Input** : Data $\mathcal{S}_{1:T}$; number of mixture components $M$; number of iterations $K$; number of local steps $J$; mixing matrix distributions $\mathcal{W}^k$ for $k \in [K]$
**Output:** $\theta_{m,t}^K$ for $m \in [M]$ and $t \in [T]$; $\pi_t$ for $t \in [T]$

// Initialization

1 **for** tasks $t = 1, \ldots, T$ **in parallel over** $T$ **clients do**
2      Randomly initialize $\Theta_t = (\theta_{m,t})_{1 \le m \le M} \in \mathbb{R}^{M \times d}$ ;
3      Randomly initialize $\pi_t^0 \in \Delta^M$;

// Main loop

4 **for** iterations $k = 1, \ldots, K$ **do**
     // Select the communication topology and the aggregation weights
5      Sample $W^{k-1} \sim \mathcal{W}^{k-1}$;
6      **for** tasks $t = 1, \ldots, T$ **in parallel over** $T$ **clients do**
7          **for** component $m = 1, \ldots, M$ **do**
             // E-step
8              **for** sample $i = 1, \ldots, n_t$ **do**
9                  $q_t^k\left(z_t^{(i)} = m\right) \leftarrow \dfrac{\pi_{tm}^k \cdot \exp\left(-l(h_{\theta_m^k}(\mathbf{x}_t^{(i)}), y_t^{(i)})\right)}{\sum_{m'=1}^M \pi_{tm'}^k \cdot \exp\left(-l(h_{\theta_{m'}^k}(\mathbf{x}_t^{(i)}), y_t^{(i)})\right)}$;
             // M-step
10              $\pi_{tm}^k \leftarrow \dfrac{\sum_{i=1}^{n_t} q_t^k(z_t^{(i)} = m)}{n_t}$ ;
11              $\theta_{m,t}^{k-\frac{1}{2}} \leftarrow \texttt{LocalSolver}(J, m, \theta_{m,t}^{k-1}, q_t^k, \mathcal{S}_t, t)$;
12          Send $\theta_{m,t}^{k-\frac{1}{2}}$, $1 \le m \le M$ to neighbors;
13          Receive $\theta_{m,s}^{k-\frac{1}{2}}$, $1 \le m \le M$ from neighbors;
14          **for** component $m = 1, \ldots, M$ **do**
15              $\theta_{m,t}^k \leftarrow \sum_{s=1}^T w_{s,t}^{k-1} \cdot \theta_{m,s}^{k-\frac{1}{2}}$;

16 **Function** $\texttt{LocalSolver}(J, m, \theta, q, \mathcal{S}, t)$**:**
17      **for** $j = 0, \ldots, J - 1$ **do**
18          Sample indexes $\mathcal{I}$ uniformly from $1, \ldots, |\mathcal{S}|$;
19          $\theta \leftarrow \theta - \frac{n_t}{n} \cdot \eta_{k-1,j} \sum_{i \in \mathcal{I}} q(z^{(i)} = m) \cdot \nabla_\theta l\left(h_\theta\left(\mathbf{x}^{(i)}\right), y^{(i)}\right)$;
20      **return** $\theta$;

---

---

**Algorithm 5:** Fully-Decentralized Federated Surrogate Optimization
___

**Input** : $\mathbf{u}^0 \in \mathbb{R}^{d_u}$; $\mathbf{V}^0 = \left(\mathbf{v}_t^0\right)_{1 \le t \le T} \in \mathcal{V}^T$; number of iterations $K$; number of local step $J$;
mixing matrix distributions $\mathcal{W}^k$ for $k \in [K]$

**Output :** $\mathbf{u}_t^K$ for $t \in [T]$; $\mathbf{v}_t^K$ for $t \in [T]$

**1 for iterations** $k = 1, \ldots, K$ **do**

     // Select the communication topology and the aggregation weights

**2**      Sample $W^{k-1} \sim \mathcal{W}^{k-1}$;

**3**      **for tasks** $t = 1, \ldots, T$ **in parallel over** $T$ **clients do**

**4**          compute partial first-order surrogate function $g_t^k$ of $f_t$ near $\left\{\mathbf{u}_t^{k-1}, \mathbf{v}_t^{k-1}\right\}$;

**5**          $\mathbf{v}_t^k \leftarrow \underset{v \in \mathcal{V}}{\arg\min}\, g_t^k\left(\mathbf{u}_t^{k-1}, \mathbf{v}\right)$;

**6**          $\mathbf{u}_t^{k-\frac{1}{2}} \leftarrow \texttt{LocalSolver}(J, \mathbf{u}_t^{k-1}, \mathbf{v}_t^{k-1}, g_t^k, t)$;

**7**          Send $\mathbf{u}_t^{k-\frac{1}{2}}$ to neighbors;

**8**          Receive $\mathbf{u}_s^{k-\frac{1}{2}}$ from neighbors;

**9**          $\mathbf{u}_t^k \leftarrow \sum_{s=1}^T w_{ts}^{k-1} \times \mathbf{u}_s^{k-\frac{1}{2}}$;

**10 Function** $\texttt{LocalSolver}(J, \mathbf{u}, \mathbf{v}, g, \mathcal{S}, t)$**:**

**11**      **for** $j = 0, \ldots, J-1$ **do**

**12**          sample $\xi^{k-1,j}$ from $\mathcal{S}$ ;

**13**          $\mathbf{u} \leftarrow \mathbf{u} - \omega_t \cdot \eta_{k-1,j} \nabla_{\mathbf{u}} g(\mathbf{u}, \mathbf{v}, \xi^{k-1,j})$;

**14**      **return** $\mathbf{u}$;

___

# E   Details on the Fully Decentralized Setting

As mentioned in Section 3.3, the convergence of decentralized optimization schemes requires certain assumptions on the sequence of mixing matrices $(W^k)_{k>0}$, to guarantee that each client can influence the estimates of other clients over time. In our paper, we consider the following general assumption.

**Assumption 8** ([31, Assumption 4]). *Symmetric doubly stochastic mixing matrices are drawn at each round k from (potentially different) distributions $W^k \sim \mathcal{W}^k$ and there exists two constants $p \in (0, 1]$, and integer $\tau \geq 1$ such that for all $\Xi \in \mathbb{R}^{M \times d \times T}$ and all integers $l \in \{0, \ldots, K/\tau\}$:*

$$\mathbb{E} \left\| \Xi W_{l,\tau} - \bar{\Xi} \right\|_{\mathcal{F}}^2 \leq (1-p) \left\| \Xi - \bar{\Xi} \right\|_{\mathcal{F}}^2, \tag{87}$$

*where $W_{l,\tau} \triangleq W^{(l+1)\tau-1} \ldots W^{l\tau}$, $\bar{\Xi} \triangleq \Xi \frac{\mathbf{1}\mathbf{1}^{\mathsf{T}}}{T}$, and the expectation is taken over the random distributions $W^k \sim \mathcal{W}^k$.*

Assumption 8 expresses the fact that the sequence of mixing matrices, on average and every $\tau$ communication rounds, brings the values in the columns of $\Xi$ closer to their row-wise average (thereby mixing the clients' updates over time). For instance, the assumption is satisfied if the communication graph is strongly connected every $\tau$ rounds, i.e., the graph $([T], \mathcal{E})$, where the edge $(i, j)$ belongs to the graph if $w_{i,j}^h > 0$ for some $h \in \{k+1, \ldots, k+\tau\}$ is connected.

We provide below the rigorous statement of Theorem 3.3, which was informally presented in Section 3.3. It shows that `D-FedEM` converges to a consensus stationary point of $f$ (proof in App. G.2).

**Theorem 3.3.** *Under Assumptions 1–8, when clients use SGD as local solver with learning rate $\eta = \frac{a_0}{\sqrt{K}}$, `D-FedEM`'s iterates satisfy the following inequalities after a large enough number of communication rounds $K$:*

$$\frac{1}{K} \sum_{k=1}^{K} \mathbb{E} \left\| \nabla_\Theta f \left( \bar{\Theta}^k, \Pi^k \right) \right\|_F^2 \leq \mathcal{O} \left( \frac{1}{\sqrt{K}} \right), \quad \frac{1}{K} \sum_{k=1}^{K} \sum_{t=1}^{T} \frac{n_t}{n} \mathcal{KL} \left( \pi_t^k, \pi_t^{k-1} \right) \leq \mathcal{O} \left( \frac{1}{K} \right), \tag{88}$$

*where $\bar{\Theta}^k = \left[ \Theta_1^k, \ldots \Theta_T^k \right] \cdot \frac{\mathbf{1}\mathbf{1}^{\mathsf{T}}}{T}$. Moreover, individual estimates $\left( \Theta_t^k \right)_{1 \leq t \leq T}$ converge to consensus, i.e., to $\bar{\Theta}^k$:*

$$\min_{k \in [K]} \mathbb{E} \sum_{t=1}^{T} \left\| \Theta_t^k - \bar{\Theta}^k \right\|_F^2 \leq \mathcal{O} \left( \frac{1}{\sqrt{K}} \right).$$

# F  Federated Surrogate Optimization

In this appendix, we give more details on the federated surrogate optimization framework introduced in Section 3.4. In particular, we provide the assumptions under which Alg. 3 and Alg. 5 converge. We also illustrate how our framework can be used to study existing algorithms.

## F.1  Reminder on Basic (Centralized) Surrogate Optimization

In this appendix, we recall the (centralized) *first-order surrogate optimization* framework introduced in [43]. In this framework, given a continuous function $f : \mathbb{R}^d \mapsto \mathbb{R}$, we are interested in solving

$$\min_{\theta \in \mathbb{R}^d} f(\theta)$$

using the majoration-minimization scheme presented in Alg. 6.

---
**Algorithm 6:** Basic Surrogate Optimization

**Input**  : $\theta^0 \in \mathbb{R}^d$; number of iterations $K$;
**Output :** $\theta^K$
1 **for iterations** $k = 1, \dots, K$ **do**
2 $\quad$ Compute $g^k$, a surrogate function of $f$ near $\theta^{k-1}$;
3 $\quad$ Update solution: $\theta^k \in \arg\min_\theta g^k(\theta)$;

---

This procedure relies on surrogate functions, that approximate well the objective function in a neighborhood of a point. Reference [43] focuses on *first-order surrogate functions* defined below.

**Definition F.1** (First-Order Surrogate [43]). A function $g : \mathbb{R}^d \mapsto \mathbb{R}$ is a first order surrogate of $f$ near $\theta^k \in \mathbb{R}^d$ when the following is satisfied:

- **Majorization:** we have $g(\theta') \geq f(\theta')$ for all $\theta' \in \arg\min_{\theta \in \mathbb{R}^d} g(\theta)$. When the more general condition $g \geq f$ holds, we say that $g$ is a **majorant** function.

- **Smoothness:** the approximation error $r \triangleq g - f$ is differentiable, and its gradient is $L$-Lipschitz. Moreover, we have $r(\theta^k) = 0$ and $\nabla r(\theta^k) = 0$.

## F.2  Novel Federated Version

As discussed in Section 3.4, our novel federated surrogate optimization framework minimizes an objective function $(\mathbf{u}, \mathbf{v}_{1:T}) \mapsto f(\mathbf{u}, \mathbf{v}_{1:T})$ that can be written as a weighted sum $f(\mathbf{u}, \mathbf{v}_{1:T}) = \sum_{t=1}^T \omega_t f_t(\mathbf{u}, \mathbf{v}_t)$ of $T$ functions. We suppose that each client $t \in [T]$ can compute a partial first order surrogate of $f_t$, defined as follows.

**Definition 1** (Partial first-order surrogate). A function $g(\mathbf{u}, \mathbf{v}) : \mathbb{R}^{d_u} \times \mathcal{V} \to \mathbb{R}$ is a partial first-order surrogate of $f(\mathbf{u}, \mathbf{v})$ wrt $\mathbf{u}$ near $(\mathbf{u}_0, \mathbf{v}_0) \in R^{d_u} \times \mathcal{V}$ when the following conditions are satisfied:

1. $g(\mathbf{u}, \mathbf{v}) \geq f(\mathbf{u}, \mathbf{v})$ for all $\mathbf{u} \in \mathbb{R}^{d_u}$ and $\mathbf{v} \in \mathcal{V}$;
2. $r(\mathbf{u}, \mathbf{v}) \triangleq g(\mathbf{u}, \mathbf{v}) - f(\mathbf{u}, \mathbf{v})$ is differentiable and $L$-smooth with respect to $\mathbf{u}$. Moreover, we have $r(\mathbf{u}_0, \mathbf{v}_0) = 0$ and $\nabla_{\mathbf{u}} r(\mathbf{u}_0, \mathbf{v}_0) = 0$.
3. $g(\mathbf{u}, \mathbf{v}_0) - g(\mathbf{u}, \mathbf{v}) = d_{\mathcal{V}}(\mathbf{v}_0, \mathbf{v})$ for all $\mathbf{u} \in \mathbb{R}^{d_u}$ and $\mathbf{v} \in \arg\min_{\mathbf{v}' \in \mathcal{V}} g(\mathbf{u}, \mathbf{v}')$, where $d_{\mathcal{V}}$ is non-negative and $d_{\mathcal{V}}(v, v') = 0 \iff v = v'$.

Under the assumption that each client $t$ can compute a partial first order surrogate of $f_t$, we propose algorithms for federated surrogate optimization in both the client-server setting (Alg. 3) and the fully decentralized one (Alg. 5). Both algorithms are iterative and distributed: at each iteration $k > 0$, client $t \in [T]$ computes a partial first-order surrogate $g_t^k$ of $f_t$ near $\{u^{k-1}, v_t^{k-1}\}$ (resp. $\{u_t^{k-1}, v_t^{k-1}\}$) for federated surrogate optimization in Alg. 3 (resp. for fully decentralized surrogate optimization in Alg 5).

The convergence of those two algorithms requires the following standard assumptions. Each of them generalizes one of the Assumptions 4–7 for our EM algorithms.

**Assumption 4'.** *The objective function $f$ is bounded below by $f^* \in \mathbb{R}$.*

**Assumption 5'.** *(Smoothness) For all $t \in [T]$ and $k > 0$, $g_t^k$ is L-smooth wrt to $\mathbf{u}$.*

**Assumption 6'.** *(Unbiased gradients and bounded variance) Each client $t \in [T]$ can sample a random batch $\xi$ from $\mathcal{S}_t$ and compute an unbiased estimator $\nabla_{\mathbf{u}} g_t^k(\mathbf{u}, \mathbf{v}; \xi)$ of the local gradient with bounded variance, i.e., $\mathbb{E}_\xi[\nabla_{\mathbf{u}} g_t^k(\mathbf{u}, \mathbf{v}; \xi)] = \nabla_{\mathbf{u}} g_t^k(\mathbf{u}, \mathbf{v})$ and $\mathbb{E}_\xi \| \nabla_{\mathbf{u}} g_t^k(\mathbf{u}, \mathbf{v}; \xi) - \nabla_{\mathbf{u}} g_t^k(\mathbf{u}, \mathbf{v}) \|^2 \leq \sigma^2$.*

**Assumption 7'.** *(Bounded dissimilarity) There exist $\beta$ and $G$ such that*

$$\sum_{t=1}^{T} \omega_t \cdot \left\| \nabla_{\mathbf{u}} g_t^k(\mathbf{u}, \mathbf{v}) \right\|^2 \leq G^2 + \beta^2 \left\| \sum_{t=1}^{T} \omega_t \cdot \nabla_{\mathbf{u}} g_t^k(\mathbf{u}, \mathbf{v}) \right\|^2.$$

Under these assumptions a parallel result to Theorem. 3.2 holds for the client-server setting.

**Theorem 3.2'.** *Under Assumptions 4'–7', when clients use SGD as local solver with learning rate $\eta = \frac{a_0}{\sqrt{K}}$, after a large enough number of communication rounds $K$, the iterates of federated surrogate optimization (Alg. 3) satisfy:*

$$\frac{1}{K} \sum_{k=1}^{K} \mathbb{E} \left\| \nabla_{\mathbf{u}} f\left(\mathbf{u}^k, \mathbf{v}_{1:T}^k\right) \right\|_F^2 \leq \mathcal{O}\left(\frac{1}{\sqrt{K}}\right), \qquad \frac{1}{K} \sum_{k=1}^{K} \Delta_{\mathbf{v}} f(\mathbf{u}^k, \mathbf{v}_{1:T}^k) \leq \mathcal{O}\left(\frac{1}{K^{3/4}}\right), \quad (89)$$

*where the expectation is over the random batches samples, and $\Delta_v f(\mathbf{u}^k, \mathbf{v}_{1:T}^k) \triangleq f\left(\mathbf{u}^k, \mathbf{v}_{1:T}^k\right) - f\left(\mathbf{u}^k, \mathbf{v}_{1:T}^{k+1}\right) \geq 0$.*

In the fully decentralized setting, if in addition to Assumptions 4'-7', we suppose that Assumption 8 holds, a parallel result to Theorem. 3.3 holds.

**Theorem 3.3'.** *Under Assumptions 4'–7' and Assumption 8, when clients use SGD as local solver with learning rate $\eta = \frac{a_0}{\sqrt{K}}$, after a large enough number of communication rounds $K$, the iterates of fully decentralized federated surrogate optimization (Alg. 5) satisfy:*

$$\frac{1}{K} \sum_{k=1}^{K} \mathbb{E} \left\| \nabla_{\mathbf{u}} f\left(\bar{\mathbf{u}}^k, v_{1:T}^k\right) \right\|^2 \leq \mathcal{O}\left(\frac{1}{\sqrt{K}}\right), \qquad \frac{1}{K} \sum_{k=1}^{K} \sum_{t=1}^{T} \omega_t \cdot d_{\mathcal{V}}\left(v_t^k, v_t^{k+1}\right) \leq \mathcal{O}\left(\frac{1}{K}\right),$$

$$(90)$$

*where $\bar{\mathbf{u}}^k = \frac{1}{T} \sum_{t=1}^{T} \mathbf{u}_t^k$. Moreover, local estimates $\left(\mathbf{u}_t^k\right)_{1 \leq t \leq T}$ converge to consensus, i.e., to $\bar{\mathbf{u}}^k$:*

$$\frac{1}{K} \sum_{k=1}^{K} \sum_{t=1}^{T} \left\| \mathbf{u}_t^k - \bar{\mathbf{u}}^k \right\|^2 \leq \mathcal{O}\left(\frac{1}{\sqrt{K}}\right).$$

The proofs of Theorem 3.2' and Theorem 3.3' are in Section G.1 and Section G.2, respectively.

### F.3   Illustration: Analyzing `pFedMe` with Federated Surrogate Optimization

In this section, we show that `pFedMe` [16] can be studied through our federated surrogate optimization framework. With reference to the general formulation of `pFedMe` in [16, Eq. (2) and (3)], consider

$$g_t^k(\mathbf{w}) = f_t\left(\theta^{k-1}\right) + \frac{\lambda}{2} \cdot \left\| \theta^{k-1} - \omega \right\|^2, \tag{91}$$

where $\theta^{k-1} = \text{prox}_{\frac{f_t}{\lambda}}\left(\omega^{k-1}\right) \triangleq \arg\min_\theta \left\{ f_t(\theta) + \frac{\lambda}{2} \cdot \left\| \theta - \omega^{k-1} \right\|^2 \right\}$. We can verify that $g_t^k$ is a first-order surrogate of $f_t$ near $\theta^{k-1}$:

1. It is clear that $g_t^k\left(\theta^{k-1}\right) = f_t\left(\theta^{k-1}\right)$.

2. Since $\theta^{k-1} = \text{prox}_{\frac{f_t}{\lambda}}\left(\omega^{k-1}\right)$, using the envelope theorem (assuming that $f_t$ is proper, convex and lower semi-continuous), it follows that $\nabla f_t\left(\omega^{k-1}\right) = \lambda\left(\theta^{k-1} - \omega^{k-1}\right) = \nabla g_k^k\left(\omega^{k-1}\right)$.

Therefore, `pFedMe` can be seen as a particular case of the federated surrogate optimization algorithm (Alg. 3), to which our convergence results apply.

# G Convergence Proofs

We study the client-server setting and the fully decentralized setting in Section G.1 and Section G.2, respectively. In both cases, we first prove the more general result for the federated surrogate optimization introduced in App. F, and then derive the specific result for `FedEM` and `D-FedEM`.

## G.1 Client-Server Setting

### G.1.1 Additional Notations

**Remark 2.** *For convenience and without loss of generality, we suppose in this section that $\omega \in \Delta^T$, i.e., $\forall t \in [T]$, $\omega_t \geq 0$ and $\sum_{t'=1}^{T} \omega_{t'} = 1$.*

At iteration $k > 0$, we use $\mathbf{u}_t^{k-1,j}$ to denote the $j$-th iterate of the local solver at client $t \in [T]$, thus

$$\mathbf{u}_t^{k-1,0} = \mathbf{u}^{k-1}, \tag{92}$$

and

$$\mathbf{u}^k = \sum_{t=1}^{T} \omega_t \cdot \mathbf{u}_t^{k-1,J}. \tag{93}$$

At iteration $k > 0$, the local solver's updates at client $t \in [T]$ can be written as (for $0 \leq j \leq J-1$):

$$\mathbf{u}_t^{k-1,j+1} = \mathbf{u}_t^{k-1,j} - \eta_{k-1,j} \nabla_{\mathbf{u}} g_t^k \left( \mathbf{u}_t^{k-1,j}, \mathbf{v}_t^{k-1}; \xi_t^{k-1,j} \right), \tag{94}$$

where $\xi_t^{k-1,j}$ is the batch drawn at the $j$-th local update of $\mathbf{u}_t^{k-1}$.

We introduce $\eta_{k-1} = \sum_{j=0}^{J-1} \eta_{k-1,j}$, and we define the normalized update of the local solver at client $t \in [T]$ as,

$$\hat{\delta}_t^{k-1} \triangleq -\frac{\mathbf{u}_t^{k-1,J} - \mathbf{u}_t^{k-1,0}}{\eta_{k-1}} = \frac{\sum_{j=0}^{J-1} \eta_{k-1,j} \cdot \nabla_{\mathbf{u}} g_t^k \left( \mathbf{u}_t^{k-1,j}, \mathbf{v}_t^{k-1}; \xi_t^{k-1,j} \right)}{\sum_{j=0}^{J-1} \eta_{k-1,j}}, \tag{95}$$

and also define

$$\delta_t^{k-1} \triangleq \frac{\sum_{j=0}^{J-1} \eta_{k-1,j} \cdot \nabla_{\mathbf{u}} g_t^k \left( \mathbf{u}_t^{k-1,j}, \mathbf{v}_t^{k-1} \right)}{\eta_{k-1}}. \tag{96}$$

With this notation,

$$\mathbf{u}^k - \mathbf{u}^{k-1} = -\eta_{k-1} \cdot \sum_{t=1}^{T} \omega_t \cdot \hat{\delta}_t^{k-1}. \tag{97}$$

Finally, we define $g^k$, $k > 0$ as

$$g^k \left( \mathbf{u}, \mathbf{v}_{1:T} \right) \triangleq \sum_{t=1}^{T} \omega_t \cdot g_t^k \left( \mathbf{u}, \mathbf{v}_t \right). \tag{98}$$

Note that $g^k$ is a convex combination of functions $g_t^k$, $t \in [T]$.

### G.1.2 Proof of Theorem 3.2′

**Lemma G.1.** *Suppose that Assumptions 5′–7′ hold. Then, for $k > 0$, and $(\eta_{k,j})_{0 \leq j \leq J-1}$ such that $\eta_k \triangleq \sum_{j=0}^{J-1} \eta_{k,j} \leq \min\left\{ \frac{1}{2\sqrt{2}L}, \frac{1}{4L\beta} \right\}$, the updates of federated surrogate optimization (Alg 3) verify*

$$\mathbb{E}\left[ \frac{f(\mathbf{u}^k, \mathbf{v}_{1:T}^k) - f(\mathbf{u}^{k-1}, \mathbf{v}_{1:T}^{k-1})}{\eta_{k-1}} \right] \leq$$

$$-\frac{1}{4} \mathbb{E} \left\| \nabla_{\mathbf{u}} f \left( \mathbf{u}^{k-1}, \mathbf{v}_{1:T}^{k-1} \right) \right\|^2 - \frac{1}{\eta_{k-1}} \sum_{t=1}^{T} \omega_t \cdot d_{\mathcal{V}} \left( \mathbf{v}_t^{k-1}, \mathbf{v}_t^k \right)$$

$$+ 2\eta_{k-1}L \left( \sum_{j=0}^{J-1} \frac{\eta_{k-1,j}^2}{\eta_{k-1}} L + 1 \right) \sigma^2 + 4\eta_{k-1}^2 L^2 G^2. \tag{99}$$

*Proof.* This proof uses standard techniques from distributed stochastic optimization. It is inspired by [66, Theorem 1].

For $k > 0$, $g^k$ is $L$-smooth wrt $\mathbf{u}$, because it is a convex combination of $L$-smooth functions $g_t^k$, $t \in [T]$. Thus, we write

$$g^k \left( \mathbf{u}^k, \mathbf{v}_{1:T}^{k-1} \right) - g^k \left( \mathbf{u}^{k-1}, \mathbf{v}_{1:T}^{k-1} \right) \leq \left\langle \mathbf{u}^k - \mathbf{u}^{k-1}, \nabla_{\mathbf{u}} g^k(\mathbf{u}^{k-1}, \mathbf{v}_{1:T}^{k-1}) \right\rangle + \frac{L}{2} \left\| \mathbf{u}^k - \mathbf{u}^{k-1} \right\|^2, \tag{100}$$

where $< \mathbf{u}, \mathbf{u}' >$ denotes the scalar product of vectors $\mathbf{u}$ and $\mathbf{u}'$. Using Eq. (97), and taking the expectation over random batches $\left( \xi_t^{k-1,j} \right)_{\substack{0 \leq j \leq J-1 \\ 1 \leq t \leq T}}$, we have

$$\mathbb{E} \left[ g^k \left( \mathbf{u}^k, \mathbf{v}_{1:T}^{k-1} \right) - g^k \left( \mathbf{u}^{k-1}, \mathbf{v}_{1:T}^{k-1} \right) \right] \leq$$
$$- \eta_{k-1} \underbrace{\mathbb{E} \left\langle \sum_{t=1}^T \omega_t \cdot \hat{\delta}_t^{k-1}, \nabla_{\mathbf{u}} g^k(\mathbf{u}^{k-1}, \mathbf{v}_{1:T}^{k-1}) \right\rangle}_{\triangleq T_1} + \frac{L\eta_{k-1}^2}{2} \cdot \underbrace{\mathbb{E} \left\| \sum_{t=1}^T \omega_t \cdot \hat{\delta}_t^{k-1} \right\|^2}_{\triangleq T_2}. \tag{101}$$

We bound each of those terms separately. For $T_1$ we have

$$T_1 = \mathbb{E} \left\langle \sum_{t=1}^T \omega_t \cdot \hat{\delta}_t^{k-1}, \nabla_{\mathbf{u}} g^k \left( \mathbf{u}^{k-1}, \mathbf{v}_{1:T}^{k-1} \right) \right\rangle \tag{102}$$

$$= \mathbb{E} \left\langle \sum_{t=1}^T \omega_t \cdot \left( \hat{\delta}_t^{k-1} - \delta_t^{k-1} \right), \nabla_{\mathbf{u}} g^k \left( \mathbf{u}^{k-1}, \mathbf{v}_{1:T}^{k-1} \right) \right\rangle$$
$$+ \mathbb{E} \left\langle \sum_{t=1}^T \omega_t \cdot \delta_t^{k-1}, \nabla_{\mathbf{u}} g^k \left( \mathbf{u}^{k-1}, \mathbf{v}_{1:T}^{k-1} \right) \right\rangle. \tag{103}$$

Because stochastic gradients are unbiased (Assumption 6′), we have

$$\mathbb{E} \left[ \hat{\delta}_t^{k-1} - \delta_t^{k-1} \right] = 0, \tag{104}$$

thus,

$$T_1 = \mathbb{E} \left\langle \sum_{t=1}^T \omega_t \cdot \delta_t^{k-1}, \nabla_{\mathbf{u}} g^k \left( \mathbf{u}^{k-1}, \mathbf{v}_{1:T}^{k-1} \right) \right\rangle \tag{105}$$

$$= \frac{1}{2} \left( \left\| \nabla_{\mathbf{u}} g^k \left( \mathbf{u}^{k-1}, \mathbf{v}_{1:T}^{k-1} \right) \right\|^2 + \mathbb{E} \left\| \sum_{t=1}^T \omega_t \cdot \delta_t^{k-1} \right\|^2 \right)$$
$$- \frac{1}{2} \mathbb{E} \left\| \nabla_{\mathbf{u}} g^k \left( \mathbf{u}^{k-1}, \mathbf{v}_{1:T}^{k-1} \right) - \sum_{t=1}^T \omega_t \cdot \delta_t^{k-1} \right\|^2. \tag{106}$$

For $T_2$ we have for $k > 0$,

$$T_2 = \mathbb{E} \left\| \sum_{t=1}^T \omega_t \cdot \hat{\delta}_t^{k-1} \right\|^2 \tag{107}$$

$$= \mathbb{E} \left\| \sum_{t=1}^T \omega_t \cdot \left( \hat{\delta}_t^{k-1} - \delta_t^{k-1} \right) + \sum_{t=1}^T \omega_t \cdot \delta_t^{k-1} \right\|^2 \tag{108}$$

$$\leq 2\,\mathbb{E}\left\|\sum_{t=1}^{T}\omega_t\cdot\left(\hat{\delta}_t^{k-1}-\delta_t^{k-1}\right)\right\|^2 + 2\,\mathbb{E}\left\|\sum_{t=1}^{T}\omega_t\cdot\delta_t^{k-1}\right\|^2 \tag{109}$$

$$= 2\sum_{t=1}^{T}\omega_t^2\cdot\mathbb{E}\left\|\hat{\delta}_t^{k-1}-\delta_t^{k-1}\right\|^2 + 2\sum_{1\leq s\neq t\leq T}\omega_t\omega_s\,\mathbb{E}\left\langle\hat{\delta}_t^{k-1}-\delta_t^{k-1},\hat{\delta}_s^{k-1}-\delta_s^{k-1}\right\rangle$$

$$+ 2\,\mathbb{E}\left\|\sum_{t=1}^{T}\omega_t\delta_t^{k-1}\right\|^2. \tag{110}$$

Since clients sample batches independently, and stochastic gradients are unbiased (Assumption $6'$), we have

$$\mathbb{E}\left\langle\hat{\delta}_t^{k-1}-\delta_t^{k-1},\hat{\delta}_s^{k-1}-\delta_s^{k-1}\right\rangle = 0, \tag{111}$$

thus,

$$T_2 \leq 2\sum_{t=1}^{T}\omega_t^2\cdot\mathbb{E}\left\|\hat{\delta}_t^{k-1}-\delta_t^{k-1}\right\|^2 + 2\,\mathbb{E}\left\|\sum_{t=1}^{T}\omega_t\delta_t^{k-1}\right\|^2 \tag{112}$$

$$= 2\sum_{t=1}^{T}\omega_t^2\,\mathbb{E}\left\|\sum_{j=0}^{J-1}\frac{\eta_{k-1,j}}{\eta_{k-1}}\left[\nabla_{\mathbf{u}}g_t^k\left(\mathbf{u}_t^{k-1,j},\mathbf{v}_t^{k-1}\right)-\nabla_{\mathbf{u}}g_t^k\left(\mathbf{u}_t^{k-1,j},\mathbf{v}_t^{k-1};\xi_t^{k-1,j}\right)\right]\right\|^2$$

$$+ 2\,\mathbb{E}\left\|\sum_{t=1}^{T}\omega_t\delta_t^{k-1}\right\|^2. \tag{113}$$

Using Jensen inequality, we have

$$\left\|\sum_{j=0}^{J-1}\frac{\eta_{k-1,j}}{\eta_{k-1}}\left[\nabla_{\mathbf{u}}g_t^k\left(\mathbf{u}_t^{k-1,j},\mathbf{v}_t^{k-1}\right)-\nabla_{\mathbf{u}}g_t^k\left(\mathbf{u}_t^{k-1,j},\mathbf{v}_t^{k-1};\xi_t^{k-1,j}\right)\right]\right\|^2 \leq$$

$$\sum_{j=0}^{J-1}\frac{\eta_{k-1,j}}{\eta_{k-1}}\left\|\nabla_{\mathbf{u}}g_t^k\left(\mathbf{u}_t^{k-1,j},\mathbf{v}_t^{k-1}\right)-\nabla_{\mathbf{u}}g_t^k\left(\mathbf{u}_t^{k-1,j},\mathbf{v}_t^{k-1};\xi_t^{k-1,j}\right)\right\|^2, \tag{114}$$

and since the variance of stochastic gradients is bounded by $\sigma^2$ (Assumption $6'$), it follows that

$$\mathbb{E}\left\|\sum_{j=0}^{J-1}\frac{\eta_{k-1,j}}{\eta_{k-1}}\left[\nabla_{\mathbf{u}}g_t^k\left(\mathbf{u}_t^{k-1,j},\mathbf{v}_t^{k-1}\right)-\nabla_{\mathbf{u}}g_t^k\left(\mathbf{u}_t^{k-1,j},\mathbf{v}_t^{k-1};\xi_t^{k-1,j}\right)\right]\right\|^2$$

$$\leq \sum_{j=0}^{J-1}\frac{\eta_{k-1,j}}{\eta_{k-1}}\sigma^2 = \sigma^2. \tag{115}$$

Replacing back in the expression of $T_2$, we have

$$T_2 \leq 2\sum_{t=1}^{T}\omega_t^2\sigma^2 + 2\,\mathbb{E}\left\|\sum_{t=1}^{T}\omega_t\cdot\delta_t^{k-1}\right\|^2. \tag{116}$$

Finally, since $0\leq\omega_t\leq 1$, $t\in[T]$ and $\sum_{t=1}^{T}\omega_t = 1$, we have

$$T_2 \leq 2\sigma^2 + 2\,\mathbb{E}\left\|\sum_{t=1}^{T}\omega_t\cdot\delta_t^{k-1}\right\|^2. \tag{117}$$

Having bounded $T_1$ and $T_2$, we can replace Eq. (106) and Eq. (117) in Eq. (101), and we get

$$\mathbb{E}\left[g^k(\mathbf{u}^k,\mathbf{v}_{1:T}^{k-1})-g^k(\mathbf{u}^{k-1},\mathbf{v}_{1:T}^{k-1})\right] \leq -\frac{\eta_{k-1}}{2}\left\|\nabla_{\mathbf{u}}g^k\left(\mathbf{u}^{k-1},\mathbf{v}_{1:T}^{k-1}\right)\right\|^2 + \eta_{k-1}^2 L\sigma^2$$

$$- \frac{\eta_{k-1}}{2} \left(1 - 2L\eta_{k-1}\right) \cdot \mathbb{E} \left\| \sum_{t=1}^{T} \omega_t \cdot \delta_t^{k-1} \right\|^2$$

$$+ \frac{\eta_{k-1}}{2} \mathbb{E} \left\| \nabla_{\mathbf{u}} g^k \left(\mathbf{u}^{k-1}, \mathbf{v}_{1:T}^{k-1}\right) - \sum_{t=1}^{T} \omega_t \cdot \delta_t^{k-1} \right\|^2. \tag{118}$$

As $\eta_{k-1} \leq \frac{1}{2\sqrt{2}L} \leq \frac{1}{2L}$, we have

$$\mathbb{E} \left[ g^k(\mathbf{u}^k, \mathbf{v}_{1:T}^{k-1}) - g^k(\mathbf{u}^{k-1}, \mathbf{v}_{1:T}^{k-1}) \right] \leq - \frac{\eta_{k-1}}{2} \left\| \nabla_{\mathbf{u}} g^k \left(\mathbf{u}^{k-1}, \mathbf{v}_{1:T}^{k-1}\right) \right\|^2 + \eta_{k-1}^2 L \sigma^2$$

$$+ \frac{\eta_{k-1}}{2} \mathbb{E} \left\| \nabla_{\mathbf{u}} g^k \left(\mathbf{u}^{k-1}, \mathbf{v}_{1:T}^{k-1}\right) - \sum_{t=1}^{T} \omega_t \delta_t^{k-1} \right\|^2. \tag{119}$$

Replacing $\nabla_{\mathbf{u}} g^k \left(\mathbf{u}^{k-1}, \mathbf{v}_{1:T}^{k-1}\right) = \sum_{t=1}^{T} \omega_t \cdot \nabla_{\mathbf{u}} g_t^k \left(\mathbf{u}^{k-1}, \mathbf{v}_t^{k-1}\right)$, and using Jensen inequality to bound the last term in the RHS of Eq. (119), we have

$$\mathbb{E} \left[ g^k(\mathbf{u}^k, \mathbf{v}_{1:T}^{k-1}) - g^k(\mathbf{u}^{k-1}, \mathbf{v}_{1:T}^{k-1}) \right] \leq - \frac{\eta_{k-1}}{2} \left\| \nabla_{\mathbf{u}} g^k \left(\mathbf{u}^{k-1}, \mathbf{v}_{1:T}^{k-1}\right) \right\|^2 + \eta_{k-1}^2 L \sigma^2$$

$$+ \frac{\eta_{k-1}}{2} \sum_{t=1}^{T} \omega_t \cdot \underbrace{\mathbb{E} \left\| \nabla_{\mathbf{u}} g_t^k \left(\mathbf{u}^{k-1}, \mathbf{v}_t^{k-1}\right) - \delta_t^{k-1} \right\|^2}_{\triangleq T_3}. \tag{120}$$

We now bound the term $T_3$:

$$T_3 = \mathbb{E} \left\| \nabla_{\mathbf{u}} g_t^k \left(\mathbf{u}^{k-1}, \mathbf{v}_t^{k-1}\right) - \delta_t^{k-1} \right\|^2 \tag{121}$$

$$= \mathbb{E} \left\| \nabla_{\mathbf{u}} g_t^k \left(\mathbf{u}^{k-1}, \mathbf{v}_t^{k-1}\right) - \sum_{j=0}^{J-1} \frac{\eta_{k-1,j}}{\eta_{k-1}} \nabla_{\mathbf{u}} g_t^k \left(\mathbf{u}_t^{k-1,j}, \mathbf{v}_t^{k-1}\right) \right\|^2 \tag{122}$$

$$= \mathbb{E} \left\| \sum_{j=0}^{J-1} \frac{\eta_{k-1,j}}{\eta_{k-1}} \left[ \nabla_{\mathbf{u}} g_t^k \left(\mathbf{u}^{k-1}, \mathbf{v}_t^{k-1}\right) - \nabla_{\mathbf{u}} g_t^k \left(\mathbf{u}_t^{k-1,j}, \mathbf{v}_t^{k-1}\right) \right] \right\|^2 \tag{123}$$

$$\leq \sum_{j=0}^{J-1} \frac{\eta_{k-1,j}}{\eta_{k-1}} \mathbb{E} \left\| \nabla_{\mathbf{u}} g_t^k \left(\mathbf{u}^{k-1}, \mathbf{v}_t^{k-1}\right) - \nabla_{\mathbf{u}} g_t^k \left(\mathbf{u}_t^{k-1,j}, \mathbf{v}_t^{k-1}\right) \right\|^2 \tag{124}$$

$$\leq \sum_{j=0}^{J-1} \frac{\eta_{k-1,j}}{\eta_{k-1}} L^2 \mathbb{E} \left\| \mathbf{u}^{k-1} - \mathbf{u}_t^{k-1,j} \right\|^2, \tag{125}$$

where the first inequality follows from Jensen inequality and the second one follow from the $L$-smoothness of $g_t^k$ (Assumption 5'). We bound now the term $\mathbb{E} \left\| \mathbf{u}^{k-1} - \mathbf{u}_t^{k-1,j} \right\|$ for $j \in \{0, \ldots, J-1\}$ and $t \in [T]$,

$$\mathbb{E} \left\| \mathbf{u}^{k-1} - \mathbf{u}_t^{k-1,j} \right\|^2 = \mathbb{E} \left\| \mathbf{u}_t^{k-1,j} - \mathbf{u}_t^{k-1,0} \right\|^2 \tag{126}$$

$$= \mathbb{E} \left\| \sum_{l=0}^{j-1} \left( \mathbf{u}_t^{k-1,l+1} - \mathbf{u}_t^{k-1,l} \right) \right\|^2 \tag{127}$$

$$= \mathbb{E} \left\| \sum_{l=0}^{j-1} \eta_{k-1,l} \nabla_{\mathbf{u}} g_t^k \left(\mathbf{u}_t^{k-1,j}, \mathbf{v}_t^{k-1}; \xi_t^{k-1,l}\right) \right\|^2 \tag{128}$$

$$\leq 2\mathbb{E} \left\| \sum_{l=0}^{j-1} \eta_{k-1,l} \left[ \nabla_{\mathbf{u}} g_t^k \left(\mathbf{u}_t^{k-1,l}, \mathbf{v}_t^{k-1}; \xi_t^{k-1,l}\right) - \nabla_{\mathbf{u}} g_t^k \left(\mathbf{u}_t^{k-1,l}, \mathbf{v}_t^{k-1}\right) \right] \right\|^2$$

$$+ 2\mathbb{E} \left\| \sum_{l=0}^{j-1} \eta_{k-1,l} \nabla_{\mathbf{u}} g_t^k \left(\mathbf{u}_t^{k-1,l}, \mathbf{v}_t^{k-1}\right) \right\|^2 \tag{129}$$

$$= 2 \sum_{l=0}^{j-1} \eta_{k-1,l}^2 \mathbb{E} \left\| \nabla_{\mathbf{u}} g_t^k \left( \mathbf{u}_t^{k-1,l}, \mathbf{v}_t^{k-1}; \xi_t^{k-1,l} \right) - \nabla_{\mathbf{u}} g_t^k \left( \mathbf{u}_t^{k-1,l}, \mathbf{v}_t^{k-1} \right) \right\|^2$$

$$+ 2\mathbb{E} \left\| \sum_{l=0}^{j-1} \eta_{k-1,l} \nabla_{\mathbf{u}} g_t^k \left( \mathbf{u}_t^{k-1,l}, \mathbf{v}_t^{k-1} \right) \right\|^2 \tag{130}$$

$$\leq 2\sigma^2 \sum_{l=0}^{j-1} \eta_{k-1,l}^2 + 2\mathbb{E} \left\| \sum_{l=0}^{j-1} \eta_{k-1,l} \nabla_{\mathbf{u}} g_t^k \left( \mathbf{u}_t^{k-1,l}, \mathbf{v}_t^{k-1} \right) \right\|^2, \tag{131}$$

where, in the last two steps, we used the fact that stochastic gradients are unbiased and have bounded variance (Assumption 6′). We bound now the last term in the RHS of Eq. (131),

$$\mathbb{E} \left\| \sum_{l=0}^{j-1} \eta_{k-1,l} \nabla_{\mathbf{u}} g_t^k \left( \mathbf{u}_t^{k-1,l}, \mathbf{v}_t^{k-1} \right) \right\|^2 =$$

$$\mathbb{E} \left\| \left( \sum_{l'=0}^{j-1} \eta_{k-1,l'} \right) \cdot \sum_{l=0}^{j-1} \frac{\eta_{k-1,l}}{\sum_{l'=0}^{j-1} \eta_{k-1,l'}} \nabla_{\mathbf{u}} g_t^k \left( \mathbf{u}_t^{k-1,l}, \mathbf{v}_t^{k-1} \right) \right\|^2 \tag{132}$$

$$\leq \left( \sum_{l'=0}^{j-1} \eta_{k-1,l'} \right)^2 \cdot \sum_{l=0}^{j-1} \frac{\eta_{k-1,l}}{\sum_{l'=0}^{j-1} \eta_{k-1,l'}} \mathbb{E} \left\| \nabla_{\mathbf{u}} g_t^k \left( \mathbf{u}_t^{k-1,l}, \mathbf{v}_t^{k-1} \right) \right\|^2 \tag{133}$$

$$= \left( \sum_{l=0}^{j-1} \eta_{k-1,l} \right) \cdot \sum_{l=0}^{j-1} \eta_{k-1,l} \mathbb{E} \left\| \nabla_{\mathbf{u}} g_t^k \left( \mathbf{u}_t^{k-1,l}, \mathbf{v}_t^{k-1} \right) \right\|^2 \tag{134}$$

$$= \left( \sum_{l=0}^{j-1} \eta_{k-1,l} \right) \cdot \sum_{l=0}^{j-1} \eta_{k-1,l} \mathbb{E} \left\| \nabla_{\mathbf{u}} g_t^k \left( \mathbf{u}_t^{k-1,0}, \mathbf{v}_t^{k-1} \right) \right.$$

$$\left. - \nabla_{\mathbf{u}} g_t^k \left( \mathbf{u}_t^{k-1,0}, \mathbf{v}_t^{k-1} \right) + \nabla_{\mathbf{u}} g_t^k \left( \mathbf{u}_t^{k-1,l}, \mathbf{v}_t^{k-1} \right) \right\|^2 \tag{135}$$

$$\leq 2 \left( \sum_{l=0}^{j-1} \eta_{k-1,l} \right) \cdot \sum_{l=0}^{j-1} \eta_{k-1,l} \cdot \left[ \mathbb{E} \left\| \nabla_{\mathbf{u}} g_t^k \left( \mathbf{u}_t^{k-1,0}, \mathbf{v}_t^{k-1} \right) \right\|^2 \right.$$

$$\left. + \mathbb{E} \left\| \nabla_{\mathbf{u}} g_t^k \left( \mathbf{u}_t^{k-1,l}, \mathbf{v}_t^{k-1} \right) - \nabla_{\mathbf{u}} g_t^k \left( \mathbf{u}_t^{k-1,0}, \mathbf{v}_t^{k-1} \right) \right\|^2 \right] \tag{136}$$

$$= 2 \left( \sum_{l=0}^{j-1} \eta_{k-1,l} \right) \cdot \sum_{l=0}^{j-1} \eta_{k-1,l} \cdot \left[ \mathbb{E} \left\| \nabla_{\mathbf{u}} g_t^k \left( \mathbf{u}^{k-1}, \mathbf{v}_t^{k-1} \right) \right\|^2 \right.$$

$$\left. + \mathbb{E} \left\| \nabla_{\mathbf{u}} g_t^k \left( \mathbf{u}_t^{k-1,l}, \mathbf{v}_t^{k-1} \right) - \nabla_{\mathbf{u}} g_t^k \left( \mathbf{u}^{k-1}, \mathbf{v}_t^{k-1} \right) \right\|^2 \right] \tag{137}$$

$$\leq 2 \left( \sum_{l=0}^{j-1} \eta_{k-1,l} \right) \sum_{l=0}^{j-1} \eta_{k-1,l} \left[ \mathbb{E} \left\| \nabla_{\mathbf{u}} g_t^k \left( \mathbf{u}^{k-1}, \mathbf{v}_t^{k-1} \right) \right\|^2 + L^2 \mathbb{E} \left\| \mathbf{u}_t^{k-1,l} - \mathbf{u}^{k-1} \right\|^2 \right] \tag{138}$$

$$= 2L^2 \left( \sum_{l=0}^{j-1} \eta_{k-1,l} \right) \sum_{l=0}^{j-1} \eta_{k-1,l} \cdot \mathbb{E} \left\| \mathbf{u}_t^{k-1,l} - \mathbf{u}^{k-1} \right\|^2$$

$$+ 2 \left( \sum_{l=0}^{j-1} \eta_{k-1,l} \right)^2 \mathbb{E} \left\| \nabla_{\mathbf{u}} g_t^k \left( \mathbf{u}^{k-1}, \mathbf{v}_t^{k-1} \right) \right\|^2, \tag{139}$$

where the first inequality is obtained using Jensen inequality, and the last one is a result of the $L$-smoothness of $g_t$ (Assumption 5′). Replacing Eq. (139) in Eq. (131), we have

$$\sum_{j=0}^{J-1} \frac{\eta_{k-1,j}}{\eta_{k-1}} \cdot \mathbb{E} \left\| \mathbf{u}^{k-1} - \mathbf{u}_t^{k-1,j} \right\|^2 \leq 2\sigma^2 \left( \sum_{j=0}^{J-1} \frac{\eta_{k-1,j}}{\eta_{k-1}} \cdot \sum_{l=0}^{j-1} \eta_{k-1,l}^2 \right)$$

$$+ 4L^2 \sum_{j=0}^{J-1} \left( \frac{\eta_{k-1,j}}{\eta_{k-1}} \sum_{l=0}^{j-1} \eta_{k-1,l} \right) \cdot \left( \sum_{l=0}^{j-1} \eta_{k-1,l} \cdot \mathbb{E} \left\| \mathbf{u}_t^{k-1,l} - \mathbf{u}_t^{k-1} \right\|^2 \right)$$

$$+ 4 \left( \sum_{j=0}^{J-1} \frac{\eta_{k-1,j}}{\eta_{k-1}} \left( \sum_{l=0}^{j-1} \eta_{k-1,l} \right)^2 \right) \cdot \mathbb{E} \left\| \nabla_{\mathbf{u}} g_t^k \left( \mathbf{u}_t^{k-1}, \mathbf{v}_t^{k-1} \right) \right\|^2. \tag{140}$$

Since $\sum_{l=0}^{j-1} \eta_{k-1,l} \cdot \mathbb{E} \left\| \mathbf{u}_t^{k-1,l} - \mathbf{u}_t^{k-1} \right\|^2 \le \sum_{j=0}^{J-1} \eta_{k-1,j} \cdot \mathbb{E} \left\| \mathbf{u}_t^{k-1,j} - \mathbf{u}_t^{k-1} \right\|^2$, we have

$$\sum_{j=0}^{J-1} \frac{\eta_{k-1,j}}{\eta_{k-1}} \cdot \mathbb{E} \left\| \mathbf{u}^{k-1} - \mathbf{u}_t^{k-1,j} \right\|^2 \le 2\sigma^2 \left( \sum_{j=0}^{J-1} \frac{\eta_{k-1,j}}{\eta_{k-1}} \cdot \sum_{l=0}^{j-1} \eta_{k-1,l}^2 \right)$$

$$+ 4L^2 \left( \sum_{j=0}^{J-1} \frac{\eta_{k-1,j}}{\eta_{k-1}} \sum_{l=0}^{j-1} \eta_{k-1,l} \right) \cdot \left( \sum_{j=0}^{J-1} \eta_{k-1,j} \cdot \mathbb{E} \left\| \mathbf{u}_t^{k-1,j} - \mathbf{u}^{k-1} \right\|^2 \right)$$

$$+ 4 \left( \sum_{j=0}^{J-1} \frac{\eta_{k-1,j}}{\eta_{k-1}} \left( \sum_{l=0}^{j-1} \eta_{k-1,l} \right)^2 \right) \cdot \mathbb{E} \left\| \nabla_{\mathbf{u}} g_t^k \left( \mathbf{u}^{k-1}, \mathbf{v}_t^{k-1} \right) \right\|^2. \tag{141}$$

We use Lemma G.11 to simplify the last expression, obtaining

$$\sum_{j=0}^{J-1} \frac{\eta_{k-1,j}}{\eta_{k-1}} \cdot \mathbb{E} \left\| \mathbf{u}^{k-1} - \mathbf{u}_t^{k-1,j} \right\|^2 \le 2\sigma^2 \cdot \left\{ \sum_{j=0}^{J-1} \eta_{k-1,j}^2 \right\}$$

$$+ 4\eta_{k-1}^2 \mathbb{E} \left\| \nabla_{\mathbf{u}} g_t^k \left( \mathbf{u}^{k-1}, \mathbf{v}_t^{k-1} \right) \right\|^2 + 4\eta_{k-1} L^2 \cdot \sum_{j=0}^{J-1} \eta_{k-1,j} \mathbb{E} \left\| \mathbf{u}_t^{k-1,j} - \mathbf{u}^{k-1} \right\|^2. \tag{142}$$

Rearranging the terms, we have

$$\left( 1 - 4\eta_{k-1}^2 L^2 \right) \cdot \sum_{j=0}^{J-1} \frac{\eta_{k-1,j}}{\eta_{k-1}} \cdot \mathbb{E} \left\| \mathbf{u}^{k-1} - \mathbf{u}_t^{k-1,j} \right\|^2 \le 2\sigma^2 \cdot \left\{ \sum_{j=0}^{J-1} \eta_{k-1,j}^2 \right\}$$

$$+ 4\eta_{k-1}^2 \cdot \mathbb{E} \left\| \nabla_{\mathbf{u}} g_t^k \left( \mathbf{u}^{k-1}, \mathbf{v}_t^{k-1} \right) \right\|^2. \tag{143}$$

Finally, replacing Eq. (143) into Eq. (125), we have

$$\left( 1 - 4\eta_{k-1}^2 L^2 \right) \cdot T_3 \le 2\sigma^2 L^2 \cdot \left( \sum_{j=0}^{J-1} \eta_{k-1,j}^2 \right) + 4\eta_{k-1}^2 L^2 \cdot \mathbb{E} \left\| \nabla_{\mathbf{u}} g_t^k \left( \mathbf{u}^{k-1}, \mathbf{v}_t^{k-1} \right) \right\|^2. \tag{144}$$

For $\eta_{k-1}$ small enough, in particular if $\eta_{k-1} \le \frac{1}{2\sqrt{2}L}$, then $\frac{1}{2} \le 1 - 4\eta_{k-1}^2 L^2$, thus

$$\frac{T_3}{2} \le 2\sigma^2 L^2 \cdot \left( \sum_{j=0}^{J-1} \eta_{k-1,j}^2 \right) + 4\eta_{k-1}^2 L^2 \cdot \mathbb{E} \left\| \nabla_{\mathbf{u}} g_t^k \left( \mathbf{u}^{k-1}, \mathbf{v}_t^{k-1} \right) \right\|^2. \tag{145}$$

Replacing the bound of $T_3$ from Eq. (145) into Eq. (120), we have obtained

$$\mathbb{E} \left[ g^k(\mathbf{u}^k, \mathbf{v}_{1:T}^{k-1}) - g^k(\mathbf{u}^{k-1}, \mathbf{v}_{1:T}^{k-1}) \right] \le -\frac{\eta_{k-1}}{2} \mathbb{E} \left\| \nabla_{\mathbf{u}} g^k \left( \mathbf{u}^{k-1}, \mathbf{v}_{1:T}^{k-1} \right) \right\|^2$$

$$+ 4\eta_{k-1}^3 L^2 \sum_{t=1}^{T} \omega_t \cdot \mathbb{E} \left\| \nabla_{\mathbf{u}} g_t^k \left( \mathbf{u}^{k-1}, \mathbf{v}_t^{k-1} \right) \right\|^2$$

$$+ 2\eta_{k-1} L \left( \sum_{j=0}^{J-1} \eta_{k-1,j}^2 L + \eta_{k-1} \right) \cdot \sigma^2. \tag{146}$$

Using Assumption 7′, we have

$$\mathbb{E}\left[g^k(\mathbf{u}^k, \mathbf{v}_{1:T}^{k-1}) - g^k(\mathbf{u}^{k-1}, \mathbf{v}_{1:T}^{k-1})\right] \leq -\frac{\eta_{k-1}}{2}\,\mathbb{E}\left\|\nabla_{\mathbf{u}}g^k\left(\mathbf{u}^{k-1}, \mathbf{v}_{1:T}^{k-1}\right)\right\|^2$$

$$+ 4\eta_{k-1}^3 L^2\beta^2 \cdot \mathbb{E}\left\|\sum_{t=1}^{T}\omega_t \cdot \nabla_{\mathbf{u}}g_t^k\left(\mathbf{u}^{k-1}, \mathbf{v}_t^{k-1}\right)\right\|^2$$

$$+ 2\eta_{k-1}L\left(\sum_{j=0}^{J-1}\eta_{k-1,j}^2 L + \eta_{k-1}\right) \cdot \sigma^2 + 4\eta_{k-1}^3 L^2 G^2. \tag{147}$$

Dividing by $\eta_{k-1}$, we get

$$\mathbb{E}\left[\frac{g^k(\mathbf{u}^k, \mathbf{v}_{1:T}^{k-1}) - g^k(\mathbf{u}^{k-1}, \mathbf{v}_{1:T}^{k-1})}{\eta_{k-1}}\right] \leq \frac{8\eta_{k-1}^2 L^2\beta^2 - 1}{2}\,\mathbb{E}\left\|\nabla_{\mathbf{u}}g^k\left(\mathbf{u}^{k-1}, \mathbf{v}_{1:T}^{k-1}\right)\right\|^2$$

$$+ 2\eta_{k-1}L\left(\sum_{j=0}^{J-1}\frac{\eta_{k-1,j}^2}{\eta_{k-1}}L + 1\right) \cdot \sigma^2 + 4\eta_{k-1}^2 L^2 G^2. \tag{148}$$

For $\eta_{k-1}$ small enough, if $\eta_{k-1} \leq \frac{1}{4L\beta}$, then $8\eta_{k-1}^2 L^2\beta^2 - 1 \leq \frac{1}{2}$. Thus,

$$\mathbb{E}\left[\frac{g^k(\mathbf{u}^k, \mathbf{v}_{1:T}^{k-1}) - g^k(\mathbf{u}^{k-1}, \mathbf{v}_{1:T}^{k-1})}{\eta_{k-1}}\right] \leq -\frac{1}{4}\,\mathbb{E}\left\|\nabla_{\mathbf{u}}g^k\left(\mathbf{u}^{k-1}, \mathbf{v}_{1:T}^{k-1}\right)\right\|^2$$

$$+ 2\eta_{k-1}L\left(\sum_{j=0}^{J-1}\frac{\eta_{k-1,j}^2}{\eta_{k-1}}L + 1\right) \cdot \sigma^2 + 4\eta_{k-1}^2 L^2 G^2. \tag{149}$$

Since for $t \in [T]$, $g_t^k$ is a partial first-order surrogate of $f_t$ near $\left\{\mathbf{u}^{k-1}, \mathbf{v}_t^{k-1}\right\}$, we have (see Def. 1)

$$g_t^k\left(\mathbf{u}^{k-1}, \mathbf{v}_t^{k-1}\right) = f_t\left(\mathbf{u}^{k-1}, \mathbf{v}_t^{k-1}\right), \tag{150}$$

$$\nabla_{\mathbf{u}}g_t^k\left(\mathbf{u}^{k-1}, \mathbf{v}_t^{k-1}\right) = \nabla_{\mathbf{u}}f_t\left(\mathbf{u}^{k-1}, \mathbf{v}_t^{k-1}\right), \tag{151}$$

$$g_t^k\left(\mathbf{u}^k, \mathbf{v}_t^{k-1}\right) = g_t^k\left(\mathbf{u}^k, \mathbf{v}_t^k\right) + d_{\mathcal{V}}\left(\mathbf{v}_t^{k-1}, \mathbf{v}_t^k\right). \tag{152}$$

Multiplying by $\omega_t$ and summing over $t \in [T]$, we have

$$g^k\left(\mathbf{u}^{k-1}, \mathbf{v}_{1:T}^{k-1}\right) = f\left(\mathbf{u}^{k-1}, \mathbf{v}_{1:T}^{k-1}\right), \tag{153}$$

$$\nabla_{\mathbf{u}}g^k\left(\mathbf{u}^{k-1}, \mathbf{v}_{1:T}^{k-1}\right) = \nabla_{\mathbf{u}}f\left(\mathbf{u}^{k-1}, \mathbf{v}_{1:T}^{k-1}\right), \tag{154}$$

$$g^k\left(\mathbf{u}^k, \mathbf{v}_{1:T}^{k-1}\right) = g^k\left(\mathbf{u}^k, \mathbf{v}_{1:T}^k\right) + \sum_{t=1}^{T}\omega_t \cdot d_{\mathcal{V}}\left(\mathbf{v}_t^{k-1}, \mathbf{v}_t^k\right). \tag{155}$$

Replacing Eq. (153), Eq. (154) and Eq. (155) in Eq. (149), we have

$$\mathbb{E}\left[\frac{g^k(\mathbf{u}^k, \mathbf{v}_{1:T}^k) - f(\mathbf{u}^{k-1}, \mathbf{v}_{1:T}^{k-1})}{\eta_{k-1}}\right] \leq$$

$$-\frac{1}{4}\,\mathbb{E}\left\|\nabla_{\mathbf{u}}f\left(\mathbf{u}^{k-1}, \mathbf{v}_{1:T}^{k-1}\right)\right\|^2 - \frac{1}{\eta_{k-1}}\sum_{t=1}^{T}\omega_t \cdot d_{\mathcal{V}}\left(\mathbf{v}_t^{k-1}, \mathbf{v}_t^k\right)$$

$$+ 2\eta_{k-1}L\left(\left\{\sum_{j=0}^{J-1}\frac{\eta_{k-1,j}^2}{\eta_{k-1}}\right\}L + 1\right) \cdot \sigma^2 + 4\eta_{k-1}^2 L^2 G^2. \tag{156}$$

Using again Definition 1, we have

$$g^k(\mathbf{u}^k, \mathbf{v}_{1:T}^k) \geq f(\mathbf{u}^k, \mathbf{v}_{1:T}^k), \tag{157}$$

thus,

$$\mathbb{E}\left[\frac{f(\mathbf{u}^k, \mathbf{v}_{1:T}^k) - f(\mathbf{u}^{k-1}, \mathbf{v}_{1:T}^{k-1})}{\eta_{k-1}}\right] \leq$$

$$- \frac{1}{4} \mathbb{E} \left\| \nabla_{\mathbf{u}} f \left( \mathbf{u}^{k-1}, \mathbf{v}_{1:T}^{k-1} \right) \right\|^2 - \frac{1}{\eta_{k-1}} \sum_{t=1}^{T} \omega_t \cdot d_{\mathcal{V}} \left( \mathbf{v}_t^{k-1}, \mathbf{v}_t^k \right)$$

$$+ 2\eta_{k-1} L \left( \sum_{j=0}^{J-1} \frac{\eta_{k-1,j}^2}{\eta_{k-1}} L + 1 \right) \cdot \sigma^2 + 4\eta_{k-1}^2 L^2 G^2. \tag{158}$$

$\square$

**Lemma G.2.** *For $k \geq 0$ and $t \in [T]$, the iterates of Alg. 3 verify*

$$0 \leq d_{\mathcal{V}} \left( \mathbf{v}_t^{k+1}, \mathbf{v}_t^k \right) \leq f_t \left( \mathbf{u}^k, \mathbf{v}_t^k \right) - f_t(\mathbf{u}^k, \mathbf{v}_t^{k+1}) \tag{159}$$

*Proof.* Since $\mathbf{v}_t^{k+1} \in \arg\min_{v \in V} g_t^k \left( \mathbf{u}^{k-1}, v \right)$, and $g_t^k$ is a partial first-order surrogate of $f_t$ near $\{\mathbf{u}^{k-1}, \mathbf{v}_t^{k-1}\}$, we have

$$g_t^k \left( \mathbf{u}^{k-1}, \mathbf{v}_t^{k-1} \right) - g_t^k \left( \mathbf{u}^{k-1}, \mathbf{v}_t^k \right) = d_{\mathcal{V}} \left( \mathbf{v}_t^{k-1}, \mathbf{v}_t^k \right), \tag{160}$$

thus,

$$f_t \left( \mathbf{u}^{k-1}, \mathbf{v}_t^{k-1} \right) - f_t \left( \mathbf{u}^{k-1}, \mathbf{v}_t^k \right) \geq d_{\mathcal{V}} \left( \mathbf{v}_t^{k-1}, \mathbf{v}_t^k \right), \tag{161}$$

where we used the fact that

$$g_t^k \left( \mathbf{u}^{k-1}, \mathbf{v}_t^{k-1} \right) = f_t \left( \mathbf{u}^{k-1}, \mathbf{v}_t^{k-1} \right), \tag{162}$$

and,

$$g_t^k \left( \mathbf{u}^{k-1}, \mathbf{v}_t^k \right) \geq f_t \left( \mathbf{u}^{k-1}, \mathbf{v}_t^k \right). \tag{163}$$

$\square$

**Theorem 3.2′.** *Under Assumptions 4′–7′, when clients use SGD as local solver with learning rate $\eta = \frac{a_0}{\sqrt{K}}$, after a large enough number of communication rounds $K$, the iterates of federated surrogate optimization (Alg. 3) satisfy:*

$$\frac{1}{K} \sum_{k=1}^{K} \mathbb{E} \left\| \nabla_{\mathbf{u}} f \left( \mathbf{u}^k, \mathbf{v}_{1:T}^k \right) \right\|_F^2 \leq \mathcal{O} \left( \frac{1}{\sqrt{K}} \right), \qquad \frac{1}{K} \sum_{k=1}^{K} \mathbb{E} \left[ \Delta_{\mathbf{v}} f(\mathbf{u}^k, \mathbf{v}_{1:T}^k) \right] \leq \mathcal{O} \left( \frac{1}{K^{3/4}} \right),$$
$$\tag{89}$$

*where the expectation is over the random batches samples, and $\Delta_{\mathbf{v}} f(\mathbf{u}^k, \mathbf{v}_{1:T}^k) \triangleq f \left( \mathbf{u}^k, \mathbf{v}_{1:T}^k \right) - f \left( \mathbf{u}^k, \mathbf{v}_{1:T}^{k+1} \right) \geq 0.$*

*Proof.* For $K$ large enough, $\eta = \frac{a_0}{\sqrt{K}} \leq \frac{1}{J} \min \left\{ \frac{1}{2\sqrt{2}L}, \frac{1}{4L\beta} \right\}$, thus the assumptions of Lemma G.1 are satisfied. Lemma G.1 and non-negativity of $d_{\mathcal{V}}$ lead to

$$\mathbb{E} \left[ \frac{f(\mathbf{u}^k, \mathbf{v}_{1:T}^k) - f(\mathbf{u}^{k-1}, \mathbf{v}_{1:T}^{k-1})}{J\eta} \right] \leq -\frac{1}{4} \mathbb{E} \left\| \nabla_{\mathbf{u}} f \left( \mathbf{u}^{k-1}, \mathbf{v}_{1:T}^{k-1} \right) \right\|^2$$
$$+ 2\eta L \left( \eta L + 1 \right) \cdot \sigma^2 + 4J^2 \eta^2 L^2 G^2. \tag{164}$$

Rearranging the terms and summing for $k \in [K]$, we have

$$\frac{1}{K} \sum_{k=1}^{K} \mathbb{E} \left\| \nabla_{\mathbf{u}} f \left( \mathbf{u}^{k-1}, \mathbf{v}_{1:T}^{k-1} \right) \right\|^2$$

$$\leq 4\mathbb{E} \left[ \frac{f(\mathbf{u}^0, \mathbf{v}_{1:T}^0) - f(\mathbf{u}^K, \mathbf{v}_{1:T}^K)}{J\eta K} \right] + 8 \frac{\eta L \left( \eta L + 1 \right) \cdot \sigma^2 + 2J^2 \eta^2 L^2 G^2}{K} \tag{165}$$

$$\leq 4\mathbb{E} \left[ \frac{f(\mathbf{u}^0, \mathbf{v}_{1:T}^0) - f^*}{J\eta K} \right] + 8 \frac{\eta L \left( \eta L + 1 \right) \cdot \sigma^2 + 2J^2 \eta^2 L^2 G^2}{K}, \tag{166}$$

where we use Assumption 4′ to obtain (166). Thus,

$$\frac{1}{K} \sum_{k=1}^{K} \mathbb{E} \left\| \nabla_{\mathbf{u}} f \left( \mathbf{u}^{k-1}, \mathbf{v}_{1:T}^{k-1} \right) \right\|^2 = \mathcal{O} \left( \frac{1}{\sqrt{K}} \right). \tag{167}$$

To prove the second part of Eq. (89), we first decompose $\Delta_{\mathbf{v}} \triangleq f\left(\mathbf{u}^k, \mathbf{v}_{1:T}^k\right) - f\left(\mathbf{u}^k, \mathbf{v}_{1:T}^{k+1}\right) \geq 0$ as follow,

$$\Delta_{\mathbf{v}} = \underbrace{f\left(\mathbf{u}^k, \mathbf{v}_{1:T}^k\right) - f\left(\mathbf{u}^{k+1}, \mathbf{v}_{1:T}^{k+1}\right)}_{\triangleq T_1^k} + \underbrace{f\left(\mathbf{u}^{k+1}, \mathbf{v}_{1:T}^{k+1}\right) - f\left(\mathbf{u}^k, \mathbf{v}_{1:T}^{k+1}\right)}_{\triangleq T_2^k}. \tag{168}$$

Using again Lemma G.1 and Eq. (167), it follows that

$$\frac{1}{K}\sum_{k=1}^{K} \mathbb{E}\left[T_1^k\right] \leq \mathcal{O}\left(\frac{1}{K}\right). \tag{169}$$

For $T_2^k$, we use the fact that $f$ is $2L$-smooth (Lemma G.12) w.r.t. $u$ and Cauchy-Schwartz inequality. Thus, for $k > 0$, we write

$$T_2^k = f\left(\mathbf{u}^{k+1}, \mathbf{v}_{1:T}^{k+1}\right) - f\left(\mathbf{u}^k, \mathbf{v}_{1:T}^{k+1}\right) \tag{170}$$

$$\leq \left\|\nabla_{\mathbf{u}} f\left(\mathbf{u}^{k+1}, \mathbf{v}_{1:T}^{k+1}\right)\right\| \cdot \left\|\mathbf{u}^{k+1} - \mathbf{u}^k\right\| + 2L^2 \left\|\mathbf{u}^{k+1} - \mathbf{u}^k\right\|^2. \tag{171}$$

Summing over $k$ and taking expectation:

$$\frac{1}{K}\sum_{k=1}^{K}\mathbb{E}\left[T_2^k\right] \leq \frac{1}{K}\sum_{k=1}^{K}\mathbb{E}\left[\left\|\nabla_{\mathbf{u}} f\left(\mathbf{u}^{k+1}, \mathbf{v}_{1:T}^{k+1}\right)\right\| \cdot \left\|\mathbf{u}^{k+1} - \mathbf{u}^k\right\|\right]$$

$$+ \frac{1}{K}\sum_{k=1}^{K} 2L^2 \mathbb{E}\left[\left\|\mathbf{u}^{k+1} - \mathbf{u}^k\right\|^2\right] \tag{172}$$

$$\leq \frac{1}{K}\sqrt{\sum_{k=1}^{K}\mathbb{E}\left[\left\|\nabla_{\mathbf{u}} f\left(\mathbf{u}^{k+1}, \mathbf{v}_{1:T}^{k+1}\right)\right\|^2\right]}\sqrt{\sum_{k=1}^{K}\mathbb{E}\left[\left\|\mathbf{u}^{k+1} - \mathbf{u}^k\right\|^2\right]}$$

$$+ \frac{1}{K}\sum_{k=1}^{K} 2L^2 \mathbb{E}\left[\left\|\mathbf{u}^{k+1} - \mathbf{u}^k\right\|^2\right], \tag{173}$$

where the second inequality follows from Cauchy-Schwarz inequality. From Eq. (143), with $\eta_{k-1} = J\eta$, we have for $t \in [T]$

$$\mathbb{E}\left\|\mathbf{u}^k - \mathbf{u}_t^{k-1,J}\right\|^2 \leq 4\sigma^2 J\eta^2 + 8J^3\eta^2 \cdot \mathbb{E}\left\|\nabla_{\mathbf{u}} g_t^k\left(\mathbf{u}^{k-1}, \mathbf{v}_t^{k-1}\right)\right\|^2. \tag{174}$$

Multiplying the previous by $\omega_t$ and summing for $t \in [T]$, we have

$$\sum_{t=1}^{T} \omega_t \cdot \mathbb{E}\left\|\mathbf{u}^{k-1} - \mathbf{u}_t^{k-1,J}\right\|^2 \leq 4J^2\sigma^2\eta^2 + 8J^3\eta^2 \cdot \sum_{t=1}^{T} \omega_t \mathbb{E}\left\|\nabla_{\mathbf{u}} g_t^k\left(\mathbf{u}^{k-1}, \mathbf{v}_t^{k-1}\right)\right\|^2. \tag{175}$$

Using Assumption 7′, it follows that

$$\sum_{t=1}^{T} \omega_t \mathbb{E}\left\|\mathbf{u}^{k-1} - \mathbf{u}_t^{k-1,J}\right\|^2 \leq 4J^2\eta^2\left(2JG^2 + \sigma^2\right) + 8J^3\eta^2\beta^2\mathbb{E}\left\|\sum_{t=1}^{T} \omega_t \nabla_{\mathbf{u}} g_t^k\left(\mathbf{u}^{k-1}, \mathbf{v}_t^{k-1}\right)\right\|^2. \tag{176}$$

Finally using Jensen inequality and the fact that $g_t^k$ is a partial first-order of $f_t$ near $\left\{u^{k-1}, v_t^{k-1}\right\}$, we have

$$\mathbb{E}\left\|\mathbf{u}^{k-1} - \mathbf{u}^k\right\|^2 \leq 4J^2\eta^2\left(2JG^2 + \sigma^2\right) + 8J^3\eta^2\beta^2\mathbb{E}\left\|\nabla_{\mathbf{u}} f\left(\mathbf{u}^{k-1}, \mathbf{v}_{1:T}^{k-1}\right)\right\|^2. \tag{177}$$

From Eq. (167) and $\eta \leq \mathcal{O}(1/\sqrt{K})$, we obtain

$$\frac{1}{K}\sum_{k=1}^{K}\mathbb{E}\left\|\mathbf{u}^{k-1} - u^k\right\|^2 \leq \mathcal{O}(1), \tag{178}$$

Replacing the last inequality in Eq. (173) and using again Eq. (167), we obtain

$$\frac{1}{K}\sum_{k=1}^{K}\mathbb{E}\left[T_2^k\right] \leq \mathcal{O}\left(\frac{1}{K^{3/4}}\right). \tag{179}$$

Combining Eq. (169) and Eq. (179), it follows that

$$\frac{1}{K} \sum_{k=1}^{K} \mathbb{E}\left[\Delta_{\mathbf{v}} f(u^k, \mathbf{v}_{1:T}^k)\right] \leq \mathcal{O}\left(\frac{1}{K^{3/4}}\right). \tag{180}$$

$\square$

### G.1.3  Proof of Theorem 3.2

In this section, $f$ denotes the negative log-likelihood function defined in Eq. (6). Moreover, we introduce the negative log-likelihood at client $t$ as follows

$$f_t(\Theta, \Pi) \triangleq -\frac{\log p(\mathcal{S}_t|\Theta, \Pi)}{n} \triangleq -\frac{1}{n_t} \sum_{i=1}^{n_t} \log p(s_t^{(i)}|\Theta, \pi_t). \tag{181}$$

**Theorem 3.2.** *Under Assumptions 1–7, when clients use SGD as local solver with learning rate $\eta = \frac{a_0}{\sqrt{K}}$, after a large enough number of communication rounds $K$, FedEM's iterates satisfy:*

$$\frac{1}{K} \sum_{k=1}^{K} \mathbb{E}\left\|\nabla_\Theta f\left(\Theta^k, \Pi^k\right)\right\|_F^2 \leq \mathcal{O}\left(\frac{1}{\sqrt{K}}\right), \qquad \frac{1}{K} \sum_{k=1}^{K} \Delta_\Pi f(\Theta^k, \Pi^k) \leq \mathcal{O}\left(\frac{1}{K^{3/4}}\right), \tag{11}$$

*where the expectation is over the random batches samples, and $\Delta_\Pi f(\Theta^k, \Pi^k) \triangleq f\left(\Theta^k, \Pi^k\right) - f\left(\Theta^k, \Pi^{k+1}\right) \geq 0$.*

*Proof.* We prove this result as a particular case of Theorem 3.2′. To this purpose, in this section, we consider that $\mathcal{V} \triangleq \Delta^M$, $\mathbf{u} = \Theta \in \mathbb{R}^{dM}$, $\mathbf{v}_t = \pi_t$, and $\omega_t = n_t/n$ for $t \in [T]$. For $k > 0$, we define $g_t^k$ as follows:

$$g_t^k\left(\Theta, \pi_t\right) = \frac{1}{n_t} \sum_{i=1}^{n_t} \sum_{m=1}^{M} q_t^k\left(z_t^{(i)} = m\right) \cdot \left(l\left(h_{\theta_m}(\mathbf{x}_t^{(i)}), y_t^{(i)}\right) - \log p_m(\mathbf{x}_t^{(i)}) - \log \pi_t \right.$$
$$\left. + \log q_t^k\left(z_t^{(i)} = m\right) - c\right), \tag{182}$$

where $c$ is the same constant appearing in Assumption 3, Eq. (3). With this definition, it is easy to check that the federated surrogate optimization algorithm (Alg. 3) reduces to FedEM (Alg. 2). Theorem 3.2 then follows immediately from Theorem 3.2′, once we verify that $\left(g_t^k\right)_{1 \leq t \leq T}$ satisfy the assumptions of Theorem 3.2′.

Assumption 4′, Assumption 6′, and Assumption 7′ follow directly from Assumption 4, Assumption 6, and Assumption 7, respectively. Lemma G.3 shows that for $k > 0$, $g^k$ is smooth w.r.t. $\Theta$ and then Assumption 5′ is satisfied. Finally, Lemmas G.4–G.6 show that for $t \in [T]$ $g_t^k$ is a partial first-order surrogate of $f_t$ w.r.t. $\Theta$ near $\left\{\Theta^{k-1}, \pi_t\right\}$ with $d_{\mathcal{V}}(\cdot, \cdot) = \mathcal{KL}(\cdot\|\cdot)$. $\square$

**Lemma G.3.** *Under Assumption 5, for $t \in [T]$ and $k > 0$, $g_t^k$ is $L$-smooth w.r.t $\Theta$.*

*Proof.* $g_t^k$ is a convex combination of $L$-smooth function $\theta \mapsto l(\theta; s_t^{(i)})$, $i \in [n_t]$. Thus it is also $L$-smooth. $\square$

**Lemma G.4.** *Suppose that Assumptions 1–3, hold. Then, for $t \in [T]$, $\Theta \in \mathbb{R}^{M \times d}$ and $\pi_t \in \Delta^M$*

$$r_t^k\left(\Theta, \pi_t\right) \triangleq g_t^k\left(\Theta, \pi_t\right) - f_t\left(\Theta, \pi_t\right) = \frac{1}{n_t} \sum_{i=1}^{n_t} \mathcal{KL}\left(q_t^k\left(z_i^{(t)}\right) \| p_t\left(z_i^{(t)}|s_i^{(t)}, \Theta, \pi_t\right)\right),$$

*where $\mathcal{KL}$ is Kullback–Leibler divergence.*

*Proof.* Let $k > 0$ and $t \in [T]$, and consider $\Theta \in \mathbb{R}^{M \times d}$ and $\pi_t \in \Delta^M$, then

$$g_t^k\left(\Theta, \pi_t\right) = \frac{1}{n_t} \sum_{i=1}^{n_t} \sum_{m=1}^{M} q_t^k\left(z_t^{(i)} = m\right) \cdot \left(l\left(h_{\theta_m}(\mathbf{x}_t^{(i)}), y_t^{(i)}\right) - \log p_m(\mathbf{x}_t^{(i)}) - \log \pi_t\right.$$

$$\left. + \log q_t^k\left(z_t^{(i)} = m\right) - c\right), \tag{183}$$

$$= \frac{1}{n_t} \sum_{i=1}^{n_t} \sum_{m=1}^{M} q_t^k\left(z_t^{(i)} = m\right) \cdot \left(-\log p_m\left(y_t^{(i)} | \mathbf{x}_t^{(i)}, \theta_m\right) - \log p_m(\mathbf{x}_t^{(i)}) - \log \pi_t\right.$$

$$\left. + \log q_t^k\left(z_t^{(i)} = m\right)\right) \tag{184}$$

$$= \frac{1}{n_t} \sum_{i=1}^{n_t} \sum_{m=1}^{M} q_t^k\left(z_t^{(i)} = m\right) \cdot \left(-\log p_m\left(y_t^{(i)} | \mathbf{x}_t^{(i)}, \theta_m\right) \cdot p_m(\mathbf{x}_t^{(i)}) \cdot p_t\left(z_t^{(i)} = m\right)\right.$$

$$\left. + \log q_t^k\left(z_t^{(i)} = m\right)\right) \tag{185}$$

$$= \frac{1}{n_t} \sum_{i=1}^{n_t} \sum_{m=1}^{M} q_t^k\left(z_t^{(i)} = m\right) \cdot \left(\log q_t^k\left(z_t^{(i)} = m\right) - \log p_t\left(s_t^{(i)}, z_t^{(i)} = m \middle| \Theta, \pi_t\right)\right) \tag{186}$$

$$= \frac{1}{n_t} \sum_{t=1}^{n_t} \sum_{m=1}^{M} q_t^k\left(z_t^{(i)} = m\right) \log \frac{q_t^k\left(z_t^{(i)} = m\right)}{p_t\left(s_t^{(i)}, z_t^{(i)} = m | \Theta, \pi_t\right)}. \tag{187}$$

Thus,

$$r_t^k\left(\Theta, \pi_t\right) \triangleq g_t^k\left(\Theta, \pi_t\right) - f_t\left(\Theta, \pi_t\right) \tag{188}$$

$$= -\frac{1}{n_t} \sum_{t=1}^{n_t} \sum_{m=1}^{M} \left(q_t^k\left(z_t^{(i)} = m\right) \cdot \log \frac{p_t\left(s_t^{(i)}, z_t^{(i)} = m | \Theta, \pi_t\right)}{q_t^k\left(z_t^{(i)} = m\right)}\right)$$

$$+ \frac{1}{n_t} \sum_{i=1}^{n_t} \log p_t\left(s_t^{(i)} | \Theta, \pi_t\right) \tag{189}$$

$$= \frac{1}{n_t} \sum_{t=1}^{n_t} \sum_{m=1}^{M} q_t^k\left(z_t^{(i)} = m\right) \left(\log p_t\left(s_t^{(i)} | \Theta, \pi_t\right)\right.$$

$$\left. - \log \frac{p_t\left(s_t^{(i)}, z_t^{(i)} = m | \Theta, \pi_t\right)}{q_t^k\left(z_t^{(i)} = m\right)}\right) \tag{190}$$

$$= \frac{1}{n_t} \sum_{t=1}^{n_t} \sum_{m=1}^{M} q_t^k\left(z_t^{(i)} = m\right) \log \frac{p_t\left(s_t^{(i)} | \Theta, \pi_t\right) \cdot q_t^k\left(z_t^{(i)} = m\right)}{p_t\left(s_t^{(i)}, z_t^{(i)} = m | \Theta, \pi_t\right)} \tag{191}$$

$$= \frac{1}{n_t} \sum_{t=1}^{n_t} \sum_{m=1}^{M} q_t^k\left(z_t^{(i)} = m\right) \cdot \log \frac{q_t^k\left(z_t^{(i)} = m\right)}{p_t\left(z_t^{(i)} = m | s_t^{(i)}, \Theta, \pi_t\right)}. \tag{192}$$

Thus,

$$r_t^k\left(\Theta, \pi_t\right) = \frac{1}{n_t} \sum_{i=1}^{n_t} \mathcal{KL}\left(q_t^k(\cdot) \| p_t(\cdot | s_i^{(t)}, \Theta, \pi_t)\right) \geq 0. \tag{193}$$

$\square$

The following lemma shows that $g_t^k$ and $g^k$ (as defined in Eq. 98) satisfy the first two properties in Definition 1.

**Lemma G.5.** *Suppose that Assumptions 1–3 and Assumption 5 hold. For all $k \geq 0$ and $t \in [T]$, $g_t^k$ is a majorant of $f_t$ and $r_t^k \triangleq g_t^k - f_t$ is L-smooth in $\Theta$. Moreover $r_t^k \left( \Theta^{k-1}, \pi_t^{k-1} \right) = 0$ and $\nabla_\Theta r_t^k \left( \Theta^{k-1}, \pi_t^{k-1} \right) = 0$.*

*The same holds for $g^k$, i.e., $g^k$ is a majorant of $f$, $r^k \triangleq g^k - f$ is L-smooth in $\Theta$, $r^k \left( \Theta^{k-1}, \Pi^{k-1} \right) = 0$ and $\nabla_\Theta r^k \left( \Theta^{k-1}, \Pi^{k-1} \right) = 0$*

*Proof.* For $t \in [T]$, consider $\Theta \in \mathbb{R}^{M \times d}$ and $\pi_t \in \Delta^M$, we have (Lemma G.4)

$$r_t^k \left( \Theta, \pi_t \right) \triangleq g_t^k \left( \Theta, \pi_t \right) - f_t \left( \Theta, \pi_t \right) = \frac{1}{n_t} \sum_{i=1}^{n_t} \mathcal{KL} \left( q_t^k \left( z_i^{(t)} \right) \| p_t \left( z_t^{(i)} | s_t^{(i)}, \Theta, \pi_t \right) \right). \quad (194)$$

Since $\mathcal{KL}$ divergence is non-negative, it follows that $g_t^k$ is a majorant of $f_t$, i.e.,

$$\forall \, \Theta \in \mathbb{R}^{M \times d}, \; \pi_t \in \Delta^M : \; g_t^k \left( \Theta, \pi \right) \geq f_t \left( \Theta, \pi_t \right). \quad (195)$$

Moreover since, $q_t^k \left( z_t^{(i)} \right) = p_t \left( z_t^{(i)} | s_t^{(i)}, \Theta^{k-1}, \pi_t^{k-1} \right)$ for $k > 0$, it follows that

$$r_t^k \left( \Theta^{k-1}, \pi_t^{k-1} \right) = 0. \quad (196)$$

For $i \in [n_t]$ and $m \in [M]$, from Eq. 78, we have

$$p_t \left( z_t^{(i)} = m | s_t^{(i)}, \Theta, \pi_t \right) = \frac{p_m \left( y_t^{(i)} | \mathbf{x}_t^{(i)}, \theta_m \right) \times \pi_{tm}}{\sum_{m'=1}^{M} p_{m'} \left( y_t^{(i)} | \mathbf{x}_t^{(i)}, \theta_{m'} \right) \times \pi_{tm'}} \quad (197)$$

$$= \frac{\exp \left[ -l \left( h_{\theta_m}(\mathbf{x}_t^{(i)}), y_t^{(i)} \right) \right] \times \pi_{tm}}{\sum_{m'=1}^{M} \exp \left[ -l \left( h_{\theta_{m'}}(\mathbf{x}_t^{(i)}), y_t^{(i)} \right) \right] \times \pi_{tm'}} \quad (198)$$

$$= \frac{\exp \left[ -l \left( h_{\theta_m}(\mathbf{x}_t^{(i)}), y_t^{(i)} \right) + \log \pi_{tm} \right]}{\sum_{m'=1}^{M} \exp \left[ -l \left( h_{\theta_{m'}}(\mathbf{x}_t^{(i)}), y_t^{(i)} \right) + \log \pi_{tm'} \right]}. \quad (199)$$

For ease of notation, we introduce

$$l_i(\theta) \triangleq l \left( h_\theta(\mathbf{x}_t^{(i)}), y_t^{(i)} \right), \qquad \theta \in \mathbb{R}^d, \; m \in [M], \; i \in [n_t], \quad (200)$$

$$\gamma_m (\Theta) \triangleq p_t \left( z_t^{(i)} = m | s_t^{(i)}, \Theta, \pi_t \right), \qquad m \in [M], \quad (201)$$

and,

$$\varphi_i (\Theta) \triangleq \mathcal{KL} \left( q_t^k \left( z_i^{(t)} \right) \| p_t \left( z_t^{(i)} | s_t^{(i)}, \Theta, \pi_t \right) \right). \quad (202)$$

For $i \in [n_t]$, function $l_i$ is differentiable because smooth (Assum 5), thus $\gamma_m$, $m \in [M]$ is differentiable as the composition of the softmax function and the function $\{ \Theta \mapsto -l_i (\Theta) + \log \pi_{tm} \}$. Its gradient is given by

$$\begin{cases} \nabla_{\theta_m} \gamma_m (\Theta) = -\gamma_m (\Theta) \cdot (1 - \gamma_m (\Theta)) \cdot \nabla l_i (\theta_m), \\ \nabla_{\theta_{m'}} \gamma_m (\Theta) = \gamma_m (\Theta) \cdot \gamma_{m'} (\Theta) \cdot \nabla l_i (\theta_m), \qquad m' \neq m. \end{cases} \quad (203)$$

Thus for $m \in [M]$, we have

$$\nabla_{\theta_m} \varphi_i (\Theta) = \sum_{m'=1}^{M} q_t^k \left( z_i^{(t)} = m' \right) \cdot \frac{\nabla_{\theta_m} \gamma_{m'} (\Theta)}{\gamma_{m'} (\Theta)} \quad (204)$$

$$= \sum_{\substack{m'=1 \\ m' \neq m}} \left[ q_t^k \left( z_i^{(t)} = m' \right) \cdot \frac{\gamma_m (\Theta) \cdot \gamma_{m'} (\Theta)}{\gamma_{m'} (\Theta) \cdot} \cdot \nabla l_i (\theta_m) \right]$$

$$- q_t^k \left( z_i^{(t)} = m \right) \cdot \frac{\gamma_m \left( \Theta \right) \cdot \left( 1 - \gamma_m \left( \Theta \right) \right)}{\gamma_m \left( \Theta \right)} \cdot \nabla l_i \left( \theta_m \right). \tag{205}$$

Using the fact that $\sum_{m'=1}^{M} q_t^k \left( z_i^{(t)} = m \right) = 1$, it follows that

$$\nabla_{\theta_m} \varphi_i \left( \Theta \right) = \left( \gamma_m \left( \Theta \right) - q_t^k \left( z_i^{(t)} = m \right) \right) \cdot \nabla l_i \left( \theta_m \right). \tag{206}$$

Since $l_i$, $i \in [n_t]$ is twice continuously differentiable (Assumption 5), and $\gamma_m$, $m \in [M]$ is differentiable, then $\phi_i$, $i \in [n_t]$ is twice continuously differentiable. We use $\mathbf{H} \left( \varphi_i \left( \Theta \right) \right) \in \mathbb{R}^{dM \times dM}$ (resp. $\mathbf{H} \left( l_i \left( \theta \right) \right) \in \mathbb{R}^{d \times d}$) to denote the Hessian of $\varphi$ (resp. $l_i$) at $\Theta$ (resp. $\theta$). The Hessian of $\varphi_i$ is a block matrix given by

$$\begin{cases} \left( \mathbf{H} \left( \varphi_i \left( \Theta \right) \right) \right)_{m,m} = -\gamma_m \left( \Theta \right) \cdot \left( 1 - \gamma_m \left( \Theta \right) \right) \cdot \left( \nabla l_i (\theta_m) \right) \cdot \left( \nabla l_i (\theta_m) \right)^{\mathsf{T}} \\ \qquad\qquad\qquad + \left( \gamma_m (\Theta) - q_t^k \left( z_i^{(t)} = m \right) \right) \cdot \mathbf{H} \left( l_i \left( \theta_m \right) \right) \\ \left( \mathbf{H} \left( \varphi_i \left( \Theta \right) \right) \right)_{m,m'} = \gamma_m \left( \Theta \right) \cdot \gamma_{m'} \left( \Theta \right) \cdot \left( \nabla l_i (\theta_{m'}) \right) \cdot \left( \nabla l_i (\theta_m) \right)^{\mathsf{T}}, \qquad m' \neq m. \end{cases} \tag{207}$$

We introduce the block matrix $\tilde{\mathbf{H}} \in \mathbb{R}^{dM \times dM}$, defined by

$$\begin{cases} \tilde{\mathbf{H}}_{m,m} = -\gamma_m \left( \Theta \right) \cdot \left( 1 - \gamma_m \left( \Theta \right) \right) \cdot \left( \nabla l_i (\theta_m) \right) \cdot \left( \nabla l_i (\theta_m) \right)^{\mathsf{T}} \\ \tilde{\mathbf{H}}_{m,m'} = \gamma_m \left( \Theta \right) \cdot \gamma_m \left( \Theta \right) \cdot \left( \nabla_\theta l_i (\theta_m) \right) \cdot \left( \nabla l_i (\theta_{m'}) \right)^{\mathsf{T}}, \qquad m' \neq m, \end{cases} \tag{208}$$

Eq. (207) can be written as

$$\begin{cases} \left( \mathbf{H} \left( \varphi_i \left( \Theta \right) \right) \right)_{m,m} - \tilde{\mathbf{H}}_{m,m} = \left( \gamma_m (\Theta) - q_t^k \left( z_i^{(t)} = m \right) \right) \cdot \mathbf{H} \left( l_i \left( \theta_m \right) \right) \\ \left( \mathbf{H} \left( \varphi_i \left( \Theta \right) \right) \right)_{m,m'} - \tilde{\mathbf{H}}_{m,m'} = 0, \qquad\qquad\qquad m' \neq m. \end{cases} \tag{209}$$

We recall that a twice differentiable function is $L$ smooth if and only if the eigenvalues of its Hessian are smaller then $L$, see e.g., [52, Lemma 1.2.2] or [6, Section 3.2]. Since $l_i$ and also $-l_i$ are $L$-smooth (Assumption 5), we have for $\theta \in \mathbb{R}^d$,

$$-L \cdot I_d \preccurlyeq \mathbf{H} \left( l_i \left( \theta \right) \right) \preccurlyeq L \cdot I_d. \tag{210}$$

Using Lemma G.15, we can conclude that matrix $\tilde{\mathbf{H}}$ is semi-definite negative. Since

$$-1 \leq \gamma_m(\Theta) - q_t^k \left( z_i^{(t)} = m \right) \leq 1, \tag{211}$$

it follows that

$$\mathbf{H} \left( \varphi_i \left( \Theta \right) \right) \preccurlyeq L \cdot I_{dM}. \tag{212}$$

The last equation proves that $\varphi_i$ is $L$-smooth. Thus $r_t^k$ is $L$-smooth with respect to $\Theta$ as the average of $L$-smooth function.

Moreover, since $r_t^k(\Theta^{k-1}, \pi_t^{k-1}) = 0$ and $\forall \Theta, \Pi;$ $r_t^k(\Theta, \pi_t) \geq 0$, it follows that $\Theta^{k-1}$ is a minimizer of $\left\{ \Theta \mapsto r_t^k \left( \Theta, \pi_t^{k-1} \right) \right\}$. Thus, $\nabla_\Theta r_t^k(\Theta^{k-1}, \pi_t^{k-1}) = 0$.

For $\Theta \in \mathbb{R}^{M \times d}$ and $\Pi \in \Delta^{T \times M}$, we have

$$r^k \left( \Theta, \Pi \right) \triangleq g^k \left( \Theta, \Pi \right) - f \left( \Theta, \Pi \right) \tag{213}$$

$$\triangleq \sum_{t=1}^{T} \frac{n_t}{n} \cdot \left[ g_t^k \left( \Theta, \pi_t \right) - f_t \left( \Theta, \pi_t \right) \right] \tag{214}$$

$$= \sum_{t=1}^{T} \frac{n_t}{n} r_t^k \left( \Theta, \pi_t \right). \tag{215}$$

We see that $r^k$ is a weighted average of $\left( r_t^k \right)_{1 \leq t \leq T}$. Thus, $r_t^k$ is $L$-smooth in $\Theta$, $r^k \left( \Theta, \Pi \right) \geq 0$, moreover $r_t^k \left( \Theta^{k-1}, \Pi^{k-1} \right) = 0$ and $\nabla_\Theta r_t^k \left( \Theta^{k-1}, \Pi^{k-1} \right) = 0$. $\qquad\square$

The following lemma shows that $g_t^k$ and $g^k$ satisfy the third property in Definition 1.

**Lemma G.6.** *Suppose that Assumption 1 holds and consider $\Theta \in \mathbb{R}^{M \times d}$ and $\Pi \in \Delta^{T \times M}$, for $k > 0$, the iterates of Alg. 3 verify*

$$g^k (\Theta, \Pi) = g^k (\Theta, \Pi^k) + \sum_{t=1}^{T} \frac{n_t}{n} \mathcal{KL} (\pi_t^k, \pi_t).$$

*Proof.* For $t \in [T]$ and $k > 0$, consider $\Theta \in \mathbb{R}^{M \times d}$ and $\pi_t \in \Delta^M$ such that $\forall m \in [M]; \pi_{tm} \neq 0$, we have

$$g_t^k (\Theta, \pi_t) - g_t^k (\Theta, \pi_t^k) = \sum_{m=1}^{M} \underbrace{\left\{ \frac{1}{n_t} \sum_{i=1}^{n_t} q_t^k \left( z_t^{(i)} = m \right) \right\}}_{= \pi_{tm}^k \ \text{(Proposition 3.1)}} \times \left( \log \pi_{tm}^k - \log \pi_{tm} \right) \quad (216)$$

$$= \sum_{m=1}^{M} \pi_{tm}^k \log \frac{\pi_{tm}^k}{\pi_{tm}} \quad (217)$$

$$= \mathcal{KL} (\pi_t^k, \pi_t). \quad (218)$$

We multiply by $\frac{n_t}{n}$ and some for $t \in [T]$. It follows that

$$g^k (\Theta, \Pi^k) + \sum_{t=1}^{T} \frac{n_t}{n} \mathcal{KL} (\pi_t^k, \pi_t) = g^k (\Theta, \Pi). \quad (219)$$

$\square$

## G.2   Fully Decentralized Setting

### G.2.1   Additional Notations

**Remark 3.** *For convenience and without loss of generality, we suppose in this section that $\omega_t = 1$, $t \in [T]$.*

We introduce the following matrix notation:

$$\mathbf{U}^k \triangleq \left[ \mathbf{u}_1^k, \ldots, \mathbf{u}_T^k \right] \in \mathbb{R}^{d_u \times T} \quad (220)$$

$$\bar{\mathbf{U}}^k \triangleq \left[ \bar{\mathbf{u}}^k, \ldots, \bar{\mathbf{u}}^k \right] \in \mathbb{R}^{d_u \times T} \quad (221)$$

$$\partial g^k \left( \mathbf{U}^k, \mathbf{v}_{1:T}^k; \xi^k \right) \triangleq \left[ \nabla_{\mathbf{u}} g_1^k \left( \mathbf{u}_1^k, \mathbf{v}_1^k; \xi_1^k \right), \ldots, \nabla_{\mathbf{u}} g_T^k \left( \mathbf{u}_T^k, \mathbf{v}_T^k; \xi_T^k \right) \right] \in \mathbb{R}^{d_u \times T} \quad (222)$$

where $\bar{\mathbf{u}}^k = \frac{1}{T} \sum_{t=1}^{T} \mathbf{u}_t^k$ and $\mathbf{v}_{1:T}^k = \left( \mathbf{v}_t^k \right)_{1 \leq t \leq T} \in \mathcal{V}^T$.

We denote by $\mathbf{u}_t^{k-1,j}$ the $j$-th iterate of the local solver at global iteration $k$ at client $t \in [T]$, and by $\mathbf{U}^{k-1,j}$ the matrix whose column $t$ is $\mathbf{u}_t^{k-1,j}$, thus,

$$\mathbf{u}_t^{k-1,0} = \mathbf{u}_t^{k-1}; \qquad \mathbf{U}^{k-1,0} = \mathbf{U}^{k-1}, \quad (223)$$

and,

$$\mathbf{u}_t^k = \sum_{s=1}^{T} w_{st}^{k-1} \mathbf{u}_s^{k-1,J}; \qquad \mathbf{U}^k = \mathbf{U}^{k-1,J} W^{k-1}. \quad (224)$$

Using this notation, the updates of Alg. 5 can be summarized as

$$\mathbf{U}^k = \left[ \mathbf{U}^{k-1} - \sum_{j=0}^{J-1} \eta_{k-1,j} \partial g^k \left( \mathbf{U}^{k-1,j}, \mathbf{v}_{1:T}; \xi^{k-1,j} \right) \right] W^{k-1}. \quad (225)$$

Similarly to the client-server setting, we define the normalized update of local solver at client $t \in [T]$:

$$\hat{\delta}_t^{k-1} \triangleq -\frac{\mathbf{u}_t^{k-1,J} - \mathbf{u}_t^{k-1,0}}{\eta_{k-1}} = \frac{\sum_{j=0}^{J-1} \eta_{k-1,j} \nabla_{\mathbf{u}} g_t^k \left( \mathbf{u}_t^{k-1,j}, \mathbf{v}_t^k; \xi_t^{k-1,j} \right)}{\sum_{j=0}^{J-1} \eta_{k-1,j}}, \quad (226)$$

and

$$\delta_t^{k-1} \triangleq \frac{\sum_{j=0}^{J-1} \eta_{k-1,j} \nabla_{\mathbf{u}} g_t^k \left(\mathbf{u}_t^{k-1,j}, \mathbf{v}_t^k\right)}{\eta_{k-1}}. \tag{227}$$

Because clients updates are independent, and stochastic gradient are unbiased, it is clear that

$$\mathbb{E}\left[\delta_t^{k-1} - \hat{\delta}_t^{k-1}\right] = 0, \tag{228}$$

and that

$$\forall\, t, s \in [T] \text{ s.t. } s \neq t, \ \ \mathbb{E}\langle \delta_t^{k-1} - \hat{\delta}_t^{k-1}, \delta_s^{k-1} - \hat{\delta}_s^{k-1} \rangle = 0. \tag{229}$$

We introduce the matrix notation,

$$\hat{\Upsilon}^{k-1} \triangleq \left[\hat{\delta}_1^{k-1}, \ldots, \hat{\delta}_T^{k-1}\right] \in \mathbb{R}^{d_u \times T}; \qquad \Upsilon^{k-1} \triangleq \left[\delta_1^{k-1}, \ldots, \delta_T^{k-1}\right] \in \mathbb{R}^{d_u \times T}. \tag{230}$$

Using this notation, Eq. (225) becomes

$$\mathbf{U}^k = \left[\mathbf{U}^{k-1} - \eta_{k-1}\hat{\Upsilon}^{k-1}\right] W^{k-1}. \tag{231}$$

### G.2.2 Proof of Theorem 3.3′

In fully decentralized optimization, proving the convergence usually consists in deriving a recurrence on a term measuring the optimality of the average iterate (in our case this term is $\mathbb{E}\left\|\nabla_{\mathbf{u}} f\left(\bar{\mathbf{u}}^k, \mathbf{v}_{1:T}^k\right)\right\|^2$) and a term measuring the distance to consensus, i.e., $\mathbb{E}\sum_{t=1}^T \left\|\mathbf{u}_t^k - \bar{\mathbf{u}}^k\right\|^2$. In what follows we obtain those two recurrences, and then prove the convergence.

**Lemma G.7** (Average iterate term recursion). *Suppose that Assumptions 5′–7′ and Assumption 8 hold. Then, for $k > 0$, and $(\eta_{k,j})_{1 \leq j \leq J-1}$ such that $\eta_k \triangleq \sum_{j=0}^{J-1} \eta_{k,j} \leq \min\left\{\frac{1}{2\sqrt{2}L}, \frac{1}{8L\beta}\right\}$, the updates of fully decentralized federated surrogate optimization (Alg. 5) verify*

$$\mathbb{E}\left[f(\bar{\mathbf{u}}^k, \mathbf{v}_{1:T}^k) - f(\bar{\mathbf{u}}^{k-1}, \mathbf{v}_{1:T}^{k-1})\right] \leq -\frac{1}{T}\sum_{t=1}^T \mathbb{E}\, d_{\mathcal{V}}\left(\mathbf{v}_t^k, \mathbf{v}_t^{k-1}\right)$$

$$- \frac{\eta_{k-1}}{8} \mathbb{E}\left\|\nabla_{\mathbf{u}} f\left(\bar{\mathbf{u}}^{k-1}, \mathbf{v}_{1:T}^{k-1}\right)\right\|^2 + \frac{(12+T)\eta_{k-1}L^2}{4T} \cdot \sum_{t=1}^T \mathbb{E}\left\|\mathbf{u}_t^{k-1} - \bar{\mathbf{u}}^{k-1}\right\|^2$$

$$+ \frac{\eta_{k-1}^2 L}{T}\left(4\sum_{j=0}^{J-1}\frac{L \cdot \eta_{k-1,j}^2}{\eta_{k-1}} + 1\right)\sigma^2 + \frac{16\eta_{k-1}^3 L^2}{T}G^2. \tag{232}$$

*Proof.* We multiply both sides of Eq. (231) by $\frac{\mathbf{1}\mathbf{1}^\top}{T}$, thus for $k > 0$ we have,

$$\mathbf{U}^k \cdot \frac{\mathbf{1}\mathbf{1}^\top}{T} = \left[\mathbf{U}^{k-1} - \eta_{k-1}\hat{\Upsilon}^{k-1}\right] W^{k-1}\frac{\mathbf{1}\mathbf{1}^\top}{T}, \tag{233}$$

since $W^{k-1}$ is doubly stochastic (Assumption 8), i.e., $W^{k-1}\frac{\mathbf{1}\mathbf{1}^\top}{T} = \frac{\mathbf{1}\mathbf{1}^\top}{T}$, is follows that,

$$\bar{\mathbf{U}}^k = \bar{\mathbf{U}}^{k-1} - \eta_{k-1}\hat{\Upsilon}^{k-1} \cdot \frac{\mathbf{1}\mathbf{1}^\top}{T}, \tag{234}$$

thus,

$$\bar{\mathbf{u}}^k = \bar{\mathbf{u}}^{k-1} - \frac{\eta_{k-1}}{T} \cdot \sum_{t=1}^T \hat{\delta}_t^{k-1}. \tag{235}$$

Using the fact that $g^k$ is $L$-smooth with respect to $\mathbf{u}$ (Assumption 5′), we write

$$\mathbb{E}\left[g^k\left(\bar{\mathbf{u}}^k, \mathbf{v}_{1:T}^{k-1}\right)\right] = \mathbb{E}\left[g^k\left(\bar{\mathbf{u}}^{k-1} - \frac{\eta_{k-1}}{T}\sum_{t=1}^T \hat{\delta}_t^{k-1}, \mathbf{v}_{1:T}^{k-1}\right)\right] \tag{236}$$

$$\leq g^k(\bar{\mathbf{u}}^{k-1}, \mathbf{v}_{1:T}^{k-1}) - \mathbb{E}\left\langle \nabla_{\mathbf{u}} g^k(\bar{\mathbf{u}}^{k-1}, \mathbf{v}_{1:T}^{k-1}), \frac{\eta_{k-1}}{T}\sum_{t=1}^T \hat{\delta}_t^{k-1}\right\rangle$$

$$+ \frac{L}{2} \mathbb{E} \left\| \frac{\eta_{k-1}}{T} \sum_{t=1}^{T} \hat{\delta}_t^{k-1} \right\|^2 \tag{237}$$

$$= g^k(\bar{\mathbf{u}}^{k-1}, \mathbf{v}_{1:T}^{k-1}) - \eta_{k-1} \underbrace{\mathbb{E} \left\langle \nabla_{\mathbf{u}} g^k(\bar{\mathbf{u}}^{k-1}, \mathbf{v}_{1:T}^{k-1}), \frac{1}{T} \sum_{t=1}^{T} \hat{\delta}_t^{k-1} \right\rangle}_{\triangleq T_1}$$

$$+ \frac{\eta_{k-1}^2 \cdot L}{2T^2} \underbrace{\mathbb{E} \left\| \sum_{t=1}^{T} \hat{\delta}_t^{k-1} \right\|^2}_{\triangleq T_2}, \tag{238}$$

where the expectation is taken over local random batches. As in the client-server case, we bound the terms $T_1$ and $T_2$. First, we bound $T_1$, for $k > 0$, we have

$$T_1 = \mathbb{E} \left\langle \nabla_{\mathbf{u}} g^k(\bar{\mathbf{u}}^{k-1}, \mathbf{v}_{1:T}^{k-1}), \frac{1}{T} \sum_{t=1}^{T} \hat{\delta}_t^{k-1} \right\rangle \tag{239}$$

$$= \underbrace{\mathbb{E} \left\langle \nabla_{\mathbf{u}} g^k \left( \bar{\mathbf{u}}^{k-1}, \mathbf{v}_{1:T}^{k-1} \right), \frac{1}{T} \sum_{t=1}^{T} \left( \hat{\delta}_t^{k-1} - \delta_t^{k-1} \right) \right\rangle}_{=0, \text{ because } \mathbb{E}[\delta_t^{k-1} - \hat{\delta}_t^{k-1}] = 0}$$

$$+ \mathbb{E} \left\langle \nabla_{\mathbf{u}} g^k \left( \bar{\mathbf{u}}^{k-1}, \mathbf{v}_{1:T}^{k-1} \right), \frac{1}{T} \sum_{t=1}^{T} \delta_t^{k-1} \right\rangle \tag{240}$$

$$= \mathbb{E} \left\langle \nabla_{\mathbf{u}} g^k \left( \bar{\mathbf{u}}^{k-1}, \mathbf{v}_{1:T}^{k-1} \right), \frac{1}{T} \sum_{t=1}^{T} \delta_t^{k-1} \right\rangle \tag{241}$$

$$= \frac{1}{2} \mathbb{E} \left\| \nabla_{\mathbf{u}} g^k \left( \bar{\mathbf{u}}^{k-1}, \mathbf{v}_{1:T}^{k-1} \right) \right\|^2 + \frac{1}{2} \mathbb{E} \left\| \frac{1}{T} \sum_{t=1}^{T} \delta_t^{k-1} \right\|^2$$

$$- \frac{1}{2} \mathbb{E} \left\| \nabla_{\mathbf{u}} g^k \left( \bar{\mathbf{u}}^{k-1}, \mathbf{v}_{1:T}^{k-1} \right) - \frac{1}{T} \sum_{t=1}^{T} \delta_t^{k-1} \right\|^2. \tag{242}$$

We bound now $T_2$. For $k > 0$, we have,

$$T_2 = \mathbb{E} \left\| \sum_{t=1}^{T} \hat{\delta}_t^{k-1} \right\|^2 \tag{243}$$

$$= \mathbb{E} \left\| \sum_{t=1}^{T} \left( \hat{\delta}_t^{k-1} - \delta_t^{k-1} \right) + \sum_{t=1}^{T} \delta_t^{k-1} \right\|^2 \tag{244}$$

$$\leq 2\mathbb{E} \left\| \sum_{t=1}^{T} \left( \hat{\delta}_t^{k-1} - \delta_t^{k-1} \right) \right\|^2 + 2 \cdot \mathbb{E} \left\| \sum_{t=1}^{T} \delta_t^{k-1} \right\|^2 \tag{245}$$

$$= 2 \cdot \sum_{t=1}^{T} \mathbb{E} \left\| \hat{\delta}_t^{k-1} - \delta_t^{k-1} \right\|^2 + 2 \sum_{1 \leq t \neq s \leq T} \underbrace{\mathbb{E} \left\langle \hat{\delta}_t^{k-1} - \delta_t^{k-1}, \hat{\delta}_s^{k-1} - \delta_s^{k-1} \right\rangle}_{=0; \text{ because of Eq. (229)}}$$

$$+ 2\mathbb{E} \left\| \sum_{t=1}^{T} \delta_t^{k-1} \right\|^2 \tag{246}$$

$$= 2 \cdot \sum_{t=1}^{T} \mathbb{E} \left\| \hat{\delta}_t^{k-1} - \delta_t^{k-1} \right\|^2 + 2 \cdot \mathbb{E} \left\| \sum_{t=1}^{T} \delta_t^{k-1} \right\|^2 \tag{247}$$

$$= 2 \cdot \mathbb{E} \left\| \sum_{t=1}^{T} \delta_t^{k-1} \right\|^2 + 2 \cdot \sum_{t=1}^{T} \left( \frac{1}{\eta_{k-1}^2} \mathbb{E} \left\| \sum_{j=0}^{J-1} \eta_{k-1,j} \cdot \left[ \nabla_{\mathbf{u}} g_t^k \left( \mathbf{u}_t^{k-1,j}, \mathbf{v}_t^{k-1} \right) \right. \right. \right.$$
$$\left. \left. \left. - \nabla_{\mathbf{u}} g_t^k \left( \mathbf{u}_t^{k-1,j}, \mathbf{v}_t^{k-1}; \xi_t^{k-1,j} \right) \right] \right\|^2 \right). \tag{248}$$

Since batches are sampled independently, and stochastic gradients are unbiased with finite variance (Assumption 6′), the last term in the RHS of the previous equation can be bounded using $\sigma^2$, leading to

$$T_2 \leq 2 \cdot \sum_{t=1}^{T} \left[ \frac{\sum_{j=0}^{J-1} \eta_{k-1,j}^2}{\eta_{k-1}^2} \sigma^2 \right] + 2 \cdot \mathbb{E} \left\| \sum_{t=1}^{T} \delta_t^{k-1} \right\|^2 \tag{249}$$

$$= 2T \cdot \sigma^2 \cdot \left( \sum_{t=1}^{T} \cdot \frac{\sum_{j=0}^{J-1} \eta_{k-1,j}^2}{\eta_{k-1}^2} \right) + 2\mathbb{E} \left\| \sum_{t=1}^{T} \delta_t^{k-1} \right\|^2 \tag{250}$$

$$\leq 2T \cdot \sigma^2 + 2 \cdot \mathbb{E} \left\| \sum_{t=1}^{T} \delta_t^{k-1} \right\|^2. \tag{251}$$

Replacing Eq. (242) and Eq. (251) in Eq. (238), we have

$$\mathbb{E} \left[ g^k(\bar{\mathbf{u}}^k, \mathbf{v}_{1:T}^{k-1}) - g^k(\bar{\mathbf{u}}^{k-1}, \mathbf{v}_{1:T}^{k-1}) \right] \leq$$
$$- \frac{\eta_{k-1}}{2} \mathbb{E} \left\| \nabla_{\mathbf{u}} g^k \left( \bar{\mathbf{u}}^{k-1}, \mathbf{v}_{1:T}^{k-1} \right) \right\|^2 - \frac{\eta_{k-1}}{2} \left( 1 - 2L\eta_{k-1} \right) \mathbb{E} \left\| \frac{1}{T} \sum_{t=1}^{T} \delta_t^{k-1} \right\|^2$$
$$+ \frac{L}{T} \eta_{k-1}^2 \sigma^2 + \frac{\eta_{k-1}}{2} \mathbb{E} \left\| \nabla_{\mathbf{u}} g^k \left( \bar{\mathbf{u}}^{k-1}, \mathbf{v}_{1:T}^{k-1} \right) - \frac{1}{T} \sum_{t=1}^{T} \delta_t^{k-1} \right\|^2. \tag{252}$$

For $\eta_{k-1}$ small enough, in particular for $\eta_{k-1} \leq \frac{1}{2L}$, we have

$$\mathbb{E} \left[ g^k(\bar{\mathbf{u}}^k, \mathbf{v}_{1:T}^{k-1}) - g^k(\bar{\mathbf{u}}^{k-1}, \mathbf{v}_{1:T}^{k-1}) \right] \leq$$
$$- \frac{\eta_{k-1}}{2} \mathbb{E} \left\| \nabla_{\mathbf{u}} g^k \left( \bar{\mathbf{u}}^{k-1}, \mathbf{v}_{1:T}^{k-1} \right) \right\|^2 + \frac{L}{T} \eta_{k-1}^2 \sigma^2$$
$$+ \frac{\eta_{k-1}}{2} \mathbb{E} \left\| \frac{1}{T} \sum_{t=1}^{T} \left( \nabla_{\mathbf{u}} g_t^k \left( \bar{\mathbf{u}}^{k-1}, \mathbf{v}_t^{k-1} \right) - \delta_t^{k-1} \right) \right\|^2. \tag{253}$$

We use Jensen inequality to bound the last term in the RHS of the previous equation, leading to

$$\mathbb{E} \left[ g^k(\bar{\mathbf{u}}^k, \mathbf{v}_{1:T}^{k-1}) - g^k(\bar{\mathbf{u}}^{k-1}, \mathbf{v}_{1:T}^{k-1}) \right] \leq$$
$$- \frac{\eta_{k-1}}{2} \mathbb{E} \left\| \nabla_{\mathbf{u}} g^k \left( \bar{\mathbf{u}}^{k-1}, \mathbf{v}_{1:T}^{k-1} \right) \right\|^2 + \frac{L}{T} \eta_{k-1}^2 \sigma^2$$
$$+ \frac{\eta_{k-1}}{2T} \cdot \sum_{t=1}^{T} \underbrace{\mathbb{E} \left\| \nabla_{\mathbf{u}} g_t^k \left( \bar{\mathbf{u}}^{k-1}, \mathbf{v}_t^{k-1} \right) - \delta_t^{k-1} \right\|^2}_{T_3}. \tag{254}$$

We bound now the term $T_3$:

$$T_3 = \mathbb{E} \left\| \nabla_{\mathbf{u}} g_t^k \left( \bar{\mathbf{u}}^{k-1}, \mathbf{v}_t^{k-1} \right) - \delta_t^{k-1} \right\|^2 \tag{255}$$

$$= \mathbb{E} \left\| \nabla_{\mathbf{u}} g_t^k \left( \bar{\mathbf{u}}^{k-1}, \mathbf{v}_t^{k-1} \right) - \frac{\sum_{j=0}^{J-1} \eta_{k-1,j} \cdot \nabla_{\mathbf{u}} g_t^k \left( \mathbf{u}_t^{k-1,j}, \mathbf{v}_t^{k-1} \right)}{\eta_{k-1}} \right\|^2 \tag{256}$$

$$= \mathbb{E} \left\| \sum_{j=0}^{J-1} \frac{\eta_{k-1,j}}{\eta_{k-1}} \cdot \left[ \nabla_{\mathbf{u}} g_t^k \left( \bar{\mathbf{u}}^{k-1}, \mathbf{v}_t^{k-1} \right) - \nabla_{\mathbf{u}} g_t^k \left( \mathbf{u}_t^{k-1,j}, \mathbf{v}_t^{k-1} \right) \right] \right\|^2. \tag{257}$$

Using Jensen inequality, it follows that

$$T_3 \le \sum_{j=0}^{J-1} \frac{\eta_{k-1,j}}{\eta_{k-1}} \cdot \mathbb{E} \left\| \nabla_{\mathbf{u}} g_t^k \left( \bar{\mathbf{u}}^{k-1}, \mathbf{v}_t^{k-1} \right) - \nabla_{\mathbf{u}} g_t^k \left( \mathbf{u}_t^{k-1,j}, \mathbf{v}_t^{k-1} \right) \right\|^2 \tag{258}$$

$$= \sum_{j=0}^{J-1} \frac{\eta_{k-1,j}}{\eta_{k-1}} \cdot \mathbb{E} \left\| \nabla_{\mathbf{u}} g_t^k \left( \bar{\mathbf{u}}^{k-1}, \mathbf{v}_t^{k-1} \right) - \nabla_{\mathbf{u}} g_t^k \left( \mathbf{u}_t^{k-1}, \mathbf{v}_t^{k-1} \right) \right. $$

$$\left. + \nabla_{\mathbf{u}} g_t^k \left( \mathbf{u}_t^{k-1}, \mathbf{v}_t^{k-1} \right) - \nabla_{\mathbf{u}} g_t^k \left( \mathbf{u}_t^{k-1,j}, \mathbf{v}_t^{k-1} \right) \right\|^2 \tag{259}$$

$$\le 2 \cdot \mathbb{E} \left\| \nabla_{\mathbf{u}} g_t^k \left( \bar{\mathbf{u}}^{k-1}, \mathbf{v}_t^{k-1} \right) - \nabla_{\mathbf{u}} g_t^k \left( \mathbf{u}_t^{k-1}, \mathbf{v}_t^{k-1} \right) \right\|^2 $$

$$+ 2 \cdot \sum_{j=0}^{J-1} \frac{\eta_{k-1,j}}{\eta_{k-1}} \cdot \mathbb{E} \left\| \nabla_{\mathbf{u}} g_t^k \left( \mathbf{u}_t^{k-1}, \mathbf{v}_t^{k-1} \right) - \nabla_{\mathbf{u}} g_t^k \left( \mathbf{u}_t^{k-1,j}, \mathbf{v}_t^{k-1} \right) \right\|^2 \tag{260}$$

$$\le 2L^2 \cdot \mathbb{E} \left\| \bar{\mathbf{u}}^{k-1} - \mathbf{u}_t^{k-1} \right\|^2 + 2L^2 \cdot \sum_{j=0}^{J-1} \frac{\eta_{k-1,j}}{\eta_{k-1}} \cdot \mathbb{E} \left\| \mathbf{u}_t^{k-1,j} - \mathbf{u}_t^{k-1,0} \right\|^2, \tag{261}$$

where we used the $L$-smoothness of $g_t^k$ (Assumption 5$'$) to obtain the last inequality. As in the centralized case (Lemma G.1), we bound terms $\left\| \mathbf{u}_t^{k-1,j} - \mathbf{u}_t^{k-1,0} \right\|^2$, $j \in \{0, \ldots, J-1\}$. Using exactly the same steps as in the proof of Lemma G.1, Eq. (143) holds with $\mathbf{u}_t^{k-1,0}$ instead of $\mathbf{u}_t^{k-1}$, i.e.,

$$\left( 1 - 4\eta_{k-1}^2 L^2 \right) \cdot \sum_{j=0}^{J-1} \frac{\eta_{k-1,j}}{\eta_{k-1}} \cdot \mathbb{E} \left\| \mathbf{u}_t^{k-1,0} - \mathbf{u}_t^{k-1,j} \right\|^2 \le 2\sigma^2 \cdot \left\{ \sum_{j=0}^{J-1} \eta_{k-1,j}^2 \right\} $$

$$+ 4\eta_{k-1}^2 \cdot \mathbb{E} \left\| \nabla_u g_t^k \left( \mathbf{u}_t^{k-1,0}, \mathbf{v}_t^{k-1} \right) \right\|^2. \tag{262}$$

For $\eta_{k-1}$ small enough, in particular for $\eta_{k-1} \le \frac{1}{2\sqrt{2}L}$, we have

$$\sum_{j=0}^{J-1} \frac{\eta_{k-1,j}}{\eta_{k-1}} \cdot \mathbb{E} \left\| \mathbf{u}_t^{k-1,0} - \mathbf{u}_t^{k-1,j} \right\|^2$$

$$\le 8\eta_{k-1}^2 \cdot \mathbb{E} \left\| \nabla_u g_t^k \left( \mathbf{u}_t^{k-1,0}, \mathbf{v}_t^{k-1} \right) \right\|^2 + 4\sigma^2 \cdot \left\{ \sum_{j=0}^{J-1} \eta_{k-1,j}^2 \right\} \tag{263}$$

$$\le 8\eta_{k-1}^2 \cdot \mathbb{E} \left\| \nabla_u g_t^k \left( \mathbf{u}_t^{k-1,0}, \mathbf{v}_t^{k-1} \right) - \nabla_u g_t^k \left( \bar{\mathbf{u}}^{k-1}, \mathbf{v}_t^{k-1} \right) + \nabla_u g_t^k \left( \bar{\mathbf{u}}^{k-1}, \mathbf{v}_t^{k-1} \right) \right\|^2 $$

$$+ 4\sigma^2 \cdot \left\{ \sum_{j=0}^{J-1} \eta_{k-1,j}^2 \right\} \tag{264}$$

$$\le 16\eta_{k-1}^2 \cdot \mathbb{E} \left\| \nabla_u g_t^k \left( \mathbf{u}_t^{k-1,0}, \mathbf{v}_t^{k-1} \right) - \nabla_u g_t^k \left( \bar{\mathbf{u}}^{k-1}, \mathbf{v}_t^{k-1} \right) \right\|^2 $$

$$+ 16\eta_{k-1}^2 \cdot \left\| \nabla_u g_t^k \left( \bar{\mathbf{u}}^{k-1}, \mathbf{v}_t^{k-1} \right) \right\|^2 + 4\sigma^2 \cdot \left\{ \sum_{j=0}^{J-1} \eta_{k-1,j}^2 \right\} \tag{265}$$

$$\le 16\eta_{k-1}^2 L^2 \cdot \mathbb{E} \left\| \mathbf{u}_t^{k-1} - \bar{\mathbf{u}}^{k-1} \right\|^2 + 16\eta_{k-1}^2 \cdot \left\| \nabla_u g_t^k \left( \bar{\mathbf{u}}^{k-1}, \mathbf{v}_t^{k-1} \right) \right\|^2$$

$$+ 4\sigma^2 \cdot \left\{ \sum_{j=0}^{J-1} \eta_{k-1,j}^2 \right\}, \tag{266}$$

where the last inequality follows from the $L$-smoothness of $g_t^k$. Replacing Eq. (266) in Eq. (261), we have

$$T_3 \leq 32\eta_{k-1}^2 L^4 \cdot \mathbb{E} \left\| \mathbf{u}_t^{k-1} - \bar{\mathbf{u}}^{k-1} \right\|^2 + 8L^2\sigma^2 \cdot \left\{ \sum_{j=0}^{J-1} \eta_{k-1,j}^2 \right\}$$
$$+ 32\eta_{k-1}^2 L^2 \cdot \mathbb{E} \left\| \nabla_u g_t^k \left( \bar{\mathbf{u}}^{k-1}, \mathbf{v}_t^{k-1} \right) \right\|^2 + 2L^2 \cdot \mathbb{E} \left\| \bar{\mathbf{u}}^{k-1} - \mathbf{u}_t^{k-1} \right\|^2. \tag{267}$$

For $\eta_k$ small enough, in particular if $\eta_k \leq \frac{1}{2\sqrt{2}L}$ we have,

$$T_3 \leq 6L^2 \mathbb{E} \left\| \mathbf{u}_t^{k-1} - \bar{\mathbf{u}}^{k-1} \right\|^2 + 8L^2\sigma^2 \sum_{j=0}^{J-1} \eta_{k-1,j}^2 + 32\eta_{k-1}^2 L^2 \left\| \nabla_u g_t^k \left( \bar{\mathbf{u}}^{k-1}, \mathbf{v}_t^{k-1} \right) \right\|^2. \tag{268}$$

Replacing Eq. (268) in Eq. (254), we have

$$\mathbb{E} \left[ g^k(\bar{\mathbf{u}}^k, \mathbf{v}_{1:T}^{k-1}) - g^k(\bar{\mathbf{u}}^{k-1}, \mathbf{v}_{1:T}^{k-1}) \right] \leq$$
$$\frac{3\eta_{k-1}L^2}{T} \cdot \sum_{t=1}^{T} \mathbb{E} \left\| \mathbf{u}_t^{k-1} - \bar{\mathbf{u}}^{k-1} \right\|^2 + \frac{\eta_{k-1}^2 L}{T} \left( 4 \sum_{j=0}^{J-1} \frac{TL \cdot \eta_{k-1,j}^2}{\eta_{k-1}} + 1 \right) \sigma^2$$
$$- \frac{\eta_{k-1}}{2} \mathbb{E} \left\| \nabla_{\mathbf{u}} g^k \left( \bar{\mathbf{u}}^{k-1}, \mathbf{v}_{1:T}^{k-1} \right) \right\|^2 + \frac{16\eta_{k-1}^3 L^2}{T} \sum_{t=1}^{T} \left\| \nabla_u g_t^k \left( \bar{\mathbf{u}}^{k-1}, \mathbf{v}_t^{k-1} \right) \right\|^2. \tag{269}$$

We use now Assumption 7′ to bound the last term in the RHS of the previous equation, leading to

$$\mathbb{E} \left[ g^k(\bar{\mathbf{u}}^k, \mathbf{v}_{1:T}^{k-1}) - g^k(\bar{\mathbf{u}}^{k-1}, \mathbf{v}_{1:T}^{k-1}) \right] \leq$$
$$\frac{3\eta_{k-1}L^2}{T} \cdot \sum_{t=1}^{T} \mathbb{E} \left\| \mathbf{u}_t^{k-1} - \bar{\mathbf{u}}^{k-1} \right\|^2 + \frac{\eta_{k-1}^2 L}{T} \left( 4 \sum_{j=0}^{J-1} \frac{TL \cdot \eta_{k-1,j}^2}{\eta_{k-1}} + 1 \right) \sigma^2$$
$$- \frac{\eta_{k-1} \cdot \left( 1 - 32\eta_{k-1}^2 L^2 \beta^2 \right)}{2} \mathbb{E} \left\| \nabla_{\mathbf{u}} g^k \left( \bar{\mathbf{u}}^{k-1}, \mathbf{v}_{1:T}^{k-1} \right) \right\|^2 + \frac{16\eta_{k-1}^3 L^2}{T} G^2. \tag{270}$$

For $\eta_{k-1}$ small enough, in particular, if $\eta_{k-1} \leq \frac{1}{8L\beta}$, we have

$$\mathbb{E} \left[ g^k(\bar{\mathbf{u}}^k, \mathbf{v}_{1:T}^{k-1}) - g^k(\bar{\mathbf{u}}^{k-1}, \mathbf{v}_{1:T}^{k-1}) \right] \leq$$
$$- \frac{\eta_{k-1}}{4} \mathbb{E} \left\| \nabla_{\mathbf{u}} g^k \left( \bar{\mathbf{u}}^{k-1}, \mathbf{v}_{1:T}^{k-1} \right) \right\|^2 + \frac{3\eta_{k-1}L^2}{T} \cdot \sum_{t=1}^{T} \mathbb{E} \left\| \mathbf{u}_t^{k-1} - \bar{\mathbf{u}}^{k-1} \right\|^2$$
$$+ \frac{\eta_{k-1}^2 L}{T} \left( 4 \sum_{j=0}^{J-1} \frac{TL \cdot \eta_{k-1,j}^2}{\eta_{k-1}} + 1 \right) \sigma^2 + \frac{16\eta_{k-1}^3 L^2}{T} G^2. \tag{271}$$

We use Lemma G.14 to get

$$\mathbb{E} \left[ g^k(\bar{\mathbf{u}}^k, \mathbf{v}_{1:T}^{k-1}) - f(\bar{\mathbf{u}}^{k-1}, \mathbf{v}_{1:T}^{k-1}) \right] \leq$$
$$- \frac{\eta_{k-1}}{8} \mathbb{E} \left\| \nabla_{\mathbf{u}} f \left( \bar{\mathbf{u}}^{k-1}, \mathbf{v}_{1:T}^{k-1} \right) \right\|^2 + \frac{(12+T)\eta_{k-1}L^2}{4T} \cdot \sum_{t=1}^{T} \mathbb{E} \left\| \mathbf{u}_t^{k-1} - \bar{\mathbf{u}}^{k-1} \right\|^2$$

$$+ \frac{\eta_{k-1}^2 L}{T} \left( 4 \sum_{j=0}^{J-1} \frac{L \cdot \eta_{k-1,j}^2}{\eta_{k-1}} + 1 \right) \sigma^2 + \frac{16\eta_{k-1}^3 L^2}{T} G^2. \tag{272}$$

Finally, since $g_t^k$ is a partial first-order surrogate of $f_t$ near $\{\mathbf{u}^{k-1}, \mathbf{v}_t^{k-1}\}$, we have

$$\mathbb{E}\left[ f(\bar{\mathbf{u}}^k, \mathbf{v}_{1:T}^k) - f(\bar{\mathbf{u}}^{k-1}, \mathbf{v}_{1:T}^{k-1}) \right] \leq -\frac{1}{T} \sum_{t=1}^{T} \mathbb{E}\, d_{\mathcal{V}}\left( \mathbf{v}_t^k, \mathbf{v}_t^{k-1} \right)$$

$$- \frac{\eta_{k-1}}{8} \mathbb{E}\left\| \nabla_{\mathbf{u}} f\left( \bar{\mathbf{u}}^{k-1}, \mathbf{v}_{1:T}^{k-1} \right) \right\|^2 + \frac{(12+T)\,\eta_{k-1} L^2}{4T} \cdot \sum_{t=1}^{T} \mathbb{E}\left\| \mathbf{u}_t^{k-1} - \bar{\mathbf{u}}^{k-1} \right\|^2$$

$$+ \frac{\eta_{k-1}^2 L}{T} \left( 4 \sum_{j=0}^{J-1} \frac{L \cdot \eta_{k-1,j}^2}{\eta_{k-1}} + 1 \right) \sigma^2 + \frac{16\eta_{k-1}^3 L^2}{T} G^2. \tag{273}$$

$\square$

**Lemma G.8** (Recursion for consensus distance, part 1). *Suppose that Assumptions 5′–7′ and Assumption 8 hold. For $k \geq \tau$, consider $m = \lfloor \frac{k}{\tau} \rfloor - 1$ and $(\eta_{k,j})_{1 \leq j \leq J-1}$ such that $\eta_k \triangleq \sum_{j=0}^{J-1} \eta_{k,j} \leq \min\left\{ \frac{1}{4L}, \frac{1}{4L\beta} \right\}$ then, the updates of fully decentralized federated surrogate optimization (Alg 5) verify*

$$\mathbb{E} \sum_{t=1}^{T} \left\| \mathbf{u}_t^k - \bar{\mathbf{u}}^k \right\|_F^2 \leq$$

$$(1 - \frac{p}{2})\mathbb{E}\left\| \mathbf{U}^{m\tau} - \bar{\mathbf{U}}^{m\tau} \right\|_F^2 + 44\tau \left( 1 + \frac{2}{p} \right) L^2 \sum_{l=m\tau}^{k-1} \eta_l^2 \, \mathbb{E}\left\| \mathbf{U}^l - \bar{\mathbf{U}}^l \right\|_F^2$$

$$+ T \cdot \sigma^2 \cdot \sum_{l=m\tau}^{k-1} \left\{ \eta_l^2 + 16\tau L^2 \left( 1 + \frac{2}{p} \right) \cdot \left\{ \sum_{j=0}^{J-1} \eta_{l,j}^2 \right\} \right\} + 16\tau \left( 1 + \frac{2}{p} \right) G^2 \sum_{l=m\tau}^{k-1} \eta_l^2$$

$$+ 16\tau \left( 1 + \frac{2}{p} \right) \beta^2 \sum_{l=m\tau}^{k-1} \eta_l^2 \, \mathbb{E}\left\| \nabla_{\mathbf{u}} f\left( \bar{\mathbf{u}}^{l,j}, \mathbf{v}_{1:T}^l \right) \right\|^2.$$

*Proof.* For $k \geq \tau$, and $m = \lfloor \frac{k}{\tau} \rfloor - 1$, we have

$$\mathbb{E} \sum_{t=1}^{T} \left\| \mathbf{u}_t^k - \bar{\mathbf{u}}^k \right\|_F^2 = \mathbb{E} \left\| \mathbf{U}^k - \bar{\mathbf{U}}^k \right\|_F^2 \tag{274}$$

$$= \mathbb{E} \left\| \mathbf{U}^k - \bar{\mathbf{U}}^{m\tau} - \left( \bar{\mathbf{U}}^k - \bar{\mathbf{U}}^{m\tau} \right) \right\|_F^2 \tag{275}$$

$$\leq \mathbb{E} \left\| \mathbf{U}^k - \bar{\mathbf{U}}^{m\tau} \right\|_F^2, \tag{276}$$

where we used the fact that $\left\| A - \bar{A} \right\|_F^2 = \left\| A \cdot \left( I - \frac{\mathbf{1}\mathbf{1}^\mathsf{T}}{T} \right) \right\|_F \leq \left\| I - \frac{\mathbf{1}\mathbf{1}^\mathsf{T}}{T} \right\|_2 \cdot \left\| A \right\|_F^2 = \left\| A \right\|_F^2$ to obtain the last inequality. Using Eq. (231) recursively, we have

$$\mathbf{U}^k = \mathbf{U}^{m\tau} \left\{ \prod_{l'=m\tau}^{k-1} W^{l'} \right\} - \sum_{l=m\tau}^{k-1} \eta_l \hat{\Upsilon}^l \left\{ \prod_{l'=l}^{k-1} W^{l'} \right\}. \tag{277}$$

Thus,

$$\mathbb{E} \sum_{t=1}^{T} \left\| \mathbf{u}_t^k - \bar{\mathbf{u}}^k \right\|_F^2 \leq \mathbb{E} \left\| \mathbf{U}^{m\tau} \left\{ \prod_{l'=m\tau}^{k-1} W^{l'} \right\} - \bar{\mathbf{U}}^{m\tau} - \sum_{l=m\tau}^{k-1} \eta_l \hat{\Upsilon}^l \left\{ \prod_{l'=l}^{k-1} W^{l'} \right\} \right\|_F^2 \tag{278}$$

$$= \mathbb{E} \left\| \mathbf{U}^{m\tau} \left\{ \prod_{l'=m\tau}^{k-1} W^{l'} \right\} - \bar{\mathbf{U}}^{m\tau} - \sum_{l=m\tau}^{k-1} \eta_l \Upsilon^l \left\{ \prod_{l'=l}^{k-1} W^{l'} \right\} \right.$$

$$+ \sum_{l=m\tau}^{k-1} \eta_l \left( \Upsilon^l - \hat{\Upsilon}^l \right) \left\{ \prod_{l'=l}^{k-1} W^{l'} \right\} \Bigg\|_F^2 \tag{279}$$

$$= \mathbb{E} \left\| \mathbf{U}^{m\tau} \left\{ \prod_{l'=m\tau}^{k-1} W^{l'} \right\} - \bar{\mathbf{U}}^{m\tau} - \sum_{l=m\tau}^{k-1} \eta_l \Upsilon^l \left\{ \prod_{l'=l}^{k-1} W^{l'} \right\} \right\|_F^2$$

$$+ \mathbb{E} \left\| \sum_{l=m\tau}^{k-1} \eta_l \left( \Upsilon^l - \hat{\Upsilon}^l \right) \left\{ \prod_{l'=l}^{k-1} W^{l'} \right\} \right\|_F^2$$

$$+ 2\mathbb{E} \left\langle \mathbf{U}^{m\tau} \left\{ \prod_{l'=m\tau}^{k-1} W^{l'} \right\} - \bar{\mathbf{U}}^{m\tau} - \sum_{l=m\tau}^{k-1} \eta_l \Upsilon^l \left\{ \prod_{l'=l}^{k-1} W^{l'} \right\} , \right.$$

$$\left. \sum_{l=m\tau}^{k-1} \eta_l \left( \Upsilon^l - \hat{\Upsilon}^l \right) \left\{ \prod_{l'=l}^{k-1} W^{l'} \right\} \right\rangle_F . \tag{280}$$

Since stochastic gradients are unbiased, the last term in the RHS of the previous equation is equal to zero. Using the following standard inequality for Euclidean norm with $\alpha > 0$,

$$\|\mathbf{a} + \mathbf{b}\|^2 \leq (1+\alpha) \|\mathbf{a}\|^2 + \left( 1 + \alpha^{-1} \right) \|\mathbf{b}\|^2 , \tag{281}$$

we have

$$\mathbb{E} \sum_{t=1}^{T} \left\| \mathbf{u}_t^k - \bar{\mathbf{u}}^k \right\|_F^2 \leq \tag{282}$$

$$(1+\alpha) \mathbb{E} \left\| \mathbf{U}^{m\tau} \left\{ \prod_{l'=m\tau}^{k-1} W^{l'} \right\} - \bar{\mathbf{U}}^{m\tau} \right\|_F^2 + \left( 1 + \alpha^{-1} \right) \mathbb{E} \left\| \sum_{l=m\tau}^{k-1} \eta_l \Upsilon^l \left\{ \prod_{l'=l}^{k-1} W^{l'} \right\} \right\|_F^2$$

$$+ \sum_{l=m\tau}^{k-1} \eta_l^2 \mathbb{E} \left\| \left( \Upsilon^l - \hat{\Upsilon}^l \right) \left\{ \prod_{l'=l}^{k-1} W^{l'} \right\} \right\|_F^2 . \tag{283}$$

Since $k \geq (m+1)\tau$ and matrices $\left( W^l \right)_{l \geq 0}$ are doubly stochastic, we have

$$\mathbb{E} \sum_{t=1}^{T} \left\| \mathbf{u}_t^k - \bar{\mathbf{u}}^k \right\|_F^2 \leq$$

$$(1+\alpha) \mathbb{E} \left\| \mathbf{U}^{m\tau} \left\{ \prod_{l'=m\tau}^{(m+1)\tau-1} W^{l'} \right\} - \bar{\mathbf{U}}^{m\tau} \right\|_F^2 + \left( 1 + \alpha^{-1} \right) \mathbb{E} \left\| \sum_{l=m\tau}^{k-1} \eta_l \Upsilon^l \right\|_F^2$$

$$+ \sum_{l=m\tau}^{k-1} \eta_l^2 \mathbb{E} \left\| \Upsilon^l - \hat{\Upsilon}^l \right\|_F^2 \tag{284}$$

$$\leq (1+\alpha) \mathbb{E} \left\| \mathbf{U}^{m\tau} \left\{ \prod_{l'=m\tau}^{(m+1)\tau-1} W^{l'} \right\} - \bar{\mathbf{U}}^{m\tau} \right\|_F^2 + \left( 1 + \alpha^{-1} \right) \cdot (k - m\tau) \sum_{l=m\tau}^{k-1} \eta_l^2 \mathbb{E} \left\| \Upsilon^l \right\|_F^2$$

$$+ \sum_{l=m\tau}^{k-1} \eta_l^2 \mathbb{E} \left\| \Upsilon^l - \hat{\Upsilon}^l \right\|_F^2 , \tag{285}$$

where we use the fact that $\|AB\|_F \leq \|A\|_2 \|B\|_F$ and that $\|A\| = 1$ when $A$ is a doubly stochastic matrix to obtain the first inequality, and Cauchy-Schwarz inequality to obtain the second one. Using Assumption 8 to bound the first term of the RHS of the previous equation and the fact that that $k \leq (m+2)\tau$, it follows that

$$\mathbb{E} \sum_{t=1}^{T} \left\| \mathbf{u}_t^k - \bar{\mathbf{u}}^k \right\|_F^2 \leq$$

$$
(1+\alpha)(1-p)\mathbb{E}\left\|\mathbf{U}^{m\tau}-\bar{\mathbf{U}}^{m\tau}\right\|_F^2 + 2\tau\left(1+\alpha^{-1}\right)\sum_{l=m\tau}^{k-1}\eta_l^2\mathbb{E}\left\|\Upsilon^l\right\|_F^2
$$

$$
+\sum_{l=m\tau}^{k-1}\eta_l^2\mathbb{E}\left\|\Upsilon^l-\hat{\Upsilon}^l\right\|_F^2. \tag{286}
$$

We use the fact that stochastic gradients have bounded variance (Assumption 6′) to bound $\mathbb{E}\left\|\Upsilon^l-\hat{\Upsilon}^l\right\|_F^2$ as follows,

$$
\mathbb{E}\left\|\Upsilon^l-\hat{\Upsilon}^l\right\|_F^2 = \sum_{t=1}^{T}\mathbb{E}\left\|\delta_t^l-\hat{\delta}_t^l\right\|^2 \tag{287}
$$

$$
= \sum_{t=1}^{T}\mathbb{E}\left\|\sum_{j=0}^{J-1}\frac{\eta_{l,j}}{\eta_l}\cdot\left(\nabla_{\mathbf{u}}g_t^{l+1}\left(\mathbf{u}_t^{l,j},\mathbf{v}_t^{k-1}\right)-\nabla_{\mathbf{u}}g_t^{l+1}\left(\mathbf{u}_t^{l,j},\mathbf{v}_t^l;\xi_t^{l,j}\right)\right)\right\|^2 \tag{288}
$$

$$
\leq \sum_{t=1}^{T}\sum_{j=0}^{J-1}\frac{\eta_{l,j}}{\eta_l}\cdot\mathbb{E}\left\|\left(\nabla_{\mathbf{u}}g_t^{l+1}\left(\mathbf{u}_t^{l,j},\mathbf{v}_t^{k-1}\right)-\nabla_{\mathbf{u}}g_t^{l+1}\left(\mathbf{u}_t^{l,j},\mathbf{v}_t^l;\xi_t^{l,j}\right)\right)\right\|^2 \tag{289}
$$

$$
\leq \sum_{t=1}^{T}\sum_{j=0}^{J-1}\frac{\eta_{l,j}}{\eta_l}\sigma^2 \tag{290}
$$

$$
= T\cdot\sigma^2, \tag{291}
$$

where we used Jensen inequality to obtain the first inequality and Assumption 6′ to obtain the second inequality. Replacing back in Eq. (286), we have

$$
\mathbb{E}\sum_{t=1}^{T}\left\|\mathbf{u}_t^k-\bar{\mathbf{u}}^k\right\|_F^2 \leq
$$

$$
(1+\alpha)(1-p)\mathbb{E}\left\|\mathbf{U}^{m\tau}-\bar{\mathbf{U}}^{m\tau}\right\|_F^2 + 2\tau\left(1+\alpha^{-1}\right)\sum_{l=m\tau}^{k-1}\eta_l^2\mathbb{E}\left\|\Upsilon^l\right\|_F^2 + T\cdot\sigma^2\cdot\left\{\sum_{l=m\tau}^{k-1}\eta_l^2\right\}. \tag{292}
$$

The last step of the proof consists in bounding $\mathbb{E}\left\|\Upsilon^l\right\|_F^2$ for $l\in\{m\tau,\dots,k-1\}$,

$$
\mathbb{E}\left\|\Upsilon^l\right\|_F^2 = \sum_{t=1}^{T}\mathbb{E}\left\|\delta_t^l\right\|^2 \tag{293}
$$

$$
= \sum_{t=1}^{T}\mathbb{E}\left\|\sum_{j=0}^{J-1}\frac{\eta_{l,j}}{\eta_l}\cdot\nabla_{\mathbf{u}}g_t^{l+1}\left(\mathbf{u}_t^{l,j},\mathbf{v}_t^l\right)\right\|^2 \tag{294}
$$

$$
\leq \sum_{t=1}^{T}\sum_{j=0}^{J-1}\frac{\eta_{l,j}}{\eta_l}\cdot\mathbb{E}\left\|\nabla_{\mathbf{u}}g_t^{l+1}\left(\mathbf{u}_t^{l,j},\mathbf{v}_t^l\right)\right\|^2 \tag{295}
$$

$$
\leq \sum_{t=1}^{T}\sum_{j=0}^{J-1}\frac{\eta_{l,j}}{\eta_l}\cdot\mathbb{E}\left\|\nabla_{\mathbf{u}}g_t^{l+1}\left(\mathbf{u}_t^{l,j},\mathbf{v}_t^l\right)-\nabla_{\mathbf{u}}f_t\left(\mathbf{u}_t^l,\mathbf{v}_t^l\right)+\nabla_{\mathbf{u}}f_t\left(\mathbf{u}_t^l,\mathbf{v}_t^l\right)\right\|^2 \tag{296}
$$

$$
\leq 2\sum_{t=1}^{T}\sum_{j=0}^{J-1}\frac{\eta_{l,j}}{\eta_l}\cdot\mathbb{E}\left\|\nabla_{\mathbf{u}}g_t^{l+1}\left(\mathbf{u}_t^{l,j},\mathbf{v}_t^l\right)-\nabla_{\mathbf{u}}f_t\left(\mathbf{u}_t^l,\mathbf{v}_t^l\right)\right\|^2
$$

$$
+2\sum_{t=1}^{T}\mathbb{E}\left\|\nabla_{\mathbf{u}}f_t\left(\mathbf{u}_t^l,\mathbf{v}_t^l\right)\right\|^2. \tag{297}
$$

Since $g_t^{l+1}$ is a first order surrogate of $f$ near $\{\mathbf{u}_t^l, \mathbf{v}_t^l\}$, we have

$$\mathbb{E}\left\|\Upsilon^l\right\|_F^2 \leq 2\sum_{t=1}^{T}\sum_{j=0}^{J-1}\frac{\eta_{l,j}}{\eta_l} \cdot \mathbb{E}\left\|\nabla_{\mathbf{u}}g_t^{l+1}\left(\mathbf{u}_t^{l,j}, \mathbf{v}_t^l\right) - \nabla_{\mathbf{u}}g_t^{l+1}\left(\mathbf{u}_t^{l,0}, \mathbf{v}_t^l\right)\right\|^2$$

$$+ 2\sum_{t=1}^{T}\mathbb{E}\left\|\nabla_{\mathbf{u}}f_t\left(\mathbf{u}_t^l, \mathbf{v}_t^l\right) - \nabla_{\mathbf{u}}f_t\left(\bar{\mathbf{u}}^l, \mathbf{v}_t^l\right) + \nabla_{\mathbf{u}}f_t\left(\bar{\mathbf{u}}^l, \mathbf{v}_t^l\right)\right\|^2 \qquad (298)$$

$$\leq 2\sum_{t=1}^{T}\sum_{j=0}^{J-1}\frac{\eta_{l,j}}{\eta_l} \cdot \mathbb{E}\left\|\nabla_{\mathbf{u}}g_t^{l+1}\left(\mathbf{u}_t^{l,j}, \mathbf{v}_t^l\right) - \nabla_{\mathbf{u}}g_t^{l+1}\left(\mathbf{u}_t^{l,0}, \mathbf{v}_t^l\right)\right\|^2$$

$$+ 4\sum_{t=1}^{T}\mathbb{E}\left\|\nabla_{\mathbf{u}}f_t\left(\mathbf{u}_t^l, \mathbf{v}_t^l\right) - \nabla_{\mathbf{u}}f_t\left(\bar{\mathbf{u}}^l, \mathbf{v}_t^l\right)\right\|^2 + 4\sum_{t=1}^{T}\mathbb{E}\left\|\nabla_{\mathbf{u}}f_t\left(\bar{\mathbf{u}}^l, \mathbf{v}_t^l\right)\right\|^2. \qquad (299)$$

Since $f$ is $2L$-smooth w.r.t $\mathbf{u}$ (Lemma G.12) and $g$ is $L$-smooth w.r.t $\mathbf{u}$ (Assumption 5′), we have

$$\mathbb{E}\left\|\Upsilon^l\right\|_F^2 \leq 2\sum_{t=1}^{T}\sum_{j=0}^{J-1}\frac{\eta_{l,j}}{\eta_l} \cdot L^2\,\mathbb{E}\left\|\mathbf{u}_t^{l,j} - \mathbf{u}_t^{l,0}\right\|^2 + 16L^2 \cdot \sum_{t=1}^{T}\mathbb{E}\left\|\mathbf{u}_t^l - \bar{\mathbf{u}}^l\right\|^2$$

$$+ 4\sum_{t=1}^{T}\mathbb{E}\left\|\nabla_{\mathbf{u}}f_t\left(\bar{\mathbf{u}}^l, \mathbf{v}_t^l\right)\right\|^2. \qquad (300)$$

We use Eq. (266) to bound the first term in the RHS of the previous equation, leading to

$$\mathbb{E}\left\|\Upsilon^l\right\|_F^2 \leq 32\eta_l^2 L^2\sum_{t=1}^{T}\mathbb{E}\left\|\nabla_{\mathbf{u}}g_t^{l+1}\left(\bar{\mathbf{u}}^{l,j}, \mathbf{v}_t^l\right)\right\|^2 + 16L^2\left(1 + 2\eta_l^2 L^2\right) \cdot \sum_{t=1}^{T}\mathbb{E}\left\|\mathbf{u}_t^l - \bar{\mathbf{u}}^l\right\|^2$$

$$+ 4\sum_{t=1}^{T}\mathbb{E}\left\|\nabla_{\mathbf{u}}f_t\left(\bar{\mathbf{u}}^l, \mathbf{v}_t^l\right)\right\|^2 + 8TL^2\sigma^2 \cdot \left\{\sum_{j=0}^{J-1}\eta_{l,j}^2\right\}. \qquad (301)$$

Using Lemma G.14, we have

$$\mathbb{E}\left\|\Upsilon^l\right\|_F^2 \leq 4\left(1 + 16\eta_l^2 L^2\right) \cdot \sum_{t=1}^{T}\mathbb{E}\left\|\nabla_{\mathbf{u}}f_t\left(\bar{\mathbf{u}}^{l,j}, \mathbf{v}_t^l\right)\right\|^2$$

$$+ 16L^2\left(1 + 6\eta_l^2 L^2\right) \cdot \sum_{t=1}^{T}\mathbb{E}\left\|\mathbf{u}_t^l - \bar{\mathbf{u}}^l\right\|^2 + 8L^2\sigma^2 T \cdot \left\{\sum_{j=0}^{J-1}\eta_{l,j}^2\right\}. \qquad (302)$$

For $\eta_l$ small enough, in particular, for $\eta_l \leq \frac{1}{4L}$, we have

$$\mathbb{E}\left\|\Upsilon^l\right\|_F^2 \leq 8\sum_{t=1}^{T}\mathbb{E}\left\|\nabla_{\mathbf{u}}f_t\left(\bar{\mathbf{u}}^{l,j}, \mathbf{v}_t^l\right)\right\|^2 + 22L^2\,\mathbb{E}\left\|\mathbf{U}^l - \bar{\mathbf{U}}^l\right\|_F^2 + 8L^2\sigma^2 T\left\{\sum_{j=0}^{J-1}\eta_{l,j}^2\right\}. \qquad (303)$$

Replacing Eq. (303) in Eq. (292), we have

$$\mathbb{E}\sum_{t=1}^{T}\left\|\mathbf{u}_t^k - \bar{\mathbf{u}}^k\right\|_F^2 \leq$$

$$(1+\alpha)(1-p)\mathbb{E}\left\|\mathbf{U}^{m\tau} - \bar{\mathbf{U}}^{m\tau}\right\|_F^2 + 44\tau\left(1 + \alpha^{-1}\right)L^2\sum_{l=m\tau}^{k-1}\eta_l^2\,\mathbb{E}\left\|\mathbf{U}^l - \bar{\mathbf{U}}^l\right\|_F^2$$

$$+ 16\tau\left(1 + \alpha^{-1}\right)\sum_{l=m\tau}^{k-1}\eta_l^2\sum_{t=1}^{T}\mathbb{E}\left\|\nabla_{\mathbf{u}}f_t\left(\bar{\mathbf{u}}^{l,j}, \mathbf{v}_t^l\right)\right\|^2$$

$$+ T \cdot \sigma^2 \cdot \sum_{l=m\tau}^{k-1}\left\{\eta_l^2 + 16\tau L^2\left(1 + \alpha^{-1}\right) \cdot \left\{\sum_{j=0}^{J-1}\eta_{l,j}^2\right\}\right\}. \qquad (304)$$

Using Lemma G.13 and considering $\alpha = \frac{p}{2}$, we have

$$\mathbb{E} \sum_{t=1}^{T} \left\| \mathbf{u}_t^k - \bar{\mathbf{u}}^k \right\|_F^2 \leq$$

$$(1 - \frac{p}{2})\mathbb{E} \left\| \mathbf{U}^{m\tau} - \bar{\mathbf{U}}^{m\tau} \right\|_F^2 + 44\tau \left( 1 + \frac{2}{p} \right) L^2 \sum_{l=m\tau}^{k-1} \eta_l^2 \, \mathbb{E} \left\| \mathbf{U}^l - \bar{\mathbf{U}}^l \right\|_F^2$$

$$+ T \cdot \sigma^2 \cdot \sum_{l=m\tau}^{k-1} \left\{ \eta_l^2 + 16\tau L^2 \left( 1 + \frac{2}{p} \right) \cdot \left\{ \sum_{j=0}^{J-1} \eta_{l,j}^2 \right\} \right\} + 16\tau \left( 1 + \frac{2}{p} \right) G^2 \sum_{l=m\tau}^{k-1} \eta_l^2$$

$$+ 16\tau \left( 1 + \frac{2}{p} \right) \beta^2 \sum_{l=m\tau}^{k-1} \eta_l^2 \, \mathbb{E} \left\| \nabla_{\mathbf{u}} f \left( \bar{\mathbf{u}}^{l,j}, \mathbf{v}_{1:T}^l \right) \right\|^2. \tag{305}$$

$$\square$$

**Lemma G.9** (Recursion for consensus distance, part 2). *Suppose that Assumptions 5′–7′ and Assumption 8 hold. Consider $m = \lfloor \frac{k}{\tau} \rfloor$, then, for $(\eta_{k,j})_{1 \leq j \leq J-1}$ such that $\eta_k \triangleq \sum_{j=0}^{J-1} \eta_{k,j} \leq \min\left\{ \frac{1}{4L}, \frac{1}{4L\beta} \right\}$, the updates of fully decentralized federated surrogate optimization (Alg 5) verify*

$$\mathbb{E} \sum_{t=1}^{T} \left\| \mathbf{u}_t^k - \bar{\mathbf{u}}^k \right\|_F^2 \leq$$

$$(1 + \frac{p}{2})\mathbb{E} \left\| \mathbf{U}^{m\tau} - \bar{\mathbf{U}}^{m\tau} \right\|_F^2 + 44\tau \left( 1 + \frac{2}{p} \right) L^2 \sum_{l=m\tau}^{k-1} \eta_l^2 \, \mathbb{E} \left\| \mathbf{U}^l - \bar{\mathbf{U}}^l \right\|_F^2$$

$$+ T \cdot \sigma^2 \cdot \sum_{l=m\tau}^{k-1} \left\{ \eta_l^2 + 16\tau L^2 \left( 1 + \frac{2}{p} \right) \cdot \left\{ \sum_{j=0}^{J-1} \eta_{l,j}^2 \right\} \right\} + 16\tau \left( 1 + \frac{2}{p} \right) G^2 \sum_{l=m\tau}^{k-1} \eta_l^2$$

$$+ 16\tau \left( 1 + \frac{2}{p} \right) \beta^2 \sum_{l=m\tau}^{k-1} \eta_l^2 \, \mathbb{E} \left\| \nabla_{\mathbf{u}} f \left( \bar{\mathbf{u}}^{l,j}, \mathbf{v}_{1:T}^l \right) \right\|^2. \tag{306}$$

*Proof.* We use exactly the same proof as in Lemma G.8, with the only difference that Eq. (284)–Eq. (286) is replaced by

$$\mathbb{E} \sum_{t=1}^{T} \left\| \mathbf{u}_t^k - \bar{\mathbf{u}}^k \right\|_F^2 \leq$$

$$(1 + \alpha)\mathbb{E} \left\| \mathbf{U}^{m\tau} - \bar{\mathbf{U}}^{m\tau} \right\|_F^2 + 2\tau \left( 1 + \alpha^{-1} \right) \sum_{l=m\tau}^{k-1} \eta_l^2 \mathbb{E} \left\| \Upsilon^l \right\|_F^2$$

$$+ \sum_{l=m\tau}^{k-1} \eta_l^2 \mathbb{E} \left\| \Upsilon^l - \hat{\Upsilon}^l \right\|_F^2, \tag{307}$$

resulting from the fact that $\left\{ \prod_{l'=m\tau}^{(m+1)\tau-1} W^{l'} \right\}$ is a doubly stochastic matrix. $\square$

**Lemma G.10.** *Under Assum. 5′-7′ and Assum 8. For $\eta_{k,j} = \frac{\eta}{J}$ with*

$$\eta \leq \min\left\{ \frac{1}{4L}, \frac{p}{92\tau L}, \frac{1}{4\beta L}, \frac{1}{32\sqrt{2}} \cdot \frac{p}{\tau\beta} \right\},$$

*the iterates of Alg. 5 verifies*

$$\frac{(12+T)L^2}{4T} \sum_{k=0}^{K} \mathbb{E} \left\| \mathbf{U}^k - \bar{\mathbf{U}}^k \right\|_F^2 \leq \frac{1}{16} \sum_{k=0}^{K} \mathbb{E} \left\| \nabla_{\mathbf{u}} f \left( \bar{\mathbf{u}}^k, \mathbf{v}_{1:T}^k \right) \right\|^2 + 16A \cdot \frac{12+T}{T} \cdot \frac{\tau L^2}{p} (K+1)\eta^2, \tag{308}$$

*for some constant $A > 0$ and $K > 0$.*

*Proof.* Note that for $k > 0$, $\eta_k = \sum_{j=0}^{J-1} \eta_{kj} = \eta$, and that $\sum_{l=m\tau}^{k-1} \eta_l^2 = \sum_{l=m\tau}^{k-1} \eta^2 \leq 2\tau \cdot \eta^2$

Using Lemma G.8 and Lemma G.9, and the fact that $p \leq 1$, we have for $m = \lfloor \frac{k}{\tau} \rfloor - 1$

$$\mathbb{E} \left\| \mathbf{U}^k - \bar{\mathbf{U}}^k \right\|_F^2 \leq (1 - \frac{p}{2}) \mathbb{E} \left\| \mathbf{U}^{m\tau} - \bar{\mathbf{U}}^{m\tau} \right\|_F^2 + \frac{132\tau}{p} L^2 \eta^2 \sum_{l=m\tau}^{k-1} \mathbb{E} \left\| \mathbf{U}^l - \bar{\mathbf{U}}^l \right\|_F^2$$

$$+ \eta^2 \underbrace{2\tau \left\{ T\sigma^2 \left( 1 + \frac{16\tau L^2}{J} \left( 1 + \frac{2}{p} \right) \right) + 16\tau \left( 1 + \frac{2}{p} \right) G^2 \right\}}_{\triangleq A}$$

$$+ \frac{16\tau}{p} \beta^2 \eta^2 \sum_{l=m\tau}^{k-1} \mathbb{E} \left\| \nabla_{\mathbf{u}} f \left( \bar{\mathbf{u}}^l, \mathbf{v}_{1:T}^l \right) \right\|^2. \tag{309}$$

and for $m = \lfloor \frac{k}{\tau} \rfloor$,

$$\mathbb{E} \left\| \mathbf{U}^k - \bar{\mathbf{U}}^k \right\|_F^2 \leq (1 + \frac{p}{2}) \mathbb{E} \left\| \mathbf{U}^{m\tau} - \bar{\mathbf{U}}^{m\tau} \right\|_F^2 + \frac{132\tau}{p} L^2 \eta^2 \sum_{l=m\tau}^{k-1} \mathbb{E} \left\| \mathbf{U}^l - \bar{\mathbf{U}}^l \right\|_F^2$$

$$+ \eta^2 \underbrace{2\tau \left\{ T\sigma^2 \left( 1 + \frac{16\tau L^2}{J} \left( 1 + \frac{2}{p} \right) \right) + 16\tau \left( 1 + \frac{2}{p} \right) G^2 \right\}}_{\triangleq A}$$

$$+ \underbrace{\frac{16\tau}{p} \beta^2}_{\triangleq D} \eta^2 \sum_{l=m\tau}^{k-1} \mathbb{E} \left\| \nabla_{\mathbf{u}} f \left( \bar{\mathbf{u}}^l, \mathbf{v}_{1:T}^l \right) \right\|^2. \tag{310}$$

Using the fact that $\eta \leq \frac{p}{92\tau L}$, it follows that for $m = \lfloor \frac{k}{\tau} \rfloor - 1$

$$\mathbb{E} \left\| \mathbf{U}^k - \bar{\mathbf{U}}^k \right\|_F^2 \leq (1 - \frac{p}{2}) \mathbb{E} \left\| \mathbf{U}^{m\tau} - \bar{\mathbf{U}}^{m\tau} \right\|_F^2 + \frac{p}{64\tau} \sum_{l=m\tau}^{k-1} \mathbb{E} \left\| \mathbf{U}^l - \bar{\mathbf{U}}^l \right\|^2$$

$$+ \eta^2 A + D\eta^2 \sum_{l=m\tau}^{k-1} \mathbb{E} \left\| \nabla_{\mathbf{u}} f \left( \bar{\mathbf{u}}^l, \mathbf{v}_{1:T}^l \right) \right\|^2, \tag{311}$$

and for $m = \lfloor \frac{k}{\tau} \rfloor$,

$$\mathbb{E} \left\| \mathbf{U}^k - \bar{\mathbf{U}}^k \right\|_F^2 \leq (1 + \frac{p}{2}) \mathbb{E} \left\| \mathbf{U}^{m\tau} - \bar{\mathbf{U}}^{m\tau} \right\|_F^2 + \frac{p}{64\tau} \sum_{l=m\tau}^{k-1} \mathbb{E} \left\| \mathbf{U}^l - \bar{\mathbf{U}}^l \right\|_F^2$$

$$+ \eta^2 A + D\eta^2 \sum_{l=m\tau}^{k-1} \mathbb{E} \left\| \nabla_{\mathbf{u}} f \left( \bar{\mathbf{u}}^l, \mathbf{v}_{1:T}^l \right) \right\|^2. \tag{312}$$

The rest of the proof follows using [31, Lemma 14] with $B = \frac{(12+T)L^2}{4T}$, $b = \frac{1}{8}$, constant (thus $\frac{8\tau}{p}$-slow[7]) steps-size $\eta \leq \frac{1}{32\sqrt{2}} \frac{p}{\tau\beta} = \frac{1}{16} \sqrt{\frac{p/8}{D\tau}}$ and constant weights $\omega_k = 1$. $\square$

**Theorem 3.3′.** *Under Assumptions 4′–7′ and Assumption 8, when clients use SGD as local solver with learning rate $\eta = \frac{a_0}{\sqrt{K}}$, after a large enough number of communication rounds $K$, the iterates of fully decentralized federated surrogate optimization (Alg. 5) satisfy:*

$$\frac{1}{K} \sum_{k=1}^K \mathbb{E} \left\| \nabla_{\mathbf{u}} f \left( \bar{\mathbf{u}}^k, \mathbf{v}_{1:T}^k \right) \right\|^2 \leq \mathcal{O} \left( \frac{1}{\sqrt{K}} \right), \tag{313}$$

---

[7]The notion of $\tau$-slow decreasing sequence is defined in [31, Defintion 2].

*and,*

$$\frac{1}{K}\sum_{k=1}^{K}\sum_{t=1}^{T}\omega_t \cdot \mathbb{E}\,d_{\mathcal{V}}\left(v_t^k, v_t^{k+1}\right) \leq \mathcal{O}\left(\frac{1}{K}\right), \tag{314}$$

*where* $\bar{\mathbf{u}}^k = \frac{1}{T}\sum_{t=1}^{T}\mathbf{u}_t^k$. *Moreover, local estimates* $\left(\mathbf{u}_t^k\right)_{1\leq t\leq T}$ *converge to consensus, i.e., to* $\bar{\mathbf{u}}^k$:

$$\frac{1}{K}\sum_{k=1}^{K}\sum_{t=1}^{T}\mathbb{E}\left\|\mathbf{u}_t^k - \bar{\mathbf{u}}^k\right\|^2 \leq \mathcal{O}\left(\frac{1}{\sqrt{K}}\right). \tag{315}$$

*Proof.* We prove first the convergence to a stationary point in $\mathbf{u}$, i.e. Eq. (313), using [31, Lemma 17], then we prove Eq. (314) and Eq. (315).

Note that for $K$ large enough, $\eta \leq \min\left\{\frac{1}{4L}, \frac{p}{92\tau L}, \frac{1}{4\beta L}, \frac{1}{32\sqrt{2}} \cdot \frac{p}{\tau\beta}\right\}$.

**Proof of Eq. 313.** Rearranging the terms in the result of Lemma G.7 and dividing it by $\eta$ we have

$$\frac{1}{\eta} \cdot \mathbb{E}\left[f(\bar{\mathbf{u}}^k, \mathbf{v}_{1:T}^k) - f(\bar{\mathbf{u}}^{k-1}, \mathbf{v}_{1:T}^{k-1})\right] \leq -\frac{1}{8}\mathbb{E}\left\|\nabla_{\mathbf{u}}f\left(\bar{\mathbf{u}}^{k-1}, \mathbf{v}_{1:T}^{k-1}\right)\right\|^2$$
$$+ \frac{(12+T)L^2}{4T} \cdot \mathbb{E}\left\|\mathbf{U}^{k-1} - \bar{\mathbf{U}}^{k-1}\right\|^2 + \frac{\eta L}{T}\left(\frac{4L}{J}+1\right)\sigma^2 + \frac{16\eta^2 L^2}{T}G^2. \tag{316}$$

Summing over $k \in [K+1]$, we have

$$\frac{1}{\eta} \cdot \mathbb{E}\left[f(\bar{\mathbf{u}}^{K+1}, \mathbf{v}_{1:T}^{K+1}) - f(\bar{\mathbf{u}}^0, \mathbf{v}_{1:T}^0)\right] \leq -\frac{1}{8}\sum_{k=0}^{K}\mathbb{E}\left\|\nabla_{\mathbf{u}}f\left(\bar{\mathbf{u}}^k, \mathbf{v}_{1:T}^k\right)\right\|^2$$
$$+ \frac{(12+T)L^2}{4T} \cdot \sum_{k=0}^{K}\mathbb{E}\left\|\mathbf{U}^k - \bar{\mathbf{U}}^k\right\|^2 + \frac{(K+1)\eta L}{T}\left(\frac{4L}{J}+1\right)\sigma^2$$
$$+ \frac{16(K+1)\cdot\eta^2 L^2}{T}G^2. \tag{317}$$

Using Lemma G.10, we have

$$\frac{1}{\eta} \cdot \mathbb{E}\left[f(\bar{\mathbf{u}}^{K+1}, \mathbf{v}_{1:T}^{K+1}) - f(\bar{\mathbf{u}}^0, \mathbf{v}_{1:T}^0)\right] \leq -\frac{1}{16}\sum_{k=0}^{K}\mathbb{E}\left\|\nabla_{\mathbf{u}}f\left(\bar{\mathbf{u}}^k, \mathbf{v}_{1:T}^k\right)\right\|^2$$
$$+ 16A \cdot \frac{12+T}{T} \cdot \frac{\tau L^2}{p}(K+1)\eta^2 + \frac{(K+1)\eta L}{T}\left(\frac{4L}{J}+1\right)\sigma^2$$
$$+ \frac{16(K+1)\eta^2 L^2}{T}G^2. \tag{318}$$

Using Assumption 4′, it follows that

$$\frac{1}{16}\sum_{k=0}^{K}\mathbb{E}\left\|\nabla_{\mathbf{u}}f\left(\bar{\mathbf{u}}^k, \mathbf{v}_{1:T}^k\right)\right\|^2 \leq \frac{f(\bar{\mathbf{u}}^0, \mathbf{v}_{1:T}^0) - f^*}{\eta}$$
$$+ 16A \cdot \frac{12+T}{T} \cdot \frac{\tau L^2}{p}(K+1)\eta^2 + \frac{(K+1)\eta L}{T}\left(\frac{4L}{J}+1\right)\sigma^2 + \frac{16(K+1)\eta^2 L^2}{T}G^2. \tag{319}$$

We divide by $K+1$ and we have

$$\frac{1}{16(K+1)}\sum_{k=0}^{K}\mathbb{E}\left\|\nabla_{\mathbf{u}}f\left(\bar{\mathbf{u}}^k, \mathbf{v}_{1:T}^k\right)\right\|^2 \leq \frac{f(\bar{\mathbf{u}}^0, \mathbf{v}_{1:T}^0) - f^*}{\eta(K+1)}$$
$$+ 16A \cdot \frac{12+T}{T} \cdot \frac{\tau L^2}{p}\eta^2 + \frac{\eta L}{T}\left(\frac{4L}{J}+1\right)\sigma^2 + \frac{16\eta^2 L^2}{T}G^2. \tag{320}$$

The final result follows from [31, Lemma 17].

**Proof of Eq. 315.** We multiply Eq. (308) (Lemma G.10) by $\frac{1}{K+1}$, and we have

$$\frac{1}{K+1}\sum_{k=0}^{K}\mathbb{E}\left\|\mathbf{U}^k-\bar{\mathbf{U}}^k\right\|_F^2 \leq \frac{1}{16(K+1)}\sum_{k=0}^{K}\mathbb{E}\left\|\nabla_{\mathbf{u}}f\left(\bar{\mathbf{u}}^k,\mathbf{v}_{1:T}^k\right)\right\|_F^2 + \frac{64A\tau}{p(K+1)}K\eta^2, \quad (321)$$

since $\eta \leq \mathcal{O}\left(\frac{1}{\sqrt{K}}\right)$, using Eq. (313), it follows that

$$\frac{1}{K}\sum_{k=1}^{K}\mathbb{E}\left\|\mathbf{U}^k-\bar{\mathbf{U}}^k\right\|_F^2 \leq \mathcal{O}\left(\frac{1}{\sqrt{K}}\right). \quad (322)$$

Thus,

$$\frac{1}{K}\sum_{k=1}^{K}\sum_{t=1}^{T}\mathbb{E}\left\|\mathbf{u}_t^k-\bar{\mathbf{u}}^k\right\|_F^2 \leq \mathcal{O}\left(\frac{1}{\sqrt{K}}\right). \quad (323)$$

**Proof of Eq. 314.** Using the result of Lemma G.7 we have

$$\frac{1}{T}\sum_{t=1}^{T}\mathbb{E}\left[d_{\mathcal{V}}\left(\mathbf{v}_t^k,\mathbf{v}_t^{k-1}\right)\right] \leq \mathbb{E}\left[f(\bar{\mathbf{u}}^{k-1},\mathbf{v}_{1:T}^{k-1})-f(\bar{\mathbf{u}}^k,\mathbf{v}_{1:T}^k)\right]$$

$$+ \frac{(12+T)\eta_{k-1}L^2}{4T}\cdot\sum_{t=1}^{T}\mathbb{E}\left\|\mathbf{u}_t^{k-1}-\bar{\mathbf{u}}^{k-1}\right\|^2$$

$$+ \frac{\eta_{k-1}^2 L}{T}\left(4\sum_{j=0}^{J-1}\frac{L\cdot\eta_{k-1,j}^2}{\eta_{k-1}}+1\right)\sigma^2 + \frac{16\eta_{k-1}^3 L^2}{T}G^2. \quad (324)$$

The final result follows from the fact that $\eta = \mathcal{O}\left(\frac{1}{\sqrt{K}}\right)$ and Eq. (315). $\qquad\square$

### G.2.3 Proof of Theorem 3.3

We state the formal version of Theorem 3.3, for which only an informal version was given in the main text.

**Theorem 3.3.** *Under Assumptions 1–8, when clients use SGD as local solver with learning rate $\eta = \frac{a_0}{\sqrt{K}}$, D-FedEM's iterates satisfy the following inequalities after a large enough number of communication rounds $K$:*

$$\frac{1}{K}\sum_{k=1}^{K}\mathbb{E}\left\|\nabla_{\Theta}f\left(\bar{\Theta}^k,\Pi^k\right)\right\|_F^2 \leq \mathcal{O}\left(\frac{1}{\sqrt{K}}\right), \quad \frac{1}{K}\sum_{k=1}^{K}\sum_{t=1}^{T}\frac{n_t}{n}\mathcal{KL}\left(\pi_t^k,\pi_t^{k-1}\right) \leq \mathcal{O}\left(\frac{1}{K}\right), \quad (325)$$

*where $\bar{\Theta}^k = \left[\Theta_1^k,\dots\Theta_T^k\right]\cdot\frac{\mathbf{1}\mathbf{1}^\intercal}{T}$. Moreover, individual estimates $\left(\Theta_t^k\right)_{1\leq t\leq T}$ converge to consensus, i.e., to $\bar{\Theta}^k$:*

$$\min_{k\in[K]}\mathbb{E}\sum_{t=1}^{T}\left\|\Theta_t^k-\bar{\Theta}^k\right\|_F^2 \leq \mathcal{O}\left(\frac{1}{\sqrt{K}}\right).$$

*Proof.* We prove this result as a particular case of Theorem 3.3′. To this purpose, we consider that $\mathcal{V}\triangleq\Delta^M$, $\mathbf{u}=\Theta\in\mathbb{R}^{dM}$, $\mathbf{v}_t=\pi_t$, and $\omega_t=n_t/n$ for $t\in[T]$. For $k>0$, we define $g_t^k$ as follow,

$$g_t^k\left(\Theta,\pi_t\right) = \frac{1}{n_t}\sum_{i=1}^{n_t}\sum_{m=1}^{M}q_t^k\left(z_t^{(i)}=m\right)\cdot\left(l\left(h_{\theta_m}(\mathbf{x}_t^{(i)}),y_t^{(i)}\right)-\log p_m(\mathbf{x}_t^{(i)})-\log\pi_t\right.$$

$$\left.+ \log q_t^k\left(z_t^{(i)}=m\right)-c\right), \quad (326)$$

where $c$ is the same constant appearing in Assumption 3, Eq. (3). With this definition, it is easy to check that the federated surrogate optimization algorithm (Alg. 5) reduces to D-FedEM (Alg. 4).

Theorem 3.3 then follows immediately from Theorem 3.3′, once we verify that $\left(g_t^k\right)_{1 \le t \le T}$ satisfy the assumptions of Theorem 3.3′.

Assumption 4′, Assumption 6′, and Assumption 7′ follow directly from Assumption 4, Assumption 6, and Assumption 7, respectively. Lemma G.3 shows that for $k > 0$, $g^k$ is smooth w.r.t. $\Theta$ and then Assumption 5′ is satisfied. Finally, Lemmas G.4–G.6 show that for $t \in [T]$ $g_t^k$ is a partial first-order surrogate of $f_t$ near $\left\{\Theta_t^{k-1}, \pi_t\right\}$ with $d_{\mathcal{V}}(\cdot, \cdot) = \mathcal{KL}(\cdot\|\cdot)$. $\qquad\square$

### G.3 Supporting Lemmas

**Lemma G.11.** *Consider $J \ge 2$ and positive real numbers $\eta_j$, $j = 0, \ldots, J-1$, then:*

$$\frac{1}{\sum_{j=0}^{J-1} \eta_j} \cdot \sum_{j=0}^{J-1} \left\{ \eta_j \cdot \sum_{l=0}^{j-1} \eta_l \right\} \le \sum_{j=0}^{J-2} \eta_j,$$

$$\frac{1}{\sum_{j=0}^{J-1} \eta_j} \cdot \sum_{j=0}^{J-1} \left\{ \eta_j \cdot \sum_{l=0}^{j-1} \eta_l^2 \right\} \le \sum_{j=0}^{J-2} \eta_j^2,$$

$$\frac{1}{\sum_{j=0}^{J-1} \eta_j} \cdot \sum_{j=0}^{J-1} \left\{ \eta_j \cdot \left( \sum_{l=0}^{j-1} \eta_l \right)^2 \right\} \le \sum_{j=0}^{J-1} \eta_j \cdot \sum_{j=0}^{J-2} \eta_j.$$

*Proof.* For the first inequality,

$$\frac{1}{\sum_{j=0}^{J-1} \eta_j} \cdot \sum_{j=0}^{J-1} \left\{ \eta_j \cdot \sum_{l=0}^{j-1} \eta_l \right\} \le \frac{1}{\sum_{j=0}^{J-1} \eta_j} \cdot \sum_{j=0}^{J-1} \left\{ \eta_j \cdot \sum_{l=0}^{J-2} \eta_l \right\} = \sum_{l=0}^{J-2} \eta_l. \tag{327}$$

For the second inequality

$$\frac{1}{\sum_{j=0}^{J-1} \eta_j} \cdot \sum_{j=0}^{J-1} \left\{ \eta_j \cdot \sum_{l=0}^{j-1} \eta_l^2 \right\} \le \frac{1}{\sum_{j=0}^{J-1} \eta_j} \cdot \sum_{j=0}^{J-1} \left\{ \eta_j \cdot \sum_{l=0}^{J-2} \eta_l^2 \right\} = \sum_{l=0}^{J-2} \eta_l^2. \tag{328}$$

For the third inequality,

$$\frac{1}{\sum_{j=0}^{J-1} \eta_j} \cdot \sum_{j=0}^{J-1} \left\{ \eta_j \cdot \left( \sum_{l=0}^{j-1} \eta_l \right)^2 \right\} \le \frac{1}{\sum_{j=0}^{J-1} \eta_j} \cdot \sum_{j=0}^{J-1} \left\{ \eta_j \cdot \left( \sum_{l=0}^{J-2} \eta_l \right)^2 \right\} \tag{329}$$

$$\le \left( \sum_{j=0}^{J-2} \eta_j \right)^2 \tag{330}$$

$$\le \sum_{j=0}^{J-1} \eta_j \cdot \sum_{j=0}^{J-2} \eta_j. \tag{331}$$

$\qquad\square$

**Lemma G.12.** *Suppose that $g$ is a partial first-order surrogate of $f$, and that $g$ is $L$-smooth, where $L$ is the constant appearing in Definition 1, then $f$ is $2L$-smooth.*

*Proof.* The difference between $f$ and $g$ is $L$-smooth, and $g$ is $L$-smooth, thus $f$ is $2L$-smooth as the sum of two $L$-smooth functions. $\qquad\square$

**Lemma G.13.** *Consider $f = \sum_{t=1}^{T} \omega_t \cdot f_t$, for weights $\omega \in \Delta^T$. Suppose that for all $(\mathbf{u}, \mathbf{v}) \in \mathbb{R}^{d_u} \times \mathcal{V}$, and $t \in [T]$, $f_t$ admits a partial first-order surrogate $g_t^{\{\mathbf{u}, \mathbf{v}\}}$ near $\{\mathbf{u}, \mathbf{v}\}$, and that $g^{\{\mathbf{u}, \mathbf{v}\}} = \sum_{t=1}^{T} \omega_t \cdot g_t^{\{\mathbf{u}, \mathbf{v}\}}$ verifies Assumption 7′ for $t \in [T]$. Then $f$ also verifies Assumption 7′.*

*Proof.* Consider arbitrary $\mathbf{u}, \mathbf{v} \in \mathbb{R}^{d_u} \times \mathcal{V}$, and for $t \in [T]$, consider $g^{\{\mathbf{u},\mathbf{v}\}}$ to be a partial first-order surrogate of $f_t$ near $\{\mathbf{u}, \mathbf{v}\}$. We write Assumption 7′ for $g^{\{\mathbf{u},\mathbf{v}\}}$,

$$\sum_{t=1}^{T} \omega_t \cdot \left\| \nabla_{\mathbf{u}} g_t^{\{u,v\}}(\mathbf{u}, \mathbf{v}) \right\|^2 \leq G^2 + \beta^2 \left\| \sum_{t=1}^{T} \omega_t \cdot \nabla_{\mathbf{u}} g_t^{\{u,v\}}(\mathbf{u}, \mathbf{v}) \right\|^2. \tag{332}$$

Since $g_t^{\{\mathbf{u},\mathbf{v}\}}$ is a partial first-order surrogate of $f_t$ near $\{u, v\}$, it follows that

$$\sum_{t=1}^{T} \omega_t \cdot \left\| \nabla_{\mathbf{u}} f_t(\mathbf{u}, \mathbf{v}) \right\|^2 \leq G^2 + \beta^2 \left\| \sum_{t=1}^{T} \omega_t \cdot \nabla_{\mathbf{u}} f_t(\mathbf{u}, \mathbf{v}) \right\|^2. \tag{333}$$

$\square$

**Remark 4.** *Note that the assumption of Lemma G.13 is implicitly verified in Alg. 3 and Alg. 5, where we assume that every client $t \in \mathcal{T}$ canfunction compute a partial first-order surrogate of its local objective $f_t$ near any iterate $(\mathbf{u}, \mathbf{v}) \in \mathbb{R}^{d_u} \times \mathcal{V}$.*

**Lemma G.14.** *For $k > 0$, the iterates of Alg. 5, verify the following inequalities:*

$$g^k \left( \bar{\mathbf{u}}^{k-1}, \mathbf{v}_{1:T}^{k-1} \right) \leq f \left( \bar{\mathbf{u}}^{k-1}, \mathbf{v}_{1:T}^{k-1} \right) + \frac{L}{2} \sum_{t=1}^{T} \omega_t \left\| \bar{\mathbf{u}}^{k-1} - \mathbf{u}_t^{k-1} \right\|^2,$$

$$\left\| \nabla_{\mathbf{u}} f \left( \bar{\mathbf{u}}^{k-1}, \mathbf{v}_{1:T}^{k-1} \right) \right\|^2 \leq 2 \left\| \nabla_{\mathbf{u}} g^k \left( \bar{\mathbf{u}}^{k-1}, \mathbf{v}_{1:T}^{k-1} \right) \right\|^2 + 2L^2 \sum_{t=1}^{T} \omega_t \left\| \bar{\mathbf{u}}^{k-1} + \mathbf{u}_t^{k-1} \right\|^2,$$

*and,*

$$\left\| \nabla_{\mathbf{u}} g^k \left( \bar{\mathbf{u}}^{k-1}, \mathbf{v}_{1:T}^{k-1} \right) \right\|^2 \leq 2 \left\| \nabla_{\mathbf{u}} f \left( \bar{\mathbf{u}}^{k-1}, \mathbf{v}_{1:T}^{k-1} \right) \right\|^2 + 2L^2 \sum_{t=1}^{T} \omega_t \left\| \bar{\mathbf{u}}^{k-1} - \mathbf{u}_t^{k-1} \right\|^2,$$

*Proof.* For $k > 0$ and $t \in [T]$, we have

$$g_t^k \left( \bar{\mathbf{u}}^{k-1}, \mathbf{v}_t^{k-1} \right) =$$
$$g_t^k \left( \bar{\mathbf{u}}^{k-1}, \mathbf{v}_t^{k-1} \right) + f_t \left( \bar{\mathbf{u}}^{k-1}, \mathbf{v}_t^{k-1} \right) - f_t \left( \bar{\mathbf{u}}^{k-1}, \mathbf{v}_t^{k-1} \right) \tag{334}$$
$$= f_t \left( \bar{\mathbf{u}}^{k-1}, \mathbf{v}_t^{k-1} \right) + r_t^k \left( \bar{\mathbf{u}}^{k-1}, \mathbf{v}_t^{k-1} \right) \tag{335}$$
$$= f_t \left( \bar{\mathbf{u}}^{k-1}, \mathbf{v}_t^{k-1} \right) + r_t^k \left( \bar{\mathbf{u}}^{k-1}, \mathbf{v}_t^{k-1} \right) - r_t^k \left( \mathbf{u}_t^{k-1}, \mathbf{v}_t^{k-1} \right) + r_t^k \left( \mathbf{u}_t^{k-1}, \mathbf{v}_t^{k-1} \right). \tag{336}$$

Since $g_t^k \left( \mathbf{u}_t^k, \mathbf{v}_t^{k-1} \right) = f_t \left( \mathbf{u}_t^k, \mathbf{v}_t^{k-1} \right)$ (Definition 1), it follows that

$$g_t^k \left( \bar{\mathbf{u}}^{k-1}, \mathbf{v}_t^{k-1} \right) = f_t \left( \bar{\mathbf{u}}^{k-1}, \mathbf{v}_t^{k-1} \right) + r_t^k \left( \bar{\mathbf{u}}^{k-1}, \mathbf{v}_t^{k-1} \right) - r_t^k \left( \mathbf{u}_t^{k-1}, \mathbf{v}_t^{k-1} \right). \tag{337}$$

Because $r_t^k$ is $L$-smooth in $\mathbf{u}$ (Definition 1), we have

$$r_t^k \left( \bar{\mathbf{u}}^{k-1}, \mathbf{v}_t^{k-1} \right) - r_t^k \left( \mathbf{u}_t^{k-1}, \mathbf{v}_t^{k-1} \right) \leq \left\langle \nabla_{\mathbf{u}} r_t^k \left( \mathbf{u}_t^{k-1}, \mathbf{v}_t^{k-1} \right), \bar{\mathbf{u}}^{k-1} - \mathbf{u}_t^{k-1} \right\rangle$$
$$+ \frac{L}{2} \left\| \bar{\mathbf{u}}^{k-1} - \mathbf{u}_t^{k-1} \right\|^2. \tag{338}$$

Since $g_t^k$ is a partial first order surrogate of We have $\nabla_{\mathbf{u}} r_t^k \left( \mathbf{u}_t^{k-1}, \mathbf{v}_t^{k-1} \right) = 0$, thus

$$g_t^k \left( \bar{\mathbf{u}}^{k-1}, \mathbf{v}_t^{k-1} \right) \leq f_t \left( \bar{\mathbf{u}}^{k-1}, \mathbf{v}_t^{k-1} \right) + \frac{L}{2} \left\| \bar{\mathbf{u}}^{k-1} - \mathbf{u}_t^{k-1} \right\|^2. \tag{339}$$

Multiplying by $\omega_t$ and summing for $t \in [T]$, we have

$$g^k \left( \bar{\mathbf{u}}^{k-1}, \mathbf{v}_{1:T}^{k-1} \right) \leq f \left( \bar{\mathbf{u}}^{k-1}, \mathbf{v}_{1:T}^{k-1} \right) + \frac{L}{2} \sum_{t=1}^{T} \omega_t \left\| \bar{\mathbf{u}}^{k-1} - \mathbf{u}_t^{k-1} \right\|^2, \tag{340}$$

and the first inequality is proved.

Writing the gradient of Eq. (337), we have

$$\nabla_{\mathbf{u}} g_t^k \left( \bar{\mathbf{u}}^{k-1}, \mathbf{v}_t^{k-1} \right) = \nabla_{\mathbf{u}} f_t \left( \bar{\mathbf{u}}^{k-1}, \mathbf{v}_t^{k-1} \right) + \nabla_{\mathbf{u}} r_t^k \left( \bar{\mathbf{u}}^{k-1}, \mathbf{v}_t^{k-1} \right) - \nabla_{\mathbf{u}} r_t^k \left( \mathbf{u}_t^{k-1}, \mathbf{v}_t^{k-1} \right). \quad (341)$$

Multiplying by $\omega_t$ and summing for $t \in [T]$, we have

$$\nabla_{\mathbf{u}} g^k \left( \bar{\mathbf{u}}^{k-1}, \mathbf{v}_{1:T}^{k-1} \right) = \nabla_{\mathbf{u}} f \left( \bar{\mathbf{u}}^{k-1}, \mathbf{v}_{1:T}^{k-1} \right) +$$

$$+ \sum_{t=1}^{T} \omega_t \left[ \nabla_{\mathbf{u}} r_t^k \left( \bar{\mathbf{u}}^{k-1}, \mathbf{v}_t^{k-1} \right) - \nabla_{\mathbf{u}} r_t^k \left( \mathbf{u}_t^{k-1}, \mathbf{v}_t^{k-1} \right) \right]. \quad (342)$$

Thus,

$$\left\| \nabla_{\mathbf{u}} g^k \left( \bar{\mathbf{u}}^{k-1}, \mathbf{v}_{1:T}^{k-1} \right) \right\|^2 =$$

$$\left\| \nabla_{\mathbf{u}} f \left( \bar{\mathbf{u}}^{k-1}, \mathbf{v}_{1:T}^{k-1} \right) + \sum_{t=1}^{T} \omega_t \left[ \nabla_{\mathbf{u}} r_t^k \left( \bar{\mathbf{u}}^{k-1}, \mathbf{v}_t^{k-1} \right) - \nabla_{\mathbf{u}} r_t^k \left( \mathbf{u}_t^{k-1}, \mathbf{v}_t^{k-1} \right) \right] \right\|^2 \quad (343)$$

$$\geq \frac{1}{2} \left\| \nabla_{\mathbf{u}} f \left( \bar{\mathbf{u}}^{k-1}, \mathbf{v}_{1:T}^{k-1} \right) \right\|^2 - \left\| \sum_{t=1}^{T} \omega_t \left[ \nabla_{\mathbf{u}} r_t^k \left( \bar{\mathbf{u}}^{k-1}, \mathbf{v}_t^{k-1} \right) - \nabla_{\mathbf{u}} r_t^k \left( \mathbf{u}_t^{k-1}, \mathbf{v}_t^{k-1} \right) \right] \right\|^2 \quad (344)$$

$$\geq \frac{1}{2} \left\| \nabla_{\mathbf{u}} f \left( \bar{\mathbf{u}}^{k-1}, \mathbf{v}_{1:T}^{k-1} \right) \right\|^2 - \sum_{t=1}^{T} \omega_t \left\| \nabla_{\mathbf{u}} r_t^k \left( \bar{\mathbf{u}}^{k-1}, \mathbf{v}_t^{k-1} \right) - \nabla_{\mathbf{u}} r_t^k \left( \mathbf{u}_t^{k-1}, \mathbf{v}_t^{k-1} \right) \right\|^2 \quad (345)$$

$$\geq \frac{1}{2} \left\| \nabla_{\mathbf{u}} f \left( \bar{\mathbf{u}}^{k-1}, \mathbf{v}_{1:T}^{k-1} \right) \right\|^2 - L^2 \sum_{t=1}^{T} \omega_t \left\| \bar{\mathbf{u}}^{k-1} - \mathbf{u}_t^{k-1} \right\|^2, \quad (346)$$

where (344) follows from $\|a\|^2 = \|a + b - b\|^2 \leq 2 \|a + b\|^2 + 2 \|b\|^2$. Thus,

$$\left\| \nabla_{\mathbf{u}} f_t \left( \bar{\mathbf{u}}^{k-1}, \mathbf{v}_t^{k-1} \right) \right\|^2 \leq 2 \left\| \nabla_{\mathbf{u}} g_t^k \left( \bar{\mathbf{u}}^{k-1}, \mathbf{v}_t^{k-1} \right) \right\|^2 + 2L^2 \sum_{t=1}^{T} \omega_t \left\| \bar{\mathbf{u}}^{k-1} - \mathbf{u}_t^{k-1} \right\|^2. \quad (347)$$

The proof of the last inequality is similar, it leverages $\|a + b\|^2 \leq 2 \|a\|^2 + 2 \|a\|^2$ to upper bound (343). $\qquad \square$

**Lemma G.15.** *Consider $\mathbf{u}_1, \ldots, \mathbf{u}_M \in \mathbb{R}^d$ and $\alpha = (\alpha_1, \ldots, \alpha_M) \in \Delta^M$. Define the block matrix $\mathbf{H}$ with*

$$\begin{cases} \mathbf{H}_{m,m} = -\alpha_m \cdot (1 - \alpha_m) \cdot \mathbf{u}_m \cdot \mathbf{u}_m^{\mathsf{T}} \\ \mathbf{H}_{m,m'} = \alpha_m \cdot \alpha_{m'} \cdot \mathbf{u}_m \cdot \mathbf{u}_{m'}^{\mathsf{T}}; \qquad m' \neq m, \end{cases} \quad (348)$$

*then $\mathbf{H}$ is a semi-definite negative matrix.*

*Proof.* Consider $\mathbf{x} = [\mathbf{x}_1, \ldots, \mathbf{x}_M] \in \mathbb{R}^{dM}$, we want to prove that

$$\mathbf{x}^{\mathsf{T}} \cdot \mathbf{H} \cdot \mathbf{x} \leq 0. \quad (349)$$

We have:

$$\mathbf{X}^{\mathsf{T}} \cdot \mathbf{H} \cdot \mathbf{X} = \sum_{m=1}^{M} \sum_{m'=1}^{M} \mathbf{x}_m^{\mathsf{T}} \cdot \mathbf{H}_{m,m'} \cdot \mathbf{x}_{m'} \quad (350)$$

$$= \sum_{m=1}^{M} \left[ \mathbf{x}_m^{\mathsf{T}} \cdot \mathbf{H}_{m,m} \cdot \mathbf{x}_m + \sum_{\substack{m'=1 \\ m' \neq m}}^{M} \mathbf{x}_m^{\mathsf{T}} \cdot \mathbf{H}_{m,m} \cdot \mathbf{x}_{m'} \right] \quad (351)$$

$$= \sum_{m=1}^{M} \left( -\alpha_m \cdot (1 - \alpha_m) \cdot \mathbf{x}_m^{\mathsf{T}} \cdot \mathbf{u}_m \cdot \mathbf{u}_m^{\mathsf{T}} \cdot \mathbf{x}_m \right) \quad (352)$$

$$+ \sum_{m=1}^{M} \left[ \sum_{\substack{m'=1 \\ m' \neq m}}^{M} \left( \alpha_m \cdot \alpha_{m'} \cdot \mathbf{x}_m^{\mathsf{T}} \cdot \mathbf{u}_m \cdot \mathbf{u}_{m'}^{\mathsf{T}} \cdot \mathbf{x}_{m'} \right) \right] \tag{353}$$

$$= \sum_{m=1}^{M} \left[ -\alpha_m \cdot (1 - \alpha_m) \cdot \langle \mathbf{x}_m, \mathbf{u}_m \rangle^2 + \alpha_m \cdot \langle \mathbf{x}_m, \mathbf{u}_m \rangle \sum_{\substack{m'=1 \\ m' \neq m}}^{M} \alpha_{m'} \cdot \langle \mathbf{x}_{m'}, \mathbf{u}_{m'} \rangle \right] . \tag{354}$$

Since $\alpha \in \Delta^M$,

$$\forall m \in [M], \quad \sum_{\substack{m'=1 \\ m' \neq m}}^{M} \alpha_{m'} = (1 - \alpha_m) , \tag{355}$$

thus,

$$\mathbf{x}^{\mathsf{T}} \cdot \mathbf{H} \cdot \mathbf{x} = \sum_{m=1}^{M} \alpha_m \cdot \langle \mathbf{x}_m, \mathbf{u}_m \rangle \cdot \sum_{\substack{m'=1 \\ m' \neq m}}^{M} \alpha_{m'} \left( \langle \mathbf{x}_{m'}, \mathbf{u}_{m'} \rangle - \langle \mathbf{x}_m, \mathbf{u}_m \rangle \right) \tag{356}$$

$$= \sum_{m=1}^{M} \alpha_m \cdot \langle \mathbf{x}_m, \mathbf{u}_m \rangle \cdot \sum_{m'=1}^{M} \alpha_{m'} \left( \langle \mathbf{x}_{m'}, \mathbf{u}_{m'} \rangle - \langle \mathbf{x}_m, \mathbf{u}_m \rangle \right) \tag{357}$$

$$= \left( \sum_{m=1}^{M} \alpha_m \cdot \langle \mathbf{x}_m, \mathbf{u}_m \rangle \right)^2 - \sum_{m=1}^{M} \alpha_m \cdot \langle \mathbf{x}_m, \mathbf{u}_m \rangle^2 . \tag{358}$$

Using Jensen inequality, we have $\mathbf{x}^{\mathsf{T}} \cdot \mathbf{H} \cdot \mathbf{x} \leq 0$. $\qquad \square$

# H   Distributed Surrogate Optimization with Black-Box Solver

In this section, we cover the scenario where the local SGD solver used in our algorithms (Alg. 3 and Alg. 5) is replaced by a (possibly non-iterative) black-box solver that is guaranteed to provide a *local inexact solution* of

$$\forall m \in [M], \;\; \underset{\theta \in \mathbb{R}^d}{\text{minimize}} \sum_{i=1}^{n_t} q^k(z_t^i = m) \cdot l(h_\theta(\mathbf{x}_t^{(i)}), y_t^{(i)}), \tag{359}$$

with the following approximation guarantee.

**Assumption 9** (Local $\alpha$-approximate solution). *There exists $0 < \alpha < 1$ such that for $t \in [T]$, $m \in [M]$ and $k > 0$,*

$$\sum_{i=1}^{n_t} q^k(z_t^i = m) \cdot \left\{ l(h_{\theta_{m,t}^k}(\mathbf{x}_t^{(i)}), y_t^{(i)}) - l(h_{\theta_{m,t,*}^k}(\mathbf{x}_t^{(i)}), y_t^{(i)}) \right\} \le$$

$$\alpha \cdot \sum_{i=1}^{n_t} q^k(z_t^i = m) \cdot \left\{ l(h_{\theta_m^{k-1}}(\mathbf{x}_t^{(i)}), y_t^{(i)}) - l(h_{\theta_{m,t,*}^k}(\mathbf{x}_t^{(i)}), y_t^{(i)}) \right\}, \tag{360}$$

*where $\theta_{m,t,*}^k \in \arg\min_{\theta \in \mathbb{R}^d} \sum_{i=1}^{n_t} q^k(z_t^i = m) \cdot l(h_\theta(\mathbf{x}_t^{(i)}), y_t^{(i)})$, $\theta_{m,t}^k$ is the output of the local solver at client $t$ and $\theta_m^{k-1}$ is its starting point (see Alg. 2).*

We further assume strong convexity.

**Assumption 10.** *For $t \in [T]$ and $i \in [n_t]$, we suppose that $\theta \mapsto l\left(h_\theta\left(\mathbf{x}_t^{(i)}\right), y_t^{(i)}\right)$ is $\mu$-strongly convex.*

Assumption 9 is equivalent to the $\gamma$-inexact solution used in [37] (Lemma. H.2), when local functions $(\Phi_t)_{1 \le t \le T}$ are assumed to be convex. We also need to have $G^2 = 0$ in Assumption 7 as in [38, Definition 3], in order to ensure the convergence of Alg. 2 and Alg. 4 to a stationary point of $f$, as shown by [66, Theorem. 2].[8]

**Theorem H.1.** *Suppose that Assumptions 1–7, 9 and 10 hold with $G^2 = 0$ and $\alpha < \frac{1}{\beta^2 \kappa^4}$, then the updates of federated surrogate optimization converge to a stationary point of $f$, i.e.,*

$$\lim_{k \to +\infty} \left\| \nabla_\Theta f(\Theta^k, \Pi^k) \right\|_F^2 = 0, \tag{361}$$

*and*

$$\lim_{k \to +\infty} \sum_{t=1}^T \frac{n_t}{n} \mathcal{KL}\left(\pi_t^k, \pi_t^{k-1}\right) = 0. \tag{362}$$

As in App. G, we provide the analysis for the general case of federated surrogate optimization (Alg. 3) before showing that `FedEM` (Alg. 2) is a particular case.

We suppose that, at iteration $k > 0$, the partial first-order surrogate functions $g_t^k$, $t \in [T]$ used in Alg. 3 verifies, in addition to Assumptions 4'–7', the following assumptions that generalize Assumptions 9 and 10,

**Assumption 9'** (Local $\alpha$-inexact solution). *There exists $0 < \alpha < 1$ such that for $t \in [T]$ and $k > 0$,*

$$\forall \mathbf{v} \in \mathcal{V}, \; g_t^k(\mathbf{u}_t^k, \mathbf{v}) - g_t^k(\mathbf{u}_{t,*}^k, \mathbf{v}) \le \alpha \cdot \left\{ g_t^k\left(\mathbf{u}^{k-1}, \mathbf{v}\right) - g_t^k\left(\mathbf{u}_{t,*}^k, \mathbf{v}\right) \right\}, \tag{363}$$

*where $\mathbf{u}_{t,*}^k \in \arg\min_{\mathbf{u} \in \mathbb{R}^{d_u}} g_t^k\left(\mathbf{u}, \mathbf{v}_t^k\right)$.*

**Assumption 10'.** *For $t \in [T]$ and $k > 0$, $g_t^k$ is $\mu$-strongly convex in $\mathbf{u}$.*

Under these assumptions a parallel result to Theorem. H.1 holds.

---

[8]As shown by [66, Theorem. 2], the convergence is guaranteed in two scenarios: 1) $G^2 = 0$, 2) All clients use take the same number of local steps using the same local solver. Note that we allow each client to use an arbitrary approximate local solver.

**Theorem H.1′.** *Suppose that Assumptions 4′–7′, Assumptions 9′ and 10′ hold with $G^2 = 0$ and $\alpha < \frac{1}{\beta^2 \kappa^4}$, then the updates of federated surrogate optimization converges to a stationary point of $f$, i.e.,*

$$\lim_{k \to +\infty} \left\| \nabla_{\mathbf{u}} f(\mathbf{u}^k, \mathbf{v}^k_{1:T}) \right\|^2 = 0, \tag{364}$$

*and*

$$\lim_{k \to +\infty} \sum_{t=1}^{T} \omega_t \cdot d_{\mathcal{V}} \left( \mathbf{v}^k_t, \mathbf{v}^{k-1}_t \right) = 0. \tag{365}$$

## H.1 Supporting Lemmas

First, we prove the following result.

**Lemma H.2.** *Under Assumptions 5′, 9′ and 10′, the iterates of Alg. 2 verify for $k > 0$ and $t \in [T]$,*

$$\forall \mathbf{v} \in \mathcal{V}, \ \left\| \nabla_{\mathbf{u}} g^k_t \left( \mathbf{u}^k_t, \mathbf{v} \right) \right\| \leq \sqrt{\alpha \kappa} \cdot \left\| \nabla_{\mathbf{u}} g^k_t \left( \mathbf{u}^{k-1}, \mathbf{v} \right) \right\|, \tag{366}$$

*where $\kappa = L/\mu$.*

*Proof.* Consider $\mathbf{v} \in \mathcal{V}$. Since $g^k_t$ is $L$-smooth in $\mathbf{u}$ (Assumption 5′), we have using Assumption 9′,

$$\left\| \nabla_{\mathbf{u}} g^k_t \left( \mathbf{u}^k_t, \mathbf{v} \right) \right\|^2_F \leq 2L \left( g^k_t \left( \mathbf{u}^k_t, \mathbf{v} \right) - g^k_t \left( \mathbf{u}^k_{t,*}, \mathbf{v} \right) \right) \leq 2L\alpha \left( g^k_t \left( \mathbf{u}^{k-1}, \mathbf{v} \right) - g^k_t \left( \mathbf{u}^k_{t,*}, \mathbf{v} \right) \right). \tag{367}$$

Since $\Phi^k_t$ is $\mu$-strongly convex (Assumption 10′), we can use Polyak-Lojasiewicz (PL) inequality,

$$g^k_t \left( \mathbf{u}^{k-1}_t, \mathbf{v} \right) - \frac{1}{2\mu} \left\| \nabla_{\mathbf{u}} g^k_t \left( \mathbf{u}^{k-1}, \mathbf{v} \right) \right\|^2 \leq g^k_t \left( \mathbf{u}^k_{t,*}, \mathbf{v} \right), \tag{368}$$

thus,

$$2\mu \left( g^k_t \left( \mathbf{u}^{k-1}_t, \mathbf{v} \right) - g^k_t \left( \mathbf{u}^k_{t,*}, \mathbf{v} \right) \right) \leq \left\| \nabla_{\mathbf{u}} g^k_t \left( \mathbf{u}^{k-1}, \mathbf{v} \right) \right\|^2. \tag{369}$$

Combining Eq. (367) and Eq. (369), we have

$$\left\| \nabla_{\mathbf{u}} g^k_t \left( \mathbf{u}^{k-1}, \mathbf{v} \right) \right\|^2 \leq \frac{L}{\mu} \alpha \left\| \nabla_{\mathbf{u}} g^{k-1}_t \left( \mathbf{u}^{k-1}, \mathbf{v} \right) \right\|^2, \tag{370}$$

thus,

$$\left\| \nabla_{\mathbf{u}} g^k_t (\mathbf{u}^k_t, \mathbf{v}) \right\| \leq \sqrt{\alpha \kappa} \left\| \nabla_{\mathbf{u}} g^k_t (\mathbf{u}^{k-1}, \mathbf{v}) \right\|. \tag{371}$$

$\square$

**Lemma H.3.** *Suppose that Assumptions 5′, 7′, 9′ and 10′ hold with $G^2 = 0$. Then,*

$$g^k \left( \mathbf{u}^k, \mathbf{v}^k \right) - g^k \left( \mathbf{u}^k_*, \mathbf{v}^k \right) \leq \tilde{\alpha} \times \left\{ g^k \left( \mathbf{u}^{k-1}, \mathbf{v}^{k-1} \right) - g^k \left( \mathbf{u}^k_*, \mathbf{v}^k \right) \right\}, \tag{372}$$

*where $\tilde{\alpha} = \beta^2 \kappa^4 \alpha$, and $\mathbf{u}^k_* \triangleq \arg\min_{\mathbf{u}} g^k \left( \mathbf{u}, \mathbf{v}^k_{1:T} \right)$ where $g^k$ is defined in (98)*

*Proof.* Consider $k > 0$ and $t \in [T]$. Since $g_t$ is $\mu$-convex in $\mathbf{u}$ (Assumption 10′), we write

$$\left\| \mathbf{u}^k_t - \mathbf{u}^k_* \right\|_F \leq \frac{1}{\mu} \left\| \nabla_{\mathbf{u}} g^k_t \left( \mathbf{u}^k_t, \mathbf{v}^k_t \right) - \nabla_{\mathbf{u}} g^k_t \left( \mathbf{u}^k_*, \mathbf{v}^k_t \right) \right\| \tag{373}$$

$$\leq \frac{1}{\mu} \left\| \nabla_{\mathbf{u}} g^k_t \left( \mathbf{u}^k_t, \mathbf{v}^k_t \right) \right\| + \frac{1}{\mu} \left\| \nabla_{\mathbf{u}} g^k_t \left( \mathbf{u}^k_*, \mathbf{v}^k_t \right) \right\| \tag{374}$$

$$\leq \frac{\sqrt{\alpha \kappa}}{\mu} \left\| \nabla_{\mathbf{u}} g^k_t \left( \mathbf{u}^{k-1}, \mathbf{v}^k_t \right) \right\| + \frac{1}{\mu} \left\| \nabla_{\mathbf{u}} g^k_t \left( \mathbf{u}^k_*, \mathbf{v}^k_t \right) \right\|, \tag{375}$$

where the last inequality is a result of Lemma H.2. Using Jensen inequality, we have

$$\left\| \mathbf{u}^k - \mathbf{u}^k_* \right\|_F = \left\| \sum_{t=1}^{T} \omega_t \cdot \left( \mathbf{u}^k_t - \mathbf{u}^k_* \right) \right\| \tag{376}$$

$$\leq \sum_{t=1}^{T} \omega_t \cdot \left\| \mathbf{u}_t^k - \mathbf{u}_*^k \right\| \tag{377}$$

$$\leq \sum_{t=1}^{T} \omega_t \cdot \left\{ \frac{\sqrt{\alpha\kappa}}{\mu} \left\| \nabla_{\mathbf{u}} g_t^k \left( \mathbf{u}^{k-1}, \mathbf{v}_t^k \right) \right\| + \frac{1}{\mu} \left\| \nabla_{\mathbf{u}} g_t^k \left( \mathbf{u}_*^k, \mathbf{v}_t^k \right) \right\| \right\}. \tag{378}$$

Using Assumption 7′ and Jensen inequality with the "$\sqrt{\cdot}$" function, it follows that

$$\left\| \mathbf{u}^k - \mathbf{u}_*^k \right\| \leq \sqrt{\alpha\kappa} \frac{\beta}{\mu} \left\| \nabla_{\mathbf{u}} g^k \left( \mathbf{u}^k, \mathbf{v}_{1:T}^k \right) \right\| + \frac{\beta}{\mu} \left\| \nabla_{\mathbf{u}} g^k \left( \mathbf{u}_*^k, \mathbf{v}_{1:T}^k \right) \right\| \tag{379}$$

$$= \sqrt{\alpha\kappa} \frac{\beta}{\mu} \left\| \nabla_{\mathbf{u}} g^k \left( \mathbf{u}^{k-1}, \mathbf{v}_{1:T}^k \right) \right\|. \tag{380}$$

Since $g^k$ is $L$-smooth in $\mathbf{u}$ as a convex combination of $L$-smooth function, we have

$$\left\| \nabla_{\mathbf{u}} g^k \left( \mathbf{u}^k, \mathbf{v}_{1:T}^k \right) \right\| = \left\| \nabla_{\mathbf{u}} g^k \left( \mathbf{u}^{k-1}, \mathbf{v}_{1:T}^k \right) - \nabla_{\mathbf{u}} g^k \left( \mathbf{u}_*^k, \mathbf{v}_{1:T}^k \right) \right\| \tag{381}$$

$$\leq L \left\| \mathbf{u}^k - \mathbf{u}_*^k \right\| \tag{382}$$

$$\leq \beta \sqrt{\alpha\kappa^3} \left\| \nabla_{\mathbf{u}} g^k \left( \mathbf{u}^{k-1}, \mathbf{v}_{1:T}^k \right) \right\|. \tag{383}$$

Using Polyak-Lojasiewicz (PL), we have

$$g^k \left( \mathbf{u}^k, \mathbf{v}_{1:T}^k \right) - g^k \left( \mathbf{u}_*^k, \mathbf{v}_{1:T}^k \right) \leq \frac{1}{2\mu} \left\| \nabla_{\mathbf{u}} g^k \left( \mathbf{u}^k, \mathbf{v}_{1:T}^k \right) \right\|^2 \leq \frac{\beta^2 \alpha \kappa^3}{2\mu} \left\| \nabla_{\mathbf{u}} g^k \left( \mathbf{u}^{k-1}, \mathbf{v}_{1:T}^k \right) \right\|^2. \tag{384}$$

Using the $L$-smoothness of $g^k$ in $\mathbf{u}$, we have

$$\left\| \nabla_{\mathbf{u}} g^k \left( \mathbf{u}^{k-1}, \mathbf{v}_{1:T}^k \right) \right\|^2 \leq 2L \left[ g^k \left( \mathbf{u}^{k-1}, \mathbf{v}_{1:T}^k \right) - g^k \left( \mathbf{u}_*^k, \mathbf{v}_{1:T}^k \right) \right]. \tag{385}$$

Thus,

$$g^k \left( \mathbf{u}^k, \mathbf{v}_{1:T}^k \right) - g^k \left( \mathbf{u}_*^k, \mathbf{v}_{1:T}^k \right) \leq \underbrace{\beta^2 \kappa^4 \alpha}_{\triangleq \tilde{\alpha}} \left( g^k \left( \mathbf{u}^{k-1}, \mathbf{v}_{1:T}^k \right) - g^k \left( \mathbf{u}_*^k, \mathbf{v}_{1:T}^k \right) \right). \tag{386}$$

Since $\mathbf{v}_t^k = \arg\min_{v \in \mathcal{V}} g_t^k \left( \mathbf{u}^{k-1}, \mathbf{v} \right)$, it follows that

$$g_t^k \left( \mathbf{u}^{k-1}, \mathbf{v}_t^k \right) \leq g_t^k \left( \mathbf{u}^{k-1}, \mathbf{v}_t^{k-1} \right). \tag{387}$$

Thus,

$$g^k \left( \mathbf{u}^k, \mathbf{v}_{1:T}^k \right) - g^k \left( \mathbf{u}_*^k, \mathbf{v}_{1:T}^k \right) \leq \tilde{\alpha} \times \left\{ g^k \left( \mathbf{u}^{k-1}, \mathbf{v}_{1:T}^{k-1} \right) - g^k \left( \mathbf{u}_*^k, \mathbf{v}_{1:T}^k \right) \right\}. \tag{388}$$
$$\square$$

For $t \in [T]$ and $k > 0$, we introduce $r_t^k \triangleq g_t^k - f_t$ and $r^k \triangleq g^k - f = \sum_{t=1}^{T} \omega_t \left( g_t^k - f_t \right)$. Since $g_t^k$ is a partial first-order surrogate of $f_t$, it follows that $r_t^k \left( \mathbf{u}^{k-1}, \mathbf{v}_t^{k-1} \right) = 0$ and that $r_t^k$ is non-negative and $L$-smooth in $\mathbf{u}$.

**Lemma H.4.** *Suppose that Assumptions 4′ and 5′ hold and that*

$$g^k(\mathbf{u}^k, \mathbf{v}_{1:T}^k) \leq g^k(\mathbf{u}^{k-1}, \mathbf{v}_{1:T}^{k-1}), \ \forall k > 0, \tag{389}$$

*then*

$$\lim_{k \to \infty} r^k(\mathbf{u}^k, \mathbf{v}_{1:T}^k) = 0 \tag{390}$$

$$\lim_{k \to \infty} \left\| \nabla_{\mathbf{u}} r^k(\mathbf{u}^k, \mathbf{v}_{1:T}^k) \right\|^2 = 0 \tag{391}$$

*If we moreover suppose that Assumption 10′ holds and that there exists $0 < \tilde{\alpha} < 1$ such that for all $k > 0$,*

$$g^k(\mathbf{u}^k, \mathbf{v}_{1:T}^k) - g^k(\mathbf{u}_*^k, \mathbf{v}_{1:T}^k) \leq \tilde{\alpha} \times \left( g^k(\mathbf{u}^{k-1}, \mathbf{v}_{1:T}^{k-1}) - g^k(\mathbf{u}_*^k, \mathbf{v}_{1:T}^k) \right), \tag{392}$$

*then,*

$$\lim_{k \to \infty} \left\| \mathbf{u}^k - \mathbf{u}_*^k \right\|^2 = 0 \tag{393}$$

*where $\mathbf{u}_*^k$ is the minimizer of $\mathbf{u} \mapsto g^k \left( \mathbf{u}, \mathbf{v}_{1:T}^k \right)$.*

*Proof.* Since $g_t$ is a partial first-order surrogate of $f$ near $\{\mathbf{u}^{k-1}, \mathbf{v}_t^{k-1}\}$ for $t \in [T]$ and $k > 0$, it follows that $g^k$ is a majorant of $f$ and that $g^k(\mathbf{u}^{k-1}, \mathbf{v}^{k-1}) = f(\mathbf{u}^{k-1}, \mathbf{v}^{k-1})$. Thus, the following holds,

$$f(\mathbf{u}^k, \mathbf{v}^k) \leq g^k(\mathbf{u}^k, \mathbf{v}^k) \leq g^k(\mathbf{u}^{k-1}, \mathbf{v}^{k-1}) = f(\mathbf{u}^{k-1}, \mathbf{v}^{k-1}), \tag{394}$$

It follows that the sequence $\left(f\left(\mathbf{u}^k, \mathbf{v}^k\right)\right)_{k \geq 0}$ is a non-increasing sequence. Since $f$ is bounded below (Assum. 4′), it follows that $\left(f\left(\mathbf{u}^k, \mathbf{v}^k\right)\right)_{k \geq 0}$ is convergent. Denote by $f^\infty$ its limit. The sequence $\left(g^k(\mathbf{u}^k, \mathbf{v}^k)\right)_{k \geq 0}$ also converges to $f^\infty$.

**Proof of Eq. 390**  Using the fact that $g^k(\mathbf{u}^k, \mathbf{v}^k) \leq g^k(\mathbf{u}^{k-1}, \mathbf{v}^k)$, we write for $k > 0$,

$$f(\mathbf{u}^k, \mathbf{v}_{1:T}^k) + r^k(\mathbf{u}^k, \mathbf{v}_{1:T}^k) = g^k(\mathbf{u}^k, \mathbf{v}_{1:T}^k) \leq g^k(\mathbf{u}^{k-1}, \mathbf{v}_{1:T}^{k-1}) = f(\mathbf{u}^{k-1}, \mathbf{v}_{1:T}^{k-1}), \tag{395}$$

Thus,

$$r^k(\mathbf{u}^k, \mathbf{v}_{1:T}^k) \leq f(\mathbf{u}^{k-1}, \mathbf{v}_{1:T}^{k-1}) - f(\mathbf{u}^k, \mathbf{v}^k), \tag{396}$$

By summing over $k$ then passing to the limit when $k \to +\infty$, we have

$$\sum_{k=1}^{\infty} r^k(\mathbf{u}^k, \mathbf{v}_{1:T}^k) \leq f(\mathbf{u}^0, \mathbf{v}_{1:T}^0) - f^\infty, \tag{397}$$

Finally since $r^k(\mathbf{u}^k, \mathbf{v}_{1:T}^k)$ is non negative for $k > 0$, the sequence $\left(r^k(\mathbf{u}^k, \mathbf{v}_{1:T}^k)\right)_{k \geq 0}$ necessarily converges to zero, i.e.,

$$\lim_{k \to \infty} r^k(\mathbf{u}^k, \mathbf{v}_{1:T}^k) = 0. \tag{398}$$

**Proof of Eq. 391**  Because the $L$-smoothness of $\mathbf{u} \mapsto r^k\left(\mathbf{u}, \mathbf{v}_{1:T}^k\right)$, we have

$$r^k\left(\mathbf{u}^k - \frac{1}{L}\nabla_\mathbf{u} r^k\left(\mathbf{u}^k, \mathbf{v}_{1:T}^k\right), \mathbf{v}_{1:T}^k\right) \leq r^k\left(\mathbf{u}^k, \mathbf{v}_{1:T}^k\right) - \frac{1}{2L}\left\|\nabla_\mathbf{u} r^k\left(\mathbf{u}^k, \mathbf{v}_{1:T}^k\right)\right\|^2 \tag{399}$$

Thus,

$$\left\|\nabla_\mathbf{u} r^k\left(\mathbf{u}^k, \mathbf{v}_{1:T}^k\right)\right\|_F^2 \leq 2L\left(r^k\left(\mathbf{u}^k, \mathbf{v}_{1:T}^k\right) - r^k\left(\mathbf{u}^k - \frac{1}{L}\nabla_\mathbf{u} r^k\left(\mathbf{u}^k, \mathbf{v}_{1:T}^k\right), \mathbf{v}_{1:T}^k\right)\right) \tag{400}$$

$$\leq 2L r^k\left(\mathbf{u}^k, \mathbf{v}_{1:T}^k\right), \tag{401}$$

because $r^k$ is a non-negative function (Definition 1). Finally, using Eq. (390), it follows that

$$\lim_{k \to \infty}\left\|\nabla_\mathbf{u} r^k(\mathbf{u}^k, \mathbf{v}_{1:T}^k)\right\|^2 = 0. \tag{402}$$

**Proof of Eq. 393**  We suppose now that there exists $0 < \tilde{\alpha} < 1$ such that

$$\forall k > 0, \;\; g^k(\mathbf{u}^k, \mathbf{v}_{1:T}^k) - g^k(\mathbf{u}_*^k, \mathbf{v}_{1:T}^k) \leq \tilde{\alpha}\left(g^k(\mathbf{u}^{k-1}, \mathbf{v}_{1:T}^{k-1}) - g^k(\mathbf{u}_*^k, \mathbf{v}_{1:T}^k)\right), \tag{403}$$

It follows that,

$$g^k(\mathbf{u}^k, \mathbf{v}_{1:T}^k) - \tilde{\alpha} g^k(\mathbf{u}^{k-1}, \mathbf{v}_{1:T}^{k-1}) \leq (1 - \tilde{\alpha}) g^k(\mathbf{u}_*^k, \mathbf{v}_{1:T}^k), \tag{404}$$

then,

$$g^k(\mathbf{u}_*^k, \mathbf{v}_{1:T}^k) \geq \frac{1}{1 - \tilde{\alpha}} \times \left[g^k(\mathbf{u}^k, \mathbf{v}_{1:T}^k) - \tilde{\alpha} \times g^k(\mathbf{u}^{k-1}, \mathbf{v}_{1:T}^{k-1})\right], \tag{405}$$

and by using the definition of $g^k$ we have,

$$g^k(\mathbf{u}_*^k, \mathbf{v}_{1:T}^k) \geq \frac{1}{1 - \tilde{\alpha}} \times \left[g^k(\mathbf{u}^k, \mathbf{v}_{1:T}^k) - \tilde{\alpha} \times f(\mathbf{u}^{k-1}, \mathbf{v}_{1:T}^{k-1})\right], \tag{406}$$

Since $g^k\left(\mathbf{u}_*^k, \mathbf{v}_{1:T}^k\right) \leq g^k\left(\mathbf{u}^k, \mathbf{v}_{1:T}^k\right) \leq g^k\left(\mathbf{u}^{k-1}, \mathbf{v}_{1:T}^{k-1}\right)$, we have

$$g^k(\mathbf{u}_*^k, \mathbf{v}_{1:T}^k) \leq g^k(\mathbf{u}^{k-1}, \mathbf{v}_{1:T}^{k-1}) = f(\mathbf{u}^{k-1}, \mathbf{v}_{1:T}^{k-1}). \tag{407}$$

From Eq. (406) and Eq. (407), it follows that,

$$\frac{1}{1-\tilde{\alpha}} \times \left[ g^k(\mathbf{u}^k, \mathbf{v}^k_{1:T}) - \tilde{\alpha} \times f(\mathbf{u}^{k-1}, \mathbf{v}^{k-1}_{1:T}) \right] \le g^k(\mathbf{u}^k_*, \mathbf{v}^k_{1:T}) \le f(\mathbf{u}^{k-1}, \mathbf{v}^{k-1}_{1:T}), \quad (408)$$

Finally, since $f\left(\mathbf{u}^{k-1}, \mathbf{v}^{k-1}_{1:T}\right) \xrightarrow[k\to+\infty]{} f^\infty$ and $g^k\left(\mathbf{u}^k, \mathbf{v}^k_{1:T}\right) \xrightarrow[k\to+\infty]{} f^\infty$, it follows from Eq. (408) that,

$$\lim_{k\to\infty} g^k\left(\mathbf{u}^k_*, \mathbf{v}^k_{1:T}\right) = f^\infty. \quad (409)$$

Since $g^k$ is $\mu$-strongly convex in $\mathbf{u}$ (Assumption 10), we write

$$\frac{\mu}{2} \left\| \mathbf{u}^k - \mathbf{u}^k_* \right\|^2 \le g^k\left(\mathbf{u}^k, \mathbf{v}^k_{1:T}\right) - g^k\left(\mathbf{u}^k_*, \mathbf{v}^k_{1:T}\right), \quad (410)$$

It follows that,

$$\lim_{k\to+\infty} \left\| \mathbf{u}^k - \mathbf{u}^k_* \right\|^2 = 0. \quad (411)$$

$\square$

## H.2 Proof of Theorem H.1′

Combining the previous lemmas we prove the convergence of Alg. 3 with a black box solver.

**Theorem H.1′.** *Suppose that Assumptions 4′–7′, Assumptions 9′ and 10′ hold with $G^2 = 0$ and $\alpha \le \frac{1}{\beta^2 \kappa^4}$, then the updates of federated surrogate optimization (Alg. 3) converge to a stationary point of $f$, i.e.,*

$$\lim_{k\to+\infty} \left\| \nabla_{\mathbf{u}} f(\mathbf{u}^k, \mathbf{v}^k_{1:T}) \right\|^2 = 0, \quad (412)$$

*and,*

$$\lim_{k\to+\infty} \sum_{t=1}^{T} \omega_t \cdot d_\mathcal{V}\left(\mathbf{v}^k_t, \mathbf{v}^{k-1}_t\right) = 0. \quad (413)$$

*Proof.*

$$f(\mathbf{u}^k, \mathbf{v}^k_{1:T}) = g^k(\mathbf{u}^k, \mathbf{v}^k_{1:T}) - r^k(\mathbf{u}^k, \mathbf{v}^k_{1:T}). \quad (414)$$

Computing the gradient norm, we have,

$$\left\| \nabla_{\mathbf{u}} f(\mathbf{u}^k, \mathbf{v}^k_{1:T}) \right\| = \left\| \nabla_{\mathbf{u}} g^k(\mathbf{u}^k, \mathbf{v}^k_{1:T}) - \nabla_{\mathbf{u}} r^k(\mathbf{u}^k, \mathbf{v}^k_{1:T}) \right\| \quad (415)$$

$$\le \left\| \nabla_{\mathbf{u}} g^k(\mathbf{u}^k, \mathbf{v}^k_{1:T}) \right\| + \left\| \nabla_{\mathbf{u}} r^k(\mathbf{u}^k, \mathbf{v}^k_{1:T}) \right\|. \quad (416)$$

Since $g^k$ is $L$-smooth in $\mathbf{u}$, we write

$$\left\| \nabla_{\mathbf{u}} g^k(\mathbf{u}^k, \mathbf{v}^k_{1:T}) \right\| = \left\| \nabla_{\mathbf{u}} g^k(\mathbf{u}^k, \mathbf{v}^k) - \nabla_{\mathbf{u}} g^k(\mathbf{u}^k_*, \mathbf{v}^k_{1:T}) \right\| \quad (417)$$

$$\le L \left\| \mathbf{u}^k - \mathbf{u}^k_* \right\|. \quad (418)$$

Thus by replacing Eq. (418) in Eq. (416), we have

$$\left\| \nabla_{\mathbf{u}} f(\mathbf{u}^k, \mathbf{v}^k_{1:T}) \right\| \le L^2 \left\| \mathbf{u}^k - \mathbf{u}^k_* \right\|^2 + \left\| \nabla_{\mathbf{u}} r^k(\mathbf{u}^k, \mathbf{v}^k_{1:T}) \right\|. \quad (419)$$

Using Lemma H.3, there exists $0 < \tilde{\alpha} < 1$, such that

$$\left[ g^k(\mathbf{u}^k, \mathbf{v}^k_{1:T}) - g^k(\mathbf{u}^k_*, \mathbf{v}^k_{1:T}) \right] \le \tilde{\alpha} \times \left[ g^k(\mathbf{u}^{k-1}, \mathbf{v}^{k-1}_{1:T}) - g^k(\mathbf{u}^k_*, \mathbf{v}^k_{1:T}) \right]. \quad (420)$$

Thus, the conditions of Lemma H.4 hold, and we can use Eq. (391) and (393), i.e.

$$\left\| \nabla_{\mathbf{u}} r^k(\mathbf{u}^k, \mathbf{v}^k_{1:T}) \right\|^2 \xrightarrow[k\to+\infty]{} 0 \quad (421)$$

$$\left\| \mathbf{u}^k - \mathbf{u}^k_* \right\|^2 \xrightarrow[k\to+\infty]{} 0. \quad (422)$$

Finally, combining this with Eq. (419), we get the final result

$$\lim_{k\to+\infty} \left\| \nabla_{\mathbf{u}} f(\mathbf{u}^k, \mathbf{v}^k_{1:T}) \right\| = 0. \quad (423)$$

Since $g_t^k$ is a partial first-order surrogate of $f_t$ near $\{\mathbf{u}^{k-1}, \mathbf{v}_t^{k-1}\}$ for $k > 0$ and $t \in [T]$, it follows that

$$\sum_{t=1}^{T} \omega \cdot d_{\mathcal{V}}\left(\mathbf{v}_t^k, \mathbf{v}_t^{k-1}\right) = g^k\left(\mathbf{u}^{k-1}, \mathbf{v}_{1:T}^{k-1}\right) - g^k\left(\mathbf{u}^{k-1}, \mathbf{v}_{1:T}^k\right) \tag{424}$$

$$\leq g^k\left(\mathbf{u}^{k-1}, \mathbf{v}_{1:T}^{k-1}\right) - g^k\left(\mathbf{u}^k, \mathbf{v}_{1:T}^k\right) \tag{425}$$

Thus,

$$\sum_{t=1}^{T} \omega_t \cdot d_{\mathcal{V}}\left(\mathbf{v}_t^k, \mathbf{v}_t^{k-1}\right) \leq f\left(\mathbf{u}^{k-1}, \mathbf{v}_{1:T}^{k-1}\right) - f\left(\mathbf{u}^k, \mathbf{v}_{1:T}^k\right) \tag{426}$$

Since $d_{\mathcal{V}}\left(\mathbf{v}_t^k, \mathbf{v}_t^{k-1}\right)$ is non-negative for $k > 0$ and $t \in [T]$, it follows that

$$\lim_{k \to +\infty} \sum_{t=1}^{T} \omega_t \cdot d_{\mathcal{V}}\left(\mathbf{v}_t^k, \mathbf{v}_t^{k-1}\right) = 0 \tag{427}$$

$\square$

### H.3  Proof of Theorem H.1

**Theorem H.1.** *Suppose that Assumptions 1–7 and Assumptions 9, 10 hold with $G^2 = 0$ and $\alpha \leq \frac{1}{\beta^2 \kappa^5}$, then the updates of* `FedEM` *(Alg. 2) converge to a stationary point of $f$, i.e.,*

$$\lim_{k \to +\infty} \left\|\nabla_\Theta f(\Theta^k, \Pi^k)\right\|_F^2 = 0, \tag{428}$$

*and,*

$$\lim_{k \to +\infty} \sum_{t=1}^{T} \frac{n_t}{n} \mathcal{KL}\left(\pi_t^k, \pi_t^{k-1}\right) = 0. \tag{429}$$

*Proof.* We prove this result as a particular case of Theorem H.1'. To this purpose, we consider that $\mathcal{V} \triangleq \Delta^M$, $u = \Theta \in \mathbb{R}^{dM}$, $v_t = \pi_t$, and $\omega_t = n_t/n$ for $t \in [T]$. For $k > 0$, we define $g_t^k$ as follow,

$$g_t^k\left(\Theta, \pi_t\right) = \frac{1}{n_t} \sum_{i=1}^{n_t} \sum_{m=1}^{M} q_t^k\left(z_t^{(i)} = m\right) \cdot \left( l\left(h_{\theta_m}(\mathbf{x}_t^{(i)}), y_t^{(i)}\right) - \log p_m(\mathbf{x}_t^{(i)}) - \log \pi_t \right.$$

$$\left. + \log q_t^k\left(z_t^{(i)} = m\right) - c\right), \tag{430}$$

where $c$ is the same constant appearing in Assumption 3, Eq. (3). With this definition, it is easy to check that the federated surrogate optimization algorithm (Alg. 3) reduces to `FedEM` (Alg. 2). Theorem H.1 then follows immediately from Theorem H.1', once we verify that $\left(g_t^k\right)_{1 \leq t \leq T}$ satisfy the assumptions of Theorem H.1'.

Assumption 4', Assumption 6', Assumption 7', Assumption 9' and Assumption 10' follow directly from Assumption 4, Assumption 6, Assumption 7, Assumption 9 and Assumption 10, respectively. Lemma G.3 shows that for $k > 0$, $g^k$ is smooth w.r.t. $\Theta$ and then Assumption 5' is satisfied. Finally, Lemmas G.4–G.6 show that for $t \in [T]$ $g_t^k$ is a partial first-order surrogate of $f_t$ w.r.t. $\Theta$ near $\left\{\Theta^{k-1}, \pi_t\right\}$ with $d_{\mathcal{V}}(\cdot, \cdot) = \mathcal{KL}(\cdot \| \cdot)$. $\square$

# I Details on Experimental Setup

## I.1 Datasets and Models

In this section we provide detailed description of the datasets and models used in our experiments. We used a synthetic dataset, verifying Assumptions 1-3, and five "real" datasets (CIFAR-10/CIFAR-100 [33], sub part of EMNIST [8], sub part of FEMNIST [7, 47] and Shakespeare [7, 47]) from which, two (FEMNIST and Shakespeare) has natural client partitioning. Below, we give a detailed description of the datasets and the models / tasks considered for each of them.

### I.1.1 CIFAR-10 / CIFAR-100

CIFAR-10 and CIFAR-100 are labeled subsets of the 80 million tiny images dataset. They both share the same $60,000$ input images. CIFAR-100 has a finer labeling, with 100 unique labels, in comparison to CIFAR-10, having 10 unique label. We used Dirichlet allocation [65], with parameter $\alpha = 0.4$ to partition CIFAR-10 among 80 clients. We used Pachinko allocation [54] with parameters $\alpha = 0.4$ and $\beta = 10$ to partition CIFAR-100 on 100 clients. For both of them we train MobileNet-v2 [55] architecture with an additional linear layer. We used TorchVision [45] implementation of MobileNet-v2.

### I.1.2 EMNIST

EMNIST (Extended MNIST) is a 62-class image classification dataset, extending the classic MNIST dataset. In our experiments, we consider $10\%$ of the EMNIST dataset, that we partition using Dirichlet allocation of parameter $\alpha = 0.4$ over 100 clients. We train the same convolutional network as in [54]. The network has two convolutional layers (with $3 \times 3$ kernels), max pooling, and dropout, followed by a 128 unit dense layer.

### I.1.3 FEMNIST

FEMNIST (Federated Extended MNIST) is a 62-class image classification dataset built by partitioning the data of Extended MNIST based on the writer of the digits/characters. In our experiments, we used a subset with $15\%$ of the total number of writers in FEMNIST. We train the same convolutional network as in [54]. The network has two convolutional layers (with $3 \times 3$ kernels), max pooling, and dropout, followed by a 128 unit dense layer.

### I.1.4 Shakespeare

This dataset is built from The Complete Works of William Shakespeare and is partitioned by the speaking roles [47]. In our experiments, we discarded roles with less than two sentences. We consider character-level based language modeling on this dataset. The model takes as input a sequence of 200 English characters and predicts the next character. The model embeds the 80 characters into a learnable 8-dimensional embedding space, and uses two stacked-LSTM layers with 256 hidden units, followed by a densely-connected layer. We also normalized each character by its frequency of appearance.

### I.1.5 Synthetic dataset

Our synthetic dataset has been generated according to Assumptions 1–3 as follows:

1. Sample weight $\pi_t \sim \mathrm{Dir}\,(\alpha)$, $t \in [T]$ from a symmetric Dirichlet distribution of parameter $\alpha \in \mathbb{R}^+$

2. Sample $\theta_m \in \mathbb{R}^d \sim \mathcal{U}\left([-1,1]^d\right)$, $m \in [M]$ for uniform distribution over $[-1,1]^d$.

3. Sample $m_t$, $t \in [T]$ from a log-normal distribution with mean 4 and sigma 2, then set $n_t = \min\,(50 + m_t, 1000)$.

4. For $t \in [T]$ and $i \in [n_t]$, draw $x_t^{(i)} \sim \mathcal{U}\left([-1,1]^d\right)$ and $\epsilon_t^{(i)} \sim \mathcal{N}\,(0, I_d)$.

5. For $t \in [T]$ and $i \in [n_t]$, draw $z_t^{(i)} \sim \mathcal{M}\,(\pi_t)$.

Table 4: Average computation time and used GPU for each dataset.

| Dataset | GPU | Simulation time |
|---|---|---|
| Shakespeare [7, 47] | Quadro RTX 8000 | 4h42min |
| FEMNIST [7] | Quadro RTX 8000 | 1h14min |
| EMNIST [8] | GeForce GTX 1080 Ti | 46min |
| CIFAR10 [33] | GeForce GTX 1080 Ti | 2h37min |
| CIFAR100 [33] | GeForce GTX 1080 Ti | 3h9min |
| Synthetic | GeForce GTX 1080 Ti | 20min |

Table 5: Learning rates $\eta$ used for the experiments in Table 2. Base-10 logarithms are reported.

| Dataset | FedAvg [47] | FedProx [38] | FedAvg+ [27] | Clustered FL [56] | pFedMe [16] | FedEM (Ours) |
|---|---|---|---|---|---|---|
| FEMNIST | $-1.5$ | $-1.5$ | $-1.5$ | $-1.5$ | $-1.5$ | $-1.0$ |
| EMNIST | $-1.5$ | $-1.5$ | $-1.5$ | $-1.5$ | $-1.5$ | $-1.0$ |
| CIFAR10 | $-1.5$ | $-1.5$ | $-1.5$ | $-1.5$ | $-1.0$ | $-1.0$ |
| CIFAR100 | $-1.0$ | $-1.0$ | $-1.0$ | $-1.0$ | $-1.0$ | $-0.5$ |
| Shakespeare | $-1.0$ | $-1.0$ | $-1.0$ | $-1.0$ | $-1.0$ | $-0.5$ |
| Synthetic | $-1.0$ | $-1.0$ | $-1.0$ | $-1.0$ | $-1.0$ | $-1.0$ |

6. For $\in [T]$ and $i \in [n_t]$, draw $y_t^{(i)} \sim \mathcal{B}\left(\text{sigmoid}\left(\langle x_t^{(i)}, \theta_{z_t^{(i)}}\rangle + \epsilon_t^{(i)}\right)\right)$.

## I.2 Implementation Details

### I.2.1 Machines

We ran the experiments on a CPU/GPU cluster, with different GPUs available (e.g., Nvidia Tesla V100, GeForce GTX 1080 Ti, Titan X, Quadro RTX 6000, and Quadro RTX 8000). Most experiments with CIFAR10/CIFAR-100 and EMNIST were run on GeForce GTX 1080 Ti cards, while most experiments with Shakespeare and FEMNIST were run on the Quadro RTX 8000 cards. For each dataset, we ran around 30 experiments (not counting the development/debugging time). Table 4 gives the average amount of time needed to run one simulation for each dataset. The time needed per simulation was extremely long for Shakespeare dataset, because we used a batch size of 128. We remarked that increasing the batch size beyond 128 caused the model to converge to poor local minima, where the model keeps predicting a white space as next character.

### I.2.2 Libraries

We used PyTorch [53] to build and train our models. We also used Torchvision [45] implementation of MobileNet-v2 [55], and for image datasets preprossessing. We used LEAF [7] to build FEMNIST dataset and the federated version of Shakespeare dataset.

### I.2.3 Hyperparameters

For each method and each task, the learning rate was set via grid search on the set $\{10^{-0.5}, 10^{-1}, 10^{-1.5}, 10^{-2}, 10^{-2.5}, 10^{-3}\}$. `FedProx` and `pFedMe`'s penalization parameter $\mu$ was tuned via grid search on $\{10^1, 10^0, 10^{-1}, 10^{-2}, 10^{-3}\}$. For Clustered FL, we used the same values of tolerance as the ones used in its official implementation [56]. We found tuning $\text{tol}_1$ and $\text{tol}_2$ particularly hard: no empirical rule is provided in [56], and the few random setting we tried did not show any improvement in comparison to the default ones. For each dataset and each method, Table 5 reports the learning rate $\eta$ that achieved the corresponding result in Table 2.

Table 6: Test accuracy: average across clients.

| Dataset | Local | FedAvg [47] | FedAvg+ [27] | Clustered FL [56] | pFedMe [16] | FedEM (Ours) | D-FedEM (Ours) |
|---------|-------|-------------|--------------|-------------------|-------------|--------------|----------------|
| FEMNIST | 71.0 | 78.6 | 75.3 | 73.5 | 74.9 | **79.9** | 77.2 |
| EMNIST | 71.9 | 82.6 | 83.1 | 82.7 | 83.3 | 83.5 | **83.5** |
| CIFAR10 | 70.2 | 78.2 | 82.3 | 78.6 | 81.7 | **84.3** | 77.0 |
| CIFAR100 | 31.5 | 40.9 | 39.0 | 41.5 | 41.8 | **44.1** | 43.9 |
| Shakespeare | 32.0 | **46.7** | 40.0 | 46.6 | 41.2 | **46.7** | 45.4 |
| Synthetic | 65.7 | 68.2 | 68.9 | 69.1 | 69.2 | **74.7** | 73.8 |

## J  Additional Experimental Results

### J.1  Fully Decentralized Federated Expectation-Maximization

`D-FedEM` considers the scenario where clients communicate directly in a peer-to-peer fashion instead of relying on the central server mediation. In order to simulate `D-FedEM`, we consider a binomial Erdős-Rényi graph [18] with parameter $p = 0.5$, and we set the mixing weight using *Fast Mixing Markov Chain* [5] rule. We report the result of this experiment in Table 6, showing the average weighted accuracy with weight proportional to local dataset sizes. We observe that `D-FedEM` often performs better than other FL approaches and slightly worst than `FedEM`, except on CIFAR-10 where it has low performances.

### J.2  Comparison with `MOCHA`

In the case of synthetic dataset, for which train a linear model, we compare `FedEM` with `MOCHA` [59]. We implemented `MOCHA` in Python following the official implementation [9] in MATLAB. We tuned the parameter $\lambda$ of `MOCHA` on a holdout validation set via grid search in $\{10^1, 10^0, 10^{-1}, 10^{-2}, 10^{-3}\}$, and we found that the optimal value of $\lambda$ is $10^0$. For this value, we ran `MOCHA` on the synthetic dataset with three different seeds, and we found that the average accuracy is $73.4 \pm 0.05$ in comparison to $74.7 \pm 0.01$ achieved by `FedEM`. Note that `MOCHA` is the second best method after `FedEM` on this dataset. Unfortunately, `MOCHA` only works for linear models.

### J.3  Generalization to Unseen Clients

Table 3 shows that `FedEM` allows new clients to learn a personalized model at least as good as `FedAvg`'s global one and always better than `FedAvg+`'s one. Unexpectedly, new clients achieve sometimes a significantly higher test accuracy than old clients (e.g., 47.5% against 44.1% on CIFAR100).

In order to better understand this difference, we looked at the distribution of `FedEM` personalized weights for the old clients and new ones. The average distribution entropy equals 0.27 and 0.92 for old and new clients, respectively. This difference shows that old clients tend to have more skewed distributions, suggesting that some components may be overfitting the local training dataset leading the old clients to give them a high weight.

We also considered a setting where unseen clients progressively collect their own dataset. We investigate the effect of the number of samples on the average test accuracy across unseen clients, starting from no local data (and therefore using uniform weights to mix the $M$ components) and progressively adding more labeled examples until the full local labeled training set is assumed to be available. Figure 2 shows that `FedEM` achieves a significant level of personalization as soon as clients collect a labeled dataset whose size is about $20\%$ of what the original clients used for training.

As we mentioned in the main text, it is not clear how the other personalized FL algorithms (e.g., pFedMe and Clustered FL) should be extended to handle unseen clients. For example, the global model learned by pFedMe during training can then be used to perform some "fine-tuning" at the new clients, but how exactly? The original `pFedMe` paper [16] does not even mention this issue. For example, the client could use the global model as initial vector for some local SGD steps (similarly to what done in `FedAvg+` or the MAML approaches) or it could perform a local `pFedMe` update (lines 6-9 in [16, Alg. 1]). The problem is even more complex for Clustered FL (and again not discussed in [56]). The new client should be assigned to one of the clusters identified. One can think to compute

---

[9]`https://github.com/gingsmith/fmtl`

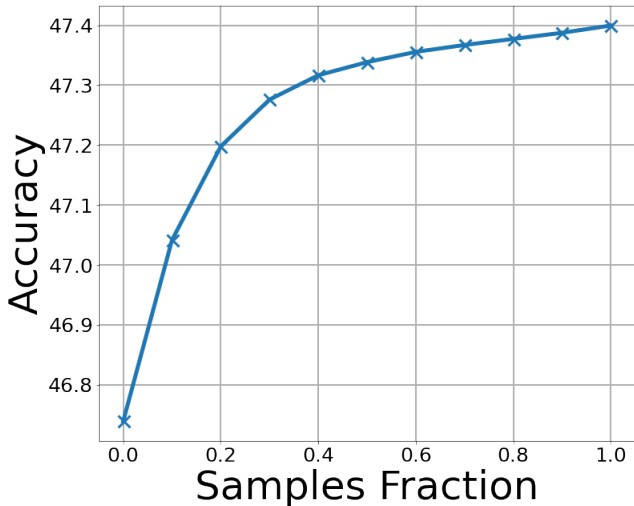

Figure 2: Effect of the number of samples on the average test accuracy across clients unseen at training on CIFAR100 dataset.

the cosine distances of the new client from those who participated in training, but this would require the server to maintain not only the model learned, but also the last-iteration gradients of all clients that participated in the training. Moreover, it is not clear which metric should be considered to assign the new client to a given cluster (perhaps the average cosine similarity from all clients in the cluster?). This is an arbitrary choice as [56] does not provide a criterion to assign clients to a cluster, but only to decide if a given cluster should be split in two new ones. It appears that many options are possible and they deserve separate investigation. Despite these considerations, we performed an additional experiment extending `pFedMe` to unseen clients as described in the second option above on CIFAR-100 dataset with a sampling rate of $20\%$. `pFedMe` achieves a test accuracy of $40.5\% \pm 1.66\%$, in comparison to $38.9\% \pm 0.97\%$ for `FedAvg` and $42.7\% \pm 0.33\%$ for `FedEM`. `FedEM` thus performs better on unseen clients, and `pFedMe`'s accuracy shows a much larger variability.

### J.4  `FedEM` and Clustering

We performed additional experiments with synthetic datasets to check if `FedEM` recovers clusters in practice. We modified the synthetic dataset generation so that the mixture weight vector $\pi_t$ of each client $t$ has a single entry equal to 1 that is selected uniformly at random. We consider two scenarios both with $T = 300$ client, the first with $M = 2$ component and the second with $M = 3$ components. In both cases `FedEM` recovered almost the correct $\Pi^*$ and $\Theta^*$: we have `cosine_distance` $\left( \Theta^*, \breve{\Theta} \right) \leq 10^{-2}$ and `cosine_distance` $\left( \Pi^*, \breve{\Pi} \right) \leq 10^{-8}$. A simple clustering algorithm that assigns each client to the component with the largest mixture weight achieves $100\%$ accuracy, i.e., it partitions the clients in sets coinciding with the original clusters.

### J.5  Effect of $M$ in Time-Constrained Setting

Recall that in `FedEM`, each client needs to update and transmit $M$ components at each round, requiring roughly $M$ times more computation and $M$ times larger messages than the competitors in our study. In this experiment, we considered a challenging time-constrained setting, where `FedEM` is limited to run one third $(= 1/M)$ of the rounds of the other methods. The results in Table 7 show that even if `FedEM` does not reach its maximum accuracy, it still outperforms the other methods on 3 datasets.

We additionally compared `FedEM` with a model having the same number of parameters in order to check if `FedEM`'s advantage comes from the additional model parameters rather than by its specific formulation. To this purpose, we trained Resnet-18 and Resnet-34 on CIFAR10. The first one has about 3 times more parameters than MobileNet-v2 and then roughly as many parameters as FedEM with $M = 3$. The second one has about 6 times more parameters than FedEM with $M = 3$. We

Table 7: Test and train accuracy comparison across different tasks. For each method, the best test accuracy is reported. For `FedEM` we run only $\frac{K}{M}$ rounds, where $K$ is the total number of rounds for other methods–$K = 80$ for Shakespeare and $K = 200$ for all other datasets–and $M = 3$ is the number of components used in `FedEM`.

| Dataset | Local | FedAvg [47] | FedProx [38] | FedAvg+ [27] | Clustered FL [56] | pFedMe [16] | FedEM (Ours) |
|---------|-------|-------------|--------------|--------------|-------------------|-------------|--------------|
| FEMNIST [7] | 71.0 (99.2) | **78.6** (79.5) | 78.6 (79.6) | 75.3 (86.0) | 73.5 (74.3) | 74.9 (91.9) | 74.0 (80.9) |
| EMNIST [8] | 71.9 (99.9) | 82.6 (86.5) | 82.7 (86.6) | 83.1 (93.5) | 82.7 (86.6) | **83.3** (91.1) | 82.7 (89.4) |
| CIFAR10 [33] | 70.2 (99.9) | 78.2 (96.8) | 78.0 (96.7) | 82.3 (98.9) | 78.6 (96.8) | 81.7 (99.8) | **82.5** (92.2) |
| CIFAR100 [33] | 31.5 (99.9) | 41.0 (78.5) | 40.9 (78.6) | 39.0 (76.7) | 41.5 (78.9) | 41.8 (99.6) | **42.0** (72.9) |
| Shakespeare [7] | 32.0 (95.3) | **46.7** (48.7) | 45.7 (47.3) | 40.0 (93.1) | 46.6 (48.7) | 41.2 (42.1) | 43.8 (44.6) |
| Synthetic | 65.7 (91.0) | 68.2 (68.7) | 68.2 (68.7) | 68.9 (71.0) | 69.1 (85.1) | 69.2 (72.8) | **73.2** (74.7) |

observed that both architectures perform even worse than MobileNet-v2, so the comparison with these larger models does not suggest that `FedEM`'s advantage comes from the larger number of parameters.

We note that there are many possible choices of (more complex) model architectures, and finding one that works well for the task at hand is quite challenging due to the large search space, the bias-variance trade-off, and the specificities of the FL setting.

Table 8: Test accuracy under 20% client sampling: average across clients with +/- standard deviation over 3 independent runs. All experiments with 1200 communication rounds.

| Dataset | FedAvg [47] | FedAvg+ [27] | pFedMe [16] | APFL [14] | FedEM (Ours) |
|---------|-------------|--------------|-------------|-----------|--------------|
| CIFAR10 [33] | $73.1 \pm 0.14$ | $77.7 \pm 0.16$ | $77.8 \pm 0.07$ | $78.2 \pm 0.27$ | $\mathbf{82.1} \pm 0.13$ |
| CIFAR100 [33] | $40.6 \pm 0.17$ | $39.7 \pm 0.75$ | $39.9 \pm 0.08$ | $40.3 \pm 0.71$ | $\mathbf{43.2} \pm 0.23$ |
| Synthetic | $68.2 \pm 0.02$ | $69.0 \pm 0.03$ | $69.1 \pm 0.03$ | $69.1 \pm 0.04$ | $\mathbf{74.7} \pm 0.01$ |

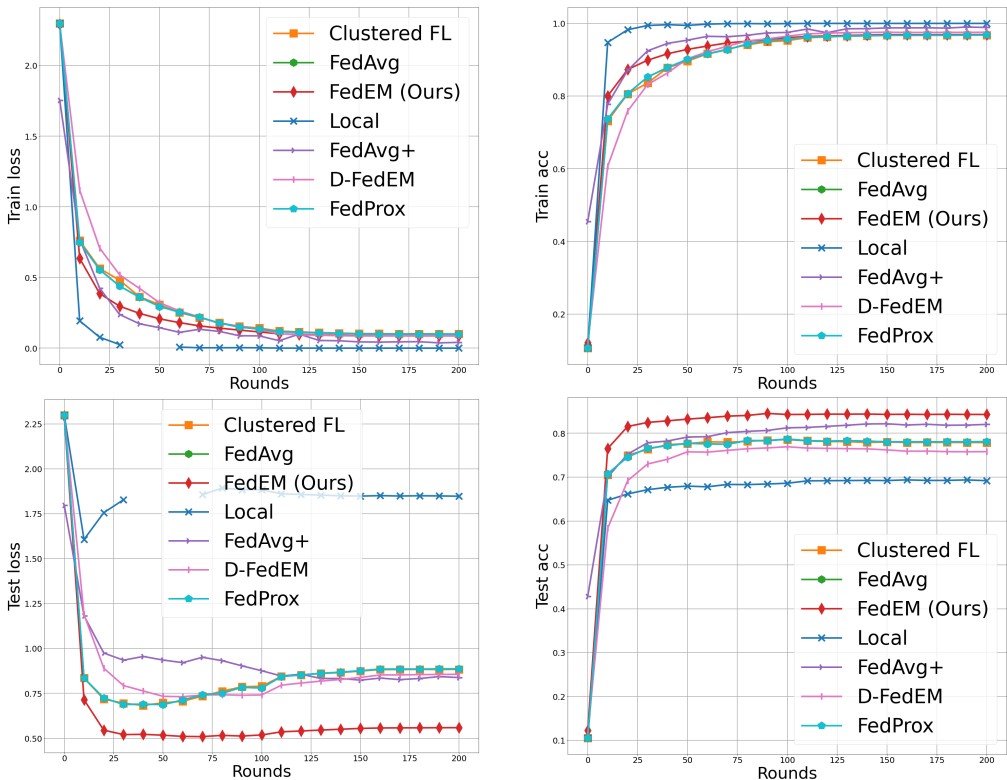

Figure 3: Train loss, train accuracy, test loss, and test accuracy for CIFAR10 [33]. .

## J.6 Additional Results under Client Sampling

In our experiments, except for Figure 1, we considered that all clients participate at each round. We run extra experiments with client sampling, by allowing only 20% of the clients to participate at each round. We also incorporate APFL [14] into the comparison. Table 8 summarizes our findings, giving the average and standard deviation of the test accuracy across 3 independent runs.

## J.7 Convergence Plots

Figures 3 to 8 show the evolution of average train loss, train accuracy, test loss, and test accuracy over time for each experiment shown in Table 2.

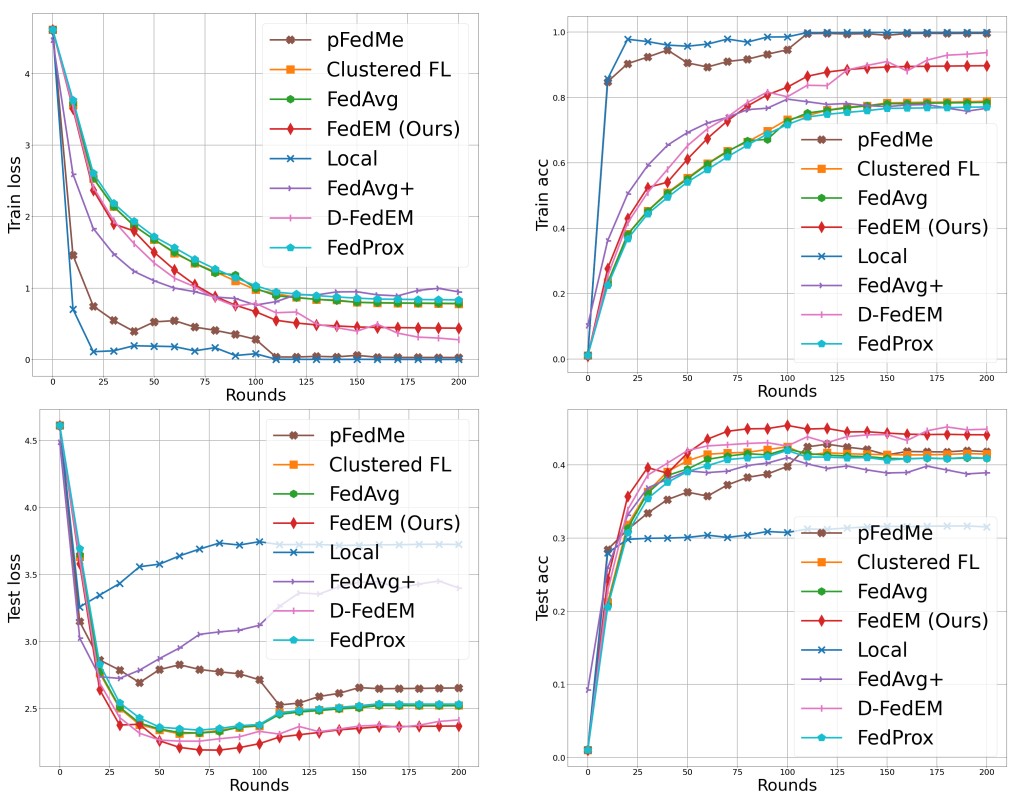

Figure 4: Train loss, train accuracy, test loss, and test accuracy for CIFAR100 [33].

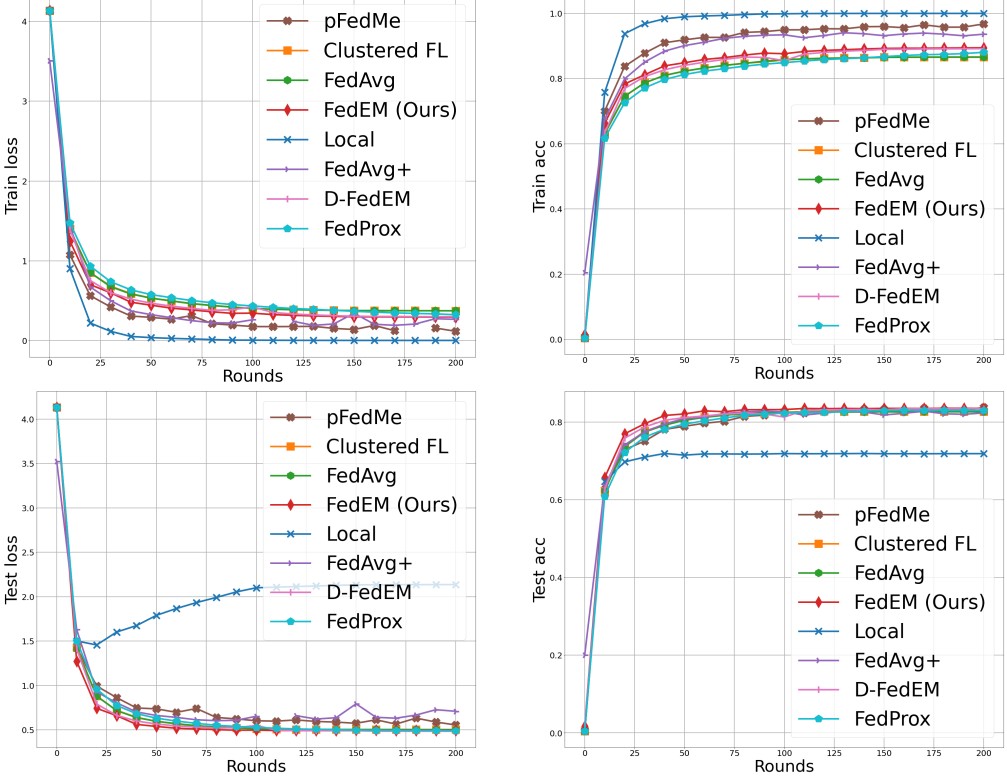

Figure 5: Train loss, train accuracy, test loss, and test accuracy for EMNIST [8].

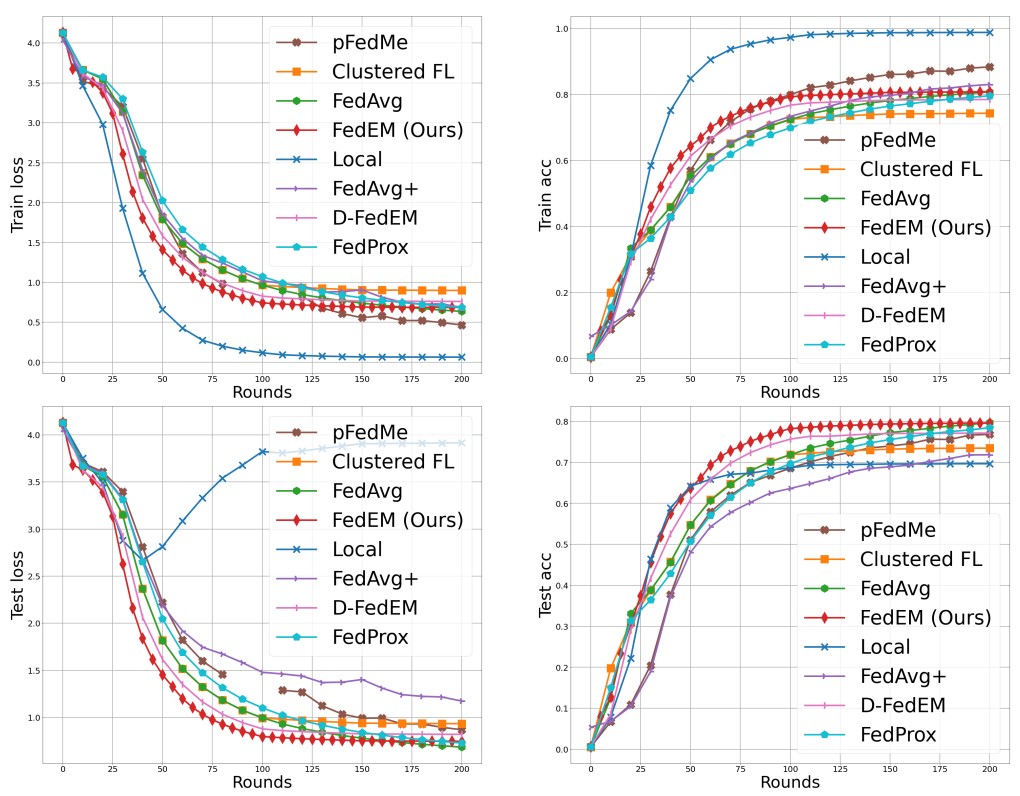

Figure 6: Train loss, train accuracy, test loss, and test accuracy for FEMNIST [7, 47].

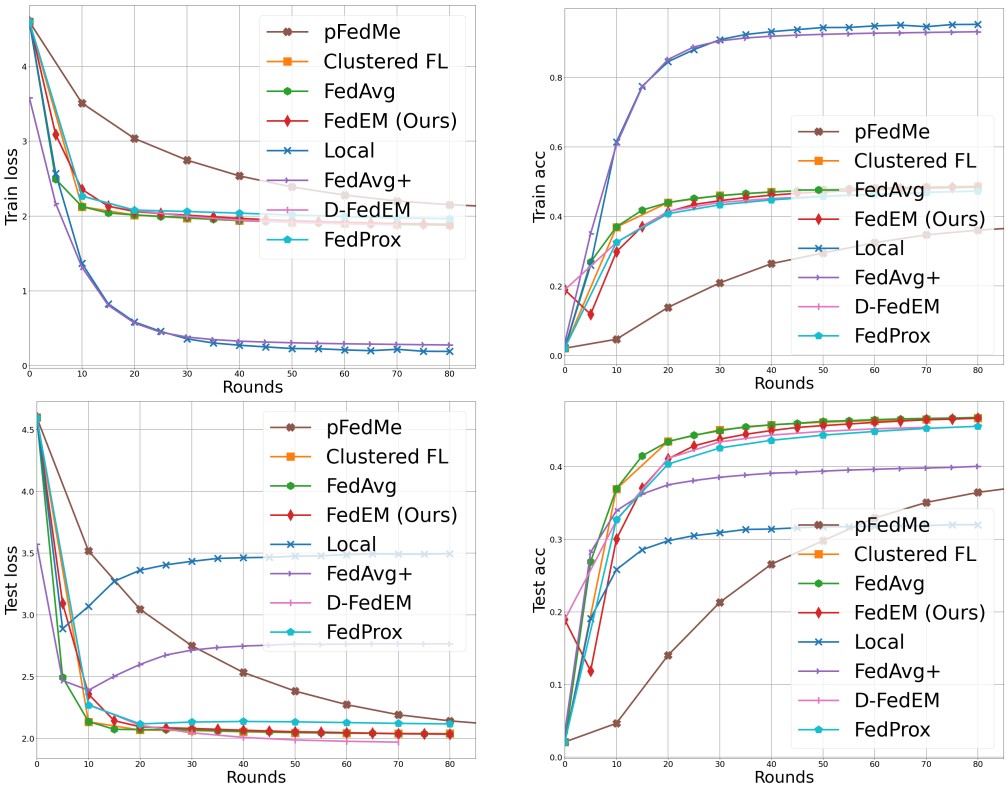

Figure 7: Train loss, train accuracy, test loss, and test accuracy for Shakespeare [7, 47].

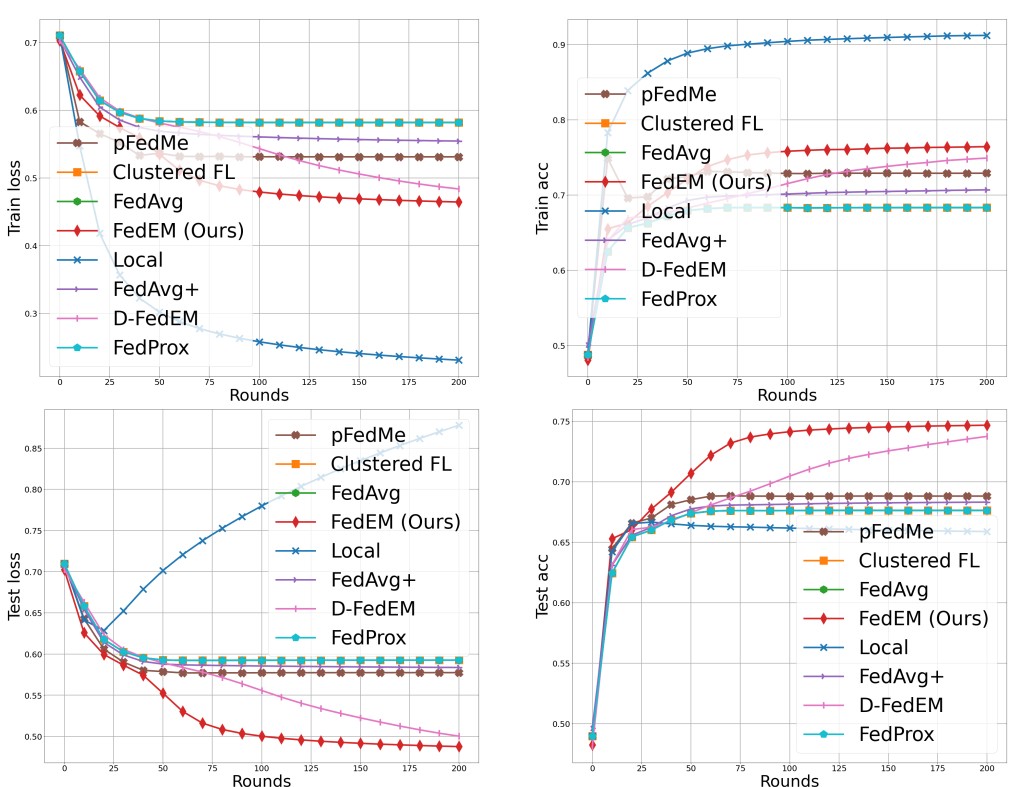

Figure 8: Train loss, train accuracy, test loss, and test accuracy for synthetic dataset.