# OpenReview forum: "Federated Multi-Task Learning under a Mixture of Distributions"
_NeurIPS.cc/2021/Conference — NeurIPS 2021 Poster_

### Official Review · Reviewer_kvAM · 2021-06-25

**Rating:** 4
**Confidence:** 4

**Summary:**

The authors propose a new approach for personalized federated learning (PFL) by formulating PFL as multi-task learning (MTL). by assuming that each local data distribution is a mixture of unknown underlying distributions the model aims to learn a linear combination of clients' models based on their data distribution.

The authors claim the following contributions:
-  Showing that federated learning (FL) is impossible without assumptions on local data distributions
- Proposing FedEM and D-FedEM an expectation-maximization (EM) based approaches for PFL
- The proposed methods achieve better accuracy and generalize better than SOTA PFL methods.





**Limitations And Societal Impact:**

The authors should discuss the possible societal impact of their work.

**Main Review:**

**Pros/Cons**:

[+] The paper is well written and structured.

[+] Good Theoretical analysis of FedEM's convergence.

[+] The proposed methods achieve higher performance compared to previous methods over different datasets and federation sizes.

[-] The authors did not address more recent approaches in PFL [1,2] and did not compare to them. FedFomo[1] aims to learn a linear combination of clients' weights W/O assuming anything about underlying data distributions and pFedHN[2] performs better than pFedMe under identical data heterogeneity learning settings.

[-] The authors should provide STD analysis on their experimental results (i.e. running on multiple seeds)

[-] "the learning rate and the other hyper-parameters were tuned via grid search" - on "Datasets and models" subparagraph the authors describe the data splitting as follow " For all tasks, we randomly split each local dataset into training (80%) and test (20%) sets. **This implies that the HPs were chosen based on the test split!**

[-] Generalization to unseen clients - “it is not clear if and how the other personalized FL algorithms can tackle the same goal.” I totally disagree with this statement. PFL methods can handle situations in which new clients join the federation. Let us take pFedMe for example, after the training process is finished the Hub/Server obtains a global model which is optimized based on the federation. This model can be sent to the new clients to be fine-tuned locally.

[-] All the clients in the federation take part in each communication round - this is a big limitation for large-scale federations, in real life FL systems federations can contain millions of clients such that training this federation will cost lots of computational power and money. Other PFL methods sample a subset of clients at each COM round (most of them sample $5$ clients per round).
This limitation raises another question, have the authors run all methods on the same number of communications? each COM round in FedEM performs $T$ communications and $T$ denotes the number of clients while for pFedMe $5$ communications occur per COM round. In FL we care about the number of communications it takes the model to converge.

**References**:

[1] Zhang, Michael, Karan Sapra, Sanja Fidler, Serena Yeung, and Jose M. Alvarez. "Personalized Federated Learning with First Order Model Optimization." arXiv preprint arXiv:2012.08565 (2020).

[2] Shamsian, Aviv, Aviv Navon, Ethan Fetaya, and Gal Chechik. "Personalized Federated Learning using Hypernetworks." arXiv preprint arXiv:2103.04628 (2021).

**Time Spent Reviewing:**

at least 5 hours

---

> ### Author Response · Authors · 2021-08-10
> **Specific answers to 4th reviewer**
>
> We thank the reviewer for the feedback and valuable references. We have done our best to address his/her concerns and hope that the reviewer will consider raising his/her score.
>
> **STD analysis** Both papers mentioned by the reviewer compute standard deviation over $3$ independent runs.
> We have re-run our experiments with $3$ different seeds.
> For example, the Table in the general answer reports accuracy's variability expressed
> as +/- standard deviation. We observe that the standard deviation is
> always less than $1\\%$ for all methods (e.g., FedEM standard deviation on CIFAR-100 is $0.23\\%$), except for pFedMe when adapted to unseen clients (see discussion below).
>
> **Tuning hyper-parameters** All experiments in the original submission have been redone by tuning the hyperparameters on a validation set. In particular we have split the previous training set into a new training set ($80\\%$) and a new validation set ($20\\%$). We have then trained each approach on this new training set, selected the best hyper-parameters on the validation set, and finally retrained on the union of training and validation sets. Final results are then evaluated on the test set (unseen during hyper-parameter tuning and training). This procedure led to selecting exactly the same hyperparameter values for all experiments except those with CIFAR-100. Even for CIFAR-100, using the new hyperparameters did not induce any significant change in terms of test accuracy (the difference was smaller than $0.05\\%$, and did not change the reported values in Table 2 of the paper).
>
> **Client sampling** The reviewer's statement that in FedEM all clients take
> part in each communication round is not correct. We confirm that FedEM
> supports client sampling as FedAvg, FedAvg+ or pFedMe do. The reviewer
> has probably missed the paragraph on client sampling (lines 315&ndash;320),
> and the corresponding Figure 1 showing the effect of sampling rate on
> the performance of FedEM and FedAvg+. The experiments in pFedMe
> paper \[15\] consider $5$ clients out of $20$ participating at each round
> corresponding to a sampling rate of $25\\%$, which is included in the
> range of values in Figure 1.
> We have run new experimental results when sampling $20\\%$ of the clients
> for CIFAR-100, CIFAR-10, and the synthetic dataset. They are reported
> in the Table in the general answer. These new results qualitatively
> confirm those in our original submission.
>
> **Extension to unseen clients** In the paper we claim that
> "with the exception of FedAvg+, it is not clear if and how the other
> personalized FL algorithms \[i.e., pFedMe and ClusteredFL\] can tackle
> the same goal", i.e., can be extended to unseen clients. We agree with
> the reviewer that the global model learned by pFedMe during training
> can then be used to perform some "fine-tuning" at the new clients,
> but how exactly? The original pFedMe paper \[15\] does not even mention
> this issue. For example, the client could use the global model as initial
> vector for some local SGD steps (similarly to what done in FedAvg+ or
> the MAML approaches) or it could perform a local pFedMe update
> (lines 6-9 in \[15, Alg. 1\]). The problem is even more complex for
> ClusteredFL (and again not discussed in \[55\]). The new client should be
> assigned to one of the clusters identified. One can think to compute
> the cosine distances of the new client from those who participated
> in training, but this would require the server to maintain not only
> the $C$ model learned, but also the last-iteration gradients of all
> clients that participated in the training. Moreover, it is not clear which
> metric should be considered to assign the new client
> to a given cluster (perhaps the average cosine similarity from all clients
> in the cluster?). This is an arbitrary choice as \[55\] does not provide
> a criterion to assign clients to a cluster, but only to decide if a
> given cluster should be split in two new ones. It appears that many
> options are possible and they deserve separate investigation. It would
> be questionable to say that one option is "the way" to extend pFedMe
> or ClusteredFL to unseen clients. On the basis of the reviewer's
> feedback, we can modify our original sentence as follows: "it is not
> clear how the other personalized FL algorithms should be extended
> to tackle the same goal."
>
> We also performed an additional experiment extending pFedMe to unseen
> clients as described in the second option above, i.e., by letting them
> perform a local pFedMe update (lines 6-9 in \[15, Alg. 1\])
> on CIFAR-100 dataset with a sampling rate of $20\\%$. pFedMe achieves
> a test accuracy of $40.5\\%$ +/- $1.66\\%$, in comparison to $38.9\\%$ +/- $ 0.97\\%$ for FedAvg
> and $42.7\\%$ +/- $0.33\\%$ for FedEM. FedEM thus performs better on unseen clients,
> and pFedMe's accuracy shows a much larger variability.
>
> **FedFomo and pFedHN** We thank the reviewer for pointing us to these two recent papers we were not aware of. We will add them to our discussion of related works. Similarly to FedEM, *FedFomo* obtains personalized models as linear combinations of different "component models," but there are many important differences:
>
> 1. FedFomo misses clear assumptions to justify such linear combination,
> while FedEM's linear combination is provably optimal under the assumptions
> in Proposition 2.1.
> 2. FedFomo has no convergence guarantee, while FedEM does.
> 3. In FedFomo there is a component model for each client, while in FedEM
> the number of component models can be set independently
> (and our experiments show that this number can be as small as 3).
> As a consequence, FedFomo requires the presence of a powerful server
> that can store all the component models and learn which models should
> be sent to a given client for personalization. On the contrary, FedEM
> can work in a fully decentralized setting.
> 4. FedFomo requires multiple communication rounds for the server to
> learn which component models are useful to a given client, as a consequence:
>    - FedFomo is not suited for cross-device FL where a client may only
>    participate in a single training round. In fact, FedFomo paper
>    explicitly mentions cross-silo FL as a motivation. Instead,
>    FedEM works in both settings.
>    - FedFomo's adaptation to new clients who did not participate to
>    training would still require these clients to train and communicate
>    with the server over multiple rounds (needed for the server to
>    identify the useful components), while in FedEM a new client only
>    needs to retrieve the trained component models and then compute its
>    own mixture weights through (8) and (9).
> 5. Finally, in FedEM even clients for which a given mixture component
> (say $\tilde{\mathcal D}_m$) has a small weight can still contribute
> to improving the corresponding component model $h_m$. They will then help
> clients whose mixture assigns a large weight to $\tilde{\mathcal D}_m$.
> On the contrary, in FedFomo each client can combine only a
> limited number of components models (the models of the most similar clients).
> FedFomo then potentially clusters groups of similar clients and limits
> knowledge transfer across clusters. We can thus expect that FedEM enjoys
> better generalization bounds than FedFomo.
>
> *pFedHN*'s paper was accepted at ICML after our submission to NeurIPS.
> It is a very interesting alternative approach to personalization
> based on the idea of hypernetworks. We observe that, differently from FedEM,
> pFedHN has no theoretical convergence guarantee (although the authors provide
> interesting generalization bounds). An advantage of pFedHN in comparison
> to FedEM is its ability to jointly train models with different complexity.
> At the same time, pFedHN needs to learn an embedding vector for each
> client through multiple server-client communication steps. This carries
> a few drawbacks that FedEM does not suffer:
>
> 1. pFedHN accuracy is likely to heavily deteriorate in a setting where
> devices participate only once (or a few times) to training,
> as it is the case for large Google-like cross-device FL training,
> because the server will not be able to learn meaningful representations
> for the clients.
> 2. Even once the hypernetwork parameters have been learned, unseen clients
> still need to communicate multiple times with the server to train their
> personalized model, while in FedEM they just need to retrieve the model
> components from the server (a single communication).
> 3. The client's individual embedding vector potentially discloses
> personal information as it allows the server to identify the client
> across multiple communication steps (secure aggregation is not possible).
>
> Also, it is not clear how easily pFedHN can scale to more complex models.
> Training personalized LeNet (about $60$k parameters) requires a
> hypernetwork with $3$ fully-connected layers with $100$ units,
> plus $100$ $\times$ $60$k = $6$ million parameters for the linear heads.
> A model like MobileNet-v2 considered in our paper has $50$ times
> more parameters, which would require to train a hypernetwork with
> $300$ million parameters, assuming that the implicit embedding in the
> $100$-dimensional space is still sufficient to capture the expressivity
> of MobileNet-v2. In comparison, Figure 1 right shows that FedEM needs
> to consider only $M=3$ times more parameters, i.e. $9$ million parameters.
> pFedHN's larger memory requirements may make this approach less suited
> for cross-silo settings, where the participating entities have large
> computation capabilities and may want to train large state-of-the-art models.

---

### Official Review · Reviewer_j2i1 · 2021-07-07

**Rating:** 7
**Confidence:** 4

**Summary:**

In this paper, the authors consider model personalisation as an important aspect of federated learning. Based on an impossibility result, the authors are motivated to learn a mixture model $p(y|x)=\sum_m^M\pi_m p(y|x, \theta_m)$ as opposed to a single mapping $p(y|x,\theta)$. As opposed to standard FedAvg approaches, the mixture weights $\pi_m$ are local, i.e. specific to each client $t$.
The authors propose an EM-type algorithm for learning the parameters $\theta_m$ and $\pi_{m,t}$ in a federated environment, coining their algorithm FedEM. Furthermore they propose a decentralised variant of FedEM.
The authors show convergence proofs under appropriate assumptions and show that FedEM and D-FedEM are special cases of  more general federated surrogate optimization.

== after rebuttal ==

I have read the rebuttal and raised my score accordingly

**Limitations And Societal Impact:**

The authors discuss the 'Fairness' of their method, both in the experimental section as well as in Appendix L. While it is clear to me what the exact definition of fairness is that the authors relate to, I suggest acknowledging the overloaded nature and general unclear definition of that term, also in connection with FL. More specifically it might be the case that "Even clients with the worst personalized models are still better off when FedEM is used for training", however it might still not be 'fair' as some clients benefit more (absolutely or relatively) than others.

**Main Review:**

Thx for presenting this interesting paper in a relevant topic in FL research. I enjoyed reading it, it is presented concisely.

General paper outline:
The paper consists of several elements, the connection of which I would like to see improved. The main technical motivation for FedEM seems to be 2.1, i.e. the impossibility result. The authors state that "The main consequence for FL is that, without further assumptions, a client cannot provably benefit from larger amounts of data available at other clients.". How does this motivate a mixture modelling approach? Further, without having read [3], how is the following not true: Imagine client 1 learning a feature extractor through some unsupervised fashion on a small vs. a huge amount of data. Client 2 will benefit from the shared feature extractor trained on a large data-set more than from the model trained on the small data-set, presumably through its better learned representation.
Going further, the authors introduce their mixture model, before showing how it generalizes existing approaches. Comparing against [47], though, FedEM requires sending all M=C models to all clients, whereas [47] only ever communicates 1 model.
The authors proceed to describe learning in the mixture model, both in the centralised and in the federated setting. Then the authors introduce a decentralised variant as well as a generalised 'federated surrogate optimization' few, without showing experiments in the later section. It seems to me that the paper would be improved by moving those two extensions to the appendix / a separate paper, as it doesn't add to the main contribution of the paper, which is the mixture model optimised with EM in the FL setting.

The experimental section is characterised by breath of data-sets considered, as well as competing methods.
There are, however, a few trivial baselines that I would like to see included:
The authors consider personalised model accuracy, i.e. not the performance of a global model at the server (as in FedAvg). As such, the authors need to include the performance of FedAvg when the last global model is allowed to be fine-tuned on the local data-sets for an appropriate amount of steps. Assuming $D_{train,t} \sim p(D_t)$ and  $D_{test,t} \sim p(D_t)$, fine-tuning should improve the performance of the global model. Moreover, due to the limited size of local training-sets, overfitting needs to be taken into account - similarly to how the authors speculate about $\pi_t$ to overfit due to its localised nature.

Another trivial baseline to compare FedEM(m) with is a model with m-times the parameter count. It needs to be established that the additional model performance is not due to additional model parameters and instead indeed due to the mixture formulation. Furthermore it would be interesting to see the entropy of $\pi$ to understand if specialisation is happening. Is there a guarantee or at least an empirical observation that EM can recover a 'true' clustering, i.e. assuming all clients within a cluster share the same data-distribution?

The authors are talking about a synthetic dataset, but do not include what that is about (also not in the appendix).

I find it peculiar that except for Figure 1, all experiments do not include client sub-sampling. Full participation is a strange assumption to make in FL and I would consider the experimental section much stronger if the authors consider the more established settings. E.g. FEMNIST usually has >3.4k clients, sampling around 100 clients per round. Otherwise the authors should re-focus their claims for the cross-silo FL setting.
Furthermore the authors make the in my opinion very strong assumption that there will be a large-enough labelled training set available at a new client. What I would like to see is the average performance of new clients with zero labelled examples (and therefore uniform $\pi$, adding progressively more labelled examples, up until the full local labelled training set is assumed to be available.

I see that the code to reproduce experiments is attached in the supplementary, however I would appreciate a table of hyper parameters in the actual paper (appendix).

I have not read the proofs of various propositions/theorems in the paper.

If the authors improve their empirical evaluation along the lines that I have suggested, I will consider raising my score.


**Time Spent Reviewing:**

5

---

> ### Author Response · Authors · 2021-08-10
> **Specific answers to 3rd reviewer (2/2)**
>
> **Clustered FL** The reviewer correctly observes that the direct
> implementation of FedEM leads to transmit $M$ times more information than
> clusteredFL at each round, but we observe that
> 1. In all experiments "the number of communication rounds
> \[($K=80$ rounds for Shakespeare and $K=200$ rounds for the other datasets)\]
> allowed all approaches to converge. \[See figures 2&ndash;7 in Appendix K\].
> As a consequence, even if other methods trained over $M = 3$ times more
> rounds&mdash;in order to have as much computation and communication as
> FedEM&mdash;the conclusions would not change" (lines 336&ndash;339),
> hence FedEM would still outperform them.
> 2. We have also "considered a time-constrained setting, where FedEM is
> limited to run one third (= $1/M$ ) of the rounds (Table 5 in App. K.3).
> Even if FedEM does not reach its maximum accuracy, it still outperforms
> all other methods on $3$ datasets" (lines 339&ndash;341) and ClusteredFL on all 6 of them.
> 3. One can consider a version of FedEM where at each communication round
> the server transmits a single component model $\theta_m$. Clients only
> update $\theta_m$ during the round and update their mixture weights every
> $M$ rounds. It is easy to check that this version of FedEM evolves exactly
> as the one presented in the paper, at the cost of being $M$ times slower
> in terms of communication rounds.
> 4. ClusteredFL requires each client to disclose to the server the
> cluster it belongs to, potentially revealing some client's sensitive
> information. On the contrary, in FedEM, the client never reveals its
> own mixture weights.
>
> **Synthetic dataset** The synthetic dataset has been generated according to Assumptions 1&ndash;3 of our model (lines 295&ndash;296). We realize that we forgot to provide the exact details in the appendix. Here they are:
> 1. Sample weights $\pi_{t},~t\in\[T\]$ from Dirichlet(\alpha)
> 2. Sample $\theta_{m},~m\in\[M\]$ from $\mathcal{U}(\[-1,1\]^{d})$
> 3. Sample $m_{t},~t\in\[T\]$ from a log-normal distribution with mean
> $4$ and sigma $2$, then set $n_{t} = \min(50 + m_{t}, 1000)$
> 4. For $t \in\[T\]$ and $i \in \[n_{t}\]$, draw
> $x_{t}^{(i)} \sim \mathcal{U}(\[-1, 1\]^{d})$, and
> $\epsilon_{t}^{(i)} \sim \mathcal{N}(0, I_{d})$
> 5. For $t \in\[T\]$ and $i \in \[n_{t}\]$, draw $z_{t}^{(i)} \sim \mathcal{M}(\pi_{t})$
> 6. For $t\in\[T\]$ and
> $i \in \[n_{t}\]$, draw $y_{t}^{(i)} \sim \mathcal{B}(\text{sigmoid}(x_{t}^{(i)}\cdot \theta_{z_{t}^{(i)}} + \epsilon_{t}^{(i)}))$
>
> **Table with hyper parameters** Indeed, Appendix J.2.3 mentions how
> hyperparameters were selected, but not their final values.
> A table with their values is ready to be included in Appendix J.
>
> **Fairness** We agree with the reviewer's comment about fairness and
> will modify the text accordingly.

---

> > ### Comment · Reviewer_j2i1 · 2021-08-12
> > **Answer to rebuttal**
> >
> > I thank the authors for their well-crafted response.
> > I missed the definition of FedEM+, I'm happy this baseline is included. I agree that it is difficult to 'make equal' FedEM(m) with a m-times larger model, since one could argue that a different model-class (Mobilenet vs. ResNet) adds additional biases that change performance independent of parameter-count. I.e. a mobilenet with the same #of parameters as a ResNet might perform better or worse in the first place. On the other hand, a mobile net with increased # of channels might be limited through it's model architecture so that the additional channel count can benefit. Regardless, I believe the additional experiments to give evidence that the performance benefit of FedEM is not due to the increase parameter count alone. Also thank you for additional experiments with client sub-sampling. Out of curiosity: Why did you choose full participation as the default setting initially?
> >
> > Thank you for your exposition on the cluster scenario. Indeed I was interested in if you observe the same predictor for two clients in the same cluster. Can you make a statement about if FedEM indeed recovers these clusters in practice? I realise that the $pi_{tm}$ are not shared with the server - assuming for a moment that that would be allowed, could you identify if two clients share the same data-characteristics, i.e. p(D_t)=p(D_t') based on those. Or more generally speaking, could you recover a similarity-matrix between clients?
> > Since the $pi_{tm}$ are not shared with the server, can you recover an estimate of $pi_{tm}$ by inspecting the update-magnitudes $\theta_m^{k+1} - \theta_m^{k}$ as received by the server? For a client with $\pi_{tm}^{k+1}=0$  for specific $m$, the magnitude would be 0 (and furthermore present a way to reduce communication cost). How do you judge the privacy implications of such a potential reconstruction?
> >
> > Thx for the new-client data set size experiments. I still find it peculiar that you argue the default setting in FL requires a labelled data-set on a 'new' client. A newly-installed device might join the federation long time after training has converged and wants to use the resulting model. In my understanding of FL, the standard assumption is not that every new device has a labelled training-set available. Do you have a source for your argumentation? Regardless about what can be considered 'standard', I believe that this new-device without labels setting is interesting and your experiments cater to that.
> >
> > I see my error in understanding the impossibility theorem - I encourage the authors to improve their exposition there, based on their rebuttal explanation and other reviewer's feedback.
> >
> > All together, I see my concerns addressed and will raise my score to 7.

---

> > > ### Author Response · Authors · 2021-08-14
> > > **Answer to reviewer j2j1**
> > >
> > > We thank the reviewer for the interesting discussion as well as for having updated the score.
> > >
> > > **Full participation vs sampling**. There was no particular reason to consider full participation as default setting and we agree it would have been better to select partial clients' participation as the default one. We were probably unconsciously influenced by papers with a focus on theory which often consider full participation in their experiments, see for example
> > > - Filip Hanzely and Peter Richtárik, Federated Learning of a Mixture of Global and Local Models, 2020
> > > - Yishay Mansour et al, Three Approaches for Personalization with Applications to Federated Learning, 2020
> > >
> > > Also the APFL paper (Deng et al, Adaptive Personalized Federated Learning 2020) first presents results under full participation and then a limited evaluation under client sampling.
> > >
> > > **FedEM and clustering**. We performed additional experiments with synthetic datasets to check if FedEM recovers clusters in practice. We modified the synthetic dataset generation (described in our previous answer) so that the mixture weight vector $\pi_t$ of each client $t$ has a single entry equal to $1$ that is selected uniformly at random. We considered $T=300$ clients and $M=2$ and $M=3$ components. In both cases FedEM recovered almost the correct $\Pi^*$ and $\Theta^*$: we have $\verb+cosine_distance+(\Theta^*, \breve{\Theta}) \leq 10^{-2}$ and   $\verb+cosine_distance+(\Pi^*, \breve{\Pi}) \leq 10^{-8}$. A simple clustering algorithm that assigns each client to the component with the largest mixture weight achieves $100\\%$ accuracy, i.e., it partitions the clients in sets coinciding with the original clusters.
> > >
> > > We also considered synthetic datasets generated as in the original submission for $10$ clients. This corresponds to the setting suggested by the reviewer, where clients are not split into disjoint clusters and have different levels of similarity with each other. We quantify the dissimilarity of clients $t$ and $t'$ using the Jensen-Shannon (JS) divergence of the joint distributions $p_t(x,y|\Theta, \pi_t)$ and $p_{t'}(x,y|\Theta, \pi_{t'})$ (denoted as JS$(p_t, p_{t'})$). We can estimate JS$(p_t, p_{t'})$ using a large set of (unlabeled) input vectors drawn from the marginal distribution $p(x)$. Let $\Omega(\Theta, \Pi)$ denote the matrix of pairwise distances, that is the entry at the $t$-th column and $t’$-th row  equals JS$(p_{t}||p_{t’})$. Our experiments show that $\Omega(\breve{\Theta}, \breve{\Pi})$ is close to $\Omega(\Theta^*, \Pi^*)$ as their cosine distance is smaller than $0.12$ (the cosine distance considers only the upper triangular parts of the two matrices).
> > >
> > > In terms of privacy leakage, we do not exclude that the server may be able to recover some information about a client's personal weights if the client participates in a large number of communication rounds. At the same time, mixture weights' inference may be more complex than it seems at first glance. For example, small updates to a given component may indicate that the client gives a small weight to it (as the reviewer correctly mentioned), but also that the component already converged for the client (see line 18 in Algorithm 2 in Appendix D.1). Secure aggregation and differential privacy techniques may be used to prevent privacy leakage. In the conclusion of the paper, we mention personalized FL approaches under privacy constraints as a promising future research direction, which seems to be quite unexplored with the notable exception of Bellet et al, Personalized and Private Peer-to-Peer Machine Learning, 2018.
> > >
> > >
> > > **New clients**. We found the idea of new clients arriving with a labeled dataset to be consistent with FL training assumption. Both Jiang et al (Improving Federated Learning Personalization via Model Agnostic Meta Learning  2019) and Shamsian et al (Personalized Federated Learning using Hypernetworks, 2021) consider that new clients already have a labeled dataset as we do.
> > > We do not have other pointers. To the best of our knowledge, the issue of model personalization for clients unseen at training is not much explored in the literature. See also the paragraph "Extension to unseen clients" in our [answer](https://openreview.net/forum?id=YCqx6zhEzRp&noteId=D6WP50MJd9X) to reviewer kvAM, where we discuss how pFedMe or ClusteredFL could be extended to unseen clients (an issue ignored in the original papers). If the reviewer has in mind some other references, we would like to read them.

---

> ### Author Response · Authors · 2021-08-10
> **Specific answers to 3rd reviewer (1/2)**
>
> We thank the reviewer for the detailed review and numerous suggestions to improve our empirical evaluation. We have tried our best to address the reviewer's requests about the empirical evaluation by providing more details and running additional experiments. We hope that he/she finds the empirical evaluation of the paper improved and will raise his/her score accordingly.
>
> **Paper reorganization** See the discussion in the general answer.
>
> **Impossibility result in Sec. 2.1** We suggest reorganizing the section
> as indicated in the general answer. About the potential counter-example
> proposed by the reviewer, the assumption that features learned by client
> $t$ are beneficial to client $t'$ can be formulated in a probabilistic way
> as follows: there exists a latent random variable $z$ (the useful feature)
> such that $p_{t}(y|z) = p_{t'}(y|z)$ or at least they are "close," which
> is indeed an assumption on the data distributions. Then, this example
> does not invalidate our conclusion, as our impossibility result is about the worst-case sample complexity (i.e., over all possible distributions, see general answer for more details). Note that, while the impossibility
> result motivates the need for *some* assumption on the local
> distributions, it does not lead to conclude that this is necessarily
> our Assumption 1. That being said, we observe in Sec. 2.3 how other
> FL approaches make&mdash;sometimes implicitly&mdash;assumptions on the local
> distributions, which can be interpreted as particular cases of ours.
> Assumption 1 then seems to provide a good level of generality.
>
> **Additional baselines** The reviewer suggests two baselines we could compare with.
> The first one is FedAvg with additional fine-tuning on the local datasets
> to personalize the model. We note that this is exactly what we call
> FedAvg+ in our paper (lines 300--302). In the submitted version, FedAvg+
> performed a single pass on the local dataset (one epoch).
> Hanzely and Richtárik (2020) provided some theoretical support for this choice.
> We have run some additional experiments to tune optimally FedAvg+'s
> number of epochs for fine-tuning. We considered $1$, $2$, $3$ and $4$ as possible
> values. For CIFAR10, CIFAR100, and Synthetic dataset, the optimal choice
> was one epoch, hence previously reported results for FedAvg+ do not change.
> We did not experiment with shakespeare and femnist, due to the time limit.
>
> The second baseline suggested by the reviewer is a model with the same number of parameters as
> FedEM to check if FedEM's advantage comes from the additional model
> parameters rather than by its specific mixture formulation. To this purpose,
> we trained Resnet-18 and Resnet-34 on CIFAR10. The first one has about $3$ times more
> parameters than MobileNet-v2 and then roughly as many parameters as FedEM with $M=3$.
> The second one has about $6$ times more parameters than FedEM with $M=3$.
> We observed that both architectures perform even worse than MobileNet-v2,
> so the comparison with these larger models does not suggest that FedEM's
> advantage comes from the larger number of parameters.
>
> We note that there are many possible choices of (more complex)
> model architectures, and finding one that works well for the task at
> hand is quite challenging due to the large search space,
> the bias-variance trade-off, and the specificities of the FL setting.
> We believe that a broader search is outside the scope of our paper.
>
> **Entropy of $\pi$** There are already some results about the entropy
> of $\pi$ in Appendix K.2 (lines 1236&ndash;1238). They show that, on CIFAR-100,
> the average entropy of the vector $\pi$ across clients is $0.27$ for old clients,
> hence there is some degree of specialization (with different clients
> assigning a large weight to different components). Indeed,
> as a reference, the entropy of the uniform distribution over the $M=3$
> components is $\ln(3)\approx 1.10$, and a distribution with $\pi_1=12/13$,
> $\pi_2=1/13$, and $\pi_3=0$ would have entropy $0.27$.
>
> **FedEM and clustering** This is an interesting question that demands an elaborate answer.
> Consider that clients are perfectly partitioned in $M$ clusters with clients in cluster $i$
> having the same optimal predictor $h_{\theta_i^*}$.
> We first want to stress that, in general, there is no one-to-one correspondence
> between the $M$ components learned by FedEM and the $M$ optimal
> predictors: each client will then combine multiple components with
> non-zero mixture weights.
> This is due to the fact that problem (4) may have multiple
> equivalent global minima, as we show in an example below.
> Nevertheless, it is still possible to recover the underlying clustering
> as all clients in the same cluster may learn close personalized predictors,
> that is they may have very close mixture weights distributions.
> In fact, proposition 2.1 guarantees that if $\breve \Theta$ and $\breve \Pi$
> solve problem (4) then the linear combinations in (5) solve problem (1).
> It follows that, if problem (1) admits a unique solution, problem (4)
> leads all clients in the same cluster to learn the same predictor,
> recovering the underlying clustering. As long as the empirical risk minimization
> in (6) is a reasonable approximation of (4), solving problem (6) can also recover the clustering.
> Unfortunately, FedEM is not guaranteed to find a global optimum of
> problem (6): even for classic applications as parameter estimation
> for a Gaussian mixture, EM methods are only guaranteed to converge
> to a stationary solution (e.g., to a local maximum of the log-likelihood)
> and not to the true parameters. FedEM shares similar theoretical guarantees.
> In principle, two clients in the same cluster can thus learn different mixture
> weights $\pi_t$ and $\pi_{t'}$, but this happens only if the pair $(\breve \Theta, \pi_t)$ is a stationary point of $\sum_{i=1}^{n_t} \log p(s_t^{(i)}| \Theta, \pi_t)$ and the pair $(\breve \Theta, \pi_{t'})$ is a stationary point of $\sum_{i=1}^{n_{t'}} \log p(s_{t'}^{(i)}| \Theta, \pi_{t'})$.
>
> *Example*: We illustrate here that there may be different values of $(\breve \Theta, \breve \Pi)$
> corresponding to different global minima of problem (4). Consider the following
> simple example: 4 clients split in two clusters, one with linear model $h_{\theta_1}$,
> the other with linear model $h_{\theta_2}$. One could expect FedEM to learn the
> components $\breve \theta_1= \theta_1$ and $\breve \theta_2= \theta_2$ with
> mixture weights $(1,0)$ for clients in the first cluster and $(0,1)$
> for clients in the second cluster.
> This is indeed a global minimum of problem (4). But another global minimum is,
> for example, $\breve \theta_1= 4\theta_1 - 3\theta_2$
> and $\breve \theta_2= 3\theta_2 - \theta_1$ with mixture weights $(1/2, 1/2)$
> and $(1/3, 2/3)$ for clients in the first and in the second cluster, respectively.
> Note how clients in the same cluster have the same mixture weights, hence
> the same local predictor is still $h_{\theta_1}$ for cluster $1$
> and $h_{\theta_2}$ for cluster $2$. It is then still possible to correctly
> recover the correct clustering, but it does not correspond to specific components.
>
> **Client sampling** We have run new experimental results when sampling $20\\%$
> of the clients for CIFAR-100, CIFAR-10, and the synthetic dataset.
> They are reported in the table in the general answer. These new results
> qualitatively confirm those in our original submission.
>
> **New clients** Note that FL literature commonly assumes that clients
> recruited at training time have a labeled dataset available
> (even if this is not large enough to allow them to autonomously train a model).
> "New" clients are simply clients who did not participate in
> training (lines 152&ndash;153), e.g. because they were not connected
> to WiFi or plugged in. Considering that they own a labeled training
> set is then aligned with standard FL assumptions.
>
> Despite the considerations above, we think the experiment suggested by
> the reviewer for clients who progressively collect their own dataset is
> an interesting one that we decided to perform. The results show that FedEM achieves
> a significant level of personalization as soon as clients collect a labeled
> dataset whose size is about $20\\%$ of what the old clients used for training.
> In particular, on CIFAR-100, new clients joining the system without labeled
> data simply average the M components and experience an initial accuracy equal
> to $46.7\\%$. The accuracy then progressively increases as clients collect more
> and more labeled data and use it to personalize their mixture weights.
> Accuracy reaches $47.5\\%$ (see Table 3 of the paper) when local datasets have size comparable to the datasets
> of old clients, and it is already $47.2\\%$ when the relative size reaches $20\\%$.

---

### Official Review · Reviewer_mgAM · 2021-07-15

**Rating:** 7
**Confidence:** 4

**Summary:**

The paper proposes an approach to learn local models in a federated learning setup where local distributions are assumed to be a mixture of a given set of ground distributions. The idea is to learn a ground model for each ground distribution and the mixture coefficients for the local distributions, and then obtain a local model as the weighted average of ground models (weights are given by the mixture coefficients).

**Ethical Concerns:**

The potential ethical concerns are related to the societal impact and thus discussed together.

**Limitations And Societal Impact:**

The paper addresses the limitations of the approach. The paper discusses societal impact in sufficient detail.

**Main Review:**

The paper is well-written, presents the contributions clearly, and discusses relevant related work. The proposed method is sound and novel. The idea of learning a set of distributions and their mixture coefficients is an interesting take on the non-iid problem in federated learning, especially given that the paper shows this to be a generalization of other frameworks. The empirical evaluation is solid with a decent number of datasets, a large number of clients (100), and a variety of model types.

The paper puts a lot into the appendix, which is fair, but I felt that some of the results (especially the fully decentralized algorithm and the federated surrogate optimization) feel a bit displaced in the main text and could be moved to the appendix to rather provide more details on the main contributions in the main text.

Overall, the paper provides a solid contribution that is backed by theory and empirical results. I vote for acceptance.

Detailed comments:

- the negative result is presented too informally and in its formulation only holds for equal feature distributions. At the same time, the result seems quite trivial: if one client holds no relevant information for another client, then no form of collaboration can improve local learning.
- way too many arxiv citations, including older ones (2016, 2018). For some, this is just due to lazy bibliography writing. E.g., [28] is actually published: Kang, Jiawen, et al. "Incentive design for efficient federated learning in mobile networks: A contract theory approach." 2019 IEEE VTS Asia Pacific Wireless Communications Symposium (APWCS). IEEE, 2019. and [32] was published at the Workshop on Private Multi-Party Machine Learning.
- In order to overcome the limitation of assumption 2, the authors might consider using the results in [1], which deal with different feature distributions p_m(x), but do not address different conditionals.

[1] Li, Xiaoxiao, et al. "FedBN: Federated Learning on Non-IID Features via Local Batch Normalization." International Conference on Learning Representations. 2020.

----------------------------------------------------------------------------------

I have read my fellow reviewer's reviews and the authors rebuttal. Many valid points of critique have been adequately addressed by the authors. While some points still remain open, e.g., the proper discussion of related work in PFL, I am still convinced that this is a good paper with an interesting take on the problem of non-iid local data. So, while this is surely not a perfect paper, I think it is a valuable contribution to the conference and I keep my vote on acceptance.

**Time Spent Reviewing:**

6

---

> ### Author Response · Authors · 2021-08-10
> **Specific answers to 2nd reviewer**
>
> We thank the reviewer for the positive evaluation of our work and useful suggestions.
>
> **Paper reorganization.** See the discussion in the general answer.
>
> **Impossibility result in Sec. 2.1** Please refer to the general answer
> for clarifications. Note that, being a negative result, the fact that
> we prove it under the benevolent assumption of equal feature
> distributions makes the result stronger: obviously if marginals are
> not equal across nodes, learning will be even more difficult. We are
> not sure this result is "quite trivial": for example, the third reviewer
> finds it quite counter-intuitive. That being said, as mentioned in the general answer, we are considering moving this section to an appendix, also to "provide more details on the main contributions in the main text" as suggested by the reviewer. What does the reviewer think about it?
>
> **FedBN** We thank the reviewer for the suggestion. FedBN is indeed
> complementary to our approach as it learns personalized models to account
> for feature shift, i.e., clients having the same $p(y|x)$, but different
> $p(x)$ or the same $p(y)$ but different $p(x|y)$. On the contrary, our
> FedEM accounts for concept shift, as it allows each client to have a
> different conditional probability distribution $p(y|x)$. While in this
> paper we focus on the case of identical marginals $p(x)$ (Assumption 3),
> we want to highlight that "Assumption 2 is not strictly required for
> our analysis to hold, but, in the most general case, solving Problem (1)
> requires to learn generative models" (lines 129&ndash;130). It is indeed
> possible to extend FedEM to generative models, but Assumption 2 allows
> us to "restrict our attention to discriminative models
> (e.g., neural networks)" (lines 130&ndash;131), which are more of interest
> for current ML applications. Having said that, combining FedEM with
> FedBN's batch normalization techniques could indeed be an alternative
> approach to tackle both types of shift.
>
> **Bibliography** We have checked again all our citations.
> We identified 8 arxiv papers which are indeed published,
> and an additional 5 papers which have been presented at workshops
> without proceedings (including \[32\]). We have corrected the
> corresponding entries. Thanks!

---

> > ### Comment · Reviewer_mgAM · 2021-08-11
> > **Answer to authors**
> >
> > Dear authors,
> >
> > Thank you for the answer and clarifications. Regarding the impossibility result, I did not mean to imply that the proof of the theorem was trivial, rather that it is a commonly known fact that if two tasks have nothing at all in common, you cannot transfer knowledge from one to the other. Thus, the result is fairly intuitive. The proof by reduction on semi-supervised learning is neat and shows why indeed the assumption of common feature distribution makes the result stronger, since the tasks do have their input distribution in common.
> >
> > I also agree that moving content to the appendix to give more details on the main contributions is a good idea and Sec. 2.1 is a good candidate. I still think that the sections on fully decentralized FL and the surrogate optimization are also good candidates (I feel they would work better in a journal version, where they have more room to breath), but I understand your decision. Plus, your argument that pFedMe is a special case of the surrogate optimization supports your decision.
> >
> > I agree with your assessment of FedBN, it is a complementary approach and I am looking forward to see future work that combines the two approaches.
> >
> > Thanks for cleaning the bibliography, I know that with a deadline there are often more important things.
> >
> > Regarding the communication-efficiency mentioned by reviewer kvAM, I think it would be good to have a brief discussion about the compatibility of FedEM with state-of-the-art communication reduction techniques (you already use client subsampling, but it seems to me that model compression, or dynamic averaging should be equally compatible).

---

> > > ### Author Response · Authors · 2021-08-13
> > > **Answer to reviewer mgAM**
> > >
> > > Dear reviewer,
> > > we are happy that you are satisfied with our answers and we thank you for the suggestion about FedEM communication efficiency. Indeed, we agree that FedEM is compatible with any communication reduction technique that can be used with FedAvg, including model compression and dynamic averaging. These techniques can simply  be applied to each model component separately. We suspect that model compression could lead to additional savings if applied cross-components, but we have not carried out any experiment in this direction. See also our [answer](https://openreview.net/forum?id=YCqx6zhEzRp&noteId=PjsN-zLa2H) to reviewer j2i1 in the "Clustered FL" paragraph, where we discuss alternative techniques to reduce FedEM communication overhead. We already briefly mentioned model compression techniques in the conclusions (lines 350-351), but we are going to add this discussion to the paper.

---

### Official Review · Reviewer_fDTX · 2021-07-15

**Rating:** 6
**Confidence:** 5

**Summary:**

This paper proposed a federated multi-task learning approach to learn personalized models.

**Limitations And Societal Impact:**

Not applied

**Main Review:**

Pros.
This paper has proposed a theretical sound algorithm for federated MTL. The idea is valid with convergence results supprt.

However, the work can be improved by addressing the following comments.

1. Section 2.1 is not well-written and self-contained. The impossibility result is not formally established and only some intuitions are given. In addition, the SSL problem is not well defined and no citation is given as well. This makes me hard to justify the motivation of the proposed work.

2. There are more recent published papers in personalized FL. For example,

a. FedBN: Federated Learning on Non-IID Features via Local Batch Normalization

b. Personalized Cross-Silo Federated Learning on Non-IID Data

These two apporaches, APFL and SCAFFOLD are able to handle clients sampling. Thus it is better to compare with them than pFedMe and clustered FL.

3. For the synethic data, the prospoed work should be compared with MOCHA which is the most closest work to the proposed work for linear model.

**Time Spent Reviewing:**

2

---

> ### Author Response · Authors · 2021-08-10
> **Specific answers to 1st reviewer**
>
> We thank the reviewer for the encouraging review and valuable references.
>
> **Impossibility result in Section 2.1** Please refer to the general answer
> for clarifications. Note that semi-supervised learning is the problem of
> learning from a training set with only a small amount of labeled data.
> We think this is well known so when the reviewer mentions "the SSL problem"
> he/she is probably referring to the impossibility result for SSL
> (for which we gave references) rather than to SSL itself.
>
> **SCAFFOLD and APFL** SCAFFOLD does not support personalized FL models.
> It uses variance reduction techniques to compensate for the potential
> client-drift due to multiple local gradient updates and improve the
> convergence properties of optimization in a federated setting.
> We see that FedAMP's paper is presenting SCAFFOLD as a personalized
> FL algorithm, but we do not know why. Algorithm 1 in SCAFFOLD's paper
> clearly shows that the local model is initialized to the global model
> at each round. Perhaps, the authors of FedAMP's paper considered that,
> after the global model is trained, each SCAFFOLD client performs a
> final fine-tuning update to obtain a personalized model, in the same
> spirit of what we call FedAvg+ in our paper. Experiments in the original
> FedAMP's paper show that this personalized version of SCAFFOLD in
> general does not perform well: its accuracy is worse than FedAvg or
> FedAvg+ for all datasets except MNIST.
>
> We have added new experiments with APFL over different datasets.
> The results are summarized in the Table provided in the general answer.
> FedEM always outperforms APFL, which is the second or the third best method.
>
> **FedBN and FedAMP** We thank the reviewer for pointing out these two very
> recent papers, which we are going to add to the related work.
>
> FedBN is complementary to our approach as it learns personalized models
> to account for feature shift, i.e., clients having the same $p(y|x)$,
> but different $p(x)$ or the same $p(y)$ but different $p(x|y)$. On the contrary,
> our FedEM approach accounts for concept shift, as it allows each client
> to have a different conditional probability distribution $p(y|x)$. While
> in this paper we focus on the case of identical marginals $p(x)$ (Assumption 3),
> we highlight that "Assumption 2 is not strictly required for our analysis
> to hold, but, in the most general case, solving Problem (1) requires to
> learn generative models" (lines 129&ndash;130). It is indeed possible to
> extend FedEM to generative models, but Assumption 2 allows us to
> "restrict our attention to discriminative models (e.g., neural networks)"
> (lines 130&ndash;311), which are more of interest for current ML applications.
> Having said that, combining FedEM with FedBN's batch normalization
> techniques could be an alternative approach to tackle both types of shift,
> as also suggested by reviewer 2.
>
> FedAMP is similar to the MTL formulation we discuss in Sec. 2.3 and Appendix B.
> It does not learn the form of the penalization term in Eq. (7) by
> learning the matrix $\Omega$&mdash;as Smith et al (2017)
> and Zantedeschi et al (2020) do&mdash;, but allows for a more general (nonlinear)
> dependence on the distance between clients' parameter vectors.
> In particular, it can be seen as a generalization of
> FedU (but FedAMP predates FedU), where the server linearly combines the
> clients' parameter vectors with coefficients that are themselves
> functions of the distance between parameter vectors.
>
> **MOCHA.** We observe that MOCHA only works for linear models.
> We did not have time to implement it (to the best of our knowledge there
> is no publicly available correct implementation in Python) and perform
> the corresponding experiments.

---

> > ### Comment · Reviewer_fDTX · 2021-08-26
> > **Answer to rebuttal**
> >
> > Thanks for addressing my comments. I agree with most of your replies. However, I believe MOCHA is worthy to be implemented and compared.

---

> > > ### Author Response · Authors · 2021-08-29
> > > **Comparison with MOCHA**
> > >
> > > We implemented MOCHA in Python following the official implementation written in MATLAB; the corresponding code will be made publicly available.
> > >
> > > We tuned the parameter $\lambda$ of MOCHA on a holdout validation set via grid search in $\\{10^{1}, 10^{0}, 10^{-1}, 10^{-2}, 10^{-3}\\}$, and we found that the optimal value of $\lambda$ is $10^{0}$. For this value, we ran MOCHA on the synthetic dataset  with three different seeds, and we found that the average accuracy is $73.4 +/- 0.05$, in comparison to $74.7 +/- 0.01$ achieved by FedEM. MOCHA is the second best method after FedEM on this dataset. Unfortunately, MOCHA only works for linear models.
> > >
> > > If the reviewer is satisfied with our previous answers and the additional experiments with APFL and MOCHA, we would appreciate it if he/she would consider updating the global score.

---

### Author Response · Authors · 2021-08-10
**General answer to all reviewers**

We thank the reviewers for their valuable comments and suggestions to improve our paper.

Overall, the reviewers liked the idea of studying personalized federated
learning under the assumption that local distributions are a mixture of a given set of ground distributions. They found this idea "valid",
"sound and novel", "an interesting take on the non-iid problem in FL",
and "interesting". They also appreciated that our FedEM algorithm is
"backed by theory" and "theoretically sound", with "convergence results
 support" and "a good theoretical analysis."

The reviewers asked us to

1. discuss the relation of FedEM with other very recent approaches like
FedBN, FedAMP, FedFOMO, and pFedHN.
2. improve the experimental evaluation with some additional experiments,
even though one reviewer observes that "the empirical evaluation is solid
with a decent number of datasets, a large number of clients ($100$),
and a variety of model types" and another that "the experimental
section is characterised by breath of data-sets considered,
as well as competing methods."
3. improve the paper presentation especially regarding the impossibility
results in Sec. 2.1.

**Relation to recent work** We discuss in detail the differences with FedBN, FedAMP, FedFOMO, and pFedHN in the specific replies to each review. In a few words, in the comparison FedEM stands out with the following features:

1. theoretical guarantees,
2. decentralized algorithm without the need for a parameter server,
3. possibility to deal with extreme cross-device settings where each
client participates in training for a few communication rounds,
4. a very simple personalization algorithm for clients unseen at
training: the new client needs only to retrieve the component models
and compute its own mixture weights.

**Additional experiments** We carried out the following additional experiments:

1. comparison with a new sota algorithm (APFL),
2. effect of client sampling on a larger number of datasets, complementing the results in Figure 1,
3. improved hyper-parameters tuning for all experiments in the paper
   (with no appreciable difference),
4. experiments were carried out multiple times to add standard deviation estimates,
5. evaluation of the average entropy of the mixture-weights across clients to
   estimate the level of "specialization",
6. comparison of FedEM training MobileNet-v2 with FedAvg training more complex
   architectures (Resnet-18 and Resnet-34),
7. FedEM improvements when clients unseen during training join the system without a labeled dataset and progressively collect it.

The following table summarizes some new experimental results related to the
first 4 points mentioned above, and is referred to in the individual replies below.

|Dataset         | FedAvg| FedAvg+ | pFedMe | APFL | FedEM (Ours)|
| --------- |------|-------|------ |----------| ---------|
| CIFAR10 | 73.1 +/- 0.14 | 77.7 +/- 0.16 | 77.8 +/- 0.07 | 78.2 +/- 0.27 | 82.1 +/- 0.13|
| CIFAR100 | 40.6 +/- 0.17 | 39.7 +/- 0.75 | 39.9 +/- 0.08 | 40.3 +/- 0.71 | 43.2 +/- 0.23|
| Synthetic |   68.2 +/- 0.02          |  69.0 +/- 0.03 |  69.1 +/- 0.03   |  69.1 +/- 0.04  | 74.7 +/- 0.01 |

Table: Test accuracy  under $20\\%$ client sampling: average across clients with
+/- standard deviation over independent
runs. All experiments with $1200$ communication rounds.

Finally, as three reviewers asked for more explanations about the impossibility result and two suggested moving some parts to the appendices, we discuss these aspects below.

**Impossibility result in Sec. 2.1** We provided the main reasoning
steps and the relevant references, but we acknowledge that this part
could be better developed. We give additional details here. We consider the classic PAC learning framework where we fix a class of models and seek a learning algorithm which is guaranteed, for all possible data distributions, to return a model with test accuracy $\epsilon$-close to the best possible error in the class (with high probability). The worst-case sample complexity then refers to the minimum amount of labeled data required by any algorithm to reach a given $\epsilon$-approximation.
Our impossibility result for FL is based on a reduction to an impossibility result for Semi-Supervised Learning (SSL), which is the problem of learning from a training set with only a
small amount of labeled data. Ben David et al (2008) conjectured that
"when the quantity of unlabeled data goes to infinity, the worst-case
sample complexity of SSL improves over supervised learning at most by
a constant factor that only depends on the hypothesis class"
(lines 112--114). This conjecture was later proved in
Darnstadt et al (2013) and Gopfert et al (2019). In the context of FL,
even if the marginal distributions $p_t(x)$ are identical,
if the distributions $p_t(y|x)$ can be arbitrarily different then
each client $t$ can learn using: 1) its own local labeled dataset,
and 2) the other clients' datasets, but only as unlabeled ones
(because their labels have no relevance for $t$). The FL problem then
reduces to $T$ parallel SSL problems, or more precisely, it is at least
as difficult as $T$ parallel SSL problems (because client $t$ has no direct
access to the other local datasets but can only learn through the
communication exchanges allowed by the FL algorithm). In other words, without any assumption on the local distributions $p_t(x,y)$, the minimum size of client $t$'s dataset
needed for $t$ to reach an $\epsilon$-approximation is reduced at most by a constant compared to the setting where client $t$ learns only with its own local dataset.
This conclusion motivates the need for specific assumptions on the
local distributions, such as our Assumption 1. We also discuss in
Sec. 2.3 how other FL approaches make&mdash;sometimes implicitly&mdash;assumptions
on the local distributions, which can be interpreted as particular cases
of ours. We could reorganize Sec. 2.1 according to the presentation
provided above, but we stress the fact that the only goal of this section is to justify why some assumptions on the distributions are needed. Therefore, if the reviewers find that such necessity is evident, we propose to move the
section to an appendix.

**Paper reorganization**. Two reviewers suggested moving the sections on
the fully decentralized algorithm and the federated surrogate
optimization framework to appendices, to be able to provide more details
about the other contributions. We agree that the paper is quite dense,
but we think the fully decentralized algorithm and the federated surrogate
optimization framework are two important contributions that should appear
in the main text. Indeed, there are not many papers describing fully
decentralized algorithms for FL with convergence guarantees. Furthermore,
the surrogate optimization framework is of general interest as it can be
used to study other federated learning methods, including for example
pFedMe (see details below). By moving Sec. 2.1 to an appendix&mdash;as we
suggest above&mdash;we will be able to expand the main contributions.

We show that pFedMe can be studied through our federated surrogate
optimization framework. With reference to the general formulation of
pFedMe problem in eqs. (2) and (3) of Dinh et al (2020), consider:

$
    g_{t}^{k}(w) = f_{t}(\theta^{k-1}) + \frac{\lambda}{2} \cdot || \theta^{k-1} - w \||^{2},
$

where,

$
    \theta^{k-1} = \arg\min_{\theta}\left\[ f_{t}(\theta) + \frac{\lambda}{2} ||\theta - w^{k-1} ||^{2}\right\].
$

It is easy then to prove that $g_{t}$ is a first order surrogate of $f_{t}$
using the envelope theorem.

pFedMe can then be seen as a particular case of the federated surrogate
optimization algorithm in our paper (Alg. 3), to which our convergence results apply.

---

> ### Author Response · Authors · 2021-08-26
> **follow-up**
>
> Dear reviewers, we just wanted to follow up to see if our responses adequately addressed
> your concerns/questions, in particular those of reviewers fDTX
> and kvAM whom we have not heard from after our first rebuttal. We are happy to discuss any point in more depth if needed.

---

### Decision · Program_Chairs · 2021-09-27

**Decision:**

Accept (Poster)

**Comment:**

The reviewers are generally in favor of accepting the paper for its algorithmic and theoretical contributions on federated multi-task learning. Based on that, I recommend acceptance. However, please make sure to incorporate the reviewers' comments and the rebuttal into the final version.